# A Generative Foundation Model for Heterogeneous Tabular Data

Xiangjian Jiang [1]   Mingxuan Liu [2]   Nikola Simidjievski [3][1]   Tassilo Klein [2]   Mateja Jamnik [1]

## Abstract

Generative modelling is a demanding test of foundation models, because it requires robust, holistic representation learning for a given data modality, rather than optimisation for a supervised prediction target alone. While recent work on tabular foundation models has achieved remarkable progress in predictive modelling, generative tabular foundation models remain underexplored. Existing tabular foundation generators, in particular, have not yet consistently matched strong dataset-specific generators in synthetic data quality. A key reason is their misalignment with the distinctive causal structural prior of heterogeneous tabular data. In this paper, we address this gap by introducing a novel tabular foundation model, **TabFORGE**, built on pretrained **Tab**ular **FO**undational **R**epresentations for **GE**neration. TabFORGE is designed to utilise the implicitly learned causal information underlying diverse tabular datasets in a unified latent space induced by a pretrained causality-aware feature encoder. It further decouples latent modelling from decoding through a two-stage design: we first pretrain a score-based diffusion transformer, and then pretrain a denoising-aligned decoder using the denoised latent embeddings. This design elegantly mitigates the distribution shifts in latent embeddings that typically arise between training and inference. We evaluate TabFORGE comprehensively against 22 benchmark methods on 45 real-world datasets. Our results show that TabFORGE effectively learns and leverages generalisable tabular representations, enabling efficient generation of high-quality synthetic tabular data, particularly with strong structural fidelity.

[1]Department of Computer Science and Technology, University of Cambridge, UK [2]SAP SE [3]Télécom Paris, Institut Polytechnique de Paris, France. Correspondence to: Xiangjian Jiang <xj265@cam.ac.uk>.

*Proceedings of the $2^{nd}$ ICML Workshop on Foundation Models for Structured Data*, Seoul, South Korea. 2026. Copyright 2026 by the author(s).

## 1. Introduction

Tabular data is ubiquitous across a wide range of real-world applications (Borisov et al., 2022a; Shwartz-Ziv & Armon, 2022; Gorishniy et al., 2021; Somvanshi et al., 2026; Jiang et al., 2024). However, acquiring high-quality tabular data can often be intricate and expensive (Hernandez et al., 2022; Shi et al., 2025b; Jiang et al., 2026), which underscores the need for powerful generative methods tailored to the tabular modality. Generative modelling has long been recognised as challenging (Fang et al., 2024; Bond-Taylor et al., 2021; Salakhutdinov, 2015), as it requires not merely recognising patterns of a single predictive target, but the learning of highly generalisable representations that capture the comprehensive data structures (Jiang et al., 2026). As such, conventional dataset-specific *tabular* generators (Xu et al., 2019; Zhang et al., 2023; Shi et al., 2025a), typically trained from scratch on each individual dataset, often yield suboptimal performance because they are unable to improve generalisability with transferable knowledge across diverse datasets (Van Breugel & Van Der Schaar, 2024). In contrast, for other modalities like text and images, several generative foundation models have been developed to address this limitation (Achiam et al., 2023; Guo et al., 2025; Lu et al., 2025; Zheng et al., 2026; Bommasani et al., 2021). This success in text and vision highlights the importance of exploring tabular foundation generators.

Prior work (Borisov et al., 2022b; Lin et al., 2025b; Margeloiu et al., 2024; Grinsztajn et al., 2025; Qu et al., 2026) has attempted to address tabular data generation via tabular foundation models. However, existing foundation models still exhibit three primary limitations: *(i) Misaligned generative process.* Some models (Borisov et al., 2022b; Grinsztajn et al., 2025; Zhang et al., 2025c) treat tables as a sequential modality (i.e., each sample as a sentence and each feature as a token), thereby performing autoregressive generation with large language models (LLMs) or tabular foundation predictors. However, autoregressive generation inevitably introduces feature-order bias. This can be problematic because the imposed order may deviate substantially from the topological order derived by the underlying causal structures (Jiang et al., 2026). *(ii) Neglect of global causal structures.* Other studies (Margeloiu et al., 2024; Ma et al., 2023) construct Energy-Based Models (EBMs) using tabular foundation predictors. However, existing EBMs are inher-

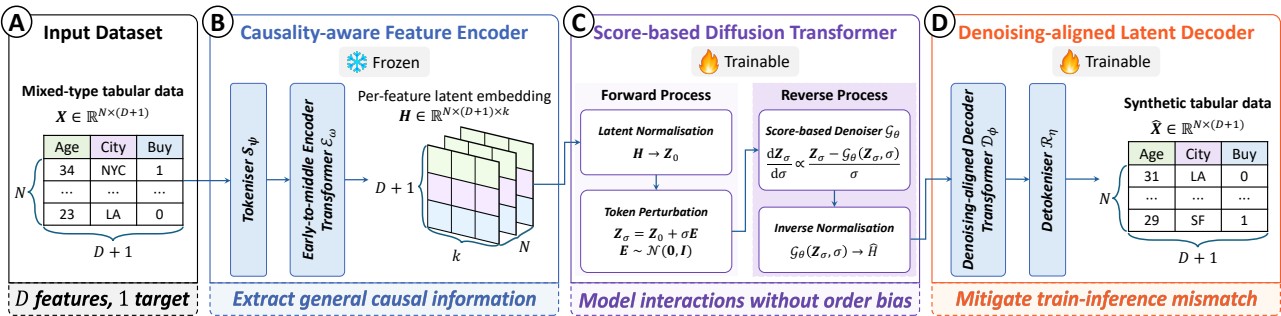

*Figure 1.* **The architecture of TabFORGE**. **(A)** Given a tabular dataset with $D$ features and one target, TabFORGE treats the target as an additional feature for generative modelling. **(B)** The frozen causality-aware feature encoder maps the table into per-feature latent embeddings by first tokenising the features and then contextualising them with inter-feature causal interactions. **(C)** The trainable score-based diffusion transformer learns the latent distribution by denosing noisy latent embeddings while preserving the token structure. **(D)** Once the diffusion transformer has been fitted, the trainable denoising-aligned latent decoder is optimised to map the denoised embeddings back to the original feature space. During inference, generation starts from random latent noise, which is progressively denoised and decoded into synthetic tabular data.

ently biased towards the local causal structures surrounding the prediction target, while largely overlooking the causal interactions across all features (Jiang et al., 2026). *(iii) Limited handling of causal information and latent space.* Earlier work (Lin et al., 2025b) proposes pretraining a cross-dataset variational autoencoder (VAE) for tabular data, conditioned on LLM embeddings. Despite their susceptibility to metadata quality and LLM capability (Lin et al., 2025b), they can break the causal structures due to obscuring feature identity. Moreover, the latent decoder is trained on clean embeddings by the encoder, whereas at inference time, it receives denoised embeddings by the diffusion model. Such a mismatch induces distribution shifts between training and inference, further degrading model performance. Full summary of related work is available in Appendix A.

In this paper, we aim to bridge these gaps by introducing TabFORGE, a novel generative tabular foundation model, which is distinguished by three core components: (i) TabFORGE encodes inter-feature causal interactions from diverse tabular datasets into a unified latent space through a causality-aware feature encoder. Specifically, the internal representations of Prior-data Fitted Network (PFN) models have been shown to contain rich, general causal information (Swelam et al., 2025), and we hypothesise that incorporating such implicit causal signals can benefit tabular data generation. As such, TabFORGE utilises the early-to-middle layers of a trained PFN model and further freezes them to avoid latent collapse or overfitting to any specific dataset. (ii) TabFORGE then models the causal interactions embedded in the latent space by pretraining a score-based diffusion transformer across diverse datasets. This diffusion-based modelling allows TabFORGE to capture global data structures effectively (Jiang et al., 2026), without imposing a potentially misaligned feature order. (iii) TabFORGE is robust to the distribution shifts in latent space between training and inference phases. As the latent space is designed to be frozen, TabFORGE can train a denoising-aligned de-

coder directly on the denoised embeddings produced by the trained diffusion transformer, rather than on clean embeddings by the feature encoder. This is fundamentally different from VAE-based latent diffusion models (Lin et al., 2025b), in which the encoder and decoder are jointly trained to establish the latent space.

Across extensive evaluation against 22 representative tabular generators on 45 real-world datasets, TabFORGE reduces the overfitting risk and efficiently generates synthetic data that better preserves the causal structures of real data, improving global utility by a clear margin over the existing tabular foundation models.

## 2. Method

### 2.1. Problem Setup (Figure 1A)

Let $\{(\mathbf{x}^{(i)}, y^{(i)})\}_{i=1}^{N} \sim p(\mathbf{x}, y)$ denote a mixed-type tabular dataset, where each sample consists of $D$ features and one target. We use $\boldsymbol{x}_d$ to denote the $d$-th feature (i.e., a column or variable), and $x_d^{(i)}$ to denote the value of the $d$-th feature in the $i$-th sample (i.e., a cell). For notational clarity, we assume that all samples are used for training, and we treat the target as an additional feature, denoted by $\boldsymbol{x}_{D+1} \coloneqq \{y^{(i)}\}_{i=1}^{N}$. Accordingly, we refer to the entire dataset by $\boldsymbol{X} \in \mathbb{R}^{N \times (D+1)}$.

### 2.2. Causality-aware Feature Encoder (Figure 1B)

TabFORGE first maps a mixed-type tabular dataset from the original feature space $\boldsymbol{X} \in \mathbb{R}^{N \times (D+1)}$ to causality-aware embeddings $\boldsymbol{H} \in \mathbb{R}^{N \times (D+1) \times k}$ within a frozen latent space. Specifically, it tokenises ($\mathcal{S}_\psi$) each feature into a shared continuous latent space and then applies the early-to-middle layers ($\mathcal{E}_\omega$) of a pretrained PFN model to contextualise the tokens with inter-feature causal interactions. This design is motivated by the observation that PFN models are intentionally exposed to abundant causal structures during their pretraining (Grinsztajn et al., 2025; Hollmann et al.,

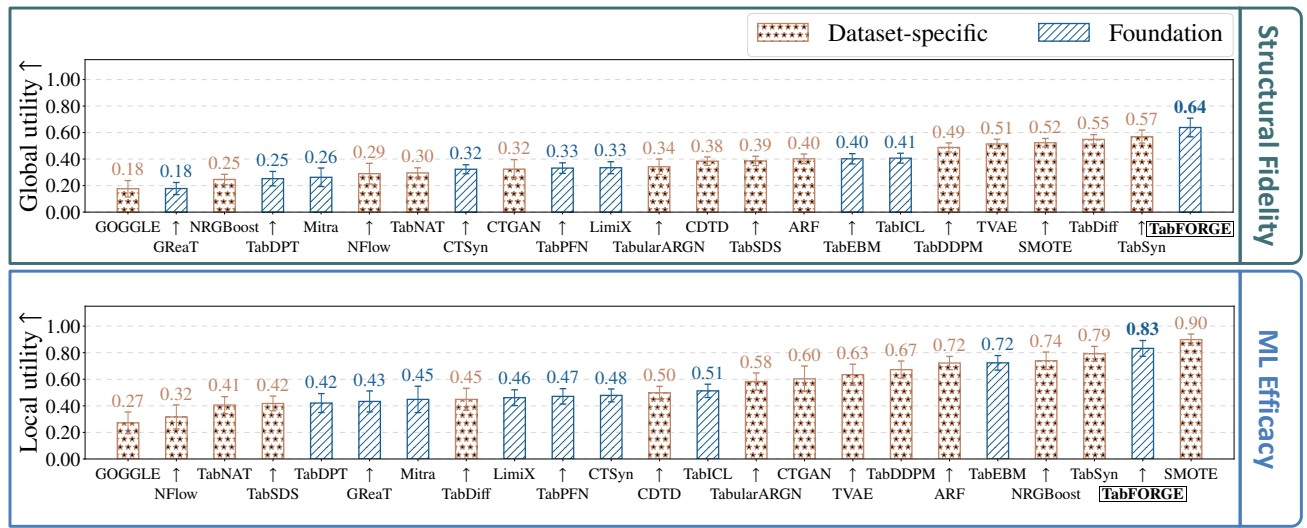

*Figure 2.* **Benchmark results of 23 generators on 45 real-world tabular datasets.** We report the normalised mean $\pm$ std metric values across datasets. **Top:** Global utility – a higher value typically indicates that the generator better captures global causal structures across all features (structural fidelity). **Bottom:** Local utility – a higher value typically indicates that synthetic data yields stronger predictive performance when used to train downstream predictors (ML efficacy). TabFORGE achieves the best overall performance, particularly surpassing benchmark methods in structural fidelity.

2025; Zhang et al., 2025b; Qu et al., 2026), and thus the early-to-middle layers of PFN models have been shown to implicitly capture general causal information (Swelam et al., 2025). As a result, TabFORGE can construct a transferable and causality-aware latent space across heterogeneous datasets. See Appendix B.2.1 for more details.

### 2.3. Score-based Diffusion Transformer (Figure 1C)

TabFORGE models latent distribution with a score-based denoiser $\mathcal{G}_\theta$ that learns to recover clean latent embeddings $\widehat{H} \in \mathbb{R}^{N \times (D+1) \times k}$ from their noisy counterparts. Specifically, the diffusion transformer maintains the one-to-one correspondence between each feature and its embedding, enabling TabFORGE to refine inter-feature interactions. Furthermore, as the transformer architecture is agnostic to the length of the token sequence (i.e., number of features), TabFORGE supports large-scale pretraining across heterogeneous datasets with varying feature sets. See Appendix B.2.2 for more details.

### 2.4. Denoising-aligned Latent Decoder (Figure 1D)

The denoising-aligned latent decoder first refines the denoised latent embeddings $\widehat{H}$ with a decoder transformer ($\mathcal{D}_\phi$) and then projects individual tokens back to the original feature space $\widehat{X} \in \mathbb{R}^{N \times (D+1)}$ with a lightweight detokeniser ($\mathcal{R}_\eta$). By training the decoder directly on denoised embeddings produced by the fitted diffusion transformer, TabFORGE aligns decoder training with the latent distribution at inference time, thereby mitigating the train-inference mismatch that commonly affects latent diffusion models. See Appendix B.2.3 for more details.

## 3. Experiments

**Real-world benchmark datasets.** We curate 45 challenging datasets from the TabArena (Erickson et al., 2025) and TabStruct (Jiang et al., 2026) benchmark suites, comprising 31 classification datasets with 748-150,000 samples and 5-119 features, and 14 regression datasets with 907-53,940 samples and 6-82 features. Full dataset descriptions are provided in Appendix C.2.

**Data preparation.** For each dataset of $N$ samples, we perform nested cross-validation with repeated shuffle (details are provided in Appendix C.3). In each repetition, we split the dataset into four disjoint subsets: 30% train set, 30% test set, 30% holdout set, and 10% validation set. We shuffle the dataset to repeat the splitting 10 times, summing up to 10 runs per dataset. All benchmark generators are fitted on the training split, denoted by $X_{\text{Train}}$, and each generator produces a synthetic dataset with $N_{\text{Train}}$ samples. We provide further details on implementation (Appendix C.4), evaluation metrics (Appendix A), and experimental setup (Appendix C.5).

### 3.1. Evaluation Results

**TabFORGE effectively preserves the global causal structures of real data.** Figure 2 (Top) shows that TabFORGE is the only foundation model that surpasses the strongest dataset-specific method, TabSyn, achieving a 12.28% (0.57→0.64) improvement in global utility. As shown in Figure 2 (Bottom), in terms of local utility, TabFORGE outperforms the previous best tabular foundation model, TabEBM, by a clear margin of 15.28% (0.72→0.83). As noted in prior work (Jiang et al., 2026), balancing global and

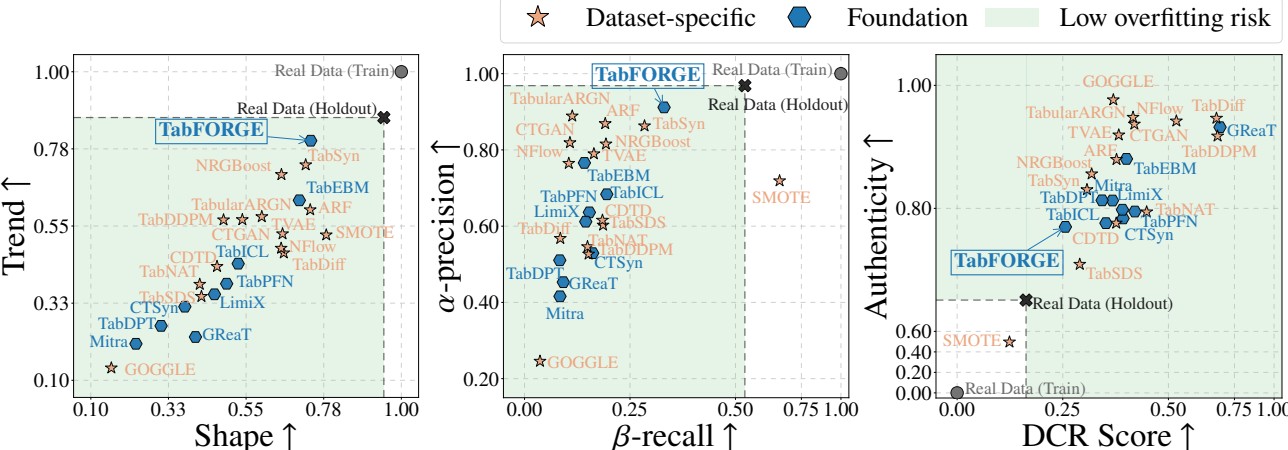

*Figure 3.* **Comparison of the fitting behaviour of 23 tabular data generators across 45 real-world datasets.** We report the normalised mean metric values across datasets, with the axis display scales adjusted for visual clarity. **Left:** Low-order density estimation, which assesses the preservation of marginal distributions (*Shape*) and inter-feature correlations (*Trend*). **Middle:** High-order density estimation, which quantifies sample-level similarity (*α-precision*) and distributional coverage (*β-recall*). **Right:** Privacy preservation, which assesses whether synthetic data avoids memorising (*Authenticity*) or closely duplicating training records (*DCR Score*). Privacy metrics should be interpreted together with the other dimensions, since overly high values may indicate a poor fit to real data. Detailed explanations are in Appendix A.2. Synthetic data that scores closer to $X_{\text{Train}}$ than $X_{\text{Holdout}}$ indicates greater similarity to the training data than holdout data, suggesting a higher risk of overfitting. Accordingly, the shaded region denotes a low risk of overfitting. TabFORGE is a strong tabular generator that achieves high performance while reducing the risk of overfitting.

local utility reflects the ability to capture both global and local causal structures. Therefore, TabFORGE preserves the global causal structures of real data more effectively than both dataset-specific generators and existing foundation models, suggesting that its design better aligns with the causal structures underpinning tabular data.

**TabFORGE reduces overfitting risk while preserving synthetic data quality.** The fitting-behaviour analysis in Figure 3 shows that TabFORGE achieves strong performance while remaining in the low-overfitting-risk region. Specifically, TabFORGE lies within the shaded region across all six metrics, covering low-order density estimation, high-order density estimation, and privacy preservation. In contrast, several high-performing benchmark methods exhibit signs of overfitting. For instance, SMOTE achieves the highest local utility (Figure 2), and outperforms the real holdout data across multiple metrics (Figure 3), including $\beta$-recall, authenticity, and DCR score. This pattern suggests that SMOTE's strong performance is likely driven by memorisation of the training data, whereas TabFORGE generalises beyond the training data rather than duplicating it.

**TabFORGE incurs low fitting and generation costs on unseen datasets.** Appendix D.3 shows that TabFORGE is substantially more efficient than most benchmark methods in both fitting and generation. In practice, TabFORGE can be readily applied to unseen datasets, with the total fitting time typically amounting to only around 9.71% of that required to train a strong dataset-specific generator (TabSyn) from scratch. Moreover, TabFORGE can generate synthetic data efficiently, requiring only around 1.66% of

the time needed by the fastest in-context tabular foundation generator (TabEBM), as TabFORGE does not require iterative in-context inference during generation by design. Moreover, the considered tabular foundation predictors use fixed prediction heads and can generate categorical features with at most 10 classes. We apply hierarchical classification (Qu et al., 2026; 2025) to enable these models to handle the considered datasets, which substantially increases their generation time. In contrast, TabFORGE employs a lightweight detokeniser that can be efficiently applied to diverse feature spaces.

**Further discussion** on synthetic data generation quality, ablation impacts of components, model practicability, and future work is available in Appendix D.

## 4. Conclusion

We introduce TabFORGE, a tabular foundation model for generative modelling. TabFORGE builds on pretrained tabular representations and introduces a causality-aware latent diffusion framework tailored to heterogeneous tabular data. Through our comprehensive evaluation against 22 benchmark generators across 45 real-world datasets, TabFORGE generates high-quality synthetic tabular data with strong structural fidelity. In particular, TabFORGE substantially improves global utility over existing tabular foundation generators and is the only foundation model in our study to surpass the strongest dataset-specific generator in global utility. Our findings further suggest that causality-aware latent representations and denoising-aligned decoding provide an effective foundation for tabular generative modelling.

## Acknowledgements

The authors would like to express their gratitude to Prof. Han-Jia Ye and Dr. Johannes Hoffart for their insightful discussions on tabular foundation models during the early stages of this project. XJ acknowledges the generous support of the Google PhD Fellowship. MJ and NS acknowledge the support of the U.S. Army Medical Research and Development Command of the Department of Defense; through the FY22 Breast Cancer Research Program of the Congressionally Directed Medical Research Programs, Clinical Research Extension Award GRANT13769713. Opinions, interpretations, conclusions, and recommendations are those of the authors and are not necessarily endorsed by the Department of Defense.

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

# Appendix

## A Generative Foundation Model for Heterogeneous Tabular Data

## Table of Contents

# A. Summary of Related Work

## A.1. Tabular Data Generator

### A.1.1. DATASET-SPECIFIC TABULAR GENERATOR

The development of dataset-specific tabular data generators has largely progressed from conventional resampling and non-parametric estimation towards increasingly expressive deep generative models (Hansen et al., 2023; Du & Li, 2024; Tu et al., 2024; Livieris et al., 2024; Lautrup et al., 2025; Kapar et al., 2025; Jiang et al., 2026; Jacob et al., 2026).

Standard and non-parametric methods generate synthetic data by directly manipulating the observed data points (Sauber-Cole & Khoshgoftaar, 2022; Chawla et al., 2002; Neto, 2025). For instance, SMOTE (Chawla et al., 2002) synthesises new samples through interpolation between real samples, while TabSDS (Neto, 2025) estimates feature-wise marginal distributions with interpolated order statistics and joint-probability-preserving shuffling. These methods are simple and often efficient, yet their expressiveness can be limited when modelling complex interactions across heterogeneous numerical and categorical features.

Another extensive line of work adapts deep generative models to the tabular modality (Xu et al., 2019; Watson et al., 2023; Kotelnikov et al., 2023; Shi et al., 2025a; Zhang et al., 2025a). VAE-based methods, such as TVAE (Xu et al., 2019) and GOGGLE (Liu et al., 2023), learn continuous latent representations of tabular data, with GOGGLE further incorporating graph neural networks to model feature dependencies. GAN-based methods, such as CTGAN (Xu et al., 2019), use adversarial training and conditional generation to handle imbalanced discrete columns. Normalising-flow methods, such as Neural Spline Flows (NFlow) (Durkan et al., 2019), learn invertible transformations between tabular data and a tractable base distribution. Tree-based density estimators, such as ARF (Watson et al., 2023), iteratively refine synthetic data through adversarial random forests. More recent methods explore generative objectives that can better capture complex mixed-type distributions. Diffusion-based generators, such as TabDDPM (Kotelnikov et al., 2023), CDTD (Mueller et al., 2025), TabSyn (Zhang et al., 2023), and TabDiff (Shi et al., 2025a), progressively denoise corrupted tabular representations and have shown strong performance on heterogeneous tabular data. Specifically, TabDDPM introduces separate diffusion processes for numerical and categorical variables; CDTD unifies mixed-type generation through continuous diffusion with learned categorical embeddings; TabSyn applies diffusion in the latent space of a VAE; and TabDiff proposes a joint continuous-time diffusion framework for numerical and categorical features. Alongside diffusion models, autoregressive tabular generators such as TabNAT (Zhang et al., 2025a) and TabularARGN (Tiwald et al., 2025) factorise the joint distribution into conditional distributions over features.

Despite this progress, dataset-specific generators remain inherently limited. Since they are fitted independently on each dataset, they cannot leverage transferable knowledge from other tabular datasets. This limitation is particularly restrictive for tabular data, where individual datasets are often small and domain-specific. Consequently, dataset-specific generators have to relearn from scratch on every dataset, which can lead to weak generalisation and overfitting to the training data.

### A.1.2. TABULAR FOUNDATION GENERATOR

The limitations of dataset-specific models motivate tabular foundation generators (Lin et al., 2025b; Grinsztajn et al., 2025; Borisov et al., 2022b), which aim to acquire generalisable representations through pretraining across broad upstream data and then adapt to downstream tasks through techniques like finetuning or in-context inference.

One popular perspective to approach tabular data generation is to leverage foundation models from other modalities (Borisov et al., 2022b; Seedat et al., 2024; Nguyen et al., 2024; Zhao et al., 2025; Li et al., 2024), such as Large Language Models (LLMs). For instance, GReaT (Borisov et al., 2022b) converts each table row into a text sequence and finetunes an autoregressive language model to generate synthetic rows. This formulation can benefit from the strong generative ability of pretrained language models. However, it treats tabular data as a sequential modality, so the imposed feature order can introduce harmful order bias. Its generation quality may also depend heavily on textual serialisation and metadata (Fang et al., 2024). As a result, LLM-based tabular foundation generators can be suboptimal for high-quality tabular data generation.

Another mainstream paradigm repurposes tabular foundation predictors as generators, such as TabPFN (Grinsztajn et al., 2025), TabDPT (Ma et al., 2025), Mitra (Zhang et al., 2025b), LimiX (Zhang et al., 2025c), and TabICL (Qu et al., 2026). These models are originally designed for in-context tabular prediction. They can be converted into autoregressive generators by treating each feature as a prediction target and iteratively sampling feature values conditioned on previously generated features (Lab, 2026; Hollmann et al., 2025). However, their autoregressive generation again imposes a feature order, which

*Table 1.* **Model design comparison between TabFORGE and prior tabular foundation models.** We use "−" to indicate dimensions that are not applicable to particular models. TabFORGE is distinguished by identity-preserved feature encoding and denoising-aligned decoding without imposing feature-order bias, and offers strong practicability for tabular data generation.

| Foundation model | Pretraining & Fitting | | | | Generation | | Practicability | |
|---|---|---|---|---|---|---|---|---|
| | Generative objective | Metadata-free | Feature identity preservation | Paradigm | Feature-order bias avoidance | Denoising-aligned latent decoding | No iterative in-context inference | Unlimited feature cardinality |
| GReaT (Borisov et al., 2022b) | ✔ | ✘ | ✔ | Autoregressive | ✘ | − | − | ✔ |
| TabPFN (Grinsztajn et al., 2025) | ✘ | ✔ | ✔ | Autoregressive | ✘ | − | ✘ | ✘ |
| TabDPT (Ma et al., 2025) | ✘ | ✔ | ✘ | Autoregressive | ✘ | − | ✘ | ✘ |
| Mitra (Zhang et al., 2025b) | ✘ | ✔ | ✔ | Autoregressive | ✘ | − | ✘ | ✘ |
| LimiX (Zhang et al., 2025c) | ✘ | ✔ | ✔ | Autoregressive | ✘ | − | ✘ | ✘ |
| TabICL (Qu et al., 2026) | ✘ | ✔ | ✘ | Autoregressive | ✘ | − | ✘ | ✘ |
| TabEBM (Margeloiu et al., 2024) | ✔ | ✔ | ✔ | Energy-based | ✔ | − | ✘ | ✔ |
| CTSyn (Lin et al., 2025b) | ✔ | ✘ | ✘ | Diffusion | ✔ | ✘ | − | ✔ |
| **TabFORGE (Ours)** | ✔ | ✔ | ✔ | Diffusion | ✔ | ✔ | ✔ | ✔ |

may conflict with the underlying causal structures. Moreover, because such an autoregressive paradigm requires repeated in-context inference to sequentially generate each feature, these methods can be computationally expensive. Furthermore, their predictive pretraining objectives can also potentially bias the learned representations towards supervised target prediction, rather than holistic modelling of all features.

A further direction converts tabular foundation predictors into energy-based generators (Margeloiu et al., 2024; Ma et al., 2023). For instance, TabEBM (Margeloiu et al., 2024) builds class-specific energy surfaces from pretrained tabular predictors and samples synthetic data through the induced energy landscape. This design can achieve strong downstream predictive utility. However, because the energy functions are derived from predictive backbones, the model tends to overly emphasise local structures around the prediction target (Jiang et al., 2026). As a result, it may overlook global causal interactions across all features, which are essential for high-quality tabular data generation.

One recent attempt at a generative tabular foundation model is CTSyn (Lin et al., 2025b), which pretrains a cross-dataset tabular generator by handling heterogeneous tables with a VAE and then modelling this latent space using diffusion, with metadata embedded by a pretrained LLM. Nevertheless, CTSyn still faces several limitations. Its dependence on metadata and LLM embeddings makes the model sensitive to metadata quality and LLM capability. Moreover, mapping variable-length feature sets into a predetermined fixed-length latent token sequence can obscure feature identity and weaken feature-specific causal semantics. In addition, as in many VAE-based latent diffusion models, the decoder is trained on clean encoder embeddings, whereas inference uses denoised embeddings produced by the diffusion model, creating a train-inference mismatch in latent space.

Due to these limitations, existing tabular foundation generators still struggle to match strong dataset-specific generators (Section 3). TabFORGE is designed to address this gap through a causality-aware latent diffusion framework for tabular foundation generation. Specifically, TabFORGE constructs a frozen causality-aware latent space and trains a denoising-aligned decoder directly on the denoised embeddings produced by the fitted diffusion transformer. Therefore, TabFORGE provides a more aligned and practical route towards tabular foundation models for generative modelling.

## A.2. Tabular Data Evaluation

Following prior studies (Zhang et al., 2023; Shi et al., 2025a; Hansen et al., 2023; Tu et al., 2024; Livieris et al., 2024; Lautrup et al., 2025; Kapar et al., 2025; Jiang et al., 2026; Margeloiu et al., 2024), we evaluate the quality of synthetic tabular data along four complementary dimensions: structural fidelity, ML efficacy, density estimation, and privacy preservation. These dimensions are selected because each captures a distinct and important aspect of data quality, as detailed below:

**Structural fidelity** evaluates whether synthetic data preserves the global causal structures of real data (Jiang et al., 2026). This dimension is particularly important for tabular generation because causal structures have been shown to provide an effective prior for tabular data, whereas preserving only distributional properties may fail to capture inter-feature causal interactions. We quantify structural fidelity using *global utility* (Jiang et al., 2026). A higher global utility indicates that the synthetic data better preserves the global causal structures of real data.

**ML efficacy** measures whether synthetic data can support downstream predictive modelling as effectively as real data (Margeloiu et al., 2024; Zhang et al., 2023). This dimension reflects the practical utility of synthetic data when it is used as a substitute for real data in downstream tasks. Following prior studies (Xu et al., 2019; Zhang et al., 2023; Shi

et al., 2025a; Jiang et al., 2026; Zhang et al., 2025a), we adopt the "train-on-synthetic, test-on-real" strategy: downstream predictors are trained on synthetic data and evaluated on real test data. To mitigate bias introduced by any single downstream model, we quantify ML efficacy using *local utility*, computed from the performance of an ensemble of nine tuned predictors (Jiang et al., 2026). A higher local utility indicates that the synthetic data better supports downstream predictive modelling, i.e., higher ML efficacy.

**Density estimation** assesses the distributional discrepancy between real and synthetic data. We include this dimension because high-quality synthetic data should capture both low-order statistics, such as feature-level marginals and pairwise correlations, and high-order distributional properties, such as sample-level similarity and diversity. Following prior studies (Zhang et al., 2023; Shi et al., 2025a; Jiang et al., 2026), we evaluate density estimation using four metrics from two categories. For low-order metrics, we use *Shape* and *Trend* (Wüst, 2011): *Shape* measures how well synthetic data replicates the marginal density of each column, and *Trend* measures how well it captures correlations between columns. For high-order metrics, we use $\alpha$-*precision* and $\beta$-*recall* (Alaa et al., 2022): $\alpha$-*precision* quantifies the similarity between synthetic and real samples, and $\beta$-*recall* evaluates the diversity and coverage of the synthetic data.

**Privacy preservation** evaluates the extent to which synthetic data avoids directly copying or memorising samples from the real training data (McKenna et al., 2019; Jordon et al., 2018; Truda, 2023; Stoian et al., 2025; Hu et al., 2024). This dimension is essential because a generator with strong overall performance may still pose privacy risks if it closely duplicates training samples. Following prior studies (Jiang et al., 2026; 2025), we measure privacy preservation using two metrics. First, we use *Authenticity* (Alaa et al., 2022), where a higher value indicates that the generator is less likely to overfit to the real training data. Second, we use the *Distance to Closest Record Score (DCR score)* (Wüst, 2011), where a higher DCR score indicates that synthetic samples are further from their nearest real training records and are therefore less likely to be direct duplications. We note that privacy metrics should be interpreted together with the other dimensions, since excessive distance from the real data may also indicate a poor fit to the real data distribution.

# B. Extended Model Design of TabFORGE

In this section, we provide the complete training and inference procedures of TabFORGE, together with detailed implementation choices and model configurations. We first present the pseudocode for pretraining, fitting, and generation (Appendix B.1). We then describe the implementation details of certain components (Appendix B.3). Next, we summarise the configurations used throughout the experiments of TabFORGE.

## B.1. Pseudocode for Training and Inference

As illustrated in Section 3, TabFORGE follows a three-phase workflow. First, during pretraining (Algorithm 1), it learns transferable latent representations from multiple real-world tabular datasets. Second, when applied to an unseen dataset (Algorithm 2), it performs lightweight fitting by updating the diffusion transformer and decoder using the available training data. Third, during inference (Algorithm 3), it generates synthetic tabular data by sampling latent noise, progressively denoising it, and decoding the final denoised embeddings into mixed-type feature values.

---

**Algorithm 1** Pretraining TabFORGE across multiple tabular datasets

---

**Input:** Pretraining datasets $\{\boldsymbol{X}_m\}_{m=1}^M$; frozen tokeniser $\mathcal{S}_\psi$; frozen encoder $\mathcal{E}_\omega$; diffusion denoiser $\mathcal{G}_\theta$; decoder transformer $\mathcal{D}_\phi$; diffusion steps $S_{\text{diff}}$; decoder steps $S_{\text{dec}}$; pretraining rounds $R$

**Output:** Pretrained diffusion denoiser $\mathcal{G}_\theta$ and decoder transformer $\mathcal{D}_\phi$

**foreach** *pretraining dataset $\boldsymbol{X}_m$* **do**

$\quad$ $\boldsymbol{X}_m \leftarrow \text{preprocess}(\boldsymbol{X}_m)$ $\qquad\qquad\qquad\qquad$ // imputation and feature encoding

$\quad$ $\boldsymbol{H}_m \leftarrow \mathcal{E}_\omega(\mathcal{S}_\psi(\boldsymbol{X}_m))$ $\qquad\qquad\qquad$ // compute frozen latent embeddings once

$\quad$ $(\boldsymbol{\mu}_{H,m}, \boldsymbol{s}_{H,m}) \leftarrow \text{statistics}(\boldsymbol{H}_m)$ $\qquad\qquad$ // dataset-specific latent statistics

$\quad$ $\boldsymbol{Z}_{0,m} \leftarrow (\boldsymbol{H}_m - \boldsymbol{\mu}_{H,m})/\boldsymbol{s}_{H,m}$ $\qquad\qquad$ // normalised clean embeddings

$\quad$ Initialise dataset-specific detokeniser $\mathcal{R}_{\eta_m}$ according to feature types and categorical cardinalities

**end**

**for** $r \leftarrow 1$ **to** $R$ **do**

$\quad$ **foreach** *pretraining dataset $\boldsymbol{X}_m$* **do**

$\quad\quad$ **for** $s \leftarrow 1$ **to** $S_{\text{diff}}$ **do**

$\quad\quad\quad$ $\boldsymbol{Z}_0 \leftarrow \text{sample\_batch}(\boldsymbol{Z}_{0,m})$

$\quad\quad\quad$ $\sigma \leftarrow \exp(p_{\text{mean}} + p_{\text{std}}\epsilon_\sigma)$, where $\epsilon_\sigma \sim \mathcal{N}(0,1)$ $\qquad$ // sample noise level

$\quad\quad\quad$ $\boldsymbol{E} \sim \mathcal{N}(\boldsymbol{0}, \boldsymbol{I})$ $\qquad\qquad\qquad\qquad\qquad$ // sample Gaussian noise

$\quad\quad\quad$ $\boldsymbol{Z}_\sigma \leftarrow \boldsymbol{Z}_0 + \sigma \boldsymbol{E}$ $\qquad\qquad\qquad\qquad\qquad$ // forward diffusion

$\quad\quad\quad$ $\widehat{\boldsymbol{Z}}_0 \leftarrow \mathcal{G}_\theta(\boldsymbol{Z}_\sigma, \sigma)$ $\qquad\qquad\qquad\qquad$ // denoise latent embeddings

$\quad\quad\quad$ $\theta \leftarrow \theta - \nabla_\theta \mathcal{L}_{\text{diff}}(\widehat{\boldsymbol{Z}}_0, \boldsymbol{Z}_0)$ $\qquad\qquad$ // update diffusion denoiser

$\quad\quad$ **end**

$\quad\quad$ **for** $s \leftarrow 1$ **to** $S_{\text{dec}}$ **do**

$\quad\quad\quad$ $(\boldsymbol{X}, \boldsymbol{Z}_0) \leftarrow \text{sample\_batch}(\boldsymbol{X}_m, \boldsymbol{Z}_{0,m})$

$\quad\quad\quad$ $\sigma \leftarrow \exp(p_{\text{mean}} + p_{\text{std}}\epsilon_\sigma)$, where $\epsilon_\sigma \sim \mathcal{N}(0,1)$

$\quad\quad\quad$ $\boldsymbol{E} \sim \mathcal{N}(\boldsymbol{0}, \boldsymbol{I})$

$\quad\quad\quad$ $\boldsymbol{Z}_\sigma \leftarrow \boldsymbol{Z}_0 + \sigma \boldsymbol{E}$

$\quad\quad\quad$ $\widehat{\boldsymbol{Z}} \leftarrow \mathcal{G}_\theta(\boldsymbol{Z}_\sigma, \sigma)$ $\qquad\qquad\qquad\qquad$ // produce denoised embeddings

$\quad\quad\quad$ $\widehat{\boldsymbol{H}} \leftarrow \widehat{\boldsymbol{Z}} \odot \boldsymbol{s}_{H,m} + \boldsymbol{\mu}_{H,m}$ $\qquad\qquad$ // inverse latent normalisation

$\quad\quad\quad$ $\boldsymbol{U} \leftarrow \mathcal{D}_\phi(\widehat{\boldsymbol{H}})$ $\qquad\qquad\qquad\qquad\qquad$ // refine denoised embeddings

$\quad\quad\quad$ $\widehat{\boldsymbol{X}} \leftarrow \mathcal{R}_{\eta_m}(\boldsymbol{U})$ $\qquad\qquad\qquad$ // detokenise into mixed-type features

$\quad\quad\quad$ $(\phi, \eta_m) \leftarrow (\phi, \eta_m) - \nabla_{\phi,\eta_m} \mathcal{L}_{\text{recon}}(\widehat{\boldsymbol{X}}, \boldsymbol{X})$ $\qquad$ // update decoder and detokeniser

$\quad\quad$ **end**

$\quad$ **end**

**end**

**return** $\mathcal{G}_\theta, \mathcal{D}_\phi$

---

---

**Algorithm 2** Fitting TabFORGE to an unseen dataset

---

**Input:** Unseen training data $\boldsymbol{X}_{\text{Train}}$; pretrained diffusion denoiser $\mathcal{G}_\theta$; pretrained decoder transformer $\mathcal{D}_\phi$; frozen tokeniser $\mathcal{S}_\psi$; frozen encoder $\mathcal{E}_\omega$; fitting steps $S_{\text{fit,diff}}$ and $S_{\text{fit,dec}}$

**Output:** Fitted diffusion denoiser $\mathcal{G}_{\theta^\star}$; fitted decoder transformer $\mathcal{D}_{\phi^\star}$; dataset-specific detokeniser $\mathcal{R}_{\eta^\star}$; latent statistics $(\boldsymbol{\mu}_{H,\star}, \boldsymbol{s}_{H,\star})$

$\boldsymbol{X}_{\text{Train}} \leftarrow \text{preprocess}(\boldsymbol{X}_{\text{Train}})$          // imputation and feature encoding

$\boldsymbol{H}_{\text{Train}} \leftarrow \mathcal{E}_\omega(\mathcal{S}_\psi(\boldsymbol{X}_{\text{Train}}))$          // compute frozen latent embeddings once

$(\boldsymbol{\mu}_{H,\star}, \boldsymbol{s}_{H,\star}) \leftarrow \text{statistics}(\boldsymbol{H}_{\text{Train}})$          // dataset-specific latent statistics

$\boldsymbol{Z}_{0,\star} \leftarrow (\boldsymbol{H}_{\text{Train}} - \boldsymbol{\mu}_{H,\star})/\boldsymbol{s}_{H,\star}$          // normalised clean embeddings

Initialise dataset-specific detokeniser $\mathcal{R}_{\eta^\star}$ according to feature types and categorical cardinalities

**for** $s \leftarrow 1$ **to** $S_{\text{fit,diff}}$ **do**

    $\boldsymbol{Z}_0 \leftarrow \text{sample\_batch}(\boldsymbol{Z}_{0,\star})$

    $\sigma \leftarrow \exp(p_{\text{mean}} + p_{\text{std}}\epsilon_\sigma)$, where $\epsilon_\sigma \sim \mathcal{N}(0,1)$          // sample noise level

    $\boldsymbol{E} \sim \mathcal{N}(\boldsymbol{0}, \boldsymbol{I})$          // sample Gaussian noise

    $\boldsymbol{Z}_\sigma \leftarrow \boldsymbol{Z}_0 + \sigma\boldsymbol{E}$          // forward diffusion

    $\widehat{\boldsymbol{Z}}_0 \leftarrow \mathcal{G}_\theta(\boldsymbol{Z}_\sigma, \sigma)$          // denoise latent embeddings

    $\theta \leftarrow \theta - \nabla_\theta \mathcal{L}_{\text{diff}}(\widehat{\boldsymbol{Z}}_0, \boldsymbol{Z}_0)$          // fit diffusion denoiser

**end**

$\theta^\star \leftarrow \theta$          // save fitted diffusion denoiser

**for** $s \leftarrow 1$ **to** $S_{\text{fit,dec}}$ **do**

    $(\boldsymbol{X}, \boldsymbol{Z}_0) \leftarrow \text{sample\_batch}(\boldsymbol{X}_{\text{Train}}, \boldsymbol{Z}_{0,\star})$

    $\sigma \leftarrow \exp(p_{\text{mean}} + p_{\text{std}}\epsilon_\sigma)$, where $\epsilon_\sigma \sim \mathcal{N}(0,1)$

    $\boldsymbol{E} \sim \mathcal{N}(\boldsymbol{0}, \boldsymbol{I})$

    $\boldsymbol{Z}_\sigma \leftarrow \boldsymbol{Z}_0 + \sigma\boldsymbol{E}$

    $\widehat{\boldsymbol{Z}} \leftarrow \mathcal{G}_{\theta^\star}(\boldsymbol{Z}_\sigma, \sigma)$          // produce denoised embeddings

    $\widehat{\boldsymbol{H}} \leftarrow \widehat{\boldsymbol{Z}} \odot \boldsymbol{s}_{H,\star} + \boldsymbol{\mu}_{H,\star}$          // inverse latent normalisation

    $\boldsymbol{U} \leftarrow \mathcal{D}_\phi(\widehat{\boldsymbol{H}})$          // refine denoised embeddings

    $\widehat{\boldsymbol{X}} \leftarrow \mathcal{R}_{\eta^\star}(\boldsymbol{U})$          // detokenise into mixed-type features

    $(\phi, \eta^\star) \leftarrow (\phi, \eta^\star) - \nabla_{\phi,\eta^\star} \mathcal{L}_{\text{recon}}(\widehat{\boldsymbol{X}}, \boldsymbol{X})$          // fit decoder and detokeniser

**end**

$\phi^\star \leftarrow \phi$          // save fitted decoder transformer

**return** $\mathcal{G}_{\theta^\star}, \mathcal{D}_{\phi^\star}, \mathcal{R}_{\eta^\star}, (\boldsymbol{\mu}_{H,\star}, \boldsymbol{s}_{H,\star})$

---

---

**Algorithm 3** Synthetic data generation with fitted TabFORGE

---

**Input:** Fitted diffusion denoiser $\mathcal{G}_{\theta^\star}$; fitted decoder transformer $\mathcal{D}_{\phi^\star}$; dataset-specific detokeniser $\mathcal{R}_{\eta^\star}$; latent statistics $(\boldsymbol{\mu}_{H,\star}, \boldsymbol{s}_{H,\star})$; desired synthetic sample size $N_{\text{syn}}$; feature count $D+1$; latent dimension $k$; reverse diffusion steps $T$

**Output:** Synthetic tabular dataset $\boldsymbol{X}_{\text{Syn}}$

$\boldsymbol{Z}_{\sigma_1} \sim \mathcal{N}(\boldsymbol{0}, \sigma_{\max}^2 \boldsymbol{I})$ with shape $\mathbb{R}^{N_{\text{syn}} \times (D+1) \times k}$          // initial latent noise

Construct decreasing schedule $\sigma_1 > \sigma_2 > \cdots > \sigma_T$, with $\sigma_1 = \sigma_{\max}$ and $\sigma_T = \sigma_{\min}$

**for** $t \leftarrow 1$ **to** $T$ **do**

    $\widehat{\boldsymbol{Z}}_t \leftarrow \mathcal{G}_{\theta^\star}(\boldsymbol{Z}_{\sigma_t}, \sigma_t)$          // predict denoised latent embeddings

    $\mathbf{d}_t \leftarrow (\boldsymbol{Z}_{\sigma_t} - \widehat{\boldsymbol{Z}}_t)/\sigma_t$          // EDM probability-flow direction

    $\boldsymbol{Z}_{\sigma_{t+1}} \leftarrow \boldsymbol{Z}_{\sigma_t} + (\sigma_{t+1} - \sigma_t)\mathbf{d}_t$          // reverse diffusion update

**end**

$\widehat{\boldsymbol{H}}_{\text{syn}} \leftarrow \boldsymbol{Z}_{\sigma_T} \odot \boldsymbol{s}_{H,\star} + \boldsymbol{\mu}_{H,\star}$          // inverse latent normalisation

$\boldsymbol{U}_{\text{syn}} \leftarrow \mathcal{D}_{\phi^\star}(\widehat{\boldsymbol{H}}_{\text{syn}})$          // generation-ready embeddings

$\widehat{\boldsymbol{X}}_{\text{syn}} \leftarrow \mathcal{R}_{\eta^\star}(\boldsymbol{U}_{\text{syn}})$          // decode mixed-type features

$\boldsymbol{X}_{\text{Syn}} \leftarrow \text{inverse\_preprocess}(\widehat{\boldsymbol{X}}_{\text{syn}})$          // recover numerical scales and categorical labels

**return** $\boldsymbol{X}_{\text{Syn}}$

---

## B.2. Model Component Details

### B.2.1. CAUSALITY-AWARE FEATURE ENCODER (FIGURE 1B)

**Tokeniser $\mathcal{S}_\psi$ handles heterogeneity without relying on metadata.** Since feature sets often vary across tabular datasets, learning a dataset-specific latent space would substantially limit cross-dataset transferability. Therefore, TabFORGE follows the TabPFN tokenisation strategy (Hollmann et al., 2025) and maps any table into a shared $k$-dimensional latent space while preserving the identity of each feature, that is, $\boldsymbol{X} \in \mathbb{R}^{N \times (D+1)} \rightarrow \boldsymbol{T} \in \mathbb{R}^{N \times (D+1) \times k}$. The per-feature tokenisation is computed directly from feature values (further details are in Appendix B.3) and does not rely on metadata such as column names or descriptions, thereby enabling flexible cross-dataset training and inference.

**Encoder layers $\mathcal{E}_\omega$ leverage implicitly learned causal information.** TabFORGE employs the early-to-middle layers of a PFN predictor (i.e., the first $L_{\text{enc}}$ transformer layers), motivated by prior work (Swelam et al., 2025) suggesting that such representations implicitly capture inter-feature causal interactions, thereby inducing the mapping $\boldsymbol{T} \rightarrow \boldsymbol{H} \in \mathbb{R}^{N \times (D+1) \times k}$. Unlike in-context PFN predictors, TabFORGE does not require additional unseen query data to obtain latent embeddings for the available training data. Concretely, TabFORGE derives latent embeddings $\boldsymbol{H}$ by applying the encoder layers with a leave-one-fold-out feature extraction strategy (Ye et al., 2025) solely over the training data (further details are in Appendix B.3). Thus, the encoder serves purely as a representation extractor for the training data, rather than as a predictor that depends on access to query samples at inference time.

A key design choice in TabFORGE is to preserve the identity of each feature embedding, operating on $\boldsymbol{H}^{(i)} \in \mathbb{R}^{(D+1) \times k}$ rather than a flattened vector (Zhang et al., 2023; Shi et al., 2025a) or a predetermined number of latent query tokens (Lin et al., 2025b). With one-to-one correspondence between each feature and its embedding, TabFORGE mitigates the disruption of per-feature semantics and the information bottleneck stemming from a fixed-length latent token sequence.

**Frozen latent space mitigates distribution shifts.** In TabFORGE, we construct a frozen latent space by inheriting the tokeniser and the first $L_{\text{enc}}$ transformer layers from a pretrained PFN model. By default, we adopt Real-TabPFN-2.5 (Grinsztajn et al., 2025), and these components remain frozen in TabFORGE. We provide ablation studies on alternative PFN models in Appendix D.2. Specifically, we freeze the latent space for three main reasons: (1) A frozen latent space uniquely allows TabFORGE to decouple latent modelling from decoding, thereby mitigating the train-inference mismatch common in latent diffusion models, as detailed in Section 2.4. (2) The early-to-middle layers of a pretrained PFN model can provide strong inductive bias for capturing general causal interactions. Freezing these layers preserves transferable and causality-aware representations. (3) A frozen encoder stabilises optimisation and mitigates latent collapse by preventing the latent space from drifting across datasets (Zheng et al., 2026; Erdogan et al., 2025).

### B.2.2. SCORE-BASED DIFFUSION TRANSFORMER (FIGURE 1C)

**Forward process preserves token structure.** The forward process constructs noisy latent embeddings by perturbing clean latent embeddings while preserving token structure. Specifically, given the clean latent embeddings $\boldsymbol{H}$, we first obtain the normalised latent embeddings $\boldsymbol{Z}_0 \in \mathbb{R}^{N \times (D+1) \times k}$ via Z-score normalisation. We then generate noisy latent embeddings:

$$\boldsymbol{Z}_\sigma = \boldsymbol{Z}_0 + \sigma \boldsymbol{E}, \quad \sigma = \exp(p_{\text{mean}} + p_{\text{std}} \, \epsilon_\sigma), \quad \epsilon_\sigma \sim \mathcal{N}(0, 1), \tag{1}$$

where $\boldsymbol{E} \sim \mathcal{N}(\boldsymbol{0}, \boldsymbol{I})$ denotes isotropic Gaussian noise, and $(p_{\text{mean}}, p_{\text{std}})$ parametrise the log-normal distribution from which the noise level $\sigma$ is sampled.

**Reverse process supports varying feature sets.** Given noisy embeddings $\boldsymbol{Z}_\sigma$, the score-based denoiser estimates the clean latent embedding via the EDM (Karras et al., 2022) parametrisation: $\mathcal{G}_\theta(\boldsymbol{Z}_\sigma, \sigma) = c_{\text{skip}}(\sigma)\boldsymbol{Z}_\sigma + c_{\text{out}}(\sigma)\mathcal{F}_\theta\left(c_{\text{in}}(\sigma)\boldsymbol{Z}_\sigma, \, c_{\text{noise}}(\sigma)\right)$, where $\mathcal{F}_\theta$ is a transformer model and $\sigma_{\text{data}}$ denotes the standard deviation of the clean latent embeddings. The corresponding preconditioning coefficients are $c_{\text{skip}}(\sigma) = \frac{\sigma_{\text{data}}^2}{\sigma^2 + \sigma_{\text{data}}^2}$, $c_{\text{out}}(\sigma) = \frac{\sigma \sigma_{\text{data}}}{\sqrt{\sigma^2 + \sigma_{\text{data}}^2}}$, $c_{\text{in}}(\sigma) = \frac{1}{\sqrt{\sigma^2 + \sigma_{\text{data}}^2}}$, $c_{\text{noise}}(\sigma) = \frac{\ln \sigma}{4}$. Under this parametrisation, the denoiser induces an approximate score field $\boldsymbol{s}_\theta(\boldsymbol{Z}_\sigma, \sigma)$ and the associated reverse-time dynamics:

$$\boldsymbol{s}_\theta(\boldsymbol{Z}_\sigma, \sigma) = \nabla_{\boldsymbol{Z}_\sigma} \log p_\sigma(\boldsymbol{Z}_\sigma), \quad \frac{\mathrm{d}\boldsymbol{Z}_\sigma}{\mathrm{d}\sigma} = -\sigma \boldsymbol{s}_\theta(\boldsymbol{Z}_\sigma, \sigma) \propto \frac{\boldsymbol{Z}_\sigma - \mathcal{G}_\theta(\boldsymbol{Z}_\sigma, \sigma)}{\sigma}. \tag{2}$$

As $\sigma$ decreases, the latent sample is progressively guided from a noisy state towards the clean latent manifold. Once the diffusion model has been trained, we apply the inverse Z-score normalisation to $\mathcal{G}_\theta(\boldsymbol{Z}_\sigma, \sigma)$ to recover the latent embeddings $\widehat{\boldsymbol{H}}$ on the same scale as the clean embeddings.

**Objective of latent modelling.** The score-based diffusion transformer is optimised using the EDM weighted denoising objective (Karras et al., 2022), which trains the denoiser to recover the clean normalised latent embeddings from their noisy counterparts. The diffusion loss is computed via

$$\mathcal{L}_{\mathrm{diff}} = \frac{1}{N} \sum_{i=1}^{N} \frac{\sigma_i^2 + \sigma_{\mathrm{data}}^2}{(\sigma_i \sigma_{\mathrm{data}})^2} \left\| \mathcal{G}_\theta(\boldsymbol{Z}_{\sigma_i}^{(i)}, \sigma_i) - \boldsymbol{Z}_0^{(i)} \right\|_2^2 \tag{3}$$

which reweights training samples according to their noise levels, yielding a balanced learning signal across the full range of noise scales considered during diffusion training.

### B.2.3. DENOISING-ALIGNED LATENT DECODER (FIGURE 1D)

**Decoder transformer $\mathcal{D}_\phi$ refines embeddings for generation.** Although the denoised latent embeddings encode general causal information, they can still be too coarse to directly generate accurate values for individual features (see Appendix D.2). Therefore, TabFORGE further refines them with a transformer $\mathcal{D}_\phi$ of $L_{\mathrm{dec}}$ layers to obtain generation-ready token representations $\boldsymbol{U} = [\boldsymbol{u}^{(1)}, \ldots, \boldsymbol{u}^{(N)}]^\mathsf{T} \in \mathbb{R}^{N \times (D+1) \times k}$. As a result, $\mathcal{D}_\phi$ remains agnostic to the number of features and can thus be pretrained across heterogeneous datasets with varying feature sets.

**Detokeniser $\mathcal{R}_\eta$ enables flexible generation.** TabFORGE employs a lightweight detokeniser to map the refined embeddings $\boldsymbol{U}$ back to the original feature space. For the $j$-th feature of the $i$-th sample, the detokeniser produces either a reconstructed numerical value or a categorical logit vector:

$$\widehat{o}_j^{(i)} = \begin{cases} \langle \boldsymbol{u}_j^{(i)}, \boldsymbol{w}_j \rangle, & \text{if } \boldsymbol{x}_j \in \boldsymbol{x}_{\mathrm{num}}, \\ \boldsymbol{W}_j \boldsymbol{u}_j^{(i)} + \boldsymbol{b}_j, & \text{if } \boldsymbol{x}_j, \end{cases} \tag{4}$$

where $\langle \cdot, \cdot \rangle$ denotes the dot product, $\boldsymbol{w}_j \in \mathbb{R}^k$ is a learnable reconstruction vector for the $j$-th numerical feature, and $\boldsymbol{W}_j \in \mathbb{R}^{C_j \times k}$ and $\boldsymbol{b}_j \in \mathbb{R}^{C_j}$ are the parameters of the linear head for the $j$-th categorical feature, whose cardinality is $C_j$.

Unlike the fixed output heads in PFN models (Grinsztajn et al., 2025; Qu et al., 2026; Zhang et al., 2025c), which are often limited to feature cardinalities no greater than 10, TabFORGE is designed to use a lightweight detokeniser for each dataset, while keeping the decoder transformer shared across datasets. As a result, TabFORGE can naturally generate high-cardinality categorical features without more complex techniques such as hierarchical classification (Qu et al., 2026; 2025), which can add non-trivial computational overhead. This dataset-specific detokeniser therefore improves flexibility in synthetic data generation at only minimal computational cost.

**Objective of denoising-aligned decoding.** The denoising-aligned latent decoder is optimised using a mixed-type reconstruction objective defined over the decoder outputs:

$$\mathcal{L}_{\mathrm{recon}} = \frac{1}{N} \sum_{i=1}^{N} \left( \left\| \widehat{\boldsymbol{x}}_{\mathrm{num}}^{(i)} - \boldsymbol{x}_{\mathrm{num}}^{(i)} \right\|_2^2 + \sum_{j=1}^{|\boldsymbol{x}_{\mathrm{cat}}|} \mathrm{CE}\left( \widehat{\boldsymbol{p}}_{\mathrm{cat},j}^{(i)}, x_{\mathrm{cat},j}^{(i)} \right) \right) \tag{5}$$

where $\boldsymbol{x}_{\mathrm{num}}$ and $\boldsymbol{x}_{\mathrm{cat}}$ denote the sets of numerical and categorical features, respectively, and $\mathrm{CE}(\cdot, \cdot)$ denotes cross-entropy, and $\widehat{\boldsymbol{p}}_{\mathrm{cat},j}^{(i)} = \mathrm{softmax}\left( \widehat{\boldsymbol{o}}_{\mathrm{cat},j}^{(i)} \right) \in \mathbb{R}^{C_j}$, applied independently to each categorical feature. A core characteristic of TabFORGE is that $\mathcal{D}_\phi$ are trained solely on denoised embeddings $\widehat{\boldsymbol{H}}$ by the fitted diffusion transformer $\mathcal{G}_\theta$. As such, $\mathcal{D}_\phi$ is directly trained on the latent distribution consistent with what it will encounter at inference time.

### B.3. Implementation Details

#### B.3.1. FEATURE TOKENISATION

Following TabPFN (Grinsztajn et al., 2025), the tokeniser constructs distinguishable feature tokens within a shared latent space by introducing a learnable base vector $\boldsymbol{u} \in \mathbb{R}^k$ that is shared across all features. It then builds feature-specific perturbations by constructing

$$\boldsymbol{R} = \boldsymbol{W}\boldsymbol{P} \in \mathbb{R}^{k \times (D+1)}, \tag{6}$$

where $\boldsymbol{W} \in \mathbb{R}^{k \times k'}$ is a learnable projection matrix with $k' < k$, and $\boldsymbol{P} \in \mathbb{R}^{k' \times (D+1)}$ is a randomly generated matrix. The $j$-th column $\boldsymbol{r}_j \in \mathbb{R}^k$ of $\boldsymbol{R}$ serves as a feature-specific perturbation for the $j$-th feature. For the $i$-th sample, the token corresponding to the $j$-th feature is then computed as

$$\boldsymbol{t}_j^{(i)} = x_j^{(i)}(\boldsymbol{u} + \boldsymbol{r}_j) \in \mathbb{R}^k. \tag{7}$$

Thus, the tokenised representation of the $i$-th sample is

$$\boldsymbol{T}^{(i)} = [\boldsymbol{t}_1^{(i)}, \ldots, \boldsymbol{t}_{D+1}^{(i)}]^{\mathsf{T}} \in \mathbb{R}^{(D+1) \times k}, \tag{8}$$

and the tokenised representation of the full dataset is

$$\boldsymbol{T} = [\boldsymbol{T}^{(1)}, \ldots, \boldsymbol{T}^{(N)}]^{\mathsf{T}} \in \mathbb{R}^{N \times (D+1) \times k}. \tag{9}$$

This construction allows all features to be embedded into a common latent space while retaining feature identity through feature-specific perturbations, without relying on metadata such as column names or textual descriptions.

### B.3.2. LEAVE-ONE-FOLD-OUT LATENT EXTRACTION

Standard PFN models are originally designed for in-context prediction (Grinsztajn et al., 2025; Qu et al., 2026; Ma et al., 2025), where query samples (i.e., test data) are embedded in the context of observed samples (i.e., training data). Thus, standard PFN models require query samples to extract latent embeddings. To obtain latent embeddings for the available training samples themselves, TabFORGE adopts a leave-one-fold-out extraction strategy (Ye et al., 2025). Specifically, the training rows are partitioned into multiple folds. For each fold, the remaining folds are used as context rows, while the holdout fold is treated as query rows whose latent representations are extracted. The latent representations from all holdout folds are then concatenated to form the complete latent embeddings. For classification datasets, we use the TabPFN classifier checkpoint to extract latent embeddings; for regression datasets, we use the TabPFN regressor checkpoint. Both settings use $L_{\text{enc}} = 12$ layers. This strategy enables TabFORGE to extract latent embeddings for the training data without requiring additional unseen query data.

### B.3.3. LATENT NORMALISATION AND CACHING

For each dataset, the frozen latent embeddings are computed only once and cached before optimising the diffusion transformer or the decoder. This caching strategy avoids repeated PFN forward passes, thereby substantially reducing the computational cost of both pretraining and fitting.

Given cached embeddings $\boldsymbol{H} \in \mathbb{R}^{N \times (D+1) \times k}$, we compute the latent normalisation statistics, i.e., the mean and standard deviation, over the sample dimension:

$$\boldsymbol{\mu}_{H,j,a} = \frac{1}{N} \sum_{i=1}^{N} H_{i,j,a}, \quad \boldsymbol{s}_{H,j,a} = \sqrt{\frac{1}{N} \sum_{i=1}^{N} (H_{i,j,a} - \boldsymbol{\mu}_{H,j,a})^2 + \epsilon} \tag{10}$$

where $j$ indexes features and $a$ indexes latent dimensions. The normalised latent embeddings are then computed as

$$\boldsymbol{Z}_0 = \frac{\boldsymbol{H} - \boldsymbol{\mu}_H}{\boldsymbol{s}_H}. \tag{11}$$

The diffusion model is trained in this normalised latent space. During decoding, denoised latent embeddings are mapped back to the original latent scale by

$$\widehat{\boldsymbol{H}} = \widehat{\boldsymbol{Z}} \odot \boldsymbol{s}_H + \boldsymbol{\mu}_H \tag{12}$$

where $\odot$ denotes element-wise multiplication (i.e., the Hadamard product). The latent normalisation statistics are saved with the fitted generator, since they are required during synthetic data generation.

### B.3.4. PRETRAINING AND INFERENCE

**Pretraining setup.** We pretrain TabFORGE on the same 43 real-world datasets (see Appendix C.1) as used by Real-TabPFN-2.5 (Grinsztajn et al., 2025), avoiding additional real-world datasets to reduce the risk of data leakage when

evaluating on unseen datasets. For each dataset, we compute frozen latent embeddings for all samples only once using the causality-aware feature encoder and cache the embeddings. Then we optimise TabFORGE in two sequential stages: (1) we train the score-based diffusion transformer for 5,000 steps; (2) we train the denoising-aligned decoder for another 5,000 steps. One full pass over all 43 datasets constitutes a pretraining round, and we run 10 rounds in total. We provide detailed model implementation details in Appendix B.3.

**Fitting to unseen datasets.** Given an unseen dataset, we first compute frozen latent embeddings for the available training data only once using the causality-aware feature encoder, then load the pretrained diffusion transformer $\mathcal{G}_\theta$ and decoder transformer $\mathcal{D}_\phi$, and initialise a dataset-specific detokeniser $\mathcal{R}_\eta$. We first finetune the diffusion transformer for 100 steps. We then finetune the decoder for 100 steps, jointly updating the transferable decoder transformer and the newly instantiated detokeniser. We provide detailed optimisation configurations in Appendix B.3.6.

**Synthetic data generation.** After fitting, TabFORGE generates synthetic mixed-type tabular data by denoising random latent noise and therefore no longer requires the causality-aware feature encoder. This yields a key practical advantage over other in-context tabular foundation generators: as the encoder performs in-context inference only once to obtain the embeddings during fitting, TabFORGE avoids the substantial cost of repeated in-context inference when generating data.

### B.3.5. PRETRAINING COST

All experiments were conducted on a machine with four NVIDIA H100 Tensor Core GPUs (80 GB), a 64-core Intel Xeon CPU at 2.20 GHz, and Ubuntu 24.04.4 LTS. Under this setup, pretraining TabFORGE required approximately 605 GPU hours (i.e., about 25 H100-days).

### B.3.6. DETAILED MODEL CONFIGURATIONS

We further provide the default configurations of TabFORGE, including the model architecture (Table 2), diffusion-process setup (Table 3), and optimisation settings (Table 4).

*Table 2.* Default architecture configurations of TabFORGE.

| Component | Default configuration |
|---|---|
| Feature encoder backbone | Real-TabPFN-2.5 |
| Encoder depth $L_{\text{enc}}$ | 12 |
| Latent dimension $k$ | 192 |
| Diffusion transformer layers | 4 |
| Diffusion attention heads | 4 |
| Decoder transformer layers $L_{\text{dec}}$ | 4 |

*Table 3.* Default diffusion and inference configurations of TabFORGE.

| Hyperparameter | Default value |
|---|---|
| $\sigma_{\min}$ | 0.002 |
| $\sigma_{\max}$ | 80.0 |
| $\sigma_{\text{init}}$ | 0.1 |
| $\rho$ | 7.0 |
| $\sigma_{\text{data}}$ | 1.0 |
| Noise conditioning | $\log \sigma / 4$ |
| Reverse diffusion steps | 10 |
| Inference-time encoder usage | None |

*Table 4.* Default optimisation configurations of TabFORGE.

| Stage | Configuration | Value |
|---|---|---|
| Pretraining | Number of rounds | 10 |
| Diffusion pretraining | Optimiser | AdamW |
| Diffusion pretraining | Learning rate | $10^{-3}$ |
| Diffusion pretraining | Weight decay | $10^{-5}$ |
| Diffusion pretraining | Steps per dataset | 5,000 |
| Decoder pretraining | Optimiser | SGD |
| Decoder pretraining | Learning rate | $10^{-1}$ |
| Decoder pretraining | Weight decay | $5 \times 10^{-5}$ |
| Decoder pretraining | Steps per dataset | 5,000 |
| Pretraining and fitting | Batch size | 3,172 |
| Pretraining and fitting | Gradient clipping | Enabled |
| Pretraining and fitting | LR scheduler | Plateau-based scheduler |
| Fitting on unseen datasets | Learning rate | $10^{-6}$ |
| Fitting diffusion on unseen datasets | Steps | 100 |
| Fitting decoder on unseen datasets | Steps | 100 |

# C. Reproducibility

## C.1. Pretraining Datasets

To reduce the risk of data leakage during evaluation on the benchmark datasets, we pretrain TabFORGE on the same 43 real-world datasets used by Real-TabPFN-2.5. Table 5 lists the datasets curated for pretraining, along with their sources and access links.

## C.2. Benchmark Datasets

We curate 45 challenging real-world benchmark datasets from public tabular-data repositories and benchmark suites, including TabArena (Erickson et al., 2025), TabStruct (Jiang et al., 2026), and OpenML (https://www.openml.org/search?type=data&sort=runs). We manually verify that none of the benchmark datasets is used during model pretraining, i.e., that they do not duplicate any datasets listed in Table 5. This ensures that the benchmark results provide a fair and comprehensive evaluation of TabFORGE. All datasets are publicly available, with further details provided in Table 7 and Table 8.

## C.3. Data Processing

**Data splitting strategies for 43 pretraining datasets.** During pretraining, TabFORGE treats all pretraining datasets as training data. In other words, all samples in the pretraining datasets are used to train the model, without further splitting.

**Detailed data splitting strategies for 45 evaluation datasets.** During evaluation, for each benchmark dataset of $N$ samples, we perform nested cross-validation with repeated shuffle, as shown in Figure 4. In each repeat, we first split the dataset into three disjoint subsets: a development set, a test set, and a holdout set, containing 40%, 30%, and 30% of all samples, respectively. We then split the development set into a training split and a validation split, containing 75% and 25% of the development samples, respectively. For classification datasets, we perform stratified splitting to preserve the class distribution. We shuffle the dataset to repeat the splitting 10 times, summing up to 10 runs per dataset. As we further preserve a holdout set with different data splitting strategies, although our benchmark shares some datasets with TabArena and TabStruct, the evaluation results are not directly comparable.

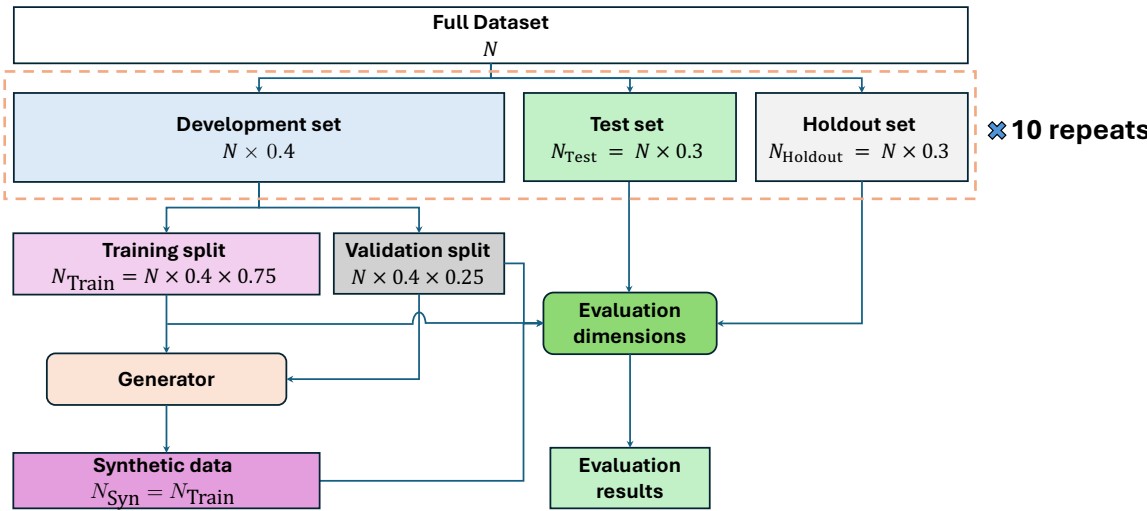

*Figure 4.* Data splitting strategies for benchmarking tabular data generators.

**Feature preprocessing for generators.** Following prior studies (Margeloiu et al., 2024; Jiang et al., 2026), we perform preprocessing in four steps. Firstly, we impute the missing values with the mean value for numerical features and the mode value for categorical features. We then compute the required statistics with training data and then transform the training split. For categorical features, we convert them into one-hot encodings. An exception is TabDiff, which tends to perform better with ordinal encoding for categorical features. For numerical features, we perform Z-score normalisation. We compute the mean and standard deviation of each feature in the training data and then transform the training samples to have a mean of zero and a variance of one for each feature. Next, we apply the same transformation to validation data and train the generators.

*Table 5.* The 43 pretraining datasets and their sources, adopted from Real-TabPFN-2.5 (Grinsztajn et al., 2025).

| Name | Source |
| --- | --- |
| artificial-characters | OpenML |
| BNG(breast-w) | OpenML |
| BNG(tic-tac-toe) | OpenML |
| connect_4 | OpenML |
| eeg-eye-state | OpenML |
| Employee-Turnover-at-TECHCO | OpenML |
| eye_movements | OpenML |
| FOREX_eurpln-hour-High | OpenML |
| gas-drift | OpenML |
| higgs | OpenML |
| Intersectional-Bias-Assessment-(Training-Data) | OpenML |
| law-school-admission-binary | OpenML |
| Medical-Appointment | OpenML |
| microaggregation2 | OpenML |
| fried | OpenML |
| mushroom | OpenML |
| NewspaperChurn | OpenML |
| nursery | OpenML |
| WBCAtt | OpenML |
| Internet Firewall Data | OpenML |
| aam_avaliacao_dataset | Kaggle |
| Air Traffic Data | Kaggle |
| ansible-defects-prediction | Kaggle |
| AV Healthcare Analytics II | Kaggle |
| Candidate Selection | Kaggle |
| Cardio Disease | Kaggle |
| Classification - Crop Damages in India (2015-2019) | Kaggle |
| CSGO Round Winner Classification | Kaggle |
| Flower Type Prediction Machine Hack | Kaggle |
| Horse Racing - Tipster Bets | Kaggle |
| How severe the accident could be | Kaggle |
| hr-comma-sep | Kaggle |
| ip-network-traffic-flows-labeled-with-87-apps | Kaggle |
| Janatahack cross-sell prediction | Kaggle |
| L&T Vehicle Loan Default Prediction | Kaggle |
| League of Legends Diamond Games (First 15 Minutes) | Kaggle |
| Richter's Predictor Modeling Earthquake Damage | Kaggle |
| Server Logs - Suspicious | Kaggle |
| Sloan Digital Sky Survey DR14 | Kaggle |
| Sloan Digital Sky Survey DR16 | Kaggle |
| Term Deposit Prediction Data Set | Kaggle |
| trajectory-based-ship-classification | Kaggle |
| Travel Insurance | Kaggle |

Once the generators are fitted, we apply the inverse transformation to recover their synthetic tabular data back to the original feature space. Finally, we conduct evaluations with training data, test data, holdout data, and corresponding synthetic data.

**Feature preprocessing for downstream predictors.** When computing global and local utility, the synthetic tabular data is inversely transformed back to the original feature space before being passed to the downstream predictors. In other words,

*Table 7.* Details of 31 real-world benchmark classification datasets.

| Dataset | Source | ID | # Samples ($N$) | # Features ($D+1$) | # Numerical | # Categorical | # Classes |
|---|---|---|---|---|---|---|---|
| BankChurn | OpenML | 46911 | 10,000 | 11 | 6 | 5 | 2 |
| Bankruptcy | OpenML | 46962 | 6,819 | 95 | 94 | 1 | 2 |
| Biodeg | OpenML | 46952 | 1,054 | 42 | 36 | 6 | 2 |
| Card | OpenML | 46919 | 30,000 | 24 | 20 | 4 | 2 |
| Churn | OpenML | 46915 | 5,000 | 20 | 15 | 5 | 2 |
| Coil2000 | OpenML | 46916 | 9,822 | 86 | 80 | 6 | 2 |
| Company | OpenML | 46950 | 5,910 | 65 | 64 | 1 | 2 |
| Coupon | OpenML | 46937 | 12,684 | 25 | 2 | 23 | 2 |
| Credit | OpenML | 46918 | 1,000 | 21 | 7 | 14 | 2 |
| Customer | OpenML | 46938 | 1,723 | 14 | 5 | 9 | 2 |
| Diabetes | OpenML | 46921 | 768 | 9 | 8 | 1 | 2 |
| Fitness | OpenML | 46927 | 1,500 | 7 | 3 | 4 | 2 |
| Give | OpenML | 46929 | 150,000 | 11 | 10 | 1 | 2 |
| HELOC | OpenML | 46932 | 10,459 | 24 | 23 | 1 | 2 |
| HR | OpenML | 46935 | 19,158 | 13 | 2 | 11 | 2 |
| Hazelnut | OpenML | 46930 | 2,400 | 31 | 30 | 1 | 2 |
| JM1 | OpenML | 46979 | 10,885 | 22 | 21 | 1 | 2 |
| Marketing | OpenML | 46940 | 2,240 | 26 | 16 | 10 | 2 |
| Maternal | OpenML | 46941 | 1,014 | 7 | 6 | 1 | 3 |
| NATICUSdroid | OpenML | 46969 | 7,491 | 87 | 0 | 87 | 2 |
| Nomao | OpenML | 45078 | 34,465 | 119 | 89 | 30 | 2 |
| Phoneme | OpenML | 1489 | 5,404 | 6 | 5 | 1 | 2 |
| Plants | OpenML | 1493 | 1,599 | 65 | 64 | 1 | 100 |
| SDSS17 | OpenML | 46955 | 78,053 | 12 | 9 | 3 | 3 |
| Satisfaction | OpenML | 46920 | 129,880 | 22 | 5 | 17 | 2 |
| Seismic | OpenML | 46956 | 2,584 | 16 | 11 | 5 | 2 |
| Shipping | OpenML | 46924 | 10,999 | 11 | 6 | 5 | 2 |
| Students | OpenML | 46960 | 4,424 | 37 | 19 | 18 | 3 |
| Transfusion | OpenML | 46913 | 748 | 5 | 4 | 1 | 2 |
| Vehicle | OpenML | 54 | 846 | 19 | 18 | 1 | 4 |
| Zernike | OpenML | 22 | 2,000 | 48 | 47 | 1 | 10 |

*Table 8.* Details of 14 real-world benchmark regression datasets.

| Dataset | Source | ID | # Samples ($N$) | # Features ($D+1$) | # Numerical | # Categorical |
|---|---|---|---|---|---|---|
| Airfoil | OpenML | 46904 | 1,503 | 6 | 5 | 1 |
| California | OpenML | 43939 | 20,640 | 10 | 9 | 1 |
| Concrete | OpenML | 46917 | 1,030 | 9 | 9 | 0 |
| Diamonds | OpenML | 46923 | 53,940 | 10 | 7 | 3 |
| Fiat500 | OpenML | 46907 | 1,538 | 8 | 7 | 1 |
| Fish | OpenML | 46954 | 907 | 7 | 7 | 0 |
| FoodDelivery | OpenML | 46928 | 45,451 | 10 | 7 | 3 |
| Healthcare | OpenML | 46931 | 1,338 | 7 | 4 | 3 |
| House16H | OpenML | 574 | 22,784 | 17 | 17 | 0 |
| Houses | OpenML | 46934 | 20,640 | 9 | 9 | 0 |
| Miami | OpenML | 46942 | 13,776 | 16 | 15 | 1 |
| Space | OpenML | 507 | 3,107 | 7 | 7 | 0 |
| Supercond | OpenML | 46961 | 21,263 | 82 | 82 | 0 |
| Wine | OpenML | 46964 | 6,497 | 13 | 12 | 1 |

the downstream models receive input data in the original, unprocessed feature space. Following TabStruct (Jiang et al., 2026), we use AutoGluon to fit downstream models, thus allowing them to apply their own model-specific preprocessing strategies for better performance.

## C.4. Technical Details of Benchmark Generators

We include 22 existing tabular data generation methods of ten different categories: (i) two non-parametric methods: SMOTE (Chawla et al., 2002) and TabSDS (Neto, 2025); (ii) two Variational Autoencoders (VAE) based methods: TVAE (Xu et al., 2019) and GOGGLE (Liu et al., 2023); (iii) a Generative Adversarial Networks (GAN) method CTGAN (Xu et al., 2019); (iv) a normalising flow model Neural Spine Flows (NFlow) (Durkan et al., 2019); (v) a tree-based method Adversarial Random Forests (ARF) (Watson et al., 2023); (vi) five diffusion models: TabDDPM (Kotelnikov et al., 2023), CDTD (Mueller et al., 2025), TabSyn (Zhang et al., 2023), TabDiff (Shi et al., 2025a), and CTSyn (Lin et al., 2025b); (vii) two energy-based models: TabEBM (Margeloiu et al., 2024) and NRGBoost (Bravo, 2025); (viii) two tabular-specific autoregressive models: TabNAT (Zhang et al., 2025a) and TabularARGN (Tiwald et al., 2025). (ix) a Large Language Model (LLM) based foundation model GReaT (Borisov et al., 2022b). (x) five PFN predictor-based foundation models: TabPFN (i.e., Real-TabPFN-2.5 (Grinsztajn et al., 2025)), TabDPT (Ma et al., 2025), Mitra (Zhang et al., 2025b), LimiX (Zhang et al., 2025c), and TabICL (i.e., TabICLv2 (Qu et al., 2026)).

We note that some of the benchmark generators are also evaluated in TabStruct (Jiang et al., 2026). For these methods, we follow the implementations and use the same hyperparameter ranges in TabStruct. For the remaining methods, we define the search ranges for better convergence on the benchmark datasets. The technical details and hyperparameter search space for each method are described below.

**SMOTE** is an interpolation-based oversampling technique (Chawla et al., 2002), which generates synthetic samples by interpolating between existing samples. We employ the open-source implementation of SMOTE provided by Imbalanced-learn (Lemaître et al., 2017), where the number of nearest neighbours $k$ can be specified. Unless stated otherwise, we use the default setting of $k = 5$.

**TabSDS** is a fully non-parametric and model-free approach for synthetic tabular data generation (Neto, 2025). TabSDS first generates synthetic marginal distributions through interpolated order statistics and then uses sequential joint-probability-preserving data shuffling to recover the dependence structure of the real data. Unlike deep generative models, TabSDS has a small tuning space, primarily controlled by the number of discretisation levels $n_c$, which determines the trade-off between data fidelity and privacy. Unless stated otherwise, we use the default setting of $n_c = 200$.

**TVAE** is a variational autoencoder (VAE) designed for tabular data (Xu et al., 2019). TVAE employs mode-specific normalisation to handle the complex distributions of numerical features. To address the class imbalance problem, TVAE conditions on specific categorical features during generation.

*Table 9.* Hyperparameter search space of TVAE.

| Hyperparameter | Range |
|---|---:|
| encoder_n_layers_hidden | $[1, 5]$ |
| encoder_n_units_hidden | $[50, 500]$ |
| encoder_nonlin | $\{\text{relu}, \text{leaky\_relu}, \text{tanh}, \text{elu}\}$ |
| n_units_embedding | $[50, 500]$ |
| decoder_n_layers_hidden | $[1, 5]$ |
| decoder_n_units_hidden | $[50, 500]$ |
| decoder_nonlin | $\{\text{relu}, \text{leaky\_relu}, \text{tanh}, \text{elu}\}$ |
| n_iter | $[100, 1000]$ |
| lr | $[10^{-4}, 10^{-3}]$ (log) |
| weight_decay | $[10^{-4}, 10^{-3}]$ (log) |

**GOGGLE** is a VAE-based tabular data generator designed to model the dependence relationships between features (Liu et al., 2023). GOGGLE proposes to learn an adjacency matrix to model the dependence relationships between features. In line with prior studies (Margeloiu et al., 2024; Jiang et al., 2026) and its official codebase (Qian et al., 2023), we observe that the official implementation of GOGGLE can be unstable and may fail to converge when fitted on large tabular datasets (e.g., more than 10,000 samples).

*Table 10.* Hyperparameter search space of GOGGLE.

| Hyperparameter | Range |
|---|---|
| encoder_dim | [32, 128] |
| encoder_l | [1, 5] |
| decoder_dim | [32, 128] |
| decoder_arch | {gcn, het, sage} |
| n_iter | [100, 500] |
| learning_rate | $[10^{-4}, 5 \times 10^{-3}]$ (log) |
| weight_decay | $[10^{-4}, 10^{-3}]$ (log) |
| alpha | [0.0, 1.0] |
| beta | [0.0, 1.0] |
| iter_opt | {True, False} |
| threshold | [0.0, 1.0] |

**CTGAN** is a conditional generative adversarial network (GAN) designed for tabular data (Xu et al., 2019). CTGAN leverages PacGAN (Lin et al., 2018) framework to mitigate mode collapse. In addition, CTGAN employs the same mode-specific normalisation technique as TVAE.

*Table 11.* Hyperparameter search space of CTGAN.

| Hyperparameter | Range |
|---|---|
| generator_n_layers_hidden | [1, 4] |
| generator_n_units_hidden | [50, 150] |
| generator_nonlin | {relu, leaky_relu, tanh, elu} |
| discriminator_n_layers_hidden | [1, 4] |
| discriminator_n_units_hidden | [50, 150] |
| discriminator_nonlin | {relu, leaky_relu, tanh, elu} |
| n_iter | [100, 1000] |
| discriminator_n_iter | [1, 5] |
| lr | $[10^{-4}, 10^{-3}]$ (log) |
| weight_decay | $[10^{-4}, 10^{-3}]$ (log) |

**NFlow** is a normalisation flow model designed for tabular data generation (Durkan et al., 2019). NFlow incorporates neural splines as a drop-in replacement for affine or additive transformations in coupling and autoregressive layers, which assists in the modelling of tabular data.

*Table 12.* Hyperparameter search space of NFlow.

| Hyperparameter | Range |
|---|---|
| n_layers_hidden | [1, 10] |
| n_units_hidden | [10, 100] |
| linear_transform_type | {lu, permutation, svd} |
| base_transform_type | {affine-coupling, quadratic-coupling, rq-coupling, affine-autoregressive, quadratic-autoregressive, rq-autoregressive} |
| dropout | [0.0, 0.2] |
| batch_norm | {False, True} |
| lr | $[2 \times 10^{-4}, 10^{-3}]$ (log) |
| n_iter | [100, 5000] |

**ARF** is a tree-based model for tabular data generation (Watson et al., 2023). ARF employs a recursive adaptation of unsupervised random forests for joint density estimation by iteratively refining synthetic data distributions using adversarial training principles in a differentiable way.

*Table 13.* Hyperparameter search space of ARF.

| Hyperparameter | Range |
|---|---|
| num_trees | $\{10, 20, \ldots, 100\}$ |
| delta | $\{0, 2, \ldots, 50\}$ |
| max_iters | $[1, 5]$ |
| early_stop | $\{\text{True}, \text{False}\}$ |
| min_node_size | $\{2, 4, \ldots, 20\}$ |

**TabDDPM** is a diffusion-based model for tabular data generation (Kotelnikov et al., 2023). TabDDPM introduces two core diffusion processes: (i) Gaussian noise for numerical features and (ii) multinomial diffusion with categorical noise for categorical features. TabDDPM directly concatenates numerical and categorical features as the input and output of the denoising function. We further note that the official implementation of TabDDPM can be unstable and may fail to converge when fitted on tabular data with mixed feature types, which has also been noted in prior studies (Jiang et al., 2026) and its official codebase (Research, 2023).

*Table 14.* Hyperparameter search space of TabDDPM.

| Hyperparameter | Range |
|---|---|
| n_iter | $[10^3, 10^4]$ |
| lr | $[10^{-5}, 10^{-1}]$ (log) |
| weight_decay | $[10^{-4}, 10^{-3}]$ (log) |
| num_timesteps | $[10, 10^3]$ |

**CDTD** is a continuous diffusion model for mixed-type tabular data (Mueller et al., 2025). CDTD unifies numerical and categorical generation by applying Gaussian diffusion directly to continuous features and learned categorical embeddings. To further account for feature heterogeneity, CDTD introduces adaptive noise schedules, which can be shared across all features by feature type or specified per feature.

*Table 15.* Hyperparameter search space of CDTD.

| Hyperparameter | Range |
|---|---|
| timewarp_type | $\{\texttt{single}, \texttt{bytype}, \texttt{all}\}$ |
| generation_steps | $\{100, 200, 500, 1000\}$ |
| timewarp_weight_low_noise | $\{1, 2, 3, 4\}$ |

**TabSyn** is a diffusion-based model for tabular data generation (Zhang et al., 2023). It synthesises tabular data by employing a diffusion model within the latent space of a variational autoencoder (VAE). TabSyn supports a wide range of data types by mapping them into a unified representation space and explicitly modelling inter-column dependencies.

**TabDiff** is a diffusion-based model for tabular data generation (Shi et al., 2025a). It introduces a joint diffusion framework capable of capturing the mixed-type distributions inherent in tabular data within a single model. In particular, TabDiff utilises a joint continuous-time diffusion process and leverages a transformer architecture to handle both numerical and categorical variables. Consistent with prior work (Jiang et al., 2026), we also observe that TabDiff is sensitive to the number of training samples. In particular, when the proportion of the full dataset used for training is reduced, e.g., from 72% to 30%, TabDiff exhibits a more pronounced performance degradation than other benchmark generators.

**CTSyn** is a diffusion-based generative foundation model for tabular data (Lin et al., 2025b). CTSyn first maps heterogeneous tables into a unified latent space using a variational autoencoder, with metadata embedded by a pretrained LLM. Since the open-source weights of CTSyn are unavailable, we strictly follow its official pretraining code (Lin et al., 2025a) and

*Table 16.* Hyperparameter search space of TabSyn.

| Hyperparameter | Range |
|---|---|
| vae.num_epochs | $[100, 1000]$ |
| vae.max_beta | $[10^{-3}, 10^{-2}]$ (log) |
| vae.min_beta | $[10^{-5}, 10^{-4}]$ (log) |
| vae.lambd | $[0.1, 1.0]$ |
| vae.num_layers | $[1, 4]$ |
| vae.d_token | $[1, 8]$ |
| vae.n_head | $[1, 4]$ |
| vae.factor | $[1, 64]$ |
| vae.lr | $[10^{-4}, 10^{-2}]$ (log) |
| vae.wd | $[0, 10^{-2}]$ (log) |
| tabsyn.num_epochs | $[100, 500]$ |
| tabsyn.lr | $[10^{-4}, 10^{-2}]$ (log) |
| tabsyn.wd | $[0, 10^{-2}]$ (log) |

*Table 17.* Hyperparameter search space of TabDiff.

| Hyperparameter | Range |
|---|---|
| batch_size | $\{512, 1024, 2048, 4096, 8192\}$ |
| c_lambda | $[0.1, 10.0]$ |
| check_val_every | $\{10, 20, 30, 40, 50\}$ |
| closs_weight_schedule | $\{$"constant", "anneal", "linear"$\}$ |
| d_lambda | $[0.1, 10.0]$ |
| ema_decay | $[0.9, 0.9999]$ |
| factor | $[0.1, 0.99]$ |
| lr | $[10^{-5}, 10^{-2}]$ (log) |
| lr_scheduler | $\{$"reduce_lr_on_plateau", "cosine", "none"$\}$ |
| reduce_lr_patience | $\{10, 30, 50, 70\}$ |
| steps | $\{100, 200, 300, 500\}$ |
| weight_decay | $[0, 10^{-2}]$ (log) |

pretrain CTSyn on the 86 datasets specified in its official report (Lin et al., 2025b). However, we find that pretraining only on these 86 datasets yields poor performance on the 45 benchmark datasets. We therefore further continue pretraining CTSyn on the 43 pretraining datasets used for TabFORGE (Table 5). In other words, CTSyn is given an advantage over TabFORGE, as it is exposed to 86 more real-world datasets during pretraining. Moreover, as noted in its official report (Lin et al., 2025b), CTSyn is highly sensitive to the quality of metadata and the capability of the employed LLM. We observe the same issue in our experiments: using the officially recommended GPT-4o causes failure to converge on many of the 45 benchmark datasets. We therefore switch to the more advanced GPT-5.1 throughout the pretraining and evaluation of CTSyn, which can mitigate the convergence issues.

*Table 18.* Hyperparameter search space of CTSyn.

| Hyperparameter | Range |
|---|---|
| num_latent ($\ell$) | $\{16\}$ |
| aggregated_dim ($M_{\text{agg}}$) | $\{64\}$ |
| diffusion_learning_rate | $\{10^{-4}\}$ |
| diffusion_pretrain_steps | $\{100{,}000, 200{,}000, 300{,}000\}$ |
| sampling_steps | $\{100, 200, 300, 400\}$ |
| conditioning_dropout | $\{0.1\}$ |

**TabEBM** is an energy-based model for tabular data generation (Margeloiu et al., 2024). It transforms a pretrained tabular predictor into a set of class-specific generators. While the original paper only provides TabEBM implementation for classification tasks, we adopt the version in TabStruct (Jiang et al., 2026) to extend its applicability in TabStruct to regression tasks by treating all reference samples as a single class, and then performing sampling over the energy landscape. Moreover, we update its backbone to Real-TabPFN-2.5 for a fair comparison with TabFORGE and other foundation models.

*Table 19.* Hyperparameter search space of TabEBM.

| Hyperparameter | Range |
|---|---|
| starting_point_noise_std | $[10^{-4}, 10^{-1}]$ (log) |
| sgld_step_size | $[10^{-3}, 10^{-1}]$ (log) |
| sgld_noise_std | $[10^{-4}, 10^{-1}]$ (log) |
| sgld_steps | $\{50, 100, 200, 500\}$ |

**NRGBoost** is an energy-based model for tabular data generation (Bravo, 2025). It is trained by maximising a local second-order approximation to the log-likelihood at each stage of the boosting process. NRGBoost is shown to offer generally good discriminative performance and competitive sampling performance compared to more specialised alternatives.

*Table 20.* Hyperparameter search space of NRGBoost.

| Hyperparameter | Range |
|---|---|
| num_trees | $\{1, 5, 10, 20, 50\}$ |
| shrinkage | $[0.01, 0.3]$ |
| max_leaves | $\{32, 64, 128, 256, 512\}$ |
| max_ratio_in_leaf | $[1, 5]$ |
| num_model_samples | $\{10{,}000, 40{,}000, 80{,}000, 160{,}000\}$ |
| p_refresh | $[0.01, 0.3]$ |
| num_chains | $\{4, 8, 16, 32\}$ |
| burn_in | $\{50, 100, 200, 500\}$ |

**TabNAT** is a continuous-discrete joint generative framework for tabular data (Zhang et al., 2025a). It combines masked generative modelling with a bidirectional transformer to learn conditional distributions under arbitrary column orders. This design enables TabNAT to model heterogeneous tabular data within a unified architecture and supports both unconditional generation and flexible conditional generation.

*Table 21.* Hyperparameter search space of TabNAT.

| Hyperparameter | Range |
|---|---|
| epochs | $\{3000, 5000, 6000\}$ |
| batch_size | $\{512, 1024, 2048\}$ |
| learning_rate | $[10^{-4}, 10^{-2}]$ (log) |
| weight_decay | $[10^{-7}, 10^{-5}]$ (log) |
| depth | $\{2, 4, 6, 8\}$ |

**TabularARGN** is an autoregressive framework for tabular data generation (Tiwald et al., 2025). It first converts heterogeneous features into categorical sub-columns, and then learns their joint distribution through shared embedding, permutation-masking, regressor, and predictor modules. During training, TabularARGN randomly permutes the column order and minimises the summed categorical cross-entropy with teacher forcing, enabling flexible conditional generation over arbitrary observed feature subsets.

**GReaT** leverages large language models (LLMs) to generate synthetic tabular data (Borisov et al., 2022b). GReaT converts each sample into a sentence and finetunes the language model to capture the sentence-level distributions. Additionally, GReaT shuffles the order of features to mitigate the permutation variance in sentence-level distributions.

*Table 22.* Hyperparameter search space of TabularARGN.

| Hyperparameter | Range |
|---|---|
| model_size | $\{$S, M, L$\}$ |
| max_training_epochs | $\{100,\ 300,\ 500\}$ |

*Table 23.* Hyperparameter search space of GReaT.

| Hyperparameter | Range |
|---|---|
| n_iter | $\{100,\ 300,\ 500,\ 1000\}$ |
| learning_rate | $[10^{-4},\ 10^{-2}]$ (log) |
| weight_decay | $[10^{-5},\ 10^{-2}]$ (log) |

**TabPFN** is originally designed as a tabular foundation predictor, and we repurpose it for autoregressive tabular data generation (Grinsztajn et al., 2025). Following the official implementation (Lab, 2026), we instantiate TabPFN classifiers for categorical features and TabPFN regressors for numerical features, and use them as feature-wise conditional density estimators. To generate synthetic samples, the model initialises an empty table with missing entries and fills the features sequentially. At each autoregressive step, the target feature is predicted from the previously generated features according to a sampled feature order. This procedure converts TabPFN in-context predictive capability into an autoregressive generator without updating the pretrained backbone parameters.

*Table 24.* Hyperparameter search space of TabPFN.

| Hyperparameter | Range |
|---|---|
| feature_permutation | $\{$random$\}$ |
| n_permutations | $\{1,\ 3,\ 5,\ 10\}$ |
| temperature | $\{0.1,\ 0.3,\ 0.5,\ 0.7,\ 1.0\}$ |
| categorical_feature_inference | $\{$auto$\}$ |
| fast_mode | $\{$False$\}$ |

**TabDPT** is originally designed as an in-context tabular foundation predictor (Ma et al., 2025), and we repurpose it for autoregressive tabular data generation in the same procedure as TabPFN. TabDPT uses a row-based transformer encoder with retrieval-based context selection, and performs inference by conditioning each query row on retrieved context rows without dataset-specific weight updates.

*Table 25.* Hyperparameter search space of TabDPT.

| Hyperparameter | Range |
|---|---|
| feature_permutation | $\{$random$\}$ |
| n_ensembles | $\{1,\ 4,\ 8,\ 16\}$ |
| temperature | $\{0.5,\ 0.8,\ 1.0\}$ |
| context_size | $\{128,\ 256,\ 512,\ 1024,\ 2048\}$ |
| permute_classes | $\{$True, False$\}$ |

**Mitra** is originally proposed as an in-context tabular foundation predictor (Zhang et al., 2025b), and we repurpose it for autoregressive tabular data generation, similar to TabPFN. MITRA is pretrained on a curated mixture of synthetic priors, including structural causal models and tree-based priors, to improve generalisation across classification and regression tasks.

**LimiX** is originally pretrained with context-conditional masked modelling (Zhang et al., 2025c), and we repurpose it for autoregressive tabular data generation. LimiX is designed to learn a conditional distribution through masked prediction and can generate samples by iteratively masking and refilling subsets of features. As noted in Section 7.7 of its official technical report (Zhang et al., 2025c), LimiX performs generation iteratively, like other in-context tabular foundation predictors.

*Table 26.* Hyperparameter search space of Mitra.

| Hyperparameter | Range |
|---|---|
| n_estimators | $\{1, 2, 4, 8\}$ |
| max_classes | $\{10\}$ |
| device | $\{\texttt{auto}\}$ |

*Table 27.* Hyperparameter search space of LimiX.

| Hyperparameter | Range |
|---|---|
| backbone | $\{\texttt{LimiX-2M, LimiX-16M}\}$ |
| feature_permutation | $\{\texttt{random}\}$ |
| inference_config | $\{\texttt{default, retrieval-ensemble}\}$ |
| mix_precision | $\{\texttt{True, False}\}$ |
| softmax_temperature | $\{0.7, 0.9, 1.0\}$ |
| outlier_remove_std | $\{8, 12, 16\}$ |
| categorical_features_indices | $\{\texttt{auto}\}$ |

**TabICL** is originally proposed as an in-context tabular foundation predictor (Qu et al., 2026), and we repurpose it for autoregressive tabular data generation. TabICLv2 performs prediction by conditioning a query sample on labelled in-context examples through a scalable Transformer architecture with repeated feature grouping, target-aware embedding, and query-aware scalable softmax.

*Table 28.* Hyperparameter search space of TabICLv2-AR.

| Hyperparameter | Range |
|---|---|
| n_estimators | $\{1, 4, 8, 16, 32\}$ |
| softmax_temperature | $\{0.7, 0.9, 1.0\}$ |
| outlier_threshold | $\{3.0, 4.0, 6.0\}$ |
| average_logits | $\{\texttt{True, False}\}$ |
| support_many_classes | $\{\texttt{True}\}$ |
| use_amp | $\{\texttt{auto}\}$ |

### C.5. Experimental Setup

For all benchmark generators, we tune their hyperparameters using Optuna (Akiba et al., 2019) according to their average optimisation objective across the 10 repeated runs. Specifically, we tune parametrised generators to minimise their validation loss. For remaining in-context tabular foundation predictors, whose optimisation objectives are not designed for generation, we instead tune their inference hyperparameters to maximise global utility. Each generator is given at most two hours to complete a single repeat. Unless otherwise stated, reported results are averaged over 10 repeats. Specifically, we use the average distance to the minimum (ADTM) metric with affine renormalisation between the best-performing and worst-performing models (Jiang et al., 2026; Grinsztajn et al., 2022; McElfresh et al., 2024; Hollmann et al., 2025; Margeloiu et al., 2024; Jiang et al., 2024). We note that some generators may fail to converge or may generate unexpected values that cause metric computation to fail. In such cases, we impute their metric values for each repeat using the mean value of the other methods.

### C.6. Software Implementations

**Implementation of benchmark generators.** We implemented SMOTE with Imbalanced-learn (Lemaître et al., 2017), an open-source Python library for imbalanced datasets with an MIT license. For TabSyn, TabEBM, TabularARGN, TabDPT, and LimiX, we used their open-source implementations with an Apache-2.0 license. For CDTD, TabDiff, and NRGBoost, we used their open-source implementations with an MIT license. For TabSDS, TabNAT, TabPFN, Mitra, and TabICL, we used their open-source implementations with their customised licenses. For other benchmark generators, we used their open-source implementations in TabStruct (Jiang et al., 2026), an open-source toolbox for tabular data generation with an Apache-2.0 license.

**Implementation of data processing and evaluation.** All data handling, including data loading and preprocessing, was performed with TabCamel (Jiang, 2025a), an open-source Python library for tabular data management. All data-quality evaluation was performed with TabEval (Jiang, 2025b), a comprehensive open-source Python framework for evaluating tabular data.

**Implementation of result analysis and visualisation.** All numerical plots and graphics have been generated using Matplotlib 3.7 (Hunter, 2007), a Python-based plotting library with a BSD license. The icons for "frozon" and "trainable" in Figure 1 are from `https://icons8.com/`.

**Implementation of benchmark pipeline.** Following prior studies (Jiang et al., 2024; 2026), we ensure the consistency and reproducibility of experimental results by implementing a uniform pipeline using PyTorch Lightning (team, 2026), an open-source library under an Apache-2.0 license. We further fixed the random seeds for data loading and evaluation throughout the training and evaluation process. This ensured that TabFORGE and all benchmark models were trained and evaluated on the same set of samples. The experimental environment settings, including library dependencies, are specified in the open-source library for reference and reproduction purposes.

# D. Extended Analysis and Discussion

## D.1. Extended Analysis on Generation Quality

**Existing tabular foundation generators still struggle to match strong dataset-specific generators.** In Figure 2, dataset-specific generators account for seven of the top-10 methods in global utility and eight of the top-10 methods in local utility. This dominance suggests that existing foundation generators have not yet fully realised the benefits of transferable tabular representations for generation. It further showcases the key limitations of mainstream tabular foundation generators: (i) the weak overall performance of LLM-based generators (e.g., GReaT) and tabular foundation predictors (e.g., TabICL) indicates that treating tabular data as a sequential modality can be suboptimal for generation; (ii) the high local utility but low global utility of TabEBM suggests that it overemphasises the prediction target, rather than modelling the global causal structures across all features; (iii) the limited effectiveness of CTSyn further highlights the importance of integrating causality-aware representations for tabular generative modelling. Moreover, its instability further shows how to integrate metadata into tabular data generation remains an open research question.

## D.2. Extended Analysis on Ablation Impacts

**Early-to-middle encoder layers provide effective causal information.** As shown in Figure 5 (Left), using $L_{enc} = 12$ encoder layers yields the highest global utility, whereas both shallower and deeper encoders degrade performance, with deeper encoders performing even worse than shallower ones. This suggests that early-to-middle PFN layers provide the most useful latent representations for tabular data generation. Shallow layers may lack the capacity to capture rich inter-feature interactions, while deeper layers become increasingly biased towards the prediction target. Therefore, TabFORGE can achieve high performance as it implicitly attends to global causal structures across all features, which is essential for high-quality tabular generative modelling.

**Decoder refinement is necessary for generation-ready representations.** Figure 5 (Middle) shows that using no decoder layer leads to the weakest performance, whereas $L_{dec} = 4$ achieves the highest global utility. These results demonstrate that denoised latent embeddings cannot be directly detokenised into high-quality tabular data without further refinement. This further supports our design choice of using a transferable decoder transformer to refine causality-aware latent embeddings into generation-ready representations that are better aligned with the value space of each feature.

**Denoising-aligned decoding mitigates training-inference mismatch.** As shown in Figure 5 (Right), denoised embeddings achieve the highest global utility, outperforming both clean embeddings and clean embeddings augmented with noise. Specifically, following prior work (Zheng et al., 2026), we augment clean embeddings with additive Gaussian noise. The results indicate that generic noise augmentation only partially addresses the mismatch between the training and inference latent distributions. In contrast, directly training the decoder on denoised embeddings exposes it to representations that are more consistent with those encountered during inference. This finding further supports a key principle of TabFORGE: denoising-aligned decoding mitigates the train-inference distribution shift that commonly arises in latent diffusion models.

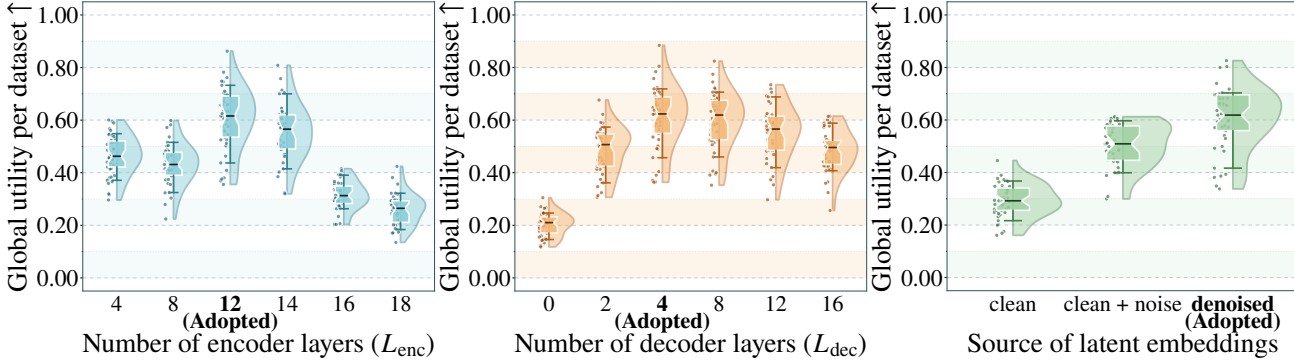

*Figure 5.* **Ablations studies of TabFORGE on 45 real-world datasets.** We report the global utility on each dataset and highlight the adopted configurations in TabFORGE. **Left:** Effect of $L_{enc}$, with the best performance at 12 layers, whereas deeper encoders perform worse because they can provide weaker signals of global causal structures. **Middle:** Effect of $L_{dec}$, where 4 layers yield the highest global utility, indicating that the decoder transformer is necessary to refine the denoised embeddings for generation. **Right:** Sources of latent embeddings, showing that denoised embeddings outperform clean embeddings, even when the latter are augmented with noise.

**The choice of feature encoder is not necessarily determined by predictive performance.** Table 29 shows that TabFORGE with TabPFN achieves the best average rank and ranks among the top three across all reported metrics. Importantly, stronger predictive performance does not necessarily imply a better encoder for generation. In particular, TabICL outperforms TabPFN in predictive performance (Qu et al., 2026), but the row-wise interaction of TabICL collapses feature embeddings for a sample into a single vector. This can substantially break per-feature semantics, making TabICL less effective for encoding the comprehensive causal information. The advantage of TabPFN suggests that per-feature representations can provide a stronger inductive bias for capturing transferable causal interactions across heterogeneous tabular datasets.

*Table 29.* **Comparison of different feature encoders in TabFORGE on 45 real-world datasets.** We report the normalised mean ± std metric values and average rank. A higher rank indicates better performance. Since the normalisation is performed only across the five variants of TabFORGE considered here, the reported numbers differ from those in Figure 2 and Figure 3. We highlight the **First**, **Second** and **Third** best performances. The adopted TabPFN-based configuration consistently ranks Top-3 across all metrics and achieves the best overall performance.

| Feature encoder type | Density Estimation | | | | Privacy Preservation | | ML Efficacy | Structural Fidelity | Average Rank |
| | Shape ↑ | Trend ↑ | $\alpha$-precision ↑ | $\beta$-recall ↑ | Authenticity ↑ | DCR Score ↑ | Local utility ↑ | Global utility ↑ | |
|---|---|---|---|---|---|---|---|---|---|
| w/ TabDPT | $0.00_{\pm0.00}$ | $0.01_{\pm0.01}$ | $0.10_{\pm0.02}$ | $0.52_{\pm0.04}$ | $0.00_{\pm0.00}$ | $0.00_{\pm0.00}$ | $0.05_{\pm0.02}$ | $0.26_{\pm0.04}$ | $4.75_{\pm0.46}$ |
| w/ Mitra | $0.51_{\pm0.02}$ | $0.49_{\pm0.07}$ | $0.51_{\pm0.08}$ | $0.57_{\pm0.08}$ | $0.51_{\pm0.03}$ | $0.30_{\pm0.03}$ | $0.53_{\pm0.07}$ | $0.63_{\pm0.08}$ | $2.38_{\pm0.52}$ |
| w/ LimiX | $0.36_{\pm0.02}$ | $0.36_{\pm0.06}$ | $0.40_{\pm0.07}$ | $0.52_{\pm0.07}$ | $0.44_{\pm0.03}$ | $0.26_{\pm0.03}$ | $0.44_{\pm0.06}$ | $0.56_{\pm0.08}$ | $3.50_{\pm0.53}$ |
| w/ TabICL | $0.71_{\pm0.02}$ | $0.44_{\pm0.05}$ | $0.35_{\pm0.06}$ | $0.08_{\pm0.01}$ | $0.92_{\pm0.00}$ | $0.93_{\pm0.00}$ | $0.39_{\pm0.06}$ | $0.09_{\pm0.03}$ | $3.12_{\pm1.64}$ |
| å w/ TabPFN (Adopted) | $0.99_{\pm0.00}$ | $0.99_{\pm0.01}$ | $1.00_{\pm0.00}$ | $0.91_{\pm0.03}$ | $0.77_{\pm0.03}$ | $0.38_{\pm0.03}$ | $1.00_{\pm0.00}$ | $0.99_{\pm0.01}$ | $1.25_{\pm0.46}$ |

### D.3. Extended Analysis on Practicability

**TabFORGE incurs a low fitting cost on unseen datasets.** Figure 6 (Left) shows that TabFORGE is substantially cheaper to fit than most benchmark methods. Specifically, the total fitting time of TabFORGE is only around 9.71% of that required to train a strong dataset-specific generator (i.e., TabSyn) from scratch. This suggests that TabFORGE offers a practical trade-off between generation quality and computational efficiency. The efficiency gain stems from the fact that most representational and generative capacity is acquired during pretraining, leaving only lightweight finetuning for unseen datasets. This further demonstrates the practical value of leveraging knowledge from diverse tabular datasets through pretraining.

**TabFORGE supports efficient and flexible synthetic data generation.** Figure 6 (Right) shows that the generation cost of TabFORGE is only around 1.66% of that required by the fastest existing in-context tabular foundation model (i.e., TabEBM). This is because TabFORGE is designed to generate synthetic data by denoising random latent noise and decoding the denoised latent embeddings, without invoking the causality-aware feature encoder. We also note that the considered tabular foundation predictors are all equipped with fixed prediction heads, and can therefore generate categorical features with at most 10 classes. To apply them to all considered datasets, we perform hierarchical classification, as suggested by prior studies (Qu et al., 2026; 2025). As a result, the generation time of these foundation predictors increases substantially. In contrast, TabFORGE employs a lightweight detokeniser that can be efficiently applied to diverse feature spaces.

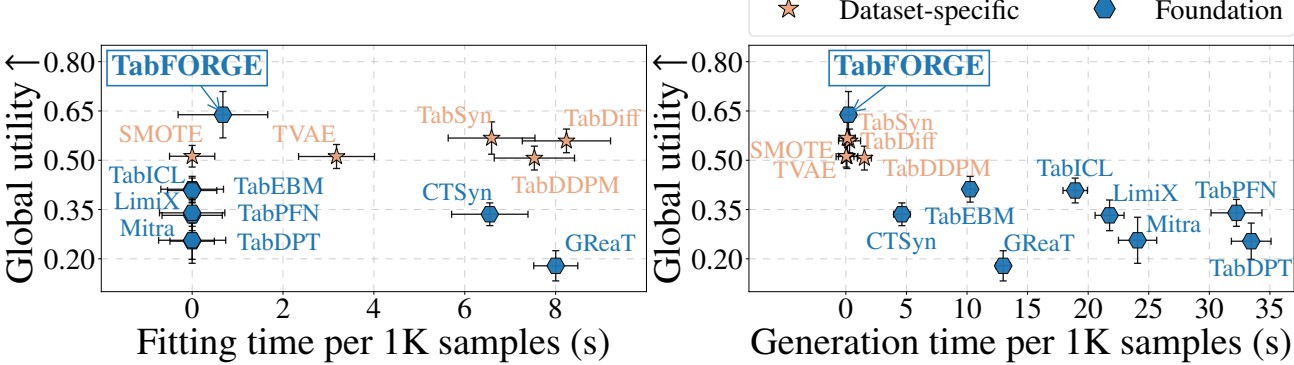

*Figure 6.* **Computation efficiency on 45 real-world datasets.** For visual clarity, we report all nine foundation models and Top-5 dataset-specific methods in global utility. **Left:** Median fitting time per 1,000 samples vs. mean normalised global utility. **Right:** Median generation time per 1,000 samples vs. mean normalised global utility. TabFORGE generally achieves better structural fidelity with higher computation efficiency than benchmark methods.

### D.4. Future Work

**Scale up the pretraining scale, including data and model size.** The current pretraining scale of TabFORGE is primarily constrained by our available computational resources. In this study, we pretrain TabFORGE on 43 real-world datasets using a moderate-size diffusion transformer and decoder architecture. Despite this limited scale, TabFORGE already demonstrates strong performance across 45 unseen real-world benchmark datasets, outperforming existing tabular foundation generators by a clear margin. Importantly, we have not observed clear signs of performance saturation in our experiments, suggesting that the current computational budget may still be far from fully exploiting the potential of the proposed framework. A promising direction is therefore to scale up TabFORGE along multiple axes. First, increasing the number and diversity of pretraining datasets may allow the model to acquire broader tabular structural priors. Second, increasing the capacity of the diffusion transformer and decoder may further improve its ability to capture complex latent distributions and refine denoised embeddings into generation-ready representations. Third, longer pretraining may further improve the stability and transferability of the learned generative representations. We therefore plan to expand the pretraining scale in future work, including larger upstream tabular corpora, larger model architectures, and a more extensive pretraining stage.

**Improve the implementation for computation efficiency.** For generalisability and stability, the current implementation of TabFORGE relies primarily on standard PyTorch modules (Paszke et al., 2019), such as standard attention mechanisms and transformer layers. This choice makes the framework easier to reproduce and extend across heterogeneous tabular datasets. However, it also leaves substantial room for further engineering optimisation. For instance, future implementations can integrate more efficient attention kernels, such as FlashAttention (Dao et al., 2022) and FlashAttention-2 (Dao, 2023), to reduce memory usage and improve throughput during both pretraining and fitting. These efficiency improvements could make large-scale pretraining more practical and enable TabFORGE to scale to broader tabular corpora and larger model variants.

**Explore tabular data generation with explicit causal structures.** As discussed in Section 1 and Section 2, TabFORGE is designed to leverage implicit structural signals encoded in latent embeddings, rather than explicitly estimating or conditioning on causal graphs. This design choice is important, as reliable causal graph discovery from real-world tabular data remains an open research problem (Jiang et al., 2026; Kaddour et al., 2022; Tu et al., 2024; Glymour et al., 2019; Nastl & Hardt, 2024). In practice, ground-truth structures are rarely available, and prior studies (Tu et al., 2024; Jiang et al., 2026) have shown that even state-of-the-art causal discovery methods may perform poorly or yield misleading structures on real-world datasets. As a result, generators that rely heavily on explicitly learned causal structures may inherit errors from imperfect causal discovery and underperform on real-world benchmarks (Jiang et al., 2026), including methods such as GOGGLE (Liu et al., 2023) and Bayesian Network (Kitson et al., 2023). TabFORGE instead uses implicit causal information embedded in pretrained latent representations as an inductive bias for modelling global causal structures, while avoiding reliance on potentially inaccurate and unverifiable explicit causal structures. Nevertheless, further bridging implicit and explicit causal modelling remains a promising and interesting direction for improving tabular data generation.

# E. Extended Experimental Results

In this section, we provide the per-dataset evaluation results for all 45 benchmark datasets. We note that, for the real holdout data, we evaluate it only in the metrics for which the training data serves as a lower/upper-bound reference (i.e., excluding local and global utility) to assess overfitting risks.

*Table 30.* **Raw benchmark results of 23 tabular generators on the "Airfoil" dataset.** We report the mean $\pm$ std of each metric across 10 repeated data splits. For benchmark generators, "$-$" denotes failed convergence of a specific model or unexpected values in the synthetic data that caused the evaluation metric computation to crash. We highlight the **First**, **Second**, and **Third** best performances for each metric. TabFORGE generally achieves competitive performance against the benchmark generators while maintaining a reduced risk of overfitting.

| Generator | Density Estimation | | | | Privacy Preservation | | ML Efficacy | Structural Fidelity |
|---|---|---|---|---|---|---|---|---|
| | Shape ↑ | Trend ↑ | $\alpha$-precision ↑ | $\beta$-recall ↑ | Authenticity ↑ | DCR Score ↑ | Local utility ↑ | Global utility ↑ |
| **Real Data** | | | | | | | | |
| Real Data (Train) | $1.00_{\pm0.00}$ | $1.00_{\pm0.00}$ | $1.00_{\pm0.00}$ | $1.00_{\pm0.00}$ | $0.00_{\pm0.00}$ | $0.00_{\pm0.00}$ | $1.00_{\pm0.00}$ | $1.00_{\pm0.00}$ |
| Real Data (Holdout) | $0.95_{\pm0.01}$ | $0.86_{\pm0.02}$ | $0.97_{\pm0.01}$ | $0.50_{\pm0.03}$ | $0.61_{\pm0.02}$ | $0.04_{\pm0.00}$ | $-$ | $-$ |
| **Dataset-specific Model** | | | | | | | | |
| SMOTE | $0.89_{\pm0.02}$ | $0.53_{\pm0.01}$ | $0.81_{\pm0.01}$ | $0.54_{\pm0.03}$ | $0.54_{\pm0.01}$ | $0.03_{\pm0.00}$ | $0.80_{\pm0.05}$ | $0.52_{\pm0.02}$ |
| TabSDS | $-$ | $-$ | $-$ | $-$ | $-$ | $-$ | $-$ | $-$ |
| TVAE | $0.77_{\pm0.02}$ | $0.41_{\pm0.03}$ | $0.91_{\pm0.03}$ | $0.11_{\pm0.02}$ | $0.94_{\pm0.01}$ | $0.41_{\pm0.03}$ | $0.54_{\pm0.03}$ | $0.52_{\pm0.02}$ |
| GOGGLE | $0.52_{\pm0.04}$ | $0.24_{\pm0.05}$ | $0.78_{\pm0.08}$ | $0.04_{\pm0.02}$ | $0.98_{\pm0.01}$ | $0.32_{\pm0.10}$ | $0.30_{\pm0.03}$ | $0.13_{\pm0.01}$ |
| CTGAN | $0.76_{\pm0.05}$ | $0.37_{\pm0.07}$ | $0.87_{\pm0.08}$ | $0.06_{\pm0.02}$ | $0.97_{\pm0.01}$ | $0.46_{\pm0.07}$ | $0.51_{\pm0.05}$ | $0.26_{\pm0.04}$ |
| NFlow | $0.82_{\pm0.02}$ | $0.39_{\pm0.07}$ | $0.91_{\pm0.06}$ | $0.06_{\pm0.02}$ | $0.97_{\pm0.01}$ | $0.50_{\pm0.04}$ | $0.41_{\pm0.01}$ | $0.27_{\pm0.02}$ |
| ARF | $-$ | $-$ | $-$ | $-$ | $-$ | $-$ | $-$ | $-$ |
| TabDDPM | $0.81_{\pm0.03}$ | $0.47_{\pm0.04}$ | $0.92_{\pm0.03}$ | $0.16_{\pm0.02}$ | $0.92_{\pm0.01}$ | $0.41_{\pm0.04}$ | $0.70_{\pm0.04}$ | $0.56_{\pm0.02}$ |
| CDTD | $0.63_{\pm0.02}$ | $0.29_{\pm0.02}$ | $0.65_{\pm0.02}$ | $0.10_{\pm0.01}$ | $0.84_{\pm0.01}$ | $0.44_{\pm0.02}$ | $0.57_{\pm0.02}$ | $0.30_{\pm0.01}$ |
| TabSyn | $0.87_{\pm0.02}$ | $0.54_{\pm0.05}$ | $0.96_{\pm0.02}$ | $0.28_{\pm0.03}$ | $0.82_{\pm0.03}$ | $0.27_{\pm0.04}$ | $0.80_{\pm0.04}$ | $0.53_{\pm0.03}$ |
| TabDiff | $0.83_{\pm0.03}$ | $0.40_{\pm0.04}$ | $0.89_{\pm0.05}$ | $0.07_{\pm0.02}$ | $0.97_{\pm0.01}$ | $0.50_{\pm0.03}$ | $0.48_{\pm0.02}$ | $0.58_{\pm0.02}$ |
| NRGBoost | $0.85_{\pm0.01}$ | $0.61_{\pm0.05}$ | $0.94_{\pm0.03}$ | $0.22_{\pm0.03}$ | $0.82_{\pm0.03}$ | $0.12_{\pm0.02}$ | $0.79_{\pm0.04}$ | $0.32_{\pm0.01}$ |
| TabNAT | $0.63_{\pm0.01}$ | $0.29_{\pm0.02}$ | $0.58_{\pm0.02}$ | $0.10_{\pm0.01}$ | $0.86_{\pm0.01}$ | $0.50_{\pm0.01}$ | $0.44_{\pm0.02}$ | $0.24_{\pm0.02}$ |
| TabularARGN | $0.76_{\pm0.01}$ | $0.61_{\pm0.07}$ | $0.96_{\pm0.02}$ | $0.14_{\pm0.04}$ | $0.92_{\pm0.02}$ | $0.31_{\pm0.07}$ | $0.46_{\pm0.02}$ | $0.30_{\pm0.04}$ |
| **Foundation Model** | | | | | | | | |
| GReaT | $0.48_{\pm0.01}$ | $0.15_{\pm0.02}$ | $0.21_{\pm0.05}$ | $0.00_{\pm0.00}$ | $1.00_{\pm0.00}$ | $1.00_{\pm0.00}$ | $0.44_{\pm0.06}$ | $0.02_{\pm0.01}$ |
| TabPFN | $-$ | $-$ | $-$ | $-$ | $-$ | $-$ | $-$ | $-$ |
| TabDPT | $0.61_{\pm0.02}$ | $0.33_{\pm0.03}$ | $0.76_{\pm0.04}$ | $0.06_{\pm0.02}$ | $0.86_{\pm0.01}$ | $0.34_{\pm0.06}$ | $0.41_{\pm0.03}$ | $0.22_{\pm0.02}$ |
| Mitra | $0.59_{\pm0.03}$ | $0.28_{\pm0.02}$ | $0.68_{\pm0.04}$ | $0.04_{\pm0.01}$ | $0.88_{\pm0.00}$ | $0.30_{\pm0.05}$ | $0.41_{\pm0.03}$ | $0.20_{\pm0.01}$ |
| LimiX | $-$ | $-$ | $-$ | $-$ | $-$ | $-$ | $-$ | $-$ |
| TabICL | $-$ | $-$ | $-$ | $-$ | $-$ | $-$ | $-$ | $-$ |
| TabEBM | $0.84_{\pm0.01}$ | $0.18_{\pm0.02}$ | $0.85_{\pm0.03}$ | $0.06_{\pm0.01}$ | $0.97_{\pm0.01}$ | $0.32_{\pm0.01}$ | $0.65_{\pm0.03}$ | $0.36_{\pm0.01}$ |
| CTSyn | $0.56_{\pm0.01}$ | $0.27_{\pm0.01}$ | $0.53_{\pm0.02}$ | $0.10_{\pm0.01}$ | $0.78_{\pm0.01}$ | $0.49_{\pm0.02}$ | $0.52_{\pm0.03}$ | $0.25_{\pm0.01}$ |
| **TabFORGE (Ours)** | $0.88_{\pm0.02}$ | $0.65_{\pm0.02}$ | $0.95_{\pm0.02}$ | $0.40_{\pm0.02}$ | $0.67_{\pm0.02}$ | $0.04_{\pm0.01}$ | $0.81_{\pm0.04}$ | $0.61_{\pm0.03}$ |

*Table 31.* **Raw benchmark results of 23 tabular generators on the "BankChurn" dataset.** We report the mean $\pm$ std of each metric across 10 repeated data splits. For benchmark generators, "$-$" denotes failed convergence of a specific model or unexpected values in the synthetic data that caused the evaluation metric computation to crash. We highlight the First, Second, and Third best performances for each metric. TabFORGE generally achieves competitive performance against the benchmark generators while maintaining a reduced risk of overfitting.

| Generator | Density Estimation | | | | Privacy Preservation | | ML Efficacy | Structural Fidelity |
|---|---|---|---|---|---|---|---|---|
| | Shape $\uparrow$ | Trend $\uparrow$ | $\alpha$-precision $\uparrow$ | $\beta$-recall $\uparrow$ | Authenticity $\uparrow$ | DCR Score $\uparrow$ | Local utility $\uparrow$ | Global utility $\uparrow$ |
| Real Data | | | | | | | | |
| Real Data (Train) | $1.00_{\pm 0.00}$ | $1.00_{\pm 0.00}$ | $1.00_{\pm 0.00}$ | $1.00_{\pm 0.00}$ | $0.00_{\pm 0.00}$ | $0.00_{\pm 0.00}$ | $1.00_{\pm 0.00}$ | $0.79_{\pm 0.07}$ |
| Real Data (Holdout) | $0.99_{\pm 0.00}$ | $0.94_{\pm 0.02}$ | $0.99_{\pm 0.01}$ | $0.50_{\pm 0.01}$ | $0.67_{\pm 0.01}$ | $0.24_{\pm 0.00}$ | $-$ | $-$ |
| Dataset-specific Model | | | | | | | | |
| SMOTE | $0.94_{\pm 0.00}$ | $0.66_{\pm 0.02}$ | $0.86_{\pm 0.01}$ | $0.67_{\pm 0.01}$ | $0.42_{\pm 0.01}$ | $0.13_{\pm 0.00}$ | $1.00_{\pm 0.01}$ | $0.64_{\pm 0.06}$ |
| TabSDS | $0.67_{\pm 0.00}$ | $0.54_{\pm 0.02}$ | $0.64_{\pm 0.00}$ | $0.23_{\pm 0.00}$ | $0.61_{\pm 0.00}$ | $0.27_{\pm 0.00}$ | $0.73_{\pm 0.01}$ | $0.51_{\pm 0.05}$ |
| TVAE | $0.92_{\pm 0.01}$ | $0.69_{\pm 0.07}$ | $0.89_{\pm 0.03}$ | $0.41_{\pm 0.02}$ | $0.75_{\pm 0.01}$ | $0.38_{\pm 0.01}$ | $0.80_{\pm 0.03}$ | $0.64_{\pm 0.06}$ |
| GOGGLE | $0.65_{\pm 0.03}$ | $0.27_{\pm 0.05}$ | $0.21_{\pm 0.05}$ | $0.10_{\pm 0.03}$ | $0.96_{\pm 0.02}$ | $0.28_{\pm 0.02}$ | $0.52_{\pm 0.18}$ | $0.31_{\pm 0.04}$ |
| CTGAN | $0.91_{\pm 0.01}$ | $0.83_{\pm 0.05}$ | $0.95_{\pm 0.03}$ | $0.35_{\pm 0.02}$ | $0.78_{\pm 0.01}$ | $0.40_{\pm 0.02}$ | $0.90_{\pm 0.04}$ | $0.66_{\pm 0.10}$ |
| NFlow | $0.87_{\pm 0.02}$ | $0.82_{\pm 0.04}$ | $0.90_{\pm 0.08}$ | $0.34_{\pm 0.02}$ | $0.80_{\pm 0.01}$ | $0.42_{\pm 0.02}$ | $0.70_{\pm 0.03}$ | $0.59_{\pm 0.06}$ |
| ARF | $0.96_{\pm 0.00}$ | $0.70_{\pm 0.07}$ | $0.99_{\pm 0.00}$ | $0.38_{\pm 0.01}$ | $0.76_{\pm 0.01}$ | $0.41_{\pm 0.01}$ | $0.91_{\pm 0.02}$ | $0.52_{\pm 0.05}$ |
| TabDDPM | $0.94_{\pm 0.01}$ | $0.91_{\pm 0.06}$ | $0.96_{\pm 0.01}$ | $0.38_{\pm 0.01}$ | $0.77_{\pm 0.01}$ | $0.39_{\pm 0.01}$ | $0.91_{\pm 0.02}$ | $0.58_{\pm 0.06}$ |
| CDTD | $0.80_{\pm 0.00}$ | $0.55_{\pm 0.02}$ | $0.80_{\pm 0.01}$ | $0.26_{\pm 0.00}$ | $0.67_{\pm 0.00}$ | $0.32_{\pm 0.01}$ | $0.71_{\pm 0.01}$ | $0.47_{\pm 0.04}$ |
| TabSyn | $0.94_{\pm 0.00}$ | $0.87_{\pm 0.02}$ | $0.96_{\pm 0.02}$ | $0.43_{\pm 0.01}$ | $0.72_{\pm 0.01}$ | $0.35_{\pm 0.01}$ | $0.97_{\pm 0.02}$ | $0.70_{\pm 0.07}$ |
| TabDiff | $0.89_{\pm 0.01}$ | $0.77_{\pm 0.01}$ | $0.79_{\pm 0.03}$ | $0.25_{\pm 0.01}$ | $0.85_{\pm 0.01}$ | $0.58_{\pm 0.02}$ | $0.75_{\pm 0.02}$ | $0.60_{\pm 0.06}$ |
| NRGBoost | $0.89_{\pm 0.00}$ | $0.68_{\pm 0.03}$ | $0.94_{\pm 0.01}$ | $0.13_{\pm 0.01}$ | $0.91_{\pm 0.01}$ | $0.48_{\pm 0.02}$ | $0.92_{\pm 0.02}$ | $0.32_{\pm 0.04}$ |
| TabNAT | $0.73_{\pm 0.00}$ | $0.53_{\pm 0.03}$ | $0.78_{\pm 0.02}$ | $0.24_{\pm 0.01}$ | $0.80_{\pm 0.01}$ | $0.42_{\pm 0.01}$ | $0.72_{\pm 0.01}$ | $0.41_{\pm 0.03}$ |
| TabularARGN | $0.80_{\pm 0.00}$ | $0.85_{\pm 0.02}$ | $0.95_{\pm 0.01}$ | $0.18_{\pm 0.00}$ | $0.89_{\pm 0.00}$ | $0.40_{\pm 0.01}$ | $0.88_{\pm 0.03}$ | $0.56_{\pm 0.07}$ |
| Foundation Model | | | | | | | | |
| GReaT | $0.88_{\pm 0.00}$ | $0.31_{\pm 0.11}$ | $0.95_{\pm 0.01}$ | $0.28_{\pm 0.01}$ | $0.85_{\pm 0.00}$ | $0.44_{\pm 0.02}$ | $0.73_{\pm 0.01}$ | $0.28_{\pm 0.07}$ |
| TabPFN | $0.80_{\pm 0.01}$ | $0.60_{\pm 0.04}$ | $0.80_{\pm 0.03}$ | $0.27_{\pm 0.01}$ | $0.74_{\pm 0.01}$ | $0.38_{\pm 0.01}$ | $0.68_{\pm 0.01}$ | $0.50_{\pm 0.04}$ |
| TabDPT | $0.63_{\pm 0.01}$ | $0.51_{\pm 0.02}$ | $0.63_{\pm 0.02}$ | $0.19_{\pm 0.01}$ | $0.69_{\pm 0.01}$ | $0.32_{\pm 0.01}$ | $0.73_{\pm 0.09}$ | $0.55_{\pm 0.05}$ |
| Mitra | $0.65_{\pm 0.02}$ | $0.39_{\pm 0.02}$ | $0.44_{\pm 0.03}$ | $0.18_{\pm 0.02}$ | $0.73_{\pm 0.01}$ | $0.27_{\pm 0.01}$ | $0.79_{\pm 0.15}$ | $0.49_{\pm 0.03}$ |
| LimiX | $0.75_{\pm 0.01}$ | $0.57_{\pm 0.02}$ | $0.70_{\pm 0.03}$ | $0.28_{\pm 0.01}$ | $0.70_{\pm 0.01}$ | $0.33_{\pm 0.01}$ | $0.70_{\pm 0.05}$ | $0.55_{\pm 0.04}$ |
| TabICL | $0.82_{\pm 0.00}$ | $0.68_{\pm 0.03}$ | $0.77_{\pm 0.02}$ | $0.33_{\pm 0.01}$ | $0.66_{\pm 0.01}$ | $0.36_{\pm 0.01}$ | $0.75_{\pm 0.01}$ | $0.58_{\pm 0.05}$ |
| TabEBM | $0.93_{\pm 0.00}$ | $0.48_{\pm 0.04}$ | $0.91_{\pm 0.02}$ | $0.18_{\pm 0.01}$ | $0.87_{\pm 0.00}$ | $0.42_{\pm 0.01}$ | $0.92_{\pm 0.03}$ | $0.51_{\pm 0.06}$ |
| CTSyn | $0.76_{\pm 0.01}$ | $0.40_{\pm 0.04}$ | $0.71_{\pm 0.01}$ | $0.24_{\pm 0.00}$ | $0.64_{\pm 0.00}$ | $0.32_{\pm 0.01}$ | $0.78_{\pm 0.07}$ | $0.44_{\pm 0.02}$ |
| **TabFORGE (Ours)** | $0.89_{\pm 0.01}$ | $0.75_{\pm 0.04}$ | $0.95_{\pm 0.02}$ | $0.36_{\pm 0.01}$ | $0.76_{\pm 0.01}$ | $0.36_{\pm 0.01}$ | $0.96_{\pm 0.05}$ | $0.72_{\pm 0.04}$ |

*Table 32.* **Raw benchmark results of 23 tabular generators on the "Bankruptcy" dataset.** We report the mean $\pm$ std of each metric across 10 repeated data splits. For benchmark generators, "$-$" denotes failed convergence of a specific model or unexpected values in the synthetic data that caused the evaluation metric computation to crash. We highlight the **First**, **Second**, and **Third** best performances for each metric. TabFORGE generally achieves competitive performance against the benchmark generators while maintaining a reduced risk of overfitting.

| Generator | Density Estimation | | | | Privacy Preservation | | ML Efficacy | Structural Fidelity |
| --- | --- | --- | --- | --- | --- | --- | --- | --- |
| | Shape ↑ | Trend ↑ | $\alpha$-precision ↑ | $\beta$-recall ↑ | Authenticity ↑ | DCR Score ↑ | Local utility ↑ | Global utility ↑ |
| **Real Data** | | | | | | | | |
| Real Data (Train) | $1.00_{\pm0.00}$ | $1.00_{\pm0.00}$ | $1.00_{\pm0.00}$ | $1.00_{\pm0.00}$ | $0.00_{\pm0.00}$ | $0.00_{\pm0.00}$ | $1.00_{\pm0.00}$ | $0.94_{\pm0.02}$ |
| Real Data (Holdout) | $0.98_{\pm0.00}$ | $0.87_{\pm0.02}$ | $0.98_{\pm0.01}$ | $0.50_{\pm0.01}$ | $0.73_{\pm0.01}$ | $0.03_{\pm0.00}$ | $-$ | $-$ |
| **Dataset-specific Model** | | | | | | | | |
| SMOTE | $0.76_{\pm0.01}$ | $0.73_{\pm0.01}$ | $0.82_{\pm0.02}$ | $0.67_{\pm0.01}$ | $0.46_{\pm0.01}$ | $0.02_{\pm0.00}$ | $1.01_{\pm0.05}$ | $0.66_{\pm0.02}$ |
| TabSDS | $0.54_{\pm0.01}$ | $0.64_{\pm0.02}$ | $0.57_{\pm0.02}$ | $0.03_{\pm0.02}$ | $0.67_{\pm0.01}$ | $0.08_{\pm0.00}$ | $0.66_{\pm0.02}$ | $0.46_{\pm0.03}$ |
| TVAE | $0.69_{\pm0.01}$ | $0.74_{\pm0.04}$ | $0.81_{\pm0.05}$ | $0.19_{\pm0.03}$ | $0.92_{\pm0.02}$ | $0.06_{\pm0.00}$ | $0.94_{\pm0.08}$ | $0.59_{\pm0.03}$ |
| GOGGLE | $-$ | $-$ | $-$ | $-$ | $-$ | $-$ | $-$ | $-$ |
| CTGAN | $0.62_{\pm0.05}$ | $0.74_{\pm0.02}$ | $0.80_{\pm0.13}$ | $0.04_{\pm0.02}$ | $0.98_{\pm0.01}$ | $0.08_{\pm0.01}$ | $0.93_{\pm0.04}$ | $0.43_{\pm0.07}$ |
| NFlow | $0.73_{\pm0.01}$ | $0.65_{\pm0.02}$ | $0.86_{\pm0.10}$ | $0.01_{\pm0.00}$ | $1.00_{\pm0.00}$ | $0.09_{\pm0.00}$ | $0.77_{\pm0.04}$ | $0.38_{\pm0.04}$ |
| ARF | $0.76_{\pm0.01}$ | $0.77_{\pm0.02}$ | $0.93_{\pm0.01}$ | $0.02_{\pm0.00}$ | $1.00_{\pm0.00}$ | $0.10_{\pm0.00}$ | $0.93_{\pm0.04}$ | $0.47_{\pm0.02}$ |
| TabDDPM | $0.43_{\pm0.01}$ | $0.81_{\pm0.04}$ | $0.00_{\pm0.00}$ | $0.00_{\pm0.00}$ | $1.00_{\pm0.00}$ | $1.00_{\pm0.00}$ | $0.93_{\pm0.10}$ | $0.58_{\pm0.02}$ |
| CDTD | $-$ | $-$ | $-$ | $-$ | $-$ | $-$ | $-$ | $-$ |
| TabSyn | $0.73_{\pm0.01}$ | $0.82_{\pm0.03}$ | $0.93_{\pm0.06}$ | $0.13_{\pm0.09}$ | $0.95_{\pm0.04}$ | $0.06_{\pm0.01}$ | $0.94_{\pm0.04}$ | $0.56_{\pm0.06}$ |
| TabDiff | $0.73_{\pm0.01}$ | $0.67_{\pm0.02}$ | $0.87_{\pm0.03}$ | $0.02_{\pm0.01}$ | $1.00_{\pm0.00}$ | $0.08_{\pm0.01}$ | $0.89_{\pm0.04}$ | $0.65_{\pm0.03}$ |
| NRGBoost | $0.73_{\pm0.01}$ | $0.78_{\pm0.02}$ | $0.11_{\pm0.02}$ | $0.00_{\pm0.00}$ | $1.00_{\pm0.00}$ | $0.16_{\pm0.01}$ | $0.88_{\pm0.03}$ | $0.16_{\pm0.02}$ |
| TabNAT | $-$ | $-$ | $-$ | $-$ | $-$ | $-$ | $-$ | $-$ |
| TabularARGN | $0.83_{\pm0.00}$ | $0.64_{\pm0.01}$ | $0.95_{\pm0.01}$ | $0.02_{\pm0.01}$ | $1.00_{\pm0.00}$ | $0.08_{\pm0.01}$ | $0.88_{\pm0.03}$ | $0.53_{\pm0.03}$ |
| **Foundation Model** | | | | | | | | |
| GReaT | $-$ | $-$ | $-$ | $-$ | $-$ | $-$ | $-$ | $-$ |
| TabPFN | $-$ | $-$ | $-$ | $-$ | $-$ | $-$ | $-$ | $-$ |
| TabDPT | $-$ | $-$ | $-$ | $-$ | $-$ | $-$ | $-$ | $-$ |
| Mitra | $-$ | $-$ | $-$ | $-$ | $-$ | $-$ | $-$ | $-$ |
| LimiX | $-$ | $-$ | $-$ | $-$ | $-$ | $-$ | $-$ | $-$ |
| TabICL | $0.59_{\pm0.01}$ | $0.61_{\pm0.01}$ | $0.80_{\pm0.04}$ | $0.04_{\pm0.02}$ | $0.85_{\pm0.01}$ | $0.07_{\pm0.00}$ | $0.78_{\pm0.03}$ | $0.50_{\pm0.03}$ |
| TabEBM | $0.75_{\pm0.01}$ | $0.89_{\pm0.02}$ | $0.80_{\pm0.02}$ | $0.12_{\pm0.01}$ | $0.88_{\pm0.01}$ | $0.07_{\pm0.00}$ | $0.98_{\pm0.05}$ | $0.56_{\pm0.02}$ |
| CTSyn | $-$ | $-$ | $-$ | $-$ | $-$ | $-$ | $-$ | $-$ |
| **TabFORGE (Ours)** | $0.75_{\pm0.01}$ | $0.89_{\pm0.02}$ | $0.97_{\pm0.01}$ | $0.44_{\pm0.01}$ | $0.67_{\pm0.02}$ | $0.03_{\pm0.00}$ | $1.01_{\pm0.03}$ | $0.69_{\pm0.03}$ |

*Table 33.* **Raw benchmark results of 23 tabular generators on the "Biodeg" dataset.** We report the mean $\pm$ std of each metric across 10 repeated data splits. For benchmark generators, "$-$" denotes failed convergence of a specific model or unexpected values in the synthetic data that caused the evaluation metric computation to crash. We highlight the **First**, **Second**, and **Third** best performances for each metric. TabFORGE generally achieves competitive performance against the benchmark generators while maintaining a reduced risk of overfitting.

| Generator | Density Estimation | | | | Privacy Preservation | | ML Efficacy | Structural Fidelity |
|---|---|---|---|---|---|---|---|---|
| | Shape ↑ | Trend ↑ | $\alpha$-precision ↑ | $\beta$-recall ↑ | Authenticity ↑ | DCR Score ↑ | Local utility ↑ | Global utility ↑ |
| **Real Data** | | | | | | | | |
| Real Data (Train) | $1.00_{\pm0.00}$ | $1.00_{\pm0.00}$ | $1.00_{\pm0.00}$ | $1.00_{\pm0.00}$ | $0.00_{\pm0.00}$ | $0.00_{\pm0.00}$ | $1.00_{\pm0.00}$ | $0.95_{\pm0.01}$ |
| Real Data (Holdout) | $0.96_{\pm0.01}$ | $0.87_{\pm0.02}$ | $0.96_{\pm0.01}$ | $0.51_{\pm0.03}$ | $0.69_{\pm0.02}$ | $0.04_{\pm0.00}$ | $-$ | $-$ |
| **Dataset-specific Model** | | | | | | | | |
| SMOTE | $0.64_{\pm0.00}$ | $0.71_{\pm0.01}$ | $0.72_{\pm0.02}$ | $0.64_{\pm0.02}$ | $0.48_{\pm0.02}$ | $0.02_{\pm0.00}$ | $0.96_{\pm0.02}$ | $0.56_{\pm0.03}$ |
| TabSDS | $0.45_{\pm0.00}$ | $0.56_{\pm0.02}$ | $0.70_{\pm0.01}$ | $0.09_{\pm0.01}$ | $0.77_{\pm0.01}$ | $0.08_{\pm0.01}$ | $0.64_{\pm0.02}$ | $0.35_{\pm0.02}$ |
| TVAE | $0.60_{\pm0.01}$ | $0.70_{\pm0.02}$ | $0.64_{\pm0.06}$ | $0.09_{\pm0.02}$ | $0.97_{\pm0.01}$ | $0.10_{\pm0.01}$ | $0.77_{\pm0.07}$ | $0.52_{\pm0.03}$ |
| GOGGLE | $0.43_{\pm0.02}$ | $0.50_{\pm0.03}$ | $0.17_{\pm0.03}$ | $0.03_{\pm0.01}$ | $0.99_{\pm0.00}$ | $0.09_{\pm0.01}$ | $-$ | $-$ |
| CTGAN | $0.59_{\pm0.02}$ | $0.77_{\pm0.02}$ | $0.92_{\pm0.02}$ | $0.03_{\pm0.01}$ | $0.99_{\pm0.01}$ | $0.14_{\pm0.01}$ | $0.79_{\pm0.05}$ | $0.27_{\pm0.03}$ |
| NFlow | $0.65_{\pm0.01}$ | $0.73_{\pm0.02}$ | $0.83_{\pm0.08}$ | $0.01_{\pm0.00}$ | $1.00_{\pm0.00}$ | $0.19_{\pm0.02}$ | $0.60_{\pm0.05}$ | $0.21_{\pm0.03}$ |
| ARF | $0.68_{\pm0.01}$ | $0.70_{\pm0.03}$ | $0.94_{\pm0.03}$ | $0.08_{\pm0.01}$ | $0.97_{\pm0.01}$ | $0.13_{\pm0.01}$ | $0.88_{\pm0.04}$ | $0.36_{\pm0.02}$ |
| TabDDPM | $0.45_{\pm0.01}$ | $0.50_{\pm0.04}$ | $0.00_{\pm0.00}$ | $0.00_{\pm0.00}$ | $1.00_{\pm0.00}$ | $1.00_{\pm0.00}$ | $0.66_{\pm0.07}$ | $0.52_{\pm0.03}$ |
| CDTD | $-$ | $-$ | $-$ | $-$ | $-$ | $-$ | $-$ | $-$ |
| TabSyn | $0.60_{\pm0.01}$ | $0.85_{\pm0.02}$ | $0.92_{\pm0.04}$ | $0.16_{\pm0.04}$ | $0.93_{\pm0.02}$ | $0.10_{\pm0.01}$ | $0.92_{\pm0.04}$ | $0.48_{\pm0.05}$ |
| TabDiff | $0.60_{\pm0.02}$ | $0.65_{\pm0.02}$ | $0.14_{\pm0.03}$ | $0.00_{\pm0.00}$ | $1.00_{\pm0.00}$ | $0.53_{\pm0.03}$ | $0.60_{\pm0.07}$ | $0.59_{\pm0.04}$ |
| NRGBoost | $0.61_{\pm0.00}$ | $0.79_{\pm0.03}$ | $0.90_{\pm0.05}$ | $0.12_{\pm0.03}$ | $0.93_{\pm0.02}$ | $0.11_{\pm0.02}$ | $0.84_{\pm0.04}$ | $0.25_{\pm0.02}$ |
| TabNAT | $-$ | $-$ | $-$ | $-$ | $-$ | $-$ | $-$ | $-$ |
| TabularARGN | $0.81_{\pm0.00}$ | $0.70_{\pm0.01}$ | $0.91_{\pm0.05}$ | $0.01_{\pm0.01}$ | $1.00_{\pm0.00}$ | $0.17_{\pm0.01}$ | $0.70_{\pm0.07}$ | $0.24_{\pm0.02}$ |
| **Foundation Model** | | | | | | | | |
| GReaT | $-$ | $-$ | $-$ | $-$ | $-$ | $-$ | $-$ | $-$ |
| TabPFN | $-$ | $-$ | $-$ | $-$ | $-$ | $-$ | $-$ | $-$ |
| TabDPT | $0.55_{\pm0.01}$ | $0.54_{\pm0.01}$ | $0.53_{\pm0.01}$ | $0.02_{\pm0.01}$ | $0.75_{\pm0.00}$ | $0.11_{\pm0.01}$ | $0.58_{\pm0.03}$ | $0.21_{\pm0.02}$ |
| Mitra | $0.43_{\pm0.01}$ | $0.60_{\pm0.01}$ | $0.42_{\pm0.03}$ | $0.03_{\pm0.01}$ | $0.78_{\pm0.00}$ | $0.11_{\pm0.01}$ | $0.55_{\pm0.02}$ | $0.19_{\pm0.02}$ |
| LimiX | $0.53_{\pm0.01}$ | $0.56_{\pm0.01}$ | $0.56_{\pm0.03}$ | $0.03_{\pm0.00}$ | $0.95_{\pm0.00}$ | $0.13_{\pm0.01}$ | $0.57_{\pm0.03}$ | $0.25_{\pm0.02}$ |
| TabICL | $0.62_{\pm0.00}$ | $0.66_{\pm0.01}$ | $0.73_{\pm0.02}$ | $0.07_{\pm0.01}$ | $0.80_{\pm0.01}$ | $0.11_{\pm0.01}$ | $0.73_{\pm0.03}$ | $0.35_{\pm0.03}$ |
| TabEBM | $0.77_{\pm0.00}$ | $0.78_{\pm0.02}$ | $0.74_{\pm0.04}$ | $0.15_{\pm0.01}$ | $0.87_{\pm0.01}$ | $0.11_{\pm0.01}$ | $0.92_{\pm0.04}$ | $0.48_{\pm0.02}$ |
| CTSyn | $-$ | $-$ | $-$ | $-$ | $-$ | $-$ | $-$ | $-$ |
| **TabFORGE (Ours)** | $0.61_{\pm0.01}$ | $0.89_{\pm0.02}$ | $0.94_{\pm0.04}$ | $0.32_{\pm0.07}$ | $0.83_{\pm0.07}$ | $0.06_{\pm0.01}$ | $0.94_{\pm0.02}$ | $0.61_{\pm0.06}$ |

*Table 34.* **Raw benchmark results of 23 tabular generators on the "California" dataset.** We report the mean $\pm$ std of each metric across 10 repeated data splits. For benchmark generators, "$-$" denotes failed convergence of a specific model or unexpected values in the synthetic data that caused the evaluation metric computation to crash. We highlight the **First**, **Second**, and **Third** best performances for each metric. TabFORGE generally achieves competitive performance against the benchmark generators while maintaining a reduced risk of overfitting.

| Generator | Density Estimation | | | | Privacy Preservation | | ML Efficacy | Structural Fidelity |
|---|---|---|---|---|---|---|---|---|
| | Shape $\uparrow$ | Trend $\uparrow$ | $\alpha$-precision $\uparrow$ | $\beta$-recall $\uparrow$ | Authenticity $\uparrow$ | DCR Score $\uparrow$ | Local utility $\uparrow$ | Global utility $\uparrow$ |
| **Real Data** | | | | | | | | |
| Real Data (Train) | $1.00_{\pm 0.00}$ | $1.00_{\pm 0.00}$ | $1.00_{\pm 0.00}$ | $1.00_{\pm 0.00}$ | $0.00_{\pm 0.00}$ | $0.00_{\pm 0.00}$ | $1.00_{\pm 0.00}$ | $1.00_{\pm 0.00}$ |
| Real Data (Holdout) | $0.99_{\pm 0.00}$ | $0.98_{\pm 0.00}$ | $0.99_{\pm 0.00}$ | $0.49_{\pm 0.01}$ | $0.69_{\pm 0.01}$ | $0.04_{\pm 0.00}$ | $-$ | $-$ |
| **Dataset-specific Model** | | | | | | | | |
| SMOTE | $0.98_{\pm 0.00}$ | $0.73_{\pm 0.00}$ | $0.79_{\pm 0.01}$ | $0.66_{\pm 0.01}$ | $0.44_{\pm 0.01}$ | $0.05_{\pm 0.00}$ | $0.91_{\pm 0.01}$ | $0.64_{\pm 0.01}$ |
| TabSDS | $0.76_{\pm 0.00}$ | $0.61_{\pm 0.01}$ | $0.74_{\pm 0.01}$ | $0.17_{\pm 0.00}$ | $0.71_{\pm 0.00}$ | $0.06_{\pm 0.01}$ | $0.58_{\pm 0.01}$ | $0.52_{\pm 0.01}$ |
| TVAE | $0.92_{\pm 0.01}$ | $0.81_{\pm 0.02}$ | $0.96_{\pm 0.01}$ | $0.16_{\pm 0.01}$ | $0.92_{\pm 0.00}$ | $0.10_{\pm 0.01}$ | $0.68_{\pm 0.04}$ | $0.57_{\pm 0.01}$ |
| GOGGLE | $0.50_{\pm 0.03}$ | $0.46_{\pm 0.02}$ | $0.58_{\pm 0.07}$ | $0.01_{\pm 0.00}$ | $1.00_{\pm 0.00}$ | $0.12_{\pm 0.03}$ | $0.28_{\pm 0.01}$ | $0.06_{\pm 0.01}$ |
| CTGAN | $0.88_{\pm 0.02}$ | $0.84_{\pm 0.02}$ | $0.96_{\pm 0.01}$ | $0.13_{\pm 0.02}$ | $0.94_{\pm 0.01}$ | $0.10_{\pm 0.01}$ | $0.65_{\pm 0.03}$ | $0.37_{\pm 0.04}$ |
| NFlow | $0.88_{\pm 0.02}$ | $0.72_{\pm 0.03}$ | $0.88_{\pm 0.02}$ | $0.10_{\pm 0.02}$ | $0.96_{\pm 0.01}$ | $0.15_{\pm 0.02}$ | $0.52_{\pm 0.03}$ | $0.33_{\pm 0.04}$ |
| ARF | $0.97_{\pm 0.00}$ | $0.83_{\pm 0.02}$ | $0.98_{\pm 0.01}$ | $0.19_{\pm 0.01}$ | $0.91_{\pm 0.00}$ | $0.11_{\pm 0.01}$ | $0.79_{\pm 0.01}$ | $0.47_{\pm 0.01}$ |
| TabDDPM | $0.98_{\pm 0.00}$ | $0.93_{\pm 0.02}$ | $0.98_{\pm 0.01}$ | $0.32_{\pm 0.01}$ | $0.83_{\pm 0.00}$ | $0.08_{\pm 0.01}$ | $0.87_{\pm 0.01}$ | $0.60_{\pm 0.01}$ |
| CDTD | $0.81_{\pm 0.00}$ | $0.71_{\pm 0.00}$ | $0.86_{\pm 0.02}$ | $0.12_{\pm 0.00}$ | $0.71_{\pm 0.00}$ | $0.11_{\pm 0.01}$ | $0.53_{\pm 0.01}$ | $0.37_{\pm 0.01}$ |
| TabSyn | $0.96_{\pm 0.01}$ | $0.96_{\pm 0.01}$ | $0.95_{\pm 0.03}$ | $0.35_{\pm 0.01}$ | $0.81_{\pm 0.00}$ | $0.07_{\pm 0.00}$ | $0.87_{\pm 0.01}$ | $0.74_{\pm 0.03}$ |
| TabDiff | $0.92_{\pm 0.01}$ | $0.73_{\pm 0.02}$ | $0.86_{\pm 0.02}$ | $0.07_{\pm 0.00}$ | $0.97_{\pm 0.00}$ | $0.18_{\pm 0.02}$ | $0.61_{\pm 0.02}$ | $0.60_{\pm 0.01}$ |
| NRGBoost | $0.96_{\pm 0.00}$ | $0.94_{\pm 0.01}$ | $0.97_{\pm 0.01}$ | $0.11_{\pm 0.00}$ | $0.94_{\pm 0.00}$ | $0.07_{\pm 0.01}$ | $0.74_{\pm 0.01}$ | $0.33_{\pm 0.01}$ |
| TabNAT | $0.71_{\pm 0.01}$ | $0.66_{\pm 0.01}$ | $0.80_{\pm 0.01}$ | $0.12_{\pm 0.00}$ | $0.68_{\pm 0.00}$ | $0.14_{\pm 0.01}$ | $0.55_{\pm 0.01}$ | $0.26_{\pm 0.01}$ |
| TabularARGN | $0.83_{\pm 0.00}$ | $0.93_{\pm 0.01}$ | $0.98_{\pm 0.01}$ | $0.05_{\pm 0.00}$ | $0.97_{\pm 0.00}$ | $0.09_{\pm 0.01}$ | $0.61_{\pm 0.01}$ | $0.40_{\pm 0.02}$ |
| **Foundation Model** | | | | | | | | |
| GReaT | $0.83_{\pm 0.00}$ | $0.49_{\pm 0.01}$ | $0.92_{\pm 0.01}$ | $0.02_{\pm 0.00}$ | $0.99_{\pm 0.00}$ | $0.21_{\pm 0.02}$ | $0.45_{\pm 0.01}$ | $0.03_{\pm 0.01}$ |
| TabPFN | $0.81_{\pm 0.01}$ | $0.62_{\pm 0.01}$ | $0.80_{\pm 0.01}$ | $0.09_{\pm 0.01}$ | $0.77_{\pm 0.00}$ | $0.12_{\pm 0.01}$ | $0.53_{\pm 0.01}$ | $0.31_{\pm 0.02}$ |
| TabDPT | $0.62_{\pm 0.01}$ | $0.58_{\pm 0.01}$ | $0.70_{\pm 0.02}$ | $0.07_{\pm 0.01}$ | $0.81_{\pm 0.01}$ | $0.09_{\pm 0.01}$ | $0.53_{\pm 0.01}$ | $0.25_{\pm 0.02}$ |
| Mitra | $0.54_{\pm 0.02}$ | $0.50_{\pm 0.01}$ | $0.50_{\pm 0.04}$ | $0.06_{\pm 0.01}$ | $0.85_{\pm 0.00}$ | $0.11_{\pm 0.02}$ | $0.50_{\pm 0.01}$ | $0.17_{\pm 0.02}$ |
| LimiX | $0.78_{\pm 0.01}$ | $0.61_{\pm 0.01}$ | $0.75_{\pm 0.02}$ | $0.09_{\pm 0.01}$ | $0.72_{\pm 0.00}$ | $0.11_{\pm 0.01}$ | $0.57_{\pm 0.01}$ | $0.30_{\pm 0.02}$ |
| TabICL | $0.77_{\pm 0.00}$ | $0.72_{\pm 0.01}$ | $0.80_{\pm 0.01}$ | $0.18_{\pm 0.01}$ | $0.76_{\pm 0.00}$ | $0.08_{\pm 0.01}$ | $0.65_{\pm 0.01}$ | $0.48_{\pm 0.02}$ |
| TabEBM | $0.92_{\pm 0.00}$ | $0.86_{\pm 0.01}$ | $0.84_{\pm 0.04}$ | $0.02_{\pm 0.00}$ | $0.99_{\pm 0.00}$ | $0.15_{\pm 0.01}$ | $0.63_{\pm 0.01}$ | $0.40_{\pm 0.01}$ |
| CTSyn | $0.72_{\pm 0.00}$ | $0.56_{\pm 0.00}$ | $0.73_{\pm 0.02}$ | $0.12_{\pm 0.00}$ | $0.83_{\pm 0.00}$ | $0.10_{\pm 0.01}$ | $0.50_{\pm 0.00}$ | $0.33_{\pm 0.01}$ |
| **TabFORGE (Ours)** | $0.96_{\pm 0.01}$ | $0.97_{\pm 0.00}$ | $0.97_{\pm 0.02}$ | $0.37_{\pm 0.03}$ | $0.73_{\pm 0.02}$ | $0.06_{\pm 0.01}$ | $0.84_{\pm 0.02}$ | $0.74_{\pm 0.03}$ |

*Table 35.* **Raw benchmark results of 23 tabular generators on the "Card" dataset.** We report the mean $\pm$ std of each metric across 10 repeated data splits. For benchmark generators, "$-$" denotes failed convergence of a specific model or unexpected values in the synthetic data that caused the evaluation metric computation to crash. We highlight the **First**, **Second**, and **Third** best performances for each metric. TabFORGE generally achieves competitive performance against the benchmark generators while maintaining a reduced risk of overfitting.

| Generator | Density Estimation | | | | Privacy Preservation | | ML Efficacy | Structural Fidelity |
|---|---|---|---|---|---|---|---|---|
| | Shape ↑ | Trend ↑ | $\alpha$-precision ↑ | $\beta$-recall ↑ | Authenticity ↑ | DCR Score ↑ | Local utility ↑ | Global utility ↑ |
| **Real Data** | | | | | | | | |
| Real Data (Train) | $1.00_{\pm 0.00}$ | $1.00_{\pm 0.00}$ | $1.00_{\pm 0.00}$ | $1.00_{\pm 0.00}$ | $0.00_{\pm 0.00}$ | $0.00_{\pm 0.00}$ | $1.00_{\pm 0.00}$ | $0.96_{\pm 0.03}$ |
| Real Data (Holdout) | $0.99_{\pm 0.00}$ | $0.99_{\pm 0.00}$ | $0.99_{\pm 0.00}$ | $0.50_{\pm 0.00}$ | $0.73_{\pm 0.00}$ | $0.02_{\pm 0.00}$ | $-$ | $-$ |
| **Dataset-specific Model** | | | | | | | | |
| SMOTE | $0.80_{\pm 0.01}$ | $-$ | $-$ | $0.65_{\pm 0.00}$ | $0.49_{\pm 0.00}$ | $0.02_{\pm 0.00}$ | $0.99_{\pm 0.01}$ | $-$ |
| TabSDS | $0.59_{\pm 0.00}$ | $0.69_{\pm 0.00}$ | $0.72_{\pm 0.01}$ | $0.17_{\pm 0.01}$ | $0.73_{\pm 0.00}$ | $0.04_{\pm 0.00}$ | $0.77_{\pm 0.01}$ | $0.48_{\pm 0.02}$ |
| TVAE | $0.80_{\pm 0.01}$ | $0.85_{\pm 0.02}$ | $0.96_{\pm 0.01}$ | $0.25_{\pm 0.01}$ | $0.89_{\pm 0.00}$ | $0.05_{\pm 0.00}$ | $0.97_{\pm 0.02}$ | $0.57_{\pm 0.02}$ |
| GOGGLE | $0.30_{\pm 0.11}$ | $0.58_{\pm 0.01}$ | $0.25_{\pm 0.06}$ | $0.01_{\pm 0.01}$ | $1.00_{\pm 0.00}$ | $0.02_{\pm 0.01}$ | $0.67_{\pm 0.22}$ | $0.16_{\pm 0.03}$ |
| CTGAN | $0.76_{\pm 0.01}$ | $0.88_{\pm 0.01}$ | $0.92_{\pm 0.08}$ | $0.22_{\pm 0.02}$ | $0.91_{\pm 0.01}$ | $0.05_{\pm 0.01}$ | $0.94_{\pm 0.05}$ | $0.41_{\pm 0.04}$ |
| NFlow | $0.78_{\pm 0.02}$ | $0.71_{\pm 0.03}$ | $0.78_{\pm 0.11}$ | $0.08_{\pm 0.02}$ | $0.97_{\pm 0.01}$ | $0.10_{\pm 0.01}$ | $0.74_{\pm 0.04}$ | $0.30_{\pm 0.06}$ |
| ARF | $0.83_{\pm 0.01}$ | $0.91_{\pm 0.02}$ | $0.98_{\pm 0.00}$ | $0.18_{\pm 0.02}$ | $0.93_{\pm 0.01}$ | $0.06_{\pm 0.00}$ | $0.94_{\pm 0.02}$ | $0.56_{\pm 0.02}$ |
| TabDDPM | $0.60_{\pm 0.07}$ | $0.80_{\pm 0.04}$ | $0.34_{\pm 0.26}$ | $0.10_{\pm 0.08}$ | $0.96_{\pm 0.03}$ | $0.51_{\pm 0.37}$ | $0.85_{\pm 0.04}$ | $0.60_{\pm 0.02}$ |
| CDTD | $-$ | $-$ | $-$ | $-$ | $-$ | $-$ | $-$ | $-$ |
| TabSyn | $0.78_{\pm 0.01}$ | $0.97_{\pm 0.02}$ | $0.95_{\pm 0.03}$ | $0.36_{\pm 0.02}$ | $0.82_{\pm 0.01}$ | $0.04_{\pm 0.00}$ | $0.96_{\pm 0.02}$ | $0.65_{\pm 0.03}$ |
| TabDiff | $0.72_{\pm 0.02}$ | $0.68_{\pm 0.02}$ | $0.23_{\pm 0.07}$ | $0.01_{\pm 0.01}$ | $1.00_{\pm 0.00}$ | $0.24_{\pm 0.06}$ | $0.77_{\pm 0.03}$ | $0.69_{\pm 0.02}$ |
| NRGBoost | $0.79_{\pm 0.00}$ | $0.94_{\pm 0.00}$ | $0.95_{\pm 0.01}$ | $0.03_{\pm 0.00}$ | $0.99_{\pm 0.00}$ | $0.08_{\pm 0.01}$ | $0.89_{\pm 0.03}$ | $-$ |
| TabNAT | $0.61_{\pm 0.00}$ | $0.62_{\pm 0.01}$ | $0.43_{\pm 0.02}$ | $0.10_{\pm 0.01}$ | $0.78_{\pm 0.00}$ | $0.10_{\pm 0.03}$ | $0.66_{\pm 0.01}$ | $0.28_{\pm 0.01}$ |
| TabularARGN | $0.82_{\pm 0.01}$ | $0.96_{\pm 0.01}$ | $0.98_{\pm 0.01}$ | $0.06_{\pm 0.00}$ | $0.97_{\pm 0.00}$ | $0.04_{\pm 0.00}$ | $0.95_{\pm 0.02}$ | $0.46_{\pm 0.04}$ |
| **Foundation Model** | | | | | | | | |
| GReaT | $0.64_{\pm 0.00}$ | $0.65_{\pm 0.03}$ | $0.58_{\pm 0.03}$ | $0.07_{\pm 0.00}$ | $0.98_{\pm 0.00}$ | $0.05_{\pm 0.00}$ | $0.78_{\pm 0.01}$ | $0.20_{\pm 0.03}$ |
| TabPFN | $0.73_{\pm 0.01}$ | $0.64_{\pm 0.02}$ | $0.68_{\pm 0.06}$ | $0.10_{\pm 0.01}$ | $0.84_{\pm 0.00}$ | $0.06_{\pm 0.01}$ | $0.67_{\pm 0.03}$ | $0.35_{\pm 0.03}$ |
| TabDPT | $0.61_{\pm 0.03}$ | $0.70_{\pm 0.01}$ | $0.66_{\pm 0.04}$ | $0.10_{\pm 0.01}$ | $0.71_{\pm 0.00}$ | $0.04_{\pm 0.00}$ | $0.86_{\pm 0.09}$ | $0.31_{\pm 0.03}$ |
| Mitra | $0.39_{\pm 0.05}$ | $0.54_{\pm 0.01}$ | $0.44_{\pm 0.04}$ | $0.06_{\pm 0.01}$ | $0.80_{\pm 0.00}$ | $0.05_{\pm 0.01}$ | $0.92_{\pm 0.13}$ | $0.26_{\pm 0.03}$ |
| LimiX | $0.56_{\pm 0.01}$ | $0.61_{\pm 0.01}$ | $0.65_{\pm 0.05}$ | $0.07_{\pm 0.01}$ | $0.89_{\pm 0.00}$ | $0.06_{\pm 0.01}$ | $0.75_{\pm 0.06}$ | $0.36_{\pm 0.04}$ |
| TabICL | $0.69_{\pm 0.01}$ | $0.72_{\pm 0.01}$ | $0.76_{\pm 0.03}$ | $0.17_{\pm 0.01}$ | $0.74_{\pm 0.00}$ | $0.06_{\pm 0.00}$ | $0.77_{\pm 0.02}$ | $0.51_{\pm 0.02}$ |
| TabEBM | $-$ | $-$ | $-$ | $-$ | $-$ | $-$ | $-$ | $-$ |
| CTSyn | $0.57_{\pm 0.02}$ | $0.59_{\pm 0.01}$ | $0.58_{\pm 0.03}$ | $0.13_{\pm 0.01}$ | $0.81_{\pm 0.00}$ | $0.03_{\pm 0.00}$ | $0.76_{\pm 0.05}$ | $0.35_{\pm 0.02}$ |
| **TabFORGE (Ours)** | $0.74_{\pm 0.01}$ | $0.93_{\pm 0.02}$ | $0.92_{\pm 0.04}$ | $0.26_{\pm 0.01}$ | $0.88_{\pm 0.01}$ | $0.04_{\pm 0.01}$ | $1.01_{\pm 0.02}$ | $0.65_{\pm 0.03}$ |

*Table 36.* **Raw benchmark results of 23 tabular generators on the "Churn" dataset.** We report the mean $\pm$ std of each metric across 10 repeated data splits. For benchmark generators, "$-$" denotes failed convergence of a specific model or unexpected values in the synthetic data that caused the evaluation metric computation to crash. We highlight the **First**, **Second**, and **Third** best performances for each metric. TabFORGE generally achieves competitive performance against the benchmark generators while maintaining a reduced risk of overfitting.

| Generator | Density Estimation | | | | Privacy Preservation | | ML Efficacy | Structural Fidelity |
|---|---|---|---|---|---|---|---|---|
| | Shape $\uparrow$ | Trend $\uparrow$ | $\alpha$-precision $\uparrow$ | $\beta$-recall $\uparrow$ | Authenticity $\uparrow$ | DCR Score $\uparrow$ | Local utility $\uparrow$ | Global utility $\uparrow$ |
| **Real Data** | | | | | | | | |
| Real Data (Train) | $1.00_{\pm0.00}$ | $1.00_{\pm0.00}$ | $1.00_{\pm0.00}$ | $1.00_{\pm0.00}$ | $0.00_{\pm0.00}$ | $0.00_{\pm0.00}$ | $1.00_{\pm0.00}$ | $0.84_{\pm0.03}$ |
| Real Data (Holdout) | $0.98_{\pm0.00}$ | $0.97_{\pm0.01}$ | $0.98_{\pm0.01}$ | $0.50_{\pm0.01}$ | $0.68_{\pm0.01}$ | $0.23_{\pm0.01}$ | $-$ | $-$ |
| **Dataset-specific Model** | | | | | | | | |
| SMOTE | $\mathbf{0.91}_{\pm0.00}$ | $0.79_{\pm0.01}$ | $0.72_{\pm0.01}$ | $\mathbf{0.66}_{\pm0.02}$ | $0.45_{\pm0.01}$ | $0.06_{\pm0.00}$ | $\mathbf{0.95}_{\pm0.02}$ | $0.49_{\pm0.02}$ |
| TabSDS | $-$ | $-$ | $-$ | $-$ | $-$ | $-$ | $-$ | $-$ |
| TVAE | $0.86_{\pm0.01}$ | $0.77_{\pm0.02}$ | $0.79_{\pm0.03}$ | $0.34_{\pm0.01}$ | $0.80_{\pm0.01}$ | $0.36_{\pm0.01}$ | $0.76_{\pm0.03}$ | $0.54_{\pm0.02}$ |
| GOGGLE | $0.55_{\pm0.02}$ | $0.43_{\pm0.03}$ | $0.08_{\pm0.01}$ | $0.04_{\pm0.01}$ | $\mathbf{0.98}_{\pm0.00}$ | $0.26_{\pm0.02}$ | $0.42_{\pm0.01}$ | $0.25_{\pm0.02}$ |
| CTGAN | $0.78_{\pm0.03}$ | $0.84_{\pm0.03}$ | $0.90_{\pm0.06}$ | $0.25_{\pm0.04}$ | $0.85_{\pm0.02}$ | $0.42_{\pm0.02}$ | $0.76_{\pm0.05}$ | $0.36_{\pm0.03}$ |
| NFlow | $0.85_{\pm0.02}$ | $0.70_{\pm0.01}$ | $0.90_{\pm0.06}$ | $0.28_{\pm0.03}$ | $0.84_{\pm0.02}$ | $0.44_{\pm0.02}$ | $0.65_{\pm0.02}$ | $0.38_{\pm0.05}$ |
| ARF | $-$ | $-$ | $-$ | $-$ | $-$ | $-$ | $-$ | $-$ |
| TabDDPM | $0.85_{\pm0.02}$ | $0.87_{\pm0.03}$ | $0.84_{\pm0.05}$ | $0.32_{\pm0.02}$ | $0.82_{\pm0.01}$ | $0.44_{\pm0.02}$ | $0.77_{\pm0.02}$ | $0.57_{\pm0.02}$ |
| CDTD | $0.70_{\pm0.00}$ | $0.61_{\pm0.01}$ | $0.78_{\pm0.01}$ | $0.24_{\pm0.01}$ | $0.66_{\pm0.01}$ | $0.33_{\pm0.01}$ | $0.71_{\pm0.01}$ | $0.44_{\pm0.01}$ |
| TabSyn | $\mathbf{0.90}_{\pm0.01}$ | $\mathbf{0.93}_{\pm0.02}$ | $\mathbf{0.95}_{\pm0.03}$ | $\mathbf{0.43}_{\pm0.02}$ | $0.73_{\pm0.02}$ | $0.37_{\pm0.01}$ | $\mathbf{0.89}_{\pm0.04}$ | $\mathbf{0.63}_{\pm0.03}$ |
| TabDiff | $0.89_{\pm0.00}$ | $0.73_{\pm0.03}$ | $0.83_{\pm0.06}$ | $0.21_{\pm0.01}$ | $\mathbf{0.88}_{\pm0.01}$ | $\mathbf{0.50}_{\pm0.02}$ | $0.73_{\pm0.02}$ | $\mathbf{0.59}_{\pm0.02}$ |
| NRGBoost | $0.88_{\pm0.01}$ | $\mathbf{0.90}_{\pm0.01}$ | $0.80_{\pm0.04}$ | $0.23_{\pm0.01}$ | $0.86_{\pm0.01}$ | $\mathbf{0.46}_{\pm0.01}$ | $\mathbf{0.91}_{\pm0.03}$ | $0.26_{\pm0.01}$ |
| TabNAT | $0.74_{\pm0.00}$ | $0.65_{\pm0.02}$ | $0.68_{\pm0.03}$ | $0.19_{\pm0.01}$ | $0.73_{\pm0.01}$ | $0.38_{\pm0.01}$ | $0.70_{\pm0.01}$ | $0.36_{\pm0.01}$ |
| TabularARGN | $0.84_{\pm0.00}$ | $0.88_{\pm0.01}$ | $\mathbf{0.97}_{\pm0.01}$ | $0.28_{\pm0.01}$ | $0.84_{\pm0.01}$ | $0.40_{\pm0.01}$ | $0.75_{\pm0.02}$ | $0.44_{\pm0.03}$ |
| **Foundation Model** | | | | | | | | |
| GReaT | $0.72_{\pm0.01}$ | $0.48_{\pm0.03}$ | $0.83_{\pm0.02}$ | $0.09_{\pm0.01}$ | $\mathbf{0.97}_{\pm0.01}$ | $\mathbf{0.47}_{\pm0.01}$ | $0.74_{\pm0.02}$ | $0.24_{\pm0.04}$ |
| TabPFN | $-$ | $-$ | $-$ | $-$ | $-$ | $-$ | $-$ | $-$ |
| TabDPT | $0.59_{\pm0.02}$ | $0.61_{\pm0.01}$ | $0.52_{\pm0.02}$ | $0.18_{\pm0.02}$ | $0.78_{\pm0.01}$ | $0.30_{\pm0.01}$ | $0.64_{\pm0.02}$ | $0.35_{\pm0.02}$ |
| Mitra | $0.62_{\pm0.01}$ | $0.45_{\pm0.01}$ | $0.39_{\pm0.02}$ | $0.15_{\pm0.01}$ | $0.80_{\pm0.01}$ | $0.30_{\pm0.01}$ | $0.73_{\pm0.01}$ | $0.39_{\pm0.03}$ |
| LimiX | $-$ | $-$ | $-$ | $-$ | $-$ | $-$ | $-$ | $-$ |
| TabICL | $-$ | $-$ | $-$ | $-$ | $-$ | $-$ | $-$ | $-$ |
| TabEBM | $\mathbf{0.93}_{\pm0.00}$ | $0.81_{\pm0.01}$ | $0.87_{\pm0.02}$ | $0.31_{\pm0.01}$ | $0.74_{\pm0.01}$ | $0.29_{\pm0.03}$ | $0.88_{\pm0.02}$ | $0.41_{\pm0.03}$ |
| CTSyn | $0.71_{\pm0.01}$ | $0.55_{\pm0.01}$ | $0.69_{\pm0.01}$ | $0.20_{\pm0.01}$ | $0.78_{\pm0.01}$ | $0.36_{\pm0.01}$ | $0.72_{\pm0.02}$ | $0.42_{\pm0.01}$ |
| **TabFORGE (Ours)** | $0.87_{\pm0.01}$ | $\mathbf{0.90}_{\pm0.01}$ | $\mathbf{0.91}_{\pm0.03}$ | $\mathbf{0.40}_{\pm0.02}$ | $0.75_{\pm0.01}$ | $0.39_{\pm0.02}$ | $0.77_{\pm0.02}$ | $\mathbf{0.63}_{\pm0.03}$ |

*Table 37.* **Raw benchmark results of 23 tabular generators on the "Coil2000" dataset.** We report the mean $\pm$ std of each metric across 10 repeated data splits. For benchmark generators, "−" denotes failed convergence of a specific model or unexpected values in the synthetic data that caused the evaluation metric computation to crash. We highlight the **First**, **Second**, and **Third** best performances for each metric. TabFORGE generally achieves competitive performance against the benchmark generators while maintaining a reduced risk of overfitting.

| Generator | Density Estimation | | | | Privacy Preservation | | ML Efficacy | Structural Fidelity |
| | Shape ↑ | Trend ↑ | $\alpha$-precision ↑ | $\beta$-recall ↑ | Authenticity ↑ | DCR Score ↑ | Local utility ↑ | Global utility ↑ |
|---|---|---|---|---|---|---|---|---|
| **Real Data** | | | | | | | | |
| Real Data (Train) | $1.00_{\pm0.00}$ | $1.00_{\pm0.00}$ | $1.00_{\pm0.00}$ | $1.00_{\pm0.00}$ | $0.01_{\pm0.00}$ | $0.00_{\pm0.00}$ | $1.00_{\pm0.00}$ | $0.95_{\pm0.01}$ |
| Real Data (Holdout) | $0.99_{\pm0.00}$ | $0.97_{\pm0.00}$ | $0.99_{\pm0.01}$ | $0.54_{\pm0.01}$ | $0.65_{\pm0.01}$ | $0.03_{\pm0.00}$ | − | − |
| **Dataset-specific Model** | | | | | | | | |
| SMOTE | $0.47_{\pm0.00}$ | $0.71_{\pm0.01}$ | $0.71_{\pm0.03}$ | $\mathbf{0.52}_{\pm0.01}$ | $0.60_{\pm0.01}$ | $0.02_{\pm0.00}$ | $\mathbf{1.01}_{\pm0.02}$ | $\mathbf{0.55}_{\pm0.01}$ |
| TabSDS | − | − | − | − | − | − | − | − |
| TVAE | $0.45_{\pm0.00}$ | $0.71_{\pm0.01}$ | $\mathbf{0.93}_{\pm0.03}$ | $0.02_{\pm0.01}$ | $0.99_{\pm0.00}$ | $0.19_{\pm0.01}$ | $0.98_{\pm0.01}$ | $0.48_{\pm0.01}$ |
| GOGGLE | $0.34_{\pm0.01}$ | $0.54_{\pm0.01}$ | $0.14_{\pm0.02}$ | $0.00_{\pm0.00}$ | $1.00_{\pm0.00}$ | $0.15_{\pm0.00}$ | − | − |
| CTGAN | $0.45_{\pm0.01}$ | $0.73_{\pm0.01}$ | $0.87_{\pm0.10}$ | $0.02_{\pm0.00}$ | $1.00_{\pm0.00}$ | $0.19_{\pm0.01}$ | $0.98_{\pm0.01}$ | $0.32_{\pm0.01}$ |
| NFlow | $0.42_{\pm0.01}$ | $0.61_{\pm0.01}$ | $0.39_{\pm0.09}$ | $0.00_{\pm0.00}$ | $1.00_{\pm0.00}$ | $0.29_{\pm0.01}$ | $0.87_{\pm0.01}$ | $0.20_{\pm0.01}$ |
| ARF | − | − | − | − | − | − | − | − |
| TabDDPM | $0.44_{\pm0.01}$ | $0.54_{\pm0.01}$ | $0.00_{\pm0.00}$ | $0.00_{\pm0.00}$ | $\mathbf{1.00}_{\pm0.00}$ | $\mathbf{0.94}_{\pm0.01}$ | $0.97_{\pm0.03}$ | $0.49_{\pm0.01}$ |
| CDTD | − | − | − | − | − | − | − | − |
| TabSyn | $0.43_{\pm0.00}$ | $0.78_{\pm0.02}$ | $0.77_{\pm0.12}$ | $0.01_{\pm0.01}$ | $1.00_{\pm0.00}$ | $0.21_{\pm0.02}$ | $0.98_{\pm0.01}$ | $0.42_{\pm0.04}$ |
| TabDiff | $0.46_{\pm0.01}$ | $0.60_{\pm0.00}$ | $0.00_{\pm0.00}$ | $0.00_{\pm0.00}$ | $\mathbf{1.00}_{\pm0.00}$ | $\mathbf{0.82}_{\pm0.01}$ | $0.97_{\pm0.02}$ | $\mathbf{0.53}_{\pm0.01}$ |
| NRGBoost | $0.40_{\pm0.00}$ | $0.79_{\pm0.01}$ | $0.06_{\pm0.01}$ | $0.00_{\pm0.00}$ | $\mathbf{1.00}_{\pm0.00}$ | $\mathbf{0.52}_{\pm0.02}$ | $0.99_{\pm0.02}$ | $0.13_{\pm0.00}$ |
| TabNAT | − | − | − | − | − | − | − | − |
| TabularARGN | $\mathbf{0.82}_{\pm0.00}$ | $\mathbf{0.89}_{\pm0.01}$ | $\mathbf{0.89}_{\pm0.03}$ | $0.07_{\pm0.01}$ | $0.98_{\pm0.00}$ | $0.15_{\pm0.00}$ | $0.98_{\pm0.01}$ | $0.46_{\pm0.01}$ |
| **Foundation Model** | | | | | | | | |
| GReaT | − | − | − | − | − | − | − | − |
| TabPFN | − | − | − | − | − | − | − | − |
| TabDPT | $\mathbf{0.48}_{\pm0.00}$ | $0.60_{\pm0.01}$ | $0.60_{\pm0.05}$ | $0.02_{\pm0.00}$ | $0.78_{\pm0.00}$ | $0.13_{\pm0.01}$ | $0.92_{\pm0.01}$ | $0.28_{\pm0.01}$ |
| Mitra | $0.31_{\pm0.01}$ | $0.51_{\pm0.01}$ | $0.26_{\pm0.03}$ | $0.01_{\pm0.00}$ | $0.77_{\pm0.00}$ | $0.17_{\pm0.00}$ | $0.87_{\pm0.01}$ | $0.23_{\pm0.00}$ |
| LimiX | − | − | − | − | − | − | − | − |
| TabICL | − | − | − | − | − | − | − | − |
| TabEBM | $\mathbf{0.68}_{\pm0.00}$ | $\mathbf{0.85}_{\pm0.01}$ | $0.72_{\pm0.02}$ | $\mathbf{0.07}_{\pm0.00}$ | $0.94_{\pm0.00}$ | $0.14_{\pm0.00}$ | $\mathbf{1.00}_{\pm0.02}$ | $0.45_{\pm0.01}$ |
| CTSyn | − | − | − | − | − | − | − | − |
| **TabFORGE (Ours)** | $0.46_{\pm0.00}$ | $\mathbf{0.93}_{\pm0.01}$ | $\mathbf{0.98}_{\pm0.01}$ | $\mathbf{0.35}_{\pm0.03}$ | $0.73_{\pm0.02}$ | $0.02_{\pm0.01}$ | $\mathbf{1.01}_{\pm0.01}$ | $\mathbf{0.69}_{\pm0.02}$ |

*Table 38.* **Raw benchmark results of 23 tabular generators on the "Company" dataset.** We report the mean $\pm$ std of each metric across 10 repeated data splits. For benchmark generators, "−" denotes failed convergence of a specific model or unexpected values in the synthetic data that caused the evaluation metric computation to crash. We highlight the **First**, **Second**, and **Third** best performances for each metric. TabFORGE generally achieves competitive performance against the benchmark generators while maintaining a reduced risk of overfitting.

| Generator | Density Estimation | | | | Privacy Preservation | | ML Efficacy | Structural Fidelity |
|---|---|---|---|---|---|---|---|---|
| | Shape ↑ | Trend ↑ | $\alpha$-precision ↑ | $\beta$-recall ↑ | Authenticity ↑ | DCR Score ↑ | Local utility ↑ | Global utility ↑ |
| **Real Data** | | | | | | | | |
| Real Data (Train) | $1.00_{\pm0.00}$ | $1.00_{\pm0.00}$ | $1.00_{\pm0.00}$ | $1.00_{\pm0.00}$ | $0.00_{\pm0.00}$ | $0.00_{\pm0.00}$ | $1.00_{\pm0.00}$ | $0.88_{\pm0.03}$ |
| Real Data (Holdout) | $0.98_{\pm0.00}$ | $0.71_{\pm0.02}$ | $0.98_{\pm0.01}$ | $0.50_{\pm0.02}$ | $0.75_{\pm0.02}$ | $0.00_{\pm0.00}$ | — | — |
| **Dataset-specific Model** | | | | | | | | |
| SMOTE | $0.94_{\pm0.01}$ | $0.65_{\pm0.04}$ | $0.79_{\pm0.01}$ | $0.68_{\pm0.02}$ | $0.47_{\pm0.02}$ | $0.00_{\pm0.00}$ | $0.99_{\pm0.03}$ | $0.66_{\pm0.03}$ |
| TabSDS | $0.68_{\pm0.00}$ | $0.46_{\pm0.02}$ | $0.57_{\pm0.01}$ | $0.14_{\pm0.01}$ | $0.67_{\pm0.01}$ | $0.00_{\pm0.00}$ | $0.71_{\pm0.04}$ | $0.46_{\pm0.02}$ |
| TVAE | $0.82_{\pm0.01}$ | $0.60_{\pm0.02}$ | $0.74_{\pm0.03}$ | $0.21_{\pm0.04}$ | $0.91_{\pm0.02}$ | $0.00_{\pm0.00}$ | $0.94_{\pm0.06}$ | $0.63_{\pm0.03}$ |
| GOGGLE | $0.44_{\pm0.00}$ | $0.58_{\pm0.03}$ | $0.24_{\pm0.22}$ | $0.01_{\pm0.00}$ | $1.00_{\pm0.00}$ | $0.00_{\pm0.00}$ | — | — |
| CTGAN | $0.67_{\pm0.04}$ | $0.58_{\pm0.02}$ | $0.86_{\pm0.11}$ | $0.07_{\pm0.04}$ | $0.97_{\pm0.02}$ | $0.01_{\pm0.00}$ | $0.91_{\pm0.05}$ | $0.61_{\pm0.07}$ |
| NFlow | $0.89_{\pm0.01}$ | $0.57_{\pm0.02}$ | $0.57_{\pm0.14}$ | $0.02_{\pm0.01}$ | $0.99_{\pm0.00}$ | $0.01_{\pm0.01}$ | $0.71_{\pm0.03}$ | $0.57_{\pm0.10}$ |
| ARF | $0.90_{\pm0.00}$ | $0.62_{\pm0.02}$ | $0.77_{\pm0.03}$ | $0.06_{\pm0.02}$ | $0.98_{\pm0.01}$ | $0.01_{\pm0.00}$ | $0.91_{\pm0.07}$ | $0.54_{\pm0.02}$ |
| TabDDPM | $0.40_{\pm0.01}$ | $0.72_{\pm0.05}$ | $0.00_{\pm0.00}$ | $0.00_{\pm0.00}$ | $1.00_{\pm0.00}$ | $0.98_{\pm0.04}$ | $0.84_{\pm0.03}$ | $0.65_{\pm0.03}$ |
| CDTD | — | — | — | — | — | — | — | — |
| TabSyn | $0.92_{\pm0.01}$ | $0.69_{\pm0.05}$ | $0.87_{\pm0.02}$ | $0.41_{\pm0.03}$ | $0.81_{\pm0.02}$ | $0.00_{\pm0.00}$ | $0.97_{\pm0.05}$ | $0.66_{\pm0.05}$ |
| TabDiff | $0.92_{\pm0.00}$ | $0.57_{\pm0.02}$ | $0.59_{\pm0.02}$ | $0.04_{\pm0.02}$ | $0.97_{\pm0.01}$ | $0.01_{\pm0.00}$ | $0.87_{\pm0.02}$ | $0.76_{\pm0.04}$ |
| NRGBoost | $0.85_{\pm0.02}$ | $0.58_{\pm0.02}$ | $0.80_{\pm0.03}$ | $0.13_{\pm0.02}$ | $0.94_{\pm0.02}$ | $0.00_{\pm0.00}$ | $0.85_{\pm0.03}$ | $0.32_{\pm0.04}$ |
| TabNAT | — | — | — | — | — | — | — | — |
| TabularARGN | $0.83_{\pm0.00}$ | $0.58_{\pm0.02}$ | $0.92_{\pm0.03}$ | $0.04_{\pm0.02}$ | $0.99_{\pm0.01}$ | $0.01_{\pm0.00}$ | $0.90_{\pm0.07}$ | $0.80_{\pm0.03}$ |
| **Foundation Model** | | | | | | | | |
| GReaT | — | — | — | — | — | — | — | — |
| TabPFN | — | — | — | — | — | — | — | — |
| TabDPT | $0.56_{\pm0.02}$ | $0.48_{\pm0.01}$ | $0.59_{\pm0.06}$ | $0.03_{\pm0.01}$ | $0.80_{\pm0.00}$ | $0.00_{\pm0.00}$ | $0.99_{\pm0.02}$ | $0.60_{\pm0.05}$ |
| Mitra | $0.51_{\pm0.00}$ | $0.53_{\pm0.02}$ | $0.39_{\pm0.02}$ | $0.03_{\pm0.00}$ | $0.90_{\pm0.00}$ | $0.00_{\pm0.00}$ | $1.00_{\pm0.00}$ | $0.58_{\pm0.07}$ |
| LimiX | $0.65_{\pm0.00}$ | $0.53_{\pm0.00}$ | $0.58_{\pm0.07}$ | $0.02_{\pm0.01}$ | $0.84_{\pm0.01}$ | $0.01_{\pm0.01}$ | $0.87_{\pm0.00}$ | $0.58_{\pm0.08}$ |
| TabICL | $0.70_{\pm0.00}$ | $0.57_{\pm0.02}$ | $0.66_{\pm0.04}$ | $0.14_{\pm0.01}$ | $0.69_{\pm0.01}$ | $0.01_{\pm0.00}$ | $0.80_{\pm0.04}$ | $0.58_{\pm0.04}$ |
| TabEBM | $0.63_{\pm0.02}$ | $0.75_{\pm0.10}$ | $0.64_{\pm0.05}$ | $0.01_{\pm0.00}$ | $0.99_{\pm0.00}$ | $0.02_{\pm0.00}$ | $0.89_{\pm0.03}$ | $0.51_{\pm0.05}$ |
| CTSyn | — | — | — | — | — | — | — | — |
| **TabFORGE (Ours)** | $0.94_{\pm0.00}$ | $0.77_{\pm0.09}$ | $0.95_{\pm0.02}$ | $0.45_{\pm0.02}$ | $0.71_{\pm0.03}$ | $0.00_{\pm0.00}$ | $0.97_{\pm0.04}$ | $0.68_{\pm0.09}$ |

*Table 39.* **Raw benchmark results of 23 tabular generators on the "Concrete" dataset.** We report the mean $\pm$ std of each metric across 10 repeated data splits. For benchmark generators, "$-$" denotes failed convergence of a specific model or unexpected values in the synthetic data that caused the evaluation metric computation to crash. We highlight the **First**, **Second**, and **Third** best performances for each metric. TabFORGE generally achieves competitive performance against the benchmark generators while maintaining a reduced risk of overfitting.

| Generator | Density Estimation | | | | Privacy Preservation | | ML Efficacy | Structural Fidelity |
|---|---|---|---|---|---|---|---|---|
| | Shape ↑ | Trend ↑ | $\alpha$-precision ↑ | $\beta$-recall ↑ | Authenticity ↑ | DCR Score ↑ | Local utility ↑ | Global utility ↑ |
| **Real Data** | | | | | | | | |
| Real Data (Train) | $1.00_{\pm 0.00}$ | $1.00_{\pm 0.00}$ | $1.00_{\pm 0.00}$ | $1.00_{\pm 0.00}$ | $0.00_{\pm 0.00}$ | $0.00_{\pm 0.00}$ | $1.00_{\pm 0.00}$ | $1.00_{\pm 0.00}$ |
| Real Data (Holdout) | $0.94_{\pm 0.01}$ | $0.98_{\pm 0.01}$ | $0.95_{\pm 0.02}$ | $0.51_{\pm 0.04}$ | $0.58_{\pm 0.03}$ | $0.07_{\pm 0.00}$ | $-$ | $-$ |
| **Dataset-specific Model** | | | | | | | | |
| SMOTE | $0.79_{\pm 0.01}$ | $-$ | $-$ | $0.50_{\pm 0.03}$ | $0.60_{\pm 0.03}$ | $0.05_{\pm 0.00}$ | $0.83_{\pm 0.03}$ | $-$ |
| TabSDS | $-$ | $-$ | $-$ | $-$ | $-$ | $-$ | $-$ | $-$ |
| TVAE | $0.76_{\pm 0.02}$ | $0.83_{\pm 0.03}$ | $0.73_{\pm 0.06}$ | $0.07_{\pm 0.01}$ | $0.98_{\pm 0.01}$ | $0.34_{\pm 0.02}$ | $0.57_{\pm 0.04}$ | $-$ |
| GOGGLE | $0.63_{\pm 0.02}$ | $0.69_{\pm 0.05}$ | $0.50_{\pm 0.07}$ | $0.04_{\pm 0.02}$ | $0.99_{\pm 0.00}$ | $0.38_{\pm 0.03}$ | $0.28_{\pm 0.01}$ | $0.17_{\pm 0.01}$ |
| CTGAN | $0.63_{\pm 0.09}$ | $0.79_{\pm 0.12}$ | $0.79_{\pm 0.13}$ | $0.04_{\pm 0.02}$ | $0.99_{\pm 0.01}$ | $0.40_{\pm 0.06}$ | $0.48_{\pm 0.05}$ | $0.33_{\pm 0.04}$ |
| NFlow | $0.79_{\pm 0.02}$ | $0.73_{\pm 0.03}$ | $0.88_{\pm 0.04}$ | $0.05_{\pm 0.01}$ | $0.99_{\pm 0.01}$ | $0.44_{\pm 0.03}$ | $0.41_{\pm 0.02}$ | $0.36_{\pm 0.02}$ |
| ARF | $0.83_{\pm 0.01}$ | $0.94_{\pm 0.03}$ | $0.91_{\pm 0.04}$ | $0.07_{\pm 0.01}$ | $0.98_{\pm 0.01}$ | $0.38_{\pm 0.02}$ | $0.64_{\pm 0.03}$ | $-$ |
| TabDDPM | $0.78_{\pm 0.02}$ | $0.97_{\pm 0.02}$ | $0.94_{\pm 0.02}$ | $0.16_{\pm 0.02}$ | $0.91_{\pm 0.01}$ | $0.23_{\pm 0.01}$ | $0.80_{\pm 0.04}$ | $-$ |
| CDTD | $-$ | $-$ | $-$ | $-$ | $-$ | $-$ | $-$ | $-$ |
| TabSyn | $-$ | $-$ | $-$ | $-$ | $-$ | $-$ | $-$ | $-$ |
| TabDiff | $-$ | $-$ | $-$ | $-$ | $-$ | $-$ | $-$ | $-$ |
| NRGBoost | $0.74_{\pm 0.01}$ | $0.93_{\pm 0.03}$ | $0.92_{\pm 0.02}$ | $0.29_{\pm 0.03}$ | $0.75_{\pm 0.03}$ | $0.04_{\pm 0.01}$ | $0.77_{\pm 0.05}$ | $-$ |
| TabNAT | $-$ | $-$ | $-$ | $-$ | $-$ | $-$ | $-$ | $-$ |
| TabularARGN | $0.76_{\pm 0.01}$ | $0.75_{\pm 0.04}$ | $0.90_{\pm 0.03}$ | $0.04_{\pm 0.01}$ | $0.99_{\pm 0.01}$ | $0.46_{\pm 0.02}$ | $0.47_{\pm 0.02}$ | $0.34_{\pm 0.03}$ |
| **Foundation Model** | | | | | | | | |
| GReaT | $0.39_{\pm 0.01}$ | $0.68_{\pm 0.02}$ | $0.00_{\pm 0.00}$ | $0.00_{\pm 0.00}$ | $1.00_{\pm 0.00}$ | $1.00_{\pm 0.00}$ | $0.45_{\pm 0.02}$ | $0.00_{\pm 0.00}$ |
| TabPFN | $0.59_{\pm 0.01}$ | $0.75_{\pm 0.02}$ | $0.64_{\pm 0.02}$ | $0.04_{\pm 0.01}$ | $0.87_{\pm 0.00}$ | $0.46_{\pm 0.01}$ | $0.46_{\pm 0.02}$ | $0.28_{\pm 0.01}$ |
| TabDPT | $0.54_{\pm 0.04}$ | $0.69_{\pm 0.06}$ | $0.60_{\pm 0.06}$ | $0.03_{\pm 0.01}$ | $0.89_{\pm 0.00}$ | $0.31_{\pm 0.03}$ | $0.46_{\pm 0.02}$ | $0.25_{\pm 0.02}$ |
| Mitra | $0.59_{\pm 0.01}$ | $0.66_{\pm 0.03}$ | $0.53_{\pm 0.04}$ | $0.04_{\pm 0.01}$ | $0.85_{\pm 0.01}$ | $0.34_{\pm 0.02}$ | $0.41_{\pm 0.02}$ | $0.31_{\pm 0.02}$ |
| LimiX | $0.66_{\pm 0.01}$ | $0.64_{\pm 0.02}$ | $0.75_{\pm 0.03}$ | $0.05_{\pm 0.01}$ | $0.81_{\pm 0.01}$ | $0.32_{\pm 0.01}$ | $0.48_{\pm 0.02}$ | $0.36_{\pm 0.02}$ |
| TabICL | $-$ | $-$ | $-$ | $-$ | $-$ | $-$ | $-$ | $-$ |
| TabEBM | $0.83_{\pm 0.01}$ | $0.94_{\pm 0.03}$ | $0.85_{\pm 0.03}$ | $0.06_{\pm 0.01}$ | $0.96_{\pm 0.01}$ | $0.34_{\pm 0.01}$ | $0.68_{\pm 0.04}$ | $-$ |
| CTSyn | $-$ | $-$ | $-$ | $-$ | $-$ | $-$ | $-$ | $-$ |
| **TabFORGE (Ours)** | $0.78_{\pm 0.02}$ | $0.98_{\pm 0.01}$ | $0.91_{\pm 0.04}$ | $0.27_{\pm 0.02}$ | $0.82_{\pm 0.02}$ | $0.10_{\pm 0.01}$ | $0.81_{\pm 0.04}$ | $0.73_{\pm 0.03}$ |

*Table 40.* **Raw benchmark results of 23 tabular generators on the "Coupon" dataset.** We report the mean $\pm$ std of each metric across 10 repeated data splits. For benchmark generators, "$-$" denotes failed convergence of a specific model or unexpected values in the synthetic data that caused the evaluation metric computation to crash. We highlight the **First**, **Second**, and **Third** best performances for each metric. TabFORGE generally achieves competitive performance against the benchmark generators while maintaining a reduced risk of overfitting.

| Generator | Density Estimation | | | | Privacy Preservation | | ML Efficacy | Structural Fidelity |
|---|---|---|---|---|---|---|---|---|
| | Shape $\uparrow$ | Trend $\uparrow$ | $\alpha$-precision $\uparrow$ | $\beta$-recall $\uparrow$ | Authenticity $\uparrow$ | DCR Score $\uparrow$ | Local utility $\uparrow$ | Global utility $\uparrow$ |
| **Real Data** | | | | | | | | |
| Real Data (Train) | $1.00_{\pm 0.00}$ | $1.00_{\pm 0.00}$ | $1.00_{\pm 0.00}$ | $1.00_{\pm 0.00}$ | $0.00_{\pm 0.00}$ | $0.00_{\pm 0.00}$ | $1.00_{\pm 0.00}$ | $0.95_{\pm 0.02}$ |
| Real Data (Holdout) | $0.99_{\pm 0.00}$ | $0.98_{\pm 0.00}$ | $0.99_{\pm 0.01}$ | $0.52_{\pm 0.01}$ | $0.61_{\pm 0.01}$ | $0.20_{\pm 0.00}$ | $-$ | $-$ |
| **Dataset-specific Model** | | | | | | | | |
| SMOTE | $0.93_{\pm 0.00}$ | $0.79_{\pm 0.01}$ | $0.68_{\pm 0.01}$ | $0.64_{\pm 0.01}$ | $0.44_{\pm 0.01}$ | $0.12_{\pm 0.00}$ | $0.96_{\pm 0.01}$ | $0.57_{\pm 0.02}$ |
| TabSDS | $-$ | $-$ | $-$ | $-$ | $-$ | $-$ | $-$ | $-$ |
| TVAE | $0.90_{\pm 0.01}$ | $0.84_{\pm 0.02}$ | $0.89_{\pm 0.03}$ | $0.02_{\pm 0.00}$ | $0.99_{\pm 0.00}$ | $0.65_{\pm 0.01}$ | $0.85_{\pm 0.02}$ | $0.47_{\pm 0.02}$ |
| GOGGLE | $0.63_{\pm 0.02}$ | $0.41_{\pm 0.03}$ | $0.03_{\pm 0.01}$ | $0.00_{\pm 0.00}$ | $1.00_{\pm 0.00}$ | $0.59_{\pm 0.01}$ | $0.46_{\pm 0.00}$ | $0.07_{\pm 0.03}$ |
| CTGAN | $0.88_{\pm 0.01}$ | $0.86_{\pm 0.04}$ | $0.96_{\pm 0.03}$ | $0.01_{\pm 0.00}$ | $1.00_{\pm 0.00}$ | $0.67_{\pm 0.00}$ | $0.82_{\pm 0.03}$ | $0.27_{\pm 0.02}$ |
| NFlow | $0.83_{\pm 0.01}$ | $0.75_{\pm 0.02}$ | $0.93_{\pm 0.03}$ | $0.00_{\pm 0.00}$ | $1.00_{\pm 0.00}$ | $0.73_{\pm 0.02}$ | $0.66_{\pm 0.01}$ | $0.18_{\pm 0.02}$ |
| ARF | $-$ | $-$ | $-$ | $-$ | $-$ | $-$ | $-$ | $-$ |
| TabDDPM | $0.72_{\pm 0.01}$ | $0.57_{\pm 0.03}$ | $0.09_{\pm 0.03}$ | $0.00_{\pm 0.00}$ | $1.00_{\pm 0.00}$ | $0.84_{\pm 0.03}$ | $0.83_{\pm 0.02}$ | $0.54_{\pm 0.02}$ |
| CDTD | $0.71_{\pm 0.00}$ | $0.69_{\pm 0.01}$ | $0.55_{\pm 0.01}$ | $0.07_{\pm 0.00}$ | $0.75_{\pm 0.00}$ | $0.43_{\pm 0.00}$ | $0.73_{\pm 0.01}$ | $0.34_{\pm 0.02}$ |
| TabSyn | $0.91_{\pm 0.01}$ | $0.91_{\pm 0.01}$ | $0.91_{\pm 0.01}$ | $0.15_{\pm 0.01}$ | $0.89_{\pm 0.01}$ | $0.53_{\pm 0.02}$ | $0.89_{\pm 0.01}$ | $0.50_{\pm 0.01}$ |
| TabDiff | $0.80_{\pm 0.01}$ | $0.67_{\pm 0.02}$ | $0.23_{\pm 0.01}$ | $0.00_{\pm 0.00}$ | $1.00_{\pm 0.00}$ | $0.91_{\pm 0.01}$ | $0.78_{\pm 0.01}$ | $0.61_{\pm 0.02}$ |
| NRGBoost | $0.90_{\pm 0.01}$ | $0.75_{\pm 0.01}$ | $0.95_{\pm 0.02}$ | $0.03_{\pm 0.01}$ | $0.99_{\pm 0.01}$ | $0.62_{\pm 0.02}$ | $0.88_{\pm 0.01}$ | $0.23_{\pm 0.01}$ |
| TabNAT | $0.67_{\pm 0.00}$ | $0.55_{\pm 0.01}$ | $0.39_{\pm 0.01}$ | $0.04_{\pm 0.00}$ | $0.86_{\pm 0.00}$ | $0.63_{\pm 0.00}$ | $0.67_{\pm 0.01}$ | $0.20_{\pm 0.01}$ |
| TabularARGN | $0.75_{\pm 0.00}$ | $0.93_{\pm 0.01}$ | $0.92_{\pm 0.02}$ | $0.02_{\pm 0.00}$ | $0.99_{\pm 0.00}$ | $0.57_{\pm 0.00}$ | $0.87_{\pm 0.01}$ | $0.37_{\pm 0.03}$ |
| **Foundation Model** | | | | | | | | |
| GReaT | $0.74_{\pm 0.00}$ | $0.51_{\pm 0.01}$ | $0.19_{\pm 0.01}$ | $0.01_{\pm 0.00}$ | $1.00_{\pm 0.00}$ | $0.67_{\pm 0.00}$ | $0.76_{\pm 0.02}$ | $0.15_{\pm 0.02}$ |
| TabPFN | $-$ | $-$ | $-$ | $-$ | $-$ | $-$ | $-$ | $-$ |
| TabDPT | $0.74_{\pm 0.01}$ | $0.61_{\pm 0.02}$ | $0.59_{\pm 0.01}$ | $0.01_{\pm 0.00}$ | $0.89_{\pm 0.00}$ | $0.53_{\pm 0.00}$ | $0.65_{\pm 0.02}$ | $0.23_{\pm 0.01}$ |
| Mitra | $0.62_{\pm 0.01}$ | $0.48_{\pm 0.02}$ | $0.39_{\pm 0.01}$ | $0.02_{\pm 0.00}$ | $0.87_{\pm 0.00}$ | $0.56_{\pm 0.00}$ | $0.66_{\pm 0.01}$ | $0.17_{\pm 0.02}$ |
| LimiX | $-$ | $-$ | $-$ | $-$ | $-$ | $-$ | $-$ | $-$ |
| TabICL | $-$ | $-$ | $-$ | $-$ | $-$ | $-$ | $-$ | $-$ |
| TabEBM | $0.91_{\pm 0.00}$ | $0.75_{\pm 0.01}$ | $0.77_{\pm 0.01}$ | $0.10_{\pm 0.00}$ | $0.91_{\pm 0.00}$ | $0.41_{\pm 0.00}$ | $0.89_{\pm 0.01}$ | $0.42_{\pm 0.03}$ |
| CTSyn | $0.67_{\pm 0.00}$ | $0.60_{\pm 0.01}$ | $0.37_{\pm 0.01}$ | $0.06_{\pm 0.00}$ | $0.77_{\pm 0.00}$ | $0.46_{\pm 0.01}$ | $0.65_{\pm 0.01}$ | $0.28_{\pm 0.01}$ |
| **TabFORGE (Ours)** | $0.93_{\pm 0.01}$ | $0.94_{\pm 0.01}$ | $0.98_{\pm 0.01}$ | $0.47_{\pm 0.03}$ | $0.57_{\pm 0.02}$ | $0.19_{\pm 0.00}$ | $0.94_{\pm 0.01}$ | $0.75_{\pm 0.04}$ |

*Table 41.* **Raw benchmark results of 23 tabular generators on the "Credit" dataset.** We report the mean $\pm$ std of each metric across 10 repeated data splits. For benchmark generators, "$-$" denotes failed convergence of a specific model or unexpected values in the synthetic data that caused the evaluation metric computation to crash. We highlight the **First**, **Second**, and **Third** best performances for each metric. TabFORGE generally achieves competitive performance against the benchmark generators while maintaining a reduced risk of overfitting.

| Generator | Density Estimation | | | | Privacy Preservation | | ML Efficacy | Structural Fidelity |
|---|---|---|---|---|---|---|---|---|
| | Shape ↑ | Trend ↑ | $\alpha$-precision ↑ | $\beta$-recall ↑ | Authenticity ↑ | DCR Score ↑ | Local utility ↑ | Global utility ↑ |
| **Real Data** | | | | | | | | |
| Real Data (Train) | $1.00_{\pm0.00}$ | $1.00_{\pm0.00}$ | $1.00_{\pm0.00}$ | $1.00_{\pm0.00}$ | $0.00_{\pm0.00}$ | $0.00_{\pm0.00}$ | $1.00_{\pm0.00}$ | $0.50_{\pm0.03}$ |
| Real Data (Holdout) | $0.96_{\pm0.01}$ | $0.92_{\pm0.03}$ | $0.95_{\pm0.02}$ | $0.48_{\pm0.04}$ | $0.75_{\pm0.02}$ | $0.26_{\pm0.01}$ | $-$ | $-$ |
| **Dataset-specific Model** | | | | | | | | |
| SMOTE | $0.94_{\pm0.01}$ | $0.75_{\pm0.01}$ | $0.80_{\pm0.03}$ | $0.62_{\pm0.03}$ | $0.50_{\pm0.01}$ | $0.02_{\pm0.00}$ | $0.97_{\pm0.04}$ | $0.31_{\pm0.02}$ |
| TabSDS | $0.74_{\pm0.00}$ | $0.62_{\pm0.02}$ | $0.59_{\pm0.02}$ | $0.23_{\pm0.01}$ | $0.56_{\pm0.01}$ | $0.26_{\pm0.01}$ | $0.68_{\pm0.02}$ | $0.26_{\pm0.02}$ |
| TVAE | $0.91_{\pm0.01}$ | $0.81_{\pm0.02}$ | $0.73_{\pm0.06}$ | $0.40_{\pm0.03}$ | $0.82_{\pm0.02}$ | $0.42_{\pm0.01}$ | $0.88_{\pm0.05}$ | $0.32_{\pm0.02}$ |
| GOGGLE | $0.61_{\pm0.01}$ | $0.42_{\pm0.07}$ | $0.06_{\pm0.02}$ | $0.14_{\pm0.02}$ | $0.94_{\pm0.01}$ | $0.29_{\pm0.02}$ | $0.98_{\pm0.02}$ | $0.29_{\pm0.03}$ |
| CTGAN | $0.90_{\pm0.01}$ | $0.81_{\pm0.03}$ | $0.90_{\pm0.06}$ | $0.29_{\pm0.04}$ | $0.88_{\pm0.02}$ | $0.47_{\pm0.02}$ | $0.91_{\pm0.05}$ | $0.32_{\pm0.04}$ |
| NFlow | $0.88_{\pm0.02}$ | $0.77_{\pm0.02}$ | $0.80_{\pm0.09}$ | $0.18_{\pm0.02}$ | $0.93_{\pm0.01}$ | $0.56_{\pm0.02}$ | $0.79_{\pm0.05}$ | $0.27_{\pm0.02}$ |
| ARF | $0.94_{\pm0.01}$ | $0.78_{\pm0.06}$ | $0.94_{\pm0.02}$ | $0.34_{\pm0.02}$ | $0.84_{\pm0.02}$ | $0.46_{\pm0.01}$ | $0.90_{\pm0.03}$ | $0.27_{\pm0.02}$ |
| TabDDPM | $0.61_{\pm0.02}$ | $0.53_{\pm0.05}$ | $0.03_{\pm0.01}$ | $0.03_{\pm0.01}$ | $1.00_{\pm0.00}$ | $0.85_{\pm0.08}$ | $0.97_{\pm0.07}$ | $0.30_{\pm0.02}$ |
| CDTD | $0.64_{\pm0.01}$ | $0.49_{\pm0.01}$ | $0.59_{\pm0.04}$ | $0.22_{\pm0.02}$ | $0.71_{\pm0.01}$ | $0.41_{\pm0.01}$ | $0.68_{\pm0.02}$ | $0.34_{\pm0.01}$ |
| TabSyn | $0.89_{\pm0.01}$ | $0.81_{\pm0.02}$ | $0.87_{\pm0.06}$ | $0.32_{\pm0.06}$ | $0.86_{\pm0.04}$ | $0.44_{\pm0.03}$ | $0.89_{\pm0.05}$ | $0.34_{\pm0.02}$ |
| TabDiff | $0.82_{\pm0.01}$ | $0.72_{\pm0.02}$ | $0.25_{\pm0.04}$ | $0.06_{\pm0.01}$ | $0.99_{\pm0.01}$ | $0.74_{\pm0.02}$ | $0.84_{\pm0.04}$ | $0.36_{\pm0.02}$ |
| NRGBoost | $0.84_{\pm0.01}$ | $0.80_{\pm0.06}$ | $0.83_{\pm0.07}$ | $0.42_{\pm0.04}$ | $0.57_{\pm0.02}$ | $0.07_{\pm0.02}$ | $0.96_{\pm0.05}$ | $0.21_{\pm0.01}$ |
| TabNAT | $0.53_{\pm0.01}$ | $0.52_{\pm0.01}$ | $0.45_{\pm0.05}$ | $0.11_{\pm0.02}$ | $0.82_{\pm0.01}$ | $0.63_{\pm0.01}$ | $0.64_{\pm0.02}$ | $0.32_{\pm0.02}$ |
| TabularARGN | $0.83_{\pm0.00}$ | $0.79_{\pm0.03}$ | $0.96_{\pm0.02}$ | $0.30_{\pm0.02}$ | $0.88_{\pm0.02}$ | $0.48_{\pm0.01}$ | $0.89_{\pm0.04}$ | $0.31_{\pm0.03}$ |
| **Foundation Model** | | | | | | | | |
| GReaT | $0.29_{\pm0.02}$ | $0.14_{\pm0.03}$ | $0.38_{\pm0.14}$ | $0.00_{\pm0.00}$ | $1.00_{\pm0.00}$ | $1.00_{\pm0.00}$ | $0.82_{\pm0.04}$ | $0.44_{\pm0.03}$ |
| TabPFN | $0.70_{\pm0.01}$ | $0.61_{\pm0.02}$ | $0.62_{\pm0.06}$ | $0.17_{\pm0.01}$ | $0.73_{\pm0.01}$ | $0.50_{\pm0.01}$ | $0.71_{\pm0.03}$ | $0.30_{\pm0.02}$ |
| TabDPT | $0.72_{\pm0.01}$ | $0.57_{\pm0.02}$ | $0.61_{\pm0.03}$ | $0.19_{\pm0.01}$ | $0.76_{\pm0.01}$ | $0.40_{\pm0.01}$ | $1.00_{\pm0.01}$ | $0.28_{\pm0.02}$ |
| Mitra | $0.66_{\pm0.01}$ | $0.56_{\pm0.04}$ | $0.33_{\pm0.03}$ | $0.16_{\pm0.01}$ | $0.78_{\pm0.01}$ | $0.32_{\pm0.01}$ | $1.00_{\pm0.00}$ | $0.33_{\pm0.02}$ |
| LimiX | $0.74_{\pm0.01}$ | $0.63_{\pm0.02}$ | $0.61_{\pm0.05}$ | $0.17_{\pm0.01}$ | $0.77_{\pm0.01}$ | $0.39_{\pm0.01}$ | $0.87_{\pm0.04}$ | $0.28_{\pm0.02}$ |
| TabICL | $0.76_{\pm0.01}$ | $0.62_{\pm0.03}$ | $0.68_{\pm0.03}$ | $0.25_{\pm0.01}$ | $0.73_{\pm0.01}$ | $0.42_{\pm0.01}$ | $0.78_{\pm0.03}$ | $0.27_{\pm0.02}$ |
| TabEBM | $0.88_{\pm0.01}$ | $0.76_{\pm0.05}$ | $0.76_{\pm0.03}$ | $0.42_{\pm0.03}$ | $0.61_{\pm0.02}$ | $0.21_{\pm0.00}$ | $0.96_{\pm0.04}$ | $0.33_{\pm0.02}$ |
| CTSyn | $0.52_{\pm0.01}$ | $0.38_{\pm0.01}$ | $0.46_{\pm0.06}$ | $0.13_{\pm0.02}$ | $0.82_{\pm0.02}$ | $0.49_{\pm0.01}$ | $0.82_{\pm0.03}$ | $0.34_{\pm0.01}$ |
| **TabFORGE (Ours)** | $0.95_{\pm0.01}$ | $0.92_{\pm0.02}$ | $0.96_{\pm0.02}$ | $0.55_{\pm0.03}$ | $0.48_{\pm0.01}$ | $0.01_{\pm0.00}$ | $0.97_{\pm0.04}$ | $0.43_{\pm0.03}$ |

*Table 42.* **Raw benchmark results of 23 tabular generators on the "Customer" dataset.** We report the mean $\pm$ std of each metric across 10 repeated data splits. For benchmark generators, "$-$" denotes failed convergence of a specific model or unexpected values in the synthetic data that caused the evaluation metric computation to crash. We highlight the **First**, **Second**, and **Third** best performances for each metric. TabFORGE generally achieves competitive performance against the benchmark generators while maintaining a reduced risk of overfitting.

| Generator | Density Estimation | | | | Privacy Preservation | | ML Efficacy | Structural Fidelity |
|---|---|---|---|---|---|---|---|---|
| | Shape ↑ | Trend ↑ | $\alpha$-precision ↑ | $\beta$-recall ↑ | Authenticity ↑ | DCR Score ↑ | Local utility ↑ | Global utility ↑ |
| **Real Data** | | | | | | | | |
| Real Data (Train) | $1.00_{\pm 0.00}$ | $1.00_{\pm 0.00}$ | $1.00_{\pm 0.00}$ | $1.00_{\pm 0.00}$ | $0.00_{\pm 0.00}$ | $0.00_{\pm 0.00}$ | $1.00_{\pm 0.00}$ | $0.66_{\pm 0.03}$ |
| Real Data (Holdout) | $0.97_{\pm 0.00}$ | $0.87_{\pm 0.04}$ | $0.96_{\pm 0.02}$ | $0.49_{\pm 0.03}$ | $0.71_{\pm 0.02}$ | $0.21_{\pm 0.01}$ | $-$ | $-$ |
| **Dataset-specific Model** | | | | | | | | |
| SMOTE | $0.94_{\pm 0.01}$ | $0.68_{\pm 0.03}$ | $0.74_{\pm 0.02}$ | $0.63_{\pm 0.01}$ | $0.46_{\pm 0.02}$ | $0.02_{\pm 0.00}$ | $1.01_{\pm 0.03}$ | $0.54_{\pm 0.03}$ |
| TabSDS | $-$ | $-$ | $-$ | $-$ | $-$ | $-$ | $-$ | $-$ |
| TVAE | $0.87_{\pm 0.01}$ | $0.59_{\pm 0.07}$ | $0.92_{\pm 0.03}$ | $0.31_{\pm 0.03}$ | $0.84_{\pm 0.02}$ | $0.35_{\pm 0.02}$ | $0.97_{\pm 0.02}$ | $0.49_{\pm 0.04}$ |
| GOGGLE | $0.63_{\pm 0.02}$ | $0.39_{\pm 0.10}$ | $0.17_{\pm 0.06}$ | $0.13_{\pm 0.04}$ | $0.94_{\pm 0.02}$ | $0.18_{\pm 0.08}$ | $1.00_{\pm 0.00}$ | $0.26_{\pm 0.03}$ |
| CTGAN | $0.82_{\pm 0.03}$ | $0.62_{\pm 0.11}$ | $0.90_{\pm 0.05}$ | $0.29_{\pm 0.06}$ | $0.86_{\pm 0.03}$ | $0.37_{\pm 0.03}$ | $0.98_{\pm 0.02}$ | $0.47_{\pm 0.05}$ |
| NFlow | $0.86_{\pm 0.01}$ | $0.62_{\pm 0.05}$ | $0.78_{\pm 0.10}$ | $0.22_{\pm 0.03}$ | $0.90_{\pm 0.01}$ | $0.43_{\pm 0.03}$ | $0.80_{\pm 0.03}$ | $0.42_{\pm 0.06}$ |
| ARF | $-$ | $-$ | $-$ | $-$ | $-$ | $-$ | $-$ | $-$ |
| TabDDPM | $0.72_{\pm 0.05}$ | $0.50_{\pm 0.06}$ | $0.53_{\pm 0.10}$ | $0.21_{\pm 0.04}$ | $0.90_{\pm 0.02}$ | $0.63_{\pm 0.14}$ | $0.97_{\pm 0.02}$ | $0.47_{\pm 0.03}$ |
| CDTD | $0.73_{\pm 0.01}$ | $0.55_{\pm 0.03}$ | $0.65_{\pm 0.03}$ | $0.30_{\pm 0.02}$ | $0.56_{\pm 0.01}$ | $0.29_{\pm 0.02}$ | $0.75_{\pm 0.01}$ | $0.47_{\pm 0.03}$ |
| TabSyn | $0.91_{\pm 0.02}$ | $0.80_{\pm 0.04}$ | $0.91_{\pm 0.06}$ | $0.41_{\pm 0.04}$ | $0.78_{\pm 0.03}$ | $0.33_{\pm 0.02}$ | $0.98_{\pm 0.02}$ | $0.52_{\pm 0.06}$ |
| TabDiff | $0.86_{\pm 0.01}$ | $0.75_{\pm 0.04}$ | $0.48_{\pm 0.05}$ | $0.16_{\pm 0.02}$ | $0.93_{\pm 0.01}$ | $0.58_{\pm 0.02}$ | $0.94_{\pm 0.06}$ | $0.50_{\pm 0.02}$ |
| NRGBoost | $0.91_{\pm 0.01}$ | $0.85_{\pm 0.02}$ | $0.91_{\pm 0.04}$ | $0.50_{\pm 0.02}$ | $0.52_{\pm 0.01}$ | $0.02_{\pm 0.00}$ | $1.01_{\pm 0.03}$ | $0.30_{\pm 0.02}$ |
| TabNAT | $0.69_{\pm 0.01}$ | $0.54_{\pm 0.03}$ | $0.51_{\pm 0.03}$ | $0.20_{\pm 0.01}$ | $0.68_{\pm 0.01}$ | $0.40_{\pm 0.02}$ | $0.92_{\pm 0.03}$ | $0.39_{\pm 0.04}$ |
| TabularARGN | $0.77_{\pm 0.00}$ | $0.79_{\pm 0.02}$ | $0.97_{\pm 0.01}$ | $0.35_{\pm 0.02}$ | $0.81_{\pm 0.01}$ | $0.35_{\pm 0.02}$ | $0.98_{\pm 0.03}$ | $0.42_{\pm 0.05}$ |
| **Foundation Model** | | | | | | | | |
| GReaT | $0.76_{\pm 0.01}$ | $0.42_{\pm 0.07}$ | $0.64_{\pm 0.04}$ | $0.27_{\pm 0.02}$ | $0.86_{\pm 0.02}$ | $0.35_{\pm 0.02}$ | $0.95_{\pm 0.01}$ | $0.41_{\pm 0.06}$ |
| TabPFN | $-$ | $-$ | $-$ | $-$ | $-$ | $-$ | $-$ | $-$ |
| TabDPT | $0.73_{\pm 0.02}$ | $0.53_{\pm 0.06}$ | $0.55_{\pm 0.02}$ | $0.23_{\pm 0.03}$ | $0.68_{\pm 0.02}$ | $0.26_{\pm 0.03}$ | $1.00_{\pm 0.00}$ | $0.39_{\pm 0.04}$ |
| Mitra | $0.57_{\pm 0.01}$ | $0.38_{\pm 0.06}$ | $0.41_{\pm 0.03}$ | $0.15_{\pm 0.02}$ | $0.73_{\pm 0.01}$ | $0.26_{\pm 0.05}$ | $1.00_{\pm 0.00}$ | $0.39_{\pm 0.03}$ |
| LimiX | $-$ | $-$ | $-$ | $-$ | $-$ | $-$ | $-$ | $-$ |
| TabICL | $-$ | $-$ | $-$ | $-$ | $-$ | $-$ | $-$ | $-$ |
| TabEBM | $0.92_{\pm 0.00}$ | $0.69_{\pm 0.05}$ | $0.80_{\pm 0.03}$ | $0.41_{\pm 0.02}$ | $0.67_{\pm 0.01}$ | $0.26_{\pm 0.03}$ | $0.99_{\pm 0.02}$ | $0.49_{\pm 0.03}$ |
| CTSyn | $0.74_{\pm 0.01}$ | $0.48_{\pm 0.04}$ | $0.60_{\pm 0.03}$ | $0.25_{\pm 0.01}$ | $0.72_{\pm 0.01}$ | $0.27_{\pm 0.02}$ | $0.85_{\pm 0.01}$ | $0.40_{\pm 0.03}$ |
| **TabFORGE (Ours)** | $0.90_{\pm 0.01}$ | $0.78_{\pm 0.03}$ | $0.93_{\pm 0.04}$ | $0.40_{\pm 0.02}$ | $0.78_{\pm 0.02}$ | $0.32_{\pm 0.02}$ | $0.98_{\pm 0.03}$ | $0.54_{\pm 0.06}$ |

*Table 43.* **Raw benchmark results of 23 tabular generators on the "Diabetes" dataset.** We report the mean $\pm$ std of each metric across 10 repeated data splits. For benchmark generators, "$-$" denotes failed convergence of a specific model or unexpected values in the synthetic data that caused the evaluation metric computation to crash. We highlight the **First**, **Second**, and **Third** best performances for each metric. TabFORGE generally achieves competitive performance against the benchmark generators while maintaining a reduced risk of overfitting.

| Generator | Density Estimation | | | | Privacy Preservation | | ML Efficacy | Structural Fidelity |
|---|---|---|---|---|---|---|---|---|
| | Shape ↑ | Trend ↑ | $\alpha$-precision ↑ | $\beta$-recall ↑ | Authenticity ↑ | DCR Score ↑ | Local utility ↑ | Global utility ↑ |
| **Real Data** | | | | | | | | |
| Real Data (Train) | $1.00_{\pm 0.00}$ | $1.00_{\pm 0.00}$ | $1.00_{\pm 0.00}$ | $1.00_{\pm 0.00}$ | $0.00_{\pm 0.00}$ | $0.00_{\pm 0.00}$ | $1.00_{\pm 0.00}$ | $0.98_{\pm 0.01}$ |
| Real Data (Holdout) | $0.94_{\pm 0.01}$ | $0.86_{\pm 0.03}$ | $0.95_{\pm 0.02}$ | $0.50_{\pm 0.04}$ | $0.70_{\pm 0.04}$ | $0.17_{\pm 0.01}$ | $-$ | $-$ |
| **Dataset-specific Model** | | | | | | | | |
| SMOTE | $0.86_{\pm 0.01}$ | $0.59_{\pm 0.04}$ | $0.81_{\pm 0.02}$ | $0.66_{\pm 0.04}$ | $0.43_{\pm 0.03}$ | $0.08_{\pm 0.01}$ | $0.98_{\pm 0.04}$ | $0.71_{\pm 0.02}$ |
| TabSDS | $0.67_{\pm 0.01}$ | $0.55_{\pm 0.03}$ | $0.64_{\pm 0.02}$ | $0.29_{\pm 0.01}$ | $0.55_{\pm 0.01}$ | $0.20_{\pm 0.02}$ | $0.71_{\pm 0.02}$ | $0.67_{\pm 0.03}$ |
| TVAE | $0.79_{\pm 0.02}$ | $0.61_{\pm 0.07}$ | $0.75_{\pm 0.09}$ | $0.36_{\pm 0.05}$ | $0.79_{\pm 0.03}$ | $0.27_{\pm 0.03}$ | $0.87_{\pm 0.04}$ | $0.76_{\pm 0.02}$ |
| GOGGLE | $0.57_{\pm 0.04}$ | $0.46_{\pm 0.06}$ | $0.80_{\pm 0.09}$ | $0.17_{\pm 0.04}$ | $0.92_{\pm 0.02}$ | $0.36_{\pm 0.05}$ | $0.48_{\pm 0.02}$ | $0.30_{\pm 0.04}$ |
| CTGAN | $0.64_{\pm 0.09}$ | $0.60_{\pm 0.07}$ | $0.78_{\pm 0.13}$ | $0.19_{\pm 0.07}$ | $0.91_{\pm 0.04}$ | $0.31_{\pm 0.06}$ | $0.80_{\pm 0.08}$ | $0.60_{\pm 0.10}$ |
| NFlow | $0.85_{\pm 0.02}$ | $0.75_{\pm 0.07}$ | $0.90_{\pm 0.05}$ | $0.28_{\pm 0.05}$ | $0.86_{\pm 0.02}$ | $0.33_{\pm 0.03}$ | $0.65_{\pm 0.04}$ | $0.68_{\pm 0.07}$ |
| ARF | $0.87_{\pm 0.01}$ | $0.69_{\pm 0.07}$ | $0.91_{\pm 0.04}$ | $0.27_{\pm 0.03}$ | $0.86_{\pm 0.02}$ | $0.35_{\pm 0.04}$ | $0.89_{\pm 0.04}$ | $0.58_{\pm 0.03}$ |
| TabDDPM | $0.82_{\pm 0.02}$ | $0.85_{\pm 0.03}$ | $0.87_{\pm 0.04}$ | $0.39_{\pm 0.02}$ | $0.74_{\pm 0.02}$ | $0.28_{\pm 0.02}$ | $1.01_{\pm 0.05}$ | $0.70_{\pm 0.02}$ |
| CDTD | $0.58_{\pm 0.01}$ | $0.54_{\pm 0.03}$ | $0.58_{\pm 0.01}$ | $0.27_{\pm 0.02}$ | $0.62_{\pm 0.02}$ | $0.32_{\pm 0.03}$ | $0.77_{\pm 0.04}$ | $0.51_{\pm 0.02}$ |
| TabSyn | $0.82_{\pm 0.01}$ | $0.77_{\pm 0.05}$ | $0.89_{\pm 0.03}$ | $0.50_{\pm 0.03}$ | $0.69_{\pm 0.04}$ | $0.23_{\pm 0.02}$ | $0.96_{\pm 0.03}$ | $0.82_{\pm 0.04}$ |
| TabDiff | $0.80_{\pm 0.02}$ | $0.77_{\pm 0.04}$ | $0.72_{\pm 0.06}$ | $0.19_{\pm 0.03}$ | $0.90_{\pm 0.01}$ | $0.40_{\pm 0.03}$ | $0.73_{\pm 0.06}$ | $0.69_{\pm 0.02}$ |
| NRGBoost | $0.83_{\pm 0.01}$ | $0.79_{\pm 0.06}$ | $0.94_{\pm 0.03}$ | $0.51_{\pm 0.02}$ | $0.54_{\pm 0.02}$ | $0.03_{\pm 0.01}$ | $0.95_{\pm 0.05}$ | $0.42_{\pm 0.01}$ |
| TabNAT | $0.59_{\pm 0.01}$ | $0.59_{\pm 0.03}$ | $0.55_{\pm 0.03}$ | $0.21_{\pm 0.02}$ | $0.83_{\pm 0.01}$ | $0.39_{\pm 0.03}$ | $0.60_{\pm 0.04}$ | $0.49_{\pm 0.03}$ |
| TabularARGN | $0.74_{\pm 0.01}$ | $0.62_{\pm 0.03}$ | $0.87_{\pm 0.02}$ | $0.19_{\pm 0.04}$ | $0.91_{\pm 0.02}$ | $0.37_{\pm 0.04}$ | $0.77_{\pm 0.05}$ | $0.61_{\pm 0.06}$ |
| **Foundation Model** | | | | | | | | |
| GReaT | $0.36_{\pm 0.01}$ | $0.35_{\pm 0.07}$ | $0.03_{\pm 0.01}$ | $0.00_{\pm 0.00}$ | $1.00_{\pm 0.00}$ | $0.80_{\pm 0.08}$ | $0.70_{\pm 0.09}$ | $0.04_{\pm 0.02}$ |
| TabPFN | $0.63_{\pm 0.01}$ | $0.63_{\pm 0.04}$ | $0.62_{\pm 0.03}$ | $0.20_{\pm 0.03}$ | $0.70_{\pm 0.01}$ | $0.38_{\pm 0.03}$ | $0.62_{\pm 0.03}$ | $0.49_{\pm 0.04}$ |
| TabDPT | $0.58_{\pm 0.04}$ | $0.51_{\pm 0.04}$ | $0.68_{\pm 0.07}$ | $0.16_{\pm 0.03}$ | $0.80_{\pm 0.02}$ | $0.25_{\pm 0.03}$ | $0.68_{\pm 0.03}$ | $0.43_{\pm 0.06}$ |
| Mitra | $0.55_{\pm 0.02}$ | $0.54_{\pm 0.03}$ | $0.73_{\pm 0.05}$ | $0.20_{\pm 0.02}$ | $0.74_{\pm 0.01}$ | $0.31_{\pm 0.04}$ | $0.63_{\pm 0.03}$ | $0.48_{\pm 0.05}$ |
| LimiX | $0.75_{\pm 0.01}$ | $0.59_{\pm 0.04}$ | $0.71_{\pm 0.03}$ | $0.23_{\pm 0.03}$ | $0.69_{\pm 0.01}$ | $0.30_{\pm 0.03}$ | $0.62_{\pm 0.02}$ | $0.56_{\pm 0.04}$ |
| TabICL | $0.77_{\pm 0.01}$ | $0.65_{\pm 0.04}$ | $0.77_{\pm 0.03}$ | $0.25_{\pm 0.02}$ | $0.77_{\pm 0.01}$ | $0.27_{\pm 0.02}$ | $0.72_{\pm 0.02}$ | $0.59_{\pm 0.03}$ |
| TabEBM | $0.87_{\pm 0.01}$ | $0.73_{\pm 0.06}$ | $0.89_{\pm 0.02}$ | $0.34_{\pm 0.04}$ | $0.72_{\pm 0.03}$ | $0.27_{\pm 0.03}$ | $0.93_{\pm 0.06}$ | $0.63_{\pm 0.02}$ |
| CTSyn | $0.48_{\pm 0.01}$ | $0.43_{\pm 0.03}$ | $0.42_{\pm 0.02}$ | $0.22_{\pm 0.01}$ | $0.78_{\pm 0.02}$ | $0.42_{\pm 0.04}$ | $0.69_{\pm 0.04}$ | $0.45_{\pm 0.03}$ |
| **TabFORGE (Ours)** | $0.79_{\pm 0.02}$ | $0.82_{\pm 0.04}$ | $0.93_{\pm 0.03}$ | $0.37_{\pm 0.04}$ | $0.76_{\pm 0.02}$ | $0.25_{\pm 0.02}$ | $0.93_{\pm 0.06}$ | $0.83_{\pm 0.04}$ |

*Table 44.* **Raw benchmark results of 23 tabular generators on the "Diamonds" dataset.** We report the mean $\pm$ std of each metric across 10 repeated data splits. For benchmark generators, "$-$" denotes failed convergence of a specific model or unexpected values in the synthetic data that caused the evaluation metric computation to crash. We highlight the **First**, **Second**, and **Third** best performances for each metric. TabFORGE generally achieves competitive performance against the benchmark generators while maintaining a reduced risk of overfitting.

| Generator | Density Estimation | | | | Privacy Preservation | | ML Efficacy | Structural Fidelity |
|---|---|---|---|---|---|---|---|---|
| | Shape ↑ | Trend ↑ | $\alpha$-precision ↑ | $\beta$-recall ↑ | Authenticity ↑ | DCR Score ↑ | Local utility ↑ | Global utility ↑ |
| **Real Data** | | | | | | | | |
| Real Data (Train) | $1.00_{\pm 0.00}$ | $1.00_{\pm 0.00}$ | $1.00_{\pm 0.00}$ | $1.00_{\pm 0.00}$ | $0.00_{\pm 0.00}$ | $0.00_{\pm 0.00}$ | $1.00_{\pm 0.00}$ | $0.99_{\pm 0.02}$ |
| Real Data (Holdout) | $0.99_{\pm 0.00}$ | $0.91_{\pm 0.04}$ | $0.99_{\pm 0.00}$ | $0.50_{\pm 0.00}$ | $0.66_{\pm 0.01}$ | $0.03_{\pm 0.00}$ | $-$ | $-$ |
| **Dataset-specific Model** | | | | | | | | |
| SMOTE | $0.98_{\pm 0.00}$ | $-$ | $-$ | $0.61_{\pm 0.01}$ | $0.50_{\pm 0.01}$ | $0.04_{\pm 0.00}$ | $0.87_{\pm 0.01}$ | $-$ |
| TabSDS | $0.70_{\pm 0.01}$ | $0.64_{\pm 0.02}$ | $0.64_{\pm 0.00}$ | $0.18_{\pm 0.01}$ | $0.66_{\pm 0.00}$ | $0.04_{\pm 0.01}$ | $0.49_{\pm 0.01}$ | $0.41_{\pm 0.01}$ |
| TVAE | $0.94_{\pm 0.00}$ | $0.84_{\pm 0.02}$ | $0.97_{\pm 0.02}$ | $0.16_{\pm 0.01}$ | $0.92_{\pm 0.01}$ | $0.08_{\pm 0.01}$ | $0.43_{\pm 0.02}$ | $0.63_{\pm 0.01}$ |
| GOGGLE | $0.57_{\pm 0.05}$ | $0.50_{\pm 0.02}$ | $0.40_{\pm 0.08}$ | $0.01_{\pm 0.00}$ | $1.00_{\pm 0.00}$ | $0.13_{\pm 0.03}$ | $0.10_{\pm 0.00}$ | $0.04_{\pm 0.02}$ |
| CTGAN | $0.90_{\pm 0.02}$ | $0.86_{\pm 0.02}$ | $0.95_{\pm 0.02}$ | $0.14_{\pm 0.02}$ | $0.93_{\pm 0.01}$ | $0.08_{\pm 0.01}$ | $0.39_{\pm 0.02}$ | $0.40_{\pm 0.03}$ |
| NFlow | $0.78_{\pm 0.06}$ | $0.70_{\pm 0.06}$ | $0.73_{\pm 0.20}$ | $0.06_{\pm 0.03}$ | $0.98_{\pm 0.01}$ | $0.18_{\pm 0.08}$ | $0.27_{\pm 0.06}$ | $0.27_{\pm 0.10}$ |
| ARF | $0.98_{\pm 0.00}$ | $0.93_{\pm 0.04}$ | $0.99_{\pm 0.00}$ | $0.25_{\pm 0.01}$ | $0.86_{\pm 0.00}$ | $0.06_{\pm 0.01}$ | $0.69_{\pm 0.01}$ | $0.58_{\pm 0.01}$ |
| TabDDPM | $0.98_{\pm 0.00}$ | $0.93_{\pm 0.04}$ | $0.99_{\pm 0.00}$ | $0.26_{\pm 0.01}$ | $0.85_{\pm 0.00}$ | $0.06_{\pm 0.01}$ | $0.73_{\pm 0.02}$ | $0.59_{\pm 0.01}$ |
| CDTD | $-$ | $-$ | $-$ | $-$ | $-$ | $-$ | $-$ | $-$ |
| TabSyn | $0.95_{\pm 0.01}$ | $0.90_{\pm 0.01}$ | $0.97_{\pm 0.01}$ | $0.28_{\pm 0.01}$ | $0.84_{\pm 0.01}$ | $0.06_{\pm 0.01}$ | $0.70_{\pm 0.03}$ | $0.67_{\pm 0.02}$ |
| TabDiff | $0.90_{\pm 0.01}$ | $0.81_{\pm 0.02}$ | $0.63_{\pm 0.05}$ | $0.04_{\pm 0.00}$ | $0.98_{\pm 0.00}$ | $0.18_{\pm 0.02}$ | $0.33_{\pm 0.02}$ | $0.60_{\pm 0.01}$ |
| NRGBoost | $0.88_{\pm 0.03}$ | $0.95_{\pm 0.03}$ | $0.96_{\pm 0.01}$ | $0.04_{\pm 0.00}$ | $0.98_{\pm 0.00}$ | $0.06_{\pm 0.01}$ | $0.49_{\pm 0.02}$ | $-$ |
| TabNAT | $0.73_{\pm 0.00}$ | $0.68_{\pm 0.02}$ | $0.71_{\pm 0.02}$ | $0.08_{\pm 0.00}$ | $0.83_{\pm 0.00}$ | $0.11_{\pm 0.01}$ | $0.36_{\pm 0.01}$ | $0.30_{\pm 0.02}$ |
| TabularARGN | $0.81_{\pm 0.00}$ | $0.91_{\pm 0.02}$ | $0.98_{\pm 0.01}$ | $0.03_{\pm 0.00}$ | $0.98_{\pm 0.00}$ | $0.06_{\pm 0.01}$ | $0.50_{\pm 0.01}$ | $0.43_{\pm 0.02}$ |
| **Foundation Model** | | | | | | | | |
| GReaT | $0.93_{\pm 0.00}$ | $0.59_{\pm 0.04}$ | $0.87_{\pm 0.01}$ | $0.03_{\pm 0.00}$ | $0.99_{\pm 0.00}$ | $0.12_{\pm 0.01}$ | $0.15_{\pm 0.00}$ | $0.21_{\pm 0.04}$ |
| TabPFN | $0.77_{\pm 0.03}$ | $0.62_{\pm 0.03}$ | $0.71_{\pm 0.10}$ | $0.09_{\pm 0.01}$ | $0.80_{\pm 0.01}$ | $0.12_{\pm 0.04}$ | $0.31_{\pm 0.03}$ | $0.33_{\pm 0.05}$ |
| TabDPT | $0.70_{\pm 0.01}$ | $0.59_{\pm 0.01}$ | $0.64_{\pm 0.03}$ | $0.08_{\pm 0.01}$ | $0.89_{\pm 0.01}$ | $0.08_{\pm 0.01}$ | $0.28_{\pm 0.01}$ | $0.28_{\pm 0.02}$ |
| Mitra | $0.58_{\pm 0.03}$ | $0.55_{\pm 0.03}$ | $0.47_{\pm 0.07}$ | $0.05_{\pm 0.01}$ | $0.81_{\pm 0.00}$ | $0.12_{\pm 0.03}$ | $0.24_{\pm 0.02}$ | $0.15_{\pm 0.05}$ |
| LimiX | $0.67_{\pm 0.02}$ | $0.66_{\pm 0.04}$ | $0.61_{\pm 0.10}$ | $0.09_{\pm 0.01}$ | $0.77_{\pm 0.01}$ | $0.11_{\pm 0.04}$ | $0.34_{\pm 0.03}$ | $0.34_{\pm 0.06}$ |
| TabICL | $0.77_{\pm 0.01}$ | $0.70_{\pm 0.03}$ | $0.75_{\pm 0.06}$ | $0.18_{\pm 0.01}$ | $0.67_{\pm 0.00}$ | $0.08_{\pm 0.02}$ | $0.50_{\pm 0.02}$ | $0.47_{\pm 0.04}$ |
| TabEBM | $-$ | $-$ | $-$ | $-$ | $-$ | $-$ | $-$ | $-$ |
| CTSyn | $0.76_{\pm 0.01}$ | $0.57_{\pm 0.01}$ | $0.70_{\pm 0.02}$ | $0.11_{\pm 0.00}$ | $0.84_{\pm 0.00}$ | $0.08_{\pm 0.01}$ | $0.33_{\pm 0.01}$ | $0.33_{\pm 0.01}$ |
| **TabFORGE (Ours)** | $0.97_{\pm 0.01}$ | $0.92_{\pm 0.03}$ | $0.99_{\pm 0.01}$ | $0.33_{\pm 0.02}$ | $0.76_{\pm 0.02}$ | $0.05_{\pm 0.01}$ | $0.74_{\pm 0.03}$ | $0.71_{\pm 0.03}$ |

*Table 45.* **Raw benchmark results of 23 tabular generators on the "Fiat500" dataset.** We report the mean $\pm$ std of each metric across 10 repeated data splits. For benchmark generators, "$-$" denotes failed convergence of a specific model or unexpected values in the synthetic data that caused the evaluation metric computation to crash. We highlight the **First**, **Second**, and **Third** best performances for each metric. TabFORGE generally achieves competitive performance against the benchmark generators while maintaining a reduced risk of overfitting.

| Generator | Density Estimation | | | | Privacy Preservation | | ML Efficacy | Structural Fidelity |
|---|---|---|---|---|---|---|---|---|
| | Shape $\uparrow$ | Trend $\uparrow$ | $\alpha$-precision $\uparrow$ | $\beta$-recall $\uparrow$ | Authenticity $\uparrow$ | DCR Score $\uparrow$ | Local utility $\uparrow$ | Global utility $\uparrow$ |
| **Real Data** | | | | | | | | |
| Real Data (Train) | $1.00_{\pm 0.00}$ | $1.00_{\pm 0.00}$ | $1.00_{\pm 0.00}$ | $1.00_{\pm 0.00}$ | $0.00_{\pm 0.00}$ | $0.00_{\pm 0.00}$ | $1.00_{\pm 0.00}$ | $0.89_{\pm 0.01}$ |
| Real Data (Holdout) | $0.96_{\pm 0.01}$ | $0.96_{\pm 0.02}$ | $0.97_{\pm 0.01}$ | $0.49_{\pm 0.02}$ | $0.68_{\pm 0.03}$ | $0.06_{\pm 0.00}$ | $-$ | $-$ |
| **Dataset-specific Model** | | | | | | | | |
| SMOTE | $0.94_{\pm 0.01}$ | $0.75_{\pm 0.02}$ | $0.79_{\pm 0.02}$ | $0.59_{\pm 0.02}$ | $0.52_{\pm 0.01}$ | $0.03_{\pm 0.00}$ | $0.93_{\pm 0.02}$ | $0.70_{\pm 0.01}$ |
| TabSDS | $0.67_{\pm 0.00}$ | $0.61_{\pm 0.03}$ | $0.68_{\pm 0.01}$ | $0.16_{\pm 0.01}$ | $0.63_{\pm 0.01}$ | $0.12_{\pm 0.01}$ | $0.69_{\pm 0.02}$ | $0.61_{\pm 0.01}$ |
| TVAE | $0.88_{\pm 0.01}$ | $0.77_{\pm 0.02}$ | $0.88_{\pm 0.04}$ | $0.18_{\pm 0.02}$ | $0.91_{\pm 0.01}$ | $0.18_{\pm 0.02}$ | $0.60_{\pm 0.05}$ | $0.63_{\pm 0.01}$ |
| GOGGLE | $0.60_{\pm 0.03}$ | $0.49_{\pm 0.04}$ | $0.52_{\pm 0.02}$ | $0.03_{\pm 0.01}$ | $0.99_{\pm 0.00}$ | $0.08_{\pm 0.05}$ | $0.21_{\pm 0.01}$ | $0.27_{\pm 0.01}$ |
| CTGAN | $0.74_{\pm 0.05}$ | $0.76_{\pm 0.10}$ | $0.75_{\pm 0.11}$ | $0.08_{\pm 0.03}$ | $0.96_{\pm 0.02}$ | $0.19_{\pm 0.07}$ | $0.56_{\pm 0.05}$ | $0.49_{\pm 0.05}$ |
| NFlow | $0.89_{\pm 0.02}$ | $0.77_{\pm 0.04}$ | $0.89_{\pm 0.07}$ | $0.13_{\pm 0.03}$ | $0.93_{\pm 0.02}$ | $0.24_{\pm 0.04}$ | $0.52_{\pm 0.06}$ | $0.56_{\pm 0.05}$ |
| ARF | $0.94_{\pm 0.00}$ | $0.84_{\pm 0.07}$ | $0.96_{\pm 0.01}$ | $0.23_{\pm 0.01}$ | $0.87_{\pm 0.01}$ | $0.16_{\pm 0.01}$ | $0.84_{\pm 0.04}$ | $0.52_{\pm 0.01}$ |
| TabDDPM | $0.87_{\pm 0.02}$ | $0.85_{\pm 0.06}$ | $0.87_{\pm 0.04}$ | $0.22_{\pm 0.02}$ | $0.89_{\pm 0.02}$ | $0.17_{\pm 0.02}$ | $0.84_{\pm 0.04}$ | $0.60_{\pm 0.01}$ |
| CDTD | $0.62_{\pm 0.00}$ | $0.65_{\pm 0.03}$ | $0.55_{\pm 0.02}$ | $0.13_{\pm 0.01}$ | $0.84_{\pm 0.01}$ | $0.39_{\pm 0.00}$ | $0.66_{\pm 0.02}$ | $0.40_{\pm 0.01}$ |
| TabSyn | $0.93_{\pm 0.01}$ | $0.95_{\pm 0.02}$ | $0.93_{\pm 0.04}$ | $0.32_{\pm 0.02}$ | $0.80_{\pm 0.02}$ | $0.10_{\pm 0.01}$ | $0.92_{\pm 0.03}$ | $0.74_{\pm 0.02}$ |
| TabDiff | $0.80_{\pm 0.01}$ | $0.75_{\pm 0.03}$ | $0.55_{\pm 0.03}$ | $0.09_{\pm 0.01}$ | $0.95_{\pm 0.01}$ | $0.70_{\pm 0.09}$ | $0.61_{\pm 0.04}$ | $0.69_{\pm 0.01}$ |
| NRGBoost | $0.69_{\pm 0.01}$ | $0.92_{\pm 0.04}$ | $0.84_{\pm 0.05}$ | $0.14_{\pm 0.05}$ | $0.87_{\pm 0.05}$ | $0.19_{\pm 0.06}$ | $0.87_{\pm 0.05}$ | $0.31_{\pm 0.02}$ |
| TabNAT | $0.60_{\pm 0.00}$ | $0.59_{\pm 0.02}$ | $0.41_{\pm 0.02}$ | $0.11_{\pm 0.01}$ | $0.74_{\pm 0.01}$ | $0.51_{\pm 0.03}$ | $0.52_{\pm 0.02}$ | $0.38_{\pm 0.01}$ |
| TabularARGN | $0.76_{\pm 0.00}$ | $0.66_{\pm 0.04}$ | $0.90_{\pm 0.02}$ | $0.10_{\pm 0.02}$ | $0.96_{\pm 0.01}$ | $0.33_{\pm 0.03}$ | $0.39_{\pm 0.02}$ | $0.45_{\pm 0.02}$ |
| **Foundation Model** | | | | | | | | |
| GReaT | $0.67_{\pm 0.01}$ | $0.57_{\pm 0.03}$ | $0.10_{\pm 0.02}$ | $0.00_{\pm 0.00}$ | $1.00_{\pm 0.00}$ | $1.00_{\pm 0.00}$ | $0.41_{\pm 0.04}$ | $0.02_{\pm 0.01}$ |
| TabPFN | $0.72_{\pm 0.01}$ | $0.59_{\pm 0.03}$ | $0.65_{\pm 0.03}$ | $0.10_{\pm 0.01}$ | $0.81_{\pm 0.01}$ | $0.34_{\pm 0.02}$ | $0.52_{\pm 0.04}$ | $0.45_{\pm 0.03}$ |
| TabDPT | $0.64_{\pm 0.03}$ | $0.53_{\pm 0.05}$ | $0.65_{\pm 0.05}$ | $0.06_{\pm 0.01}$ | $0.85_{\pm 0.01}$ | $0.14_{\pm 0.03}$ | $0.36_{\pm 0.02}$ | $0.44_{\pm 0.03}$ |
| Mitra | $0.70_{\pm 0.02}$ | $0.58_{\pm 0.02}$ | $0.66_{\pm 0.02}$ | $0.09_{\pm 0.01}$ | $0.84_{\pm 0.01}$ | $0.11_{\pm 0.02}$ | $0.34_{\pm 0.02}$ | $0.42_{\pm 0.02}$ |
| LimiX | $0.74_{\pm 0.01}$ | $0.64_{\pm 0.03}$ | $0.70_{\pm 0.03}$ | $0.11_{\pm 0.01}$ | $0.79_{\pm 0.01}$ | $0.17_{\pm 0.02}$ | $0.57_{\pm 0.04}$ | $0.56_{\pm 0.03}$ |
| TabICL | $0.81_{\pm 0.01}$ | $0.68_{\pm 0.03}$ | $0.90_{\pm 0.02}$ | $0.21_{\pm 0.01}$ | $0.81_{\pm 0.01}$ | $0.15_{\pm 0.01}$ | $0.62_{\pm 0.03}$ | $0.58_{\pm 0.02}$ |
| TabEBM | $0.79_{\pm 0.01}$ | $0.84_{\pm 0.06}$ | $0.88_{\pm 0.03}$ | $0.11_{\pm 0.02}$ | $0.92_{\pm 0.01}$ | $0.27_{\pm 0.01}$ | $0.82_{\pm 0.03}$ | $0.61_{\pm 0.01}$ |
| CTSyn | $0.71_{\pm 0.01}$ | $0.60_{\pm 0.02}$ | $0.43_{\pm 0.01}$ | $0.12_{\pm 0.01}$ | $0.73_{\pm 0.01}$ | $0.44_{\pm 0.01}$ | $0.50_{\pm 0.02}$ | $0.36_{\pm 0.01}$ |
| **TabFORGE (Ours)** | $0.94_{\pm 0.01}$ | $0.95_{\pm 0.02}$ | $0.95_{\pm 0.02}$ | $0.36_{\pm 0.08}$ | $0.74_{\pm 0.09}$ | $0.08_{\pm 0.02}$ | $0.87_{\pm 0.04}$ | $0.76_{\pm 0.03}$ |

*Table 46.* **Raw benchmark results of 23 tabular generators on the "Fish" dataset.** We report the mean $\pm$ std of each metric across 10 repeated data splits. For benchmark generators, "$-$" denotes failed convergence of a specific model or unexpected values in the synthetic data that caused the evaluation metric computation to crash. We highlight the **First**, **Second**, and **Third** best performances for each metric. TabFORGE generally achieves competitive performance against the benchmark generators while maintaining a reduced risk of overfitting.

| Generator | Density Estimation | | | | Privacy Preservation | | ML Efficacy | Structural Fidelity |
|---|---|---|---|---|---|---|---|---|
| | Shape ↑ | Trend ↑ | $\alpha$-precision ↑ | $\beta$-recall ↑ | Authenticity ↑ | DCR Score ↑ | Local utility ↑ | Global utility ↑ |
| **Real Data** | | | | | | | | |
| Real Data (Train) | $1.00_{\pm0.00}$ | $1.00_{\pm0.00}$ | $1.00_{\pm0.00}$ | $1.00_{\pm0.00}$ | $0.00_{\pm0.00}$ | $0.00_{\pm0.00}$ | $1.00_{\pm0.00}$ | $1.00_{\pm0.01}$ |
| Real Data (Holdout) | $0.94_{\pm0.01}$ | $0.98_{\pm0.02}$ | $0.95_{\pm0.03}$ | $0.55_{\pm0.05}$ | $0.63_{\pm0.05}$ | $0.09_{\pm0.01}$ | $-$ | $-$ |
| **Dataset-specific Model** | | | | | | | | |
| SMOTE | $0.74_{\pm0.03}$ | $0.80_{\pm0.02}$ | $0.81_{\pm0.03}$ | $0.52_{\pm0.03}$ | $0.59_{\pm0.02}$ | $0.05_{\pm0.01}$ | $0.93_{\pm0.02}$ | $0.69_{\pm0.02}$ |
| TabSDS | $0.49_{\pm0.02}$ | $0.71_{\pm0.01}$ | $0.69_{\pm0.01}$ | $0.21_{\pm0.01}$ | $0.65_{\pm0.02}$ | $0.13_{\pm0.01}$ | $0.68_{\pm0.02}$ | $0.58_{\pm0.02}$ |
| TVAE | $0.64_{\pm0.04}$ | $0.88_{\pm0.06}$ | $0.68_{\pm0.05}$ | $0.23_{\pm0.03}$ | $0.88_{\pm0.03}$ | $0.20_{\pm0.02}$ | $0.76_{\pm0.04}$ | $0.67_{\pm0.01}$ |
| GOGGLE | $0.47_{\pm0.05}$ | $0.73_{\pm0.08}$ | $0.62_{\pm0.15}$ | $0.07_{\pm0.03}$ | $0.96_{\pm0.02}$ | $0.22_{\pm0.03}$ | $0.35_{\pm0.04}$ | $0.23_{\pm0.03}$ |
| CTGAN | $0.53_{\pm0.08}$ | $0.86_{\pm0.08}$ | $0.79_{\pm0.11}$ | $0.15_{\pm0.04}$ | $0.94_{\pm0.02}$ | $0.25_{\pm0.05}$ | $0.68_{\pm0.07}$ | $0.47_{\pm0.08}$ |
| NFlow | $0.72_{\pm0.03}$ | $0.90_{\pm0.05}$ | $0.88_{\pm0.05}$ | $0.19_{\pm0.03}$ | $0.90_{\pm0.02}$ | $0.27_{\pm0.03}$ | $0.61_{\pm0.04}$ | $0.56_{\pm0.06}$ |
| ARF | $0.72_{\pm0.04}$ | $0.95_{\pm0.03}$ | $0.93_{\pm0.03}$ | $0.26_{\pm0.01}$ | $0.85_{\pm0.02}$ | $0.22_{\pm0.02}$ | $0.87_{\pm0.03}$ | $0.68_{\pm0.02}$ |
| TabDDPM | $0.64_{\pm0.03}$ | $0.95_{\pm0.02}$ | $0.76_{\pm0.05}$ | $0.23_{\pm0.04}$ | $0.88_{\pm0.02}$ | $0.23_{\pm0.02}$ | $0.81_{\pm0.05}$ | $0.69_{\pm0.02}$ |
| CDTD | $0.57_{\pm0.02}$ | $0.78_{\pm0.02}$ | $0.54_{\pm0.02}$ | $0.18_{\pm0.02}$ | $0.68_{\pm0.02}$ | $0.33_{\pm0.03}$ | $0.63_{\pm0.03}$ | $0.54_{\pm0.01}$ |
| TabSyn | $0.72_{\pm0.03}$ | $0.98_{\pm0.02}$ | $0.85_{\pm0.04}$ | $0.36_{\pm0.05}$ | $0.78_{\pm0.04}$ | $0.14_{\pm0.02}$ | $0.92_{\pm0.04}$ | $0.77_{\pm0.02}$ |
| TabDiff | $0.77_{\pm0.02}$ | $0.92_{\pm0.03}$ | $0.57_{\pm0.12}$ | $0.15_{\pm0.03}$ | $0.93_{\pm0.02}$ | $0.53_{\pm0.07}$ | $0.75_{\pm0.04}$ | $0.72_{\pm0.02}$ |
| NRGBoost | $0.69_{\pm0.01}$ | $0.83_{\pm0.04}$ | $0.83_{\pm0.04}$ | $0.19_{\pm0.03}$ | $0.82_{\pm0.04}$ | $0.13_{\pm0.03}$ | $0.84_{\pm0.04}$ | $0.32_{\pm0.02}$ |
| TabNAT | $0.50_{\pm0.01}$ | $0.83_{\pm0.02}$ | $0.40_{\pm0.05}$ | $0.13_{\pm0.02}$ | $0.72_{\pm0.01}$ | $0.46_{\pm0.04}$ | $0.61_{\pm0.02}$ | $0.39_{\pm0.01}$ |
| TabularARGN | $0.76_{\pm0.00}$ | $0.69_{\pm0.03}$ | $0.87_{\pm0.02}$ | $0.17_{\pm0.01}$ | $0.92_{\pm0.02}$ | $0.32_{\pm0.02}$ | $0.55_{\pm0.03}$ | $0.43_{\pm0.03}$ |
| **Foundation Model** | | | | | | | | |
| GReaT | $0.53_{\pm0.03}$ | $0.70_{\pm0.05}$ | $0.08_{\pm0.01}$ | $0.01_{\pm0.01}$ | $1.00_{\pm0.00}$ | $0.90_{\pm0.08}$ | $0.58_{\pm0.09}$ | $0.12_{\pm0.04}$ |
| TabPFN | $0.59_{\pm0.02}$ | $0.80_{\pm0.03}$ | $0.59_{\pm0.02}$ | $0.15_{\pm0.01}$ | $0.78_{\pm0.01}$ | $0.25_{\pm0.02}$ | $0.59_{\pm0.03}$ | $0.43_{\pm0.03}$ |
| TabDPT | $0.50_{\pm0.04}$ | $0.64_{\pm0.04}$ | $0.63_{\pm0.08}$ | $0.12_{\pm0.01}$ | $0.77_{\pm0.01}$ | $0.25_{\pm0.03}$ | $0.51_{\pm0.04}$ | $0.39_{\pm0.04}$ |
| Mitra | $0.47_{\pm0.03}$ | $0.71_{\pm0.04}$ | $0.63_{\pm0.06}$ | $0.11_{\pm0.02}$ | $0.77_{\pm0.01}$ | $0.21_{\pm0.02}$ | $0.64_{\pm0.04}$ | $0.43_{\pm0.03}$ |
| LimiX | $0.51_{\pm0.02}$ | $0.77_{\pm0.03}$ | $0.71_{\pm0.03}$ | $0.17_{\pm0.02}$ | $0.75_{\pm0.01}$ | $0.20_{\pm0.02}$ | $0.66_{\pm0.03}$ | $0.47_{\pm0.03}$ |
| TabICL | $0.66_{\pm0.03}$ | $0.79_{\pm0.02}$ | $0.72_{\pm0.02}$ | $0.22_{\pm0.02}$ | $0.72_{\pm0.02}$ | $0.17_{\pm0.02}$ | $0.69_{\pm0.02}$ | $0.54_{\pm0.02}$ |
| TabEBM | $0.82_{\pm0.01}$ | $0.95_{\pm0.03}$ | $0.87_{\pm0.03}$ | $0.16_{\pm0.03}$ | $0.89_{\pm0.03}$ | $0.27_{\pm0.02}$ | $0.87_{\pm0.02}$ | $0.65_{\pm0.02}$ |
| CTSyn | $0.52_{\pm0.02}$ | $0.65_{\pm0.03}$ | $0.43_{\pm0.03}$ | $0.14_{\pm0.02}$ | $0.71_{\pm0.01}$ | $0.43_{\pm0.04}$ | $0.58_{\pm0.03}$ | $0.38_{\pm0.01}$ |
| **TabFORGE (Ours)** | $0.72_{\pm0.03}$ | $0.95_{\pm0.02}$ | $0.83_{\pm0.04}$ | $0.27_{\pm0.04}$ | $0.85_{\pm0.02}$ | $0.20_{\pm0.02}$ | $0.87_{\pm0.03}$ | $0.75_{\pm0.02}$ |

*Table 47.* **Raw benchmark results of 23 tabular generators on the "Fitness" dataset.** We report the mean $\pm$ std of each metric across 10 repeated data splits. For benchmark generators, "$-$" denotes failed convergence of a specific model or unexpected values in the synthetic data that caused the evaluation metric computation to crash. We highlight the **First**, **Second**, and **Third** best performances for each metric. TabFORGE generally achieves competitive performance against the benchmark generators while maintaining a reduced risk of overfitting.

| Generator | Density Estimation | | | | Privacy Preservation | | ML Efficacy | Structural Fidelity |
|---|---|---|---|---|---|---|---|---|
| | Shape ↑ | Trend ↑ | $\alpha$-precision ↑ | $\beta$-recall ↑ | Authenticity ↑ | DCR Score ↑ | Local utility ↑ | Global utility ↑ |
| **Real Data** | | | | | | | | |
| Real Data (Train) | $1.00_{\pm0.00}$ | $1.00_{\pm0.00}$ | $1.00_{\pm0.00}$ | $1.00_{\pm0.00}$ | $0.00_{\pm0.00}$ | $0.00_{\pm0.00}$ | $1.00_{\pm0.00}$ | $0.76_{\pm0.07}$ |
| Real Data (Holdout) | $0.96_{\pm0.01}$ | $0.66_{\pm0.05}$ | $0.97_{\pm0.01}$ | $0.49_{\pm0.03}$ | $0.66_{\pm0.02}$ | $0.04_{\pm0.00}$ | $-$ | $-$ |
| **Dataset-specific Model** | | | | | | | | |
| SMOTE | $0.94_{\pm0.01}$ | $0.64_{\pm0.04}$ | $0.83_{\pm0.02}$ | $0.60_{\pm0.02}$ | $0.49_{\pm0.01}$ | $0.02_{\pm0.00}$ | $0.99_{\pm0.03}$ | $0.50_{\pm0.06}$ |
| TabSDS | $0.73_{\pm0.00}$ | $0.43_{\pm0.04}$ | $0.69_{\pm0.01}$ | $0.25_{\pm0.01}$ | $0.58_{\pm0.01}$ | $0.05_{\pm0.01}$ | $0.73_{\pm0.02}$ | $0.47_{\pm0.05}$ |
| TVAE | $0.88_{\pm0.01}$ | $0.59_{\pm0.04}$ | $0.85_{\pm0.04}$ | $0.23_{\pm0.03}$ | $0.85_{\pm0.02}$ | $0.11_{\pm0.02}$ | $0.87_{\pm0.06}$ | $0.61_{\pm0.07}$ |
| GOGGLE | $0.60_{\pm0.03}$ | $0.36_{\pm0.04}$ | $0.25_{\pm0.04}$ | $0.02_{\pm0.01}$ | $0.99_{\pm0.00}$ | $0.23_{\pm0.08}$ | $0.75_{\pm0.22}$ | $0.23_{\pm0.07}$ |
| CTGAN | $0.69_{\pm0.08}$ | $0.45_{\pm0.08}$ | $0.68_{\pm0.19}$ | $0.15_{\pm0.04}$ | $0.92_{\pm0.02}$ | $0.21_{\pm0.24}$ | $0.86_{\pm0.06}$ | $0.34_{\pm0.10}$ |
| NFlow | $0.88_{\pm0.03}$ | $0.61_{\pm0.08}$ | $0.90_{\pm0.09}$ | $0.14_{\pm0.03}$ | $0.92_{\pm0.01}$ | $0.25_{\pm0.05}$ | $0.74_{\pm0.04}$ | $0.34_{\pm0.07}$ |
| ARF | $0.95_{\pm0.01}$ | $0.58_{\pm0.09}$ | $0.96_{\pm0.02}$ | $0.29_{\pm0.02}$ | $0.81_{\pm0.02}$ | $0.11_{\pm0.02}$ | $0.93_{\pm0.04}$ | $0.44_{\pm0.05}$ |
| TabDDPM | $0.91_{\pm0.02}$ | $0.77_{\pm0.05}$ | $0.91_{\pm0.02}$ | $0.25_{\pm0.02}$ | $0.86_{\pm0.02}$ | $0.12_{\pm0.01}$ | $0.90_{\pm0.03}$ | $0.50_{\pm0.06}$ |
| CDTD | $0.79_{\pm0.00}$ | $0.48_{\pm0.04}$ | $0.70_{\pm0.02}$ | $0.22_{\pm0.01}$ | $0.69_{\pm0.01}$ | $0.10_{\pm0.02}$ | $0.75_{\pm0.01}$ | $0.46_{\pm0.05}$ |
| TabSyn | $0.93_{\pm0.01}$ | $0.72_{\pm0.12}$ | $0.95_{\pm0.03}$ | $0.42_{\pm0.02}$ | $0.70_{\pm0.03}$ | $0.05_{\pm0.01}$ | $0.98_{\pm0.03}$ | $0.66_{\pm0.08}$ |
| TabDiff | $0.88_{\pm0.01}$ | $0.65_{\pm0.02}$ | $0.82_{\pm0.03}$ | $0.09_{\pm0.01}$ | $0.95_{\pm0.01}$ | $0.33_{\pm0.05}$ | $0.79_{\pm0.06}$ | $0.58_{\pm0.07}$ |
| NRGBoost | $0.93_{\pm0.01}$ | $0.79_{\pm0.07}$ | $0.94_{\pm0.02}$ | $0.43_{\pm0.03}$ | $0.60_{\pm0.03}$ | $0.02_{\pm0.00}$ | $0.98_{\pm0.03}$ | $0.33_{\pm0.04}$ |
| TabNAT | $0.70_{\pm0.00}$ | $0.51_{\pm0.03}$ | $0.62_{\pm0.02}$ | $0.16_{\pm0.01}$ | $0.74_{\pm0.01}$ | $0.17_{\pm0.03}$ | $0.67_{\pm0.02}$ | $0.31_{\pm0.05}$ |
| TabularARGN | $0.82_{\pm0.01}$ | $0.60_{\pm0.04}$ | $0.95_{\pm0.02}$ | $0.30_{\pm0.04}$ | $0.81_{\pm0.04}$ | $0.13_{\pm0.07}$ | $0.89_{\pm0.05}$ | $0.45_{\pm0.08}$ |
| **Foundation Model** | | | | | | | | |
| GReaT | $0.79_{\pm0.01}$ | $0.36_{\pm0.04}$ | $0.63_{\pm0.04}$ | $0.10_{\pm0.02}$ | $0.95_{\pm0.01}$ | $0.23_{\pm0.04}$ | $0.74_{\pm0.04}$ | $0.29_{\pm0.07}$ |
| TabPFN | $0.74_{\pm0.02}$ | $0.47_{\pm0.03}$ | $0.79_{\pm0.04}$ | $0.14_{\pm0.01}$ | $0.78_{\pm0.01}$ | $0.16_{\pm0.03}$ | $0.74_{\pm0.03}$ | $0.37_{\pm0.05}$ |
| TabDPT | $0.62_{\pm0.04}$ | $0.39_{\pm0.04}$ | $0.44_{\pm0.08}$ | $0.12_{\pm0.01}$ | $0.80_{\pm0.01}$ | $0.17_{\pm0.11}$ | $0.88_{\pm0.09}$ | $0.35_{\pm0.06}$ |
| Mitra | $0.64_{\pm0.02}$ | $0.41_{\pm0.04}$ | $0.51_{\pm0.04}$ | $0.07_{\pm0.01}$ | $0.81_{\pm0.01}$ | $0.19_{\pm0.05}$ | $0.87_{\pm0.17}$ | $0.31_{\pm0.05}$ |
| LimiX | $0.77_{\pm0.02}$ | $0.46_{\pm0.03}$ | $0.69_{\pm0.04}$ | $0.15_{\pm0.02}$ | $0.81_{\pm0.01}$ | $0.17_{\pm0.03}$ | $0.74_{\pm0.07}$ | $0.28_{\pm0.04}$ |
| TabICL | $0.80_{\pm0.01}$ | $0.49_{\pm0.04}$ | $0.80_{\pm0.02}$ | $0.26_{\pm0.02}$ | $0.67_{\pm0.01}$ | $0.10_{\pm0.02}$ | $0.77_{\pm0.02}$ | $0.43_{\pm0.05}$ |
| TabEBM | $0.93_{\pm0.01}$ | $0.59_{\pm0.07}$ | $0.90_{\pm0.05}$ | $0.18_{\pm0.03}$ | $0.84_{\pm0.02}$ | $0.12_{\pm0.02}$ | $0.94_{\pm0.04}$ | $0.41_{\pm0.06}$ |
| CTSyn | $0.69_{\pm0.01}$ | $0.45_{\pm0.04}$ | $0.63_{\pm0.02}$ | $0.17_{\pm0.01}$ | $0.72_{\pm0.01}$ | $0.12_{\pm0.03}$ | $0.75_{\pm0.06}$ | $0.38_{\pm0.03}$ |
| **TabFORGE (Ours)** | $0.89_{\pm0.01}$ | $0.65_{\pm0.06}$ | $0.94_{\pm0.03}$ | $0.21_{\pm0.02}$ | $0.84_{\pm0.02}$ | $0.13_{\pm0.02}$ | $1.00_{\pm0.05}$ | $0.55_{\pm0.08}$ |

*Table 48.* **Raw benchmark results of 23 tabular generators on the "FoodDelivery" dataset.** We report the mean $\pm$ std of each metric across 10 repeated data splits. For benchmark generators, "$-$" denotes failed convergence of a specific model or unexpected values in the synthetic data that caused the evaluation metric computation to crash. We highlight the **First**, **Second**, and **Third** best performances for each metric. TabFORGE generally achieves competitive performance against the benchmark generators while maintaining a reduced risk of overfitting.

| Generator | Density Estimation | | | | Privacy Preservation | | ML Efficacy | Structural Fidelity |
|---|---|---|---|---|---|---|---|---|
| | Shape ↑ | Trend ↑ | $\alpha$-precision ↑ | $\beta$-recall ↑ | Authenticity ↑ | DCR Score ↑ | Local utility ↑ | Global utility ↑ |
| **Real Data** | | | | | | | | |
| Real Data (Train) | $1.00_{\pm 0.00}$ | $1.00_{\pm 0.00}$ | $1.00_{\pm 0.00}$ | $1.00_{\pm 0.00}$ | $0.00_{\pm 0.00}$ | $0.00_{\pm 0.00}$ | $1.00_{\pm 0.00}$ | $0.71_{\pm 0.00}$ |
| Real Data (Holdout) | $0.98_{\pm 0.00}$ | $0.94_{\pm 0.00}$ | $0.99_{\pm 0.00}$ | $0.49_{\pm 0.00}$ | $0.63_{\pm 0.01}$ | $0.25_{\pm 0.00}$ | $-$ | $-$ |
| **Dataset-specific Model** | | | | | | | | |
| SMOTE | $0.95_{\pm 0.00}$ | $-$ | $-$ | $0.61_{\pm 0.00}$ | $0.44_{\pm 0.00}$ | $0.41_{\pm 0.00}$ | $0.94_{\pm 0.00}$ | $-$ |
| TabSDS | $-$ | $-$ | $-$ | $-$ | $-$ | $-$ | $-$ | $-$ |
| TVAE | $0.89_{\pm 0.01}$ | $0.73_{\pm 0.02}$ | $0.98_{\pm 0.01}$ | $0.30_{\pm 0.01}$ | $0.81_{\pm 0.01}$ | $0.43_{\pm 0.01}$ | $0.92_{\pm 0.01}$ | $0.43_{\pm 0.01}$ |
| GOGGLE | $0.55_{\pm 0.02}$ | $0.40_{\pm 0.01}$ | $0.38_{\pm 0.08}$ | $0.05_{\pm 0.02}$ | $0.98_{\pm 0.01}$ | $0.42_{\pm 0.01}$ | $0.44_{\pm 0.02}$ | $0.12_{\pm 0.02}$ |
| CTGAN | $0.85_{\pm 0.02}$ | $0.77_{\pm 0.02}$ | $0.89_{\pm 0.09}$ | $0.29_{\pm 0.03}$ | $0.83_{\pm 0.02}$ | $0.43_{\pm 0.01}$ | $0.90_{\pm 0.01}$ | $0.31_{\pm 0.02}$ |
| NFlow | $0.83_{\pm 0.02}$ | $0.64_{\pm 0.04}$ | $0.85_{\pm 0.07}$ | $0.24_{\pm 0.03}$ | $0.87_{\pm 0.02}$ | $0.47_{\pm 0.02}$ | $0.73_{\pm 0.02}$ | $0.27_{\pm 0.03}$ |
| ARF | $-$ | $-$ | $-$ | $-$ | $-$ | $-$ | $-$ | $-$ |
| TabDDPM | $0.95_{\pm 0.00}$ | $0.85_{\pm 0.00}$ | $0.99_{\pm 0.00}$ | $0.35_{\pm 0.00}$ | $0.77_{\pm 0.00}$ | $0.43_{\pm 0.00}$ | $0.94_{\pm 0.00}$ | $0.35_{\pm 0.01}$ |
| CDTD | $-$ | $-$ | $-$ | $-$ | $-$ | $-$ | $-$ | $-$ |
| TabSyn | $0.93_{\pm 0.00}$ | $0.79_{\pm 0.03}$ | $0.95_{\pm 0.02}$ | $0.35_{\pm 0.01}$ | $0.78_{\pm 0.00}$ | $0.43_{\pm 0.00}$ | $0.94_{\pm 0.01}$ | $0.43_{\pm 0.02}$ |
| TabDiff | $0.92_{\pm 0.00}$ | $0.65_{\pm 0.01}$ | $0.78_{\pm 0.02}$ | $0.22_{\pm 0.01}$ | $0.89_{\pm 0.01}$ | $0.47_{\pm 0.01}$ | $0.87_{\pm 0.01}$ | $0.46_{\pm 0.01}$ |
| NRGBoost | $-$ | $-$ | $-$ | $-$ | $-$ | $-$ | $-$ | $-$ |
| TabNAT | $0.69_{\pm 0.00}$ | $0.56_{\pm 0.01}$ | $0.66_{\pm 0.01}$ | $0.18_{\pm 0.00}$ | $0.73_{\pm 0.00}$ | $0.52_{\pm 0.00}$ | $0.75_{\pm 0.00}$ | $0.29_{\pm 0.01}$ |
| TabularARGN | $0.82_{\pm 0.00}$ | $0.74_{\pm 0.02}$ | $0.89_{\pm 0.02}$ | $0.12_{\pm 0.00}$ | $0.93_{\pm 0.00}$ | $0.43_{\pm 0.00}$ | $0.78_{\pm 0.01}$ | $0.16_{\pm 0.02}$ |
| **Foundation Model** | | | | | | | | |
| GReaT | $0.67_{\pm 0.00}$ | $0.43_{\pm 0.04}$ | $0.42_{\pm 0.01}$ | $0.02_{\pm 0.00}$ | $0.99_{\pm 0.00}$ | $1.00_{\pm 0.00}$ | $0.80_{\pm 0.01}$ | $0.38_{\pm 0.01}$ |
| TabPFN | $-$ | $-$ | $-$ | $-$ | $-$ | $-$ | $-$ | $-$ |
| TabDPT | $0.67_{\pm 0.01}$ | $0.56_{\pm 0.01}$ | $0.69_{\pm 0.05}$ | $0.14_{\pm 0.01}$ | $0.77_{\pm 0.01}$ | $0.38_{\pm 0.00}$ | $0.77_{\pm 0.01}$ | $0.20_{\pm 0.02}$ |
| Mitra | $0.66_{\pm 0.01}$ | $0.42_{\pm 0.01}$ | $0.44_{\pm 0.03}$ | $0.13_{\pm 0.01}$ | $0.81_{\pm 0.01}$ | $0.35_{\pm 0.01}$ | $0.72_{\pm 0.02}$ | $0.22_{\pm 0.02}$ |
| LimiX | $-$ | $-$ | $-$ | $-$ | $-$ | $-$ | $-$ | $-$ |
| TabICL | $-$ | $-$ | $-$ | $-$ | $-$ | $-$ | $-$ | $-$ |
| TabEBM | $-$ | $-$ | $-$ | $-$ | $-$ | $-$ | $-$ | $-$ |
| CTSyn | $0.60_{\pm 0.00}$ | $0.47_{\pm 0.02}$ | $0.58_{\pm 0.01}$ | $0.13_{\pm 0.00}$ | $0.74_{\pm 0.00}$ | $0.52_{\pm 0.00}$ | $0.74_{\pm 0.01}$ | $0.34_{\pm 0.01}$ |
| **TabFORGE (Ours)** | $0.95_{\pm 0.00}$ | $0.85_{\pm 0.00}$ | $0.99_{\pm 0.00}$ | $0.35_{\pm 0.00}$ | $0.77_{\pm 0.00}$ | $0.42_{\pm 0.00}$ | $0.91_{\pm 0.01}$ | $0.41_{\pm 0.03}$ |

*Table 49.* **Raw benchmark results of 23 tabular generators on the "Give" dataset.** We report the mean $\pm$ std of each metric across 10 repeated data splits. For benchmark generators, "−" denotes failed convergence of a specific model or unexpected values in the synthetic data that caused the evaluation metric computation to crash. We highlight the **First**, **Second**, and **Third** best performances for each metric. TabFORGE generally achieves competitive performance against the benchmark generators while maintaining a reduced risk of overfitting.

| Generator | Density Estimation | | | | Privacy Preservation | | ML Efficacy | Structural Fidelity |
|---|---|---|---|---|---|---|---|---|
| | Shape ↑ | Trend ↑ | $\alpha$-precision ↑ | $\beta$-recall ↑ | Authenticity ↑ | DCR Score ↑ | Local utility ↑ | Global utility ↑ |
| **Real Data** | | | | | | | | |
| Real Data (Train) | $1.00_{\pm0.00}$ | $1.00_{\pm0.00}$ | $1.00_{\pm0.00}$ | $1.00_{\pm0.00}$ | $0.00_{\pm0.00}$ | $0.00_{\pm0.00}$ | $1.00_{\pm0.00}$ | $0.95_{\pm0.01}$ |
| Real Data (Holdout) | $1.00_{\pm0.00}$ | $1.00_{\pm0.00}$ | $1.00_{\pm0.00}$ | $0.50_{\pm0.00}$ | $0.67_{\pm0.00}$ | $0.00_{\pm0.00}$ | − | − |
| **Dataset-specific Model** | | | | | | | | |
| SMOTE | $0.64_{\pm0.00}$ | − | − | $\mathbf{0.38}_{\pm0.01}$ | $0.71_{\pm0.01}$ | $0.01_{\pm0.00}$ | $\mathbf{1.04}_{\pm0.01}$ | − |
| TabSDS | $0.54_{\pm0.00}$ | $0.47_{\pm0.02}$ | $0.73_{\pm0.01}$ | $0.13_{\pm0.01}$ | $0.63_{\pm0.00}$ | $0.01_{\pm0.00}$ | $0.70_{\pm0.01}$ | $0.48_{\pm0.03}$ |
| TVAE | $0.64_{\pm0.03}$ | $0.58_{\pm0.08}$ | $0.96_{\pm0.02}$ | $0.11_{\pm0.02}$ | $0.94_{\pm0.01}$ | $0.01_{\pm0.00}$ | $0.99_{\pm0.04}$ | $0.69_{\pm0.02}$ |
| GOGGLE | $0.36_{\pm0.04}$ | $0.51_{\pm0.00}$ | $0.54_{\pm0.14}$ | $0.00_{\pm0.00}$ | $\mathbf{1.00}_{\pm0.00}$ | $0.00_{\pm0.00}$ | − | − |
| CTGAN | $0.63_{\pm0.03}$ | $0.51_{\pm0.00}$ | $0.96_{\pm0.02}$ | $0.11_{\pm0.02}$ | $0.94_{\pm0.01}$ | $0.01_{\pm0.00}$ | $0.97_{\pm0.05}$ | $0.49_{\pm0.04}$ |
| NFlow | $0.63_{\pm0.04}$ | $0.64_{\pm0.04}$ | $0.80_{\pm0.07}$ | $0.09_{\pm0.01}$ | $0.96_{\pm0.01}$ | $\mathbf{0.02}_{\pm0.01}$ | $0.74_{\pm0.05}$ | $0.34_{\pm0.11}$ |
| ARF | $\mathbf{0.70}_{\pm0.01}$ | $0.78_{\pm0.03}$ | $0.97_{\pm0.01}$ | $0.09_{\pm0.02}$ | $0.96_{\pm0.01}$ | $0.01_{\pm0.00}$ | $0.94_{\pm0.01}$ | $0.58_{\pm0.03}$ |
| TabDDPM | $0.64_{\pm0.00}$ | $\mathbf{0.96}_{\pm0.03}$ | $\mathbf{0.98}_{\pm0.00}$ | $\mathbf{0.38}_{\pm0.00}$ | $0.77_{\pm0.00}$ | $0.01_{\pm0.00}$ | $1.00_{\pm0.01}$ | $0.68_{\pm0.01}$ |
| CDTD | − | − | − | − | − | − | − | − |
| TabSyn | $0.63_{\pm0.00}$ | $0.69_{\pm0.06}$ | $0.95_{\pm0.03}$ | $0.37_{\pm0.00}$ | $0.78_{\pm0.00}$ | $0.01_{\pm0.00}$ | $\mathbf{1.02}_{\pm0.04}$ | $\mathbf{0.70}_{\pm0.05}$ |
| TabDiff | $0.59_{\pm0.02}$ | $0.51_{\pm0.01}$ | $0.80_{\pm0.09}$ | $0.22_{\pm0.08}$ | $0.86_{\pm0.05}$ | $0.01_{\pm0.01}$ | $0.89_{\pm0.02}$ | $\mathbf{0.71}_{\pm0.02}$ |
| NRGBoost | $0.63_{\pm0.00}$ | $0.61_{\pm0.05}$ | $0.93_{\pm0.01}$ | $0.00_{\pm0.00}$ | $\mathbf{1.00}_{\pm0.00}$ | $\mathbf{0.03}_{\pm0.01}$ | $0.97_{\pm0.02}$ | − |
| TabNAT | − | − | − | − | − | − | − | − |
| TabularARGN | $\mathbf{0.78}_{\pm0.00}$ | $\mathbf{0.95}_{\pm0.12}$ | $\mathbf{0.98}_{\pm0.01}$ | $0.02_{\pm0.00}$ | $\mathbf{0.99}_{\pm0.00}$ | $0.01_{\pm0.00}$ | $0.98_{\pm0.02}$ | $0.49_{\pm0.09}$ |
| **Foundation Model** | | | | | | | | |
| GReaT | − | − | − | − | − | − | − | − |
| TabPFN | − | − | − | − | − | − | − | − |
| TabDPT | $0.48_{\pm0.01}$ | $0.51_{\pm0.02}$ | $0.69_{\pm0.04}$ | $0.05_{\pm0.01}$ | $0.79_{\pm0.00}$ | $0.01_{\pm0.00}$ | $0.87_{\pm0.02}$ | $0.39_{\pm0.03}$ |
| Mitra | $0.44_{\pm0.02}$ | $0.43_{\pm0.01}$ | $0.54_{\pm0.07}$ | $0.07_{\pm0.01}$ | $0.81_{\pm0.00}$ | $0.01_{\pm0.00}$ | $0.75_{\pm0.02}$ | $0.33_{\pm0.04}$ |
| LimiX | $0.55_{\pm0.02}$ | $0.52_{\pm0.01}$ | $0.67_{\pm0.04}$ | $0.06_{\pm0.01}$ | $0.79_{\pm0.00}$ | $\mathbf{0.01}_{\pm0.00}$ | $0.77_{\pm0.02}$ | $0.38_{\pm0.06}$ |
| TabICL | $0.55_{\pm0.01}$ | $0.61_{\pm0.01}$ | $0.80_{\pm0.02}$ | $0.15_{\pm0.01}$ | $0.81_{\pm0.00}$ | $0.01_{\pm0.00}$ | $0.83_{\pm0.02}$ | $0.50_{\pm0.04}$ |
| TabEBM | − | − | − | − | − | − | − | − |
| CTSyn | − | − | − | − | − | − | − | − |
| **TabFORGE (Ours)** | $0.62_{\pm0.01}$ | $\mathbf{0.97}_{\pm0.01}$ | $\mathbf{0.99}_{\pm0.01}$ | $\mathbf{0.41}_{\pm0.04}$ | $0.68_{\pm0.04}$ | $0.01_{\pm0.00}$ | $\mathbf{1.01}_{\pm0.01}$ | $0.61_{\pm0.05}$ |

*Table 50.* **Raw benchmark results of 23 tabular generators on the "HELOC" dataset.** We report the mean $\pm$ std of each metric across 10 repeated data splits. For benchmark generators, "$-$" denotes failed convergence of a specific model or unexpected values in the synthetic data that caused the evaluation metric computation to crash. We highlight the **First**, **Second**, and **Third** best performances for each metric. TabFORGE generally achieves competitive performance against the benchmark generators while maintaining a reduced risk of overfitting.

| Generator | Density Estimation | | | | Privacy Preservation | | ML Efficacy | Structural Fidelity |
|---|---|---|---|---|---|---|---|---|
| | Shape ↑ | Trend ↑ | $\alpha$-precision ↑ | $\beta$-recall ↑ | Authenticity ↑ | DCR Score ↑ | Local utility ↑ | Global utility ↑ |
| *Real Data* | | | | | | | | |
| Real Data (Train) | $1.00_{\pm 0.00}$ | $1.00_{\pm 0.00}$ | $1.00_{\pm 0.00}$ | $1.00_{\pm 0.00}$ | $0.00_{\pm 0.00}$ | $0.00_{\pm 0.00}$ | $1.00_{\pm 0.00}$ | $1.00_{\pm 0.00}$ |
| Real Data (Holdout) | $0.99_{\pm 0.00}$ | $0.99_{\pm 0.00}$ | $0.99_{\pm 0.01}$ | $0.49_{\pm 0.01}$ | $0.71_{\pm 0.01}$ | $0.08_{\pm 0.00}$ | $-$ | $-$ |
| *Dataset-specific Model* | | | | | | | | |
| SMOTE | $0.82_{\pm 0.01}$ | $0.84_{\pm 0.00}$ | $0.77_{\pm 0.01}$ | $\mathbf{0.72}_{\pm 0.01}$ | $0.43_{\pm 0.01}$ | $0.05_{\pm 0.00}$ | $\mathbf{0.98}_{\pm 0.01}$ | $0.60_{\pm 0.01}$ |
| TabSDS | $0.65_{\pm 0.00}$ | $0.71_{\pm 0.00}$ | $0.66_{\pm 0.01}$ | $0.20_{\pm 0.01}$ | $0.61_{\pm 0.00}$ | $0.12_{\pm 0.01}$ | $0.68_{\pm 0.00}$ | $0.52_{\pm 0.01}$ |
| TVAE | $0.81_{\pm 0.01}$ | $0.85_{\pm 0.02}$ | $0.83_{\pm 0.04}$ | $0.22_{\pm 0.02}$ | $0.92_{\pm 0.01}$ | $0.16_{\pm 0.01}$ | $0.93_{\pm 0.02}$ | $0.60_{\pm 0.01}$ |
| GOGGLE | $0.49_{\pm 0.02}$ | $0.67_{\pm 0.00}$ | $0.25_{\pm 0.05}$ | $0.03_{\pm 0.01}$ | $0.99_{\pm 0.00}$ | $0.11_{\pm 0.01}$ | $0.45_{\pm 0.02}$ | $0.06_{\pm 0.01}$ |
| CTGAN | $0.76_{\pm 0.03}$ | $0.94_{\pm 0.01}$ | $\mathbf{0.92}_{\pm 0.05}$ | $0.16_{\pm 0.02}$ | $0.94_{\pm 0.01}$ | $0.17_{\pm 0.01}$ | $0.92_{\pm 0.03}$ | $0.53_{\pm 0.04}$ |
| NFlow | $0.79_{\pm 0.02}$ | $0.73_{\pm 0.02}$ | $0.74_{\pm 0.12}$ | $0.04_{\pm 0.02}$ | $\mathbf{0.99}_{\pm 0.00}$ | $\mathbf{0.24}_{\pm 0.02}$ | $0.77_{\pm 0.03}$ | $0.29_{\pm 0.04}$ |
| ARF | $\mathbf{0.85}_{\pm 0.00}$ | $\mathbf{0.97}_{\pm 0.00}$ | $\mathbf{0.93}_{\pm 0.01}$ | $0.13_{\pm 0.01}$ | $0.96_{\pm 0.00}$ | $0.19_{\pm 0.01}$ | $\mathbf{0.96}_{\pm 0.01}$ | $0.58_{\pm 0.01}$ |
| TabDDPM | $0.74_{\pm 0.02}$ | $0.89_{\pm 0.03}$ | $0.78_{\pm 0.03}$ | $0.27_{\pm 0.01}$ | $0.89_{\pm 0.01}$ | $0.15_{\pm 0.01}$ | $0.94_{\pm 0.01}$ | $0.67_{\pm 0.01}$ |
| CDTD | $0.65_{\pm 0.00}$ | $0.73_{\pm 0.01}$ | $0.65_{\pm 0.01}$ | $0.18_{\pm 0.01}$ | $0.75_{\pm 0.01}$ | $0.13_{\pm 0.01}$ | $0.73_{\pm 0.01}$ | $0.49_{\pm 0.01}$ |
| TabSyn | $0.80_{\pm 0.01}$ | $\mathbf{0.96}_{\pm 0.01}$ | $0.81_{\pm 0.03}$ | $\mathbf{0.47}_{\pm 0.02}$ | $0.79_{\pm 0.01}$ | $0.12_{\pm 0.01}$ | $\mathbf{0.98}_{\pm 0.01}$ | $\mathbf{0.76}_{\pm 0.02}$ |
| TabDiff | $0.75_{\pm 0.01}$ | $0.70_{\pm 0.00}$ | $0.61_{\pm 0.02}$ | $0.02_{\pm 0.00}$ | $\mathbf{1.00}_{\pm 0.00}$ | $\mathbf{0.28}_{\pm 0.02}$ | $0.79_{\pm 0.05}$ | $\mathbf{0.68}_{\pm 0.01}$ |
| NRGBoost | $0.73_{\pm 0.01}$ | $0.81_{\pm 0.01}$ | $0.85_{\pm 0.03}$ | $0.13_{\pm 0.01}$ | $0.95_{\pm 0.00}$ | $0.18_{\pm 0.01}$ | $0.92_{\pm 0.02}$ | $0.24_{\pm 0.01}$ |
| TabNAT | $0.65_{\pm 0.01}$ | $0.66_{\pm 0.00}$ | $0.63_{\pm 0.01}$ | $0.11_{\pm 0.01}$ | $0.87_{\pm 0.00}$ | $0.16_{\pm 0.01}$ | $0.74_{\pm 0.02}$ | $0.30_{\pm 0.01}$ |
| TabularARGN | $\mathbf{0.85}_{\pm 0.01}$ | $0.93_{\pm 0.04}$ | $\mathbf{0.94}_{\pm 0.01}$ | $0.13_{\pm 0.01}$ | $0.90_{\pm 0.02}$ | $0.14_{\pm 0.01}$ | $0.95_{\pm 0.02}$ | $0.57_{\pm 0.02}$ |
| *Foundation Model* | | | | | | | | |
| GReaT | $0.66_{\pm 0.00}$ | $0.69_{\pm 0.00}$ | $0.75_{\pm 0.03}$ | $0.01_{\pm 0.00}$ | $\mathbf{1.00}_{\pm 0.00}$ | $0.19_{\pm 0.01}$ | $0.70_{\pm 0.02}$ | $0.10_{\pm 0.01}$ |
| TabPFN | $0.70_{\pm 0.01}$ | $0.66_{\pm 0.01}$ | $0.72_{\pm 0.06}$ | $0.05_{\pm 0.01}$ | $0.81_{\pm 0.00}$ | $0.17_{\pm 0.01}$ | $0.73_{\pm 0.02}$ | $0.29_{\pm 0.02}$ |
| TabDPT | $0.58_{\pm 0.01}$ | $0.67_{\pm 0.01}$ | $0.67_{\pm 0.03}$ | $0.10_{\pm 0.01}$ | $0.79_{\pm 0.01}$ | $0.13_{\pm 0.01}$ | $0.79_{\pm 0.01}$ | $0.37_{\pm 0.02}$ |
| Mitra | $0.57_{\pm 0.01}$ | $0.62_{\pm 0.01}$ | $0.37_{\pm 0.04}$ | $0.05_{\pm 0.01}$ | $0.85_{\pm 0.00}$ | $0.15_{\pm 0.01}$ | $0.70_{\pm 0.02}$ | $0.15_{\pm 0.02}$ |
| LimiX | $0.61_{\pm 0.01}$ | $0.65_{\pm 0.01}$ | $0.60_{\pm 0.05}$ | $0.05_{\pm 0.01}$ | $0.84_{\pm 0.00}$ | $0.18_{\pm 0.01}$ | $0.69_{\pm 0.01}$ | $0.28_{\pm 0.02}$ |
| TabICL | $0.63_{\pm 0.01}$ | $0.68_{\pm 0.01}$ | $0.69_{\pm 0.03}$ | $0.19_{\pm 0.01}$ | $0.80_{\pm 0.00}$ | $0.16_{\pm 0.01}$ | $0.80_{\pm 0.01}$ | $0.52_{\pm 0.01}$ |
| TabEBM | $0.83_{\pm 0.00}$ | $0.95_{\pm 0.02}$ | $0.78_{\pm 0.02}$ | $0.07_{\pm 0.00}$ | $0.95_{\pm 0.00}$ | $\mathbf{0.19}_{\pm 0.01}$ | $0.94_{\pm 0.01}$ | $0.46_{\pm 0.01}$ |
| CTSyn | $0.57_{\pm 0.01}$ | $0.70_{\pm 0.01}$ | $0.60_{\pm 0.02}$ | $0.18_{\pm 0.01}$ | $0.75_{\pm 0.01}$ | $0.13_{\pm 0.01}$ | $0.71_{\pm 0.01}$ | $0.35_{\pm 0.01}$ |
| **TabFORGE (Ours)** | $0.78_{\pm 0.01}$ | $0.93_{\pm 0.01}$ | $0.85_{\pm 0.06}$ | $\mathbf{0.33}_{\pm 0.01}$ | $0.86_{\pm 0.01}$ | $0.14_{\pm 0.01}$ | $0.91_{\pm 0.02}$ | $\mathbf{0.73}_{\pm 0.01}$ |

*Table 51.* **Raw benchmark results of 23 tabular generators on the "HR" dataset.** We report the mean $\pm$ std of each metric across 10 repeated data splits. For benchmark generators, "−" denotes failed convergence of a specific model or unexpected values in the synthetic data that caused the evaluation metric computation to crash. We highlight the **First**, **Second**, and **Third** best performances for each metric. TabFORGE generally achieves competitive performance against the benchmark generators while maintaining a reduced risk of overfitting.

| Generator | Density Estimation | | | | Privacy Preservation | | ML Efficacy | Structural Fidelity |
| --- | --- | --- | --- | --- | --- | --- | --- | --- |
| | Shape ↑ | Trend ↑ | $\alpha$-precision ↑ | $\beta$-recall ↑ | Authenticity ↑ | DCR Score ↑ | Local utility ↑ | Global utility ↑ |
| **Real Data** | | | | | | | | |
| Real Data (Train) | $1.00_{\pm0.00}$ | $1.00_{\pm0.00}$ | $1.00_{\pm0.00}$ | $1.00_{\pm0.00}$ | $0.00_{\pm0.00}$ | $0.00_{\pm0.00}$ | $1.00_{\pm0.00}$ | $0.72_{\pm0.09}$ |
| Real Data (Holdout) | $0.99_{\pm0.00}$ | $0.97_{\pm0.00}$ | $0.99_{\pm0.01}$ | $0.50_{\pm0.01}$ | $0.77_{\pm0.00}$ | $0.23_{\pm0.00}$ | — | — |
| **Dataset-specific Model** | | | | | | | | |
| SMOTE | $0.89_{\pm0.00}$ | $0.67_{\pm0.00}$ | $0.60_{\pm0.01}$ | $0.49_{\pm0.01}$ | $0.64_{\pm0.01}$ | $0.37_{\pm0.00}$ | $0.96_{\pm0.01}$ | $0.33_{\pm0.05}$ |
| TabSDS | — | — | — | — | — | — | — | — |
| TVAE | $0.82_{\pm0.01}$ | $0.70_{\pm0.02}$ | $0.72_{\pm0.03}$ | $0.17_{\pm0.02}$ | $0.94_{\pm0.01}$ | $0.54_{\pm0.02}$ | $0.87_{\pm0.03}$ | $0.29_{\pm0.04}$ |
| GOGGLE | $0.52_{\pm0.01}$ | $0.31_{\pm0.02}$ | $0.07_{\pm0.01}$ | $0.00_{\pm0.00}$ | $1.00_{\pm0.00}$ | $0.37_{\pm0.11}$ | $0.78_{\pm0.17}$ | $0.26_{\pm0.08}$ |
| CTGAN | $0.80_{\pm0.01}$ | $0.69_{\pm0.02}$ | $0.86_{\pm0.05}$ | $0.13_{\pm0.02}$ | $0.95_{\pm0.01}$ | $0.57_{\pm0.01}$ | $0.91_{\pm0.04}$ | $0.27_{\pm0.05}$ |
| NFlow | $0.79_{\pm0.02}$ | $0.66_{\pm0.03}$ | $0.88_{\pm0.06}$ | $0.13_{\pm0.02}$ | $0.96_{\pm0.01}$ | $0.59_{\pm0.06}$ | $0.73_{\pm0.03}$ | $0.21_{\pm0.04}$ |
| ARF | — | — | — | — | — | — | — | — |
| TabDDPM | $0.67_{\pm0.04}$ | $0.46_{\pm0.03}$ | $0.70_{\pm0.07}$ | $0.09_{\pm0.01}$ | $0.97_{\pm0.00}$ | $0.87_{\pm0.08}$ | $0.80_{\pm0.02}$ | $0.29_{\pm0.04}$ |
| CDTD | $0.76_{\pm0.01}$ | $0.57_{\pm0.01}$ | $0.58_{\pm0.01}$ | $0.13_{\pm0.00}$ | $0.85_{\pm0.00}$ | $0.43_{\pm0.02}$ | $0.76_{\pm0.01}$ | $0.29_{\pm0.03}$ |
| TabSyn | $0.88_{\pm0.01}$ | $0.78_{\pm0.01}$ | $0.72_{\pm0.02}$ | $0.24_{\pm0.01}$ | $0.91_{\pm0.00}$ | $0.46_{\pm0.04}$ | $0.92_{\pm0.02}$ | $0.31_{\pm0.05}$ |
| TabDiff | $0.79_{\pm0.01}$ | $0.57_{\pm0.01}$ | $0.57_{\pm0.02}$ | $0.06_{\pm0.00}$ | $0.98_{\pm0.00}$ | $0.76_{\pm0.01}$ | $0.81_{\pm0.01}$ | $0.34_{\pm0.04}$ |
| NRGBoost | $0.87_{\pm0.00}$ | $0.76_{\pm0.01}$ | $0.78_{\pm0.02}$ | $0.11_{\pm0.00}$ | $0.96_{\pm0.00}$ | $0.56_{\pm0.00}$ | $0.92_{\pm0.02}$ | $0.15_{\pm0.02}$ |
| TabNAT | $0.62_{\pm0.00}$ | $0.57_{\pm0.01}$ | $0.44_{\pm0.01}$ | $0.12_{\pm0.00}$ | $0.78_{\pm0.00}$ | $0.45_{\pm0.01}$ | $0.70_{\pm0.01}$ | $0.23_{\pm0.02}$ |
| TabularARGN | $0.74_{\pm0.00}$ | $0.82_{\pm0.01}$ | $0.80_{\pm0.02}$ | $0.08_{\pm0.01}$ | $0.97_{\pm0.00}$ | $0.47_{\pm0.04}$ | $0.90_{\pm0.02}$ | $0.30_{\pm0.05}$ |
| **Foundation Model** | | | | | | | | |
| GReaT | $0.79_{\pm0.00}$ | $0.58_{\pm0.04}$ | $0.43_{\pm0.01}$ | $0.15_{\pm0.01}$ | $0.95_{\pm0.00}$ | $0.44_{\pm0.03}$ | $0.76_{\pm0.01}$ | $0.48_{\pm0.05}$ |
| TabPFN | — | — | — | — | — | — | — | — |
| TabDPT | $0.62_{\pm0.01}$ | $0.50_{\pm0.01}$ | $0.54_{\pm0.03}$ | $0.07_{\pm0.01}$ | $0.84_{\pm0.00}$ | $0.42_{\pm0.03}$ | $0.93_{\pm0.07}$ | $0.31_{\pm0.03}$ |
| Mitra | $0.52_{\pm0.01}$ | $0.39_{\pm0.01}$ | $0.42_{\pm0.02}$ | $0.04_{\pm0.01}$ | $0.85_{\pm0.00}$ | $0.34_{\pm0.06}$ | $0.94_{\pm0.14}$ | $0.30_{\pm0.07}$ |
| LimiX | — | — | — | — | — | — | — | — |
| TabICL | — | — | — | — | — | — | — | — |
| TabEBM | $0.85_{\pm0.00}$ | $0.67_{\pm0.01}$ | $0.93_{\pm0.00}$ | $0.07_{\pm0.00}$ | $0.96_{\pm0.00}$ | $0.58_{\pm0.00}$ | $0.90_{\pm0.02}$ | $0.22_{\pm0.04}$ |
| CTSyn | $0.64_{\pm0.00}$ | $0.53_{\pm0.01}$ | $0.43_{\pm0.01}$ | $0.14_{\pm0.00}$ | $0.82_{\pm0.00}$ | $0.40_{\pm0.03}$ | $0.72_{\pm0.04}$ | $0.37_{\pm0.02}$ |
| **TabFORGE (Ours)** | $0.90_{\pm0.00}$ | $0.83_{\pm0.00}$ | $0.78_{\pm0.01}$ | $0.37_{\pm0.01}$ | $0.72_{\pm0.01}$ | $0.38_{\pm0.00}$ | $0.95_{\pm0.02}$ | $0.37_{\pm0.05}$ |

*Table 52.* **Raw benchmark results of 23 tabular generators on the "Hazelnut" dataset.** We report the mean $\pm$ std of each metric across 10 repeated data splits. For benchmark generators, "$-$" denotes failed convergence of a specific model or unexpected values in the synthetic data that caused the evaluation metric computation to crash. We highlight the **First**, **Second**, and **Third** best performances for each metric. TabFORGE generally achieves competitive performance against the benchmark generators while maintaining a reduced risk of overfitting.

| Generator | Density Estimation | | | | Privacy Preservation | | ML Efficacy | Structural Fidelity |
|---|---|---|---|---|---|---|---|---|
| | Shape ↑ | Trend ↑ | $\alpha$-precision ↑ | $\beta$-recall ↑ | Authenticity ↑ | DCR Score ↑ | Local utility ↑ | Global utility ↑ |
| **Real Data** | | | | | | | | |
| Real Data (Train) | $1.00_{\pm0.00}$ | $1.00_{\pm0.00}$ | $1.00_{\pm0.00}$ | $1.00_{\pm0.00}$ | $0.00_{\pm0.00}$ | $0.00_{\pm0.00}$ | $1.00_{\pm0.00}$ | $1.00_{\pm0.00}$ |
| Real Data (Holdout) | $0.96_{\pm0.00}$ | $0.97_{\pm0.00}$ | $0.97_{\pm0.01}$ | $0.50_{\pm0.02}$ | $0.65_{\pm0.02}$ | $0.07_{\pm0.00}$ | $-$ | $-$ |
| **Dataset-specific Model** | | | | | | | | |
| SMOTE | $0.95_{\pm0.01}$ | $0.84_{\pm0.00}$ | $0.77_{\pm0.02}$ | $0.72_{\pm0.02}$ | $0.39_{\pm0.01}$ | $0.04_{\pm0.00}$ | $0.97_{\pm0.01}$ | $0.70_{\pm0.02}$ |
| TabSDS | $0.78_{\pm0.00}$ | $0.63_{\pm0.00}$ | $0.73_{\pm0.01}$ | $0.12_{\pm0.02}$ | $0.69_{\pm0.01}$ | $0.16_{\pm0.01}$ | $0.74_{\pm0.01}$ | $0.54_{\pm0.03}$ |
| TVAE | $0.81_{\pm0.01}$ | $0.78_{\pm0.01}$ | $0.79_{\pm0.04}$ | $0.00_{\pm0.00}$ | $1.00_{\pm0.00}$ | $0.31_{\pm0.01}$ | $0.84_{\pm0.03}$ | $0.62_{\pm0.03}$ |
| GOGGLE | $0.49_{\pm0.02}$ | $0.60_{\pm0.01}$ | $0.22_{\pm0.07}$ | $0.00_{\pm0.00}$ | $1.00_{\pm0.00}$ | $0.31_{\pm0.01}$ | $0.31_{\pm0.03}$ | $0.10_{\pm0.01}$ |
| CTGAN | $0.79_{\pm0.01}$ | $0.91_{\pm0.01}$ | $0.74_{\pm0.07}$ | $0.00_{\pm0.00}$ | $1.00_{\pm0.00}$ | $0.34_{\pm0.02}$ | $0.84_{\pm0.02}$ | $0.33_{\pm0.06}$ |
| NFlow | $0.90_{\pm0.01}$ | $0.73_{\pm0.03}$ | $0.62_{\pm0.08}$ | $0.00_{\pm0.00}$ | $1.00_{\pm0.00}$ | $0.50_{\pm0.05}$ | $0.64_{\pm0.03}$ | $0.23_{\pm0.09}$ |
| ARF | $0.94_{\pm0.01}$ | $0.90_{\pm0.01}$ | $0.91_{\pm0.02}$ | $0.01_{\pm0.00}$ | $1.00_{\pm0.00}$ | $0.30_{\pm0.01}$ | $0.89_{\pm0.02}$ | $0.44_{\pm0.03}$ |
| TabDDPM | $0.81_{\pm0.05}$ | $0.84_{\pm0.03}$ | $0.69_{\pm0.06}$ | $0.03_{\pm0.00}$ | $0.99_{\pm0.00}$ | $0.31_{\pm0.07}$ | $0.93_{\pm0.01}$ | $0.64_{\pm0.02}$ |
| CDTD | $0.66_{\pm0.00}$ | $0.73_{\pm0.00}$ | $0.54_{\pm0.01}$ | $0.12_{\pm0.03}$ | $0.79_{\pm0.02}$ | $0.32_{\pm0.01}$ | $0.72_{\pm0.01}$ | $0.46_{\pm0.01}$ |
| TabSyn | $0.93_{\pm0.01}$ | $0.96_{\pm0.00}$ | $0.92_{\pm0.02}$ | $0.28_{\pm0.06}$ | $0.86_{\pm0.05}$ | $0.13_{\pm0.01}$ | $0.97_{\pm0.02}$ | $0.69_{\pm0.02}$ |
| TabDiff | $0.93_{\pm0.01}$ | $0.74_{\pm0.01}$ | $0.58_{\pm0.04}$ | $0.00_{\pm0.00}$ | $1.00_{\pm0.00}$ | $0.48_{\pm0.03}$ | $0.74_{\pm0.03}$ | $0.70_{\pm0.02}$ |
| NRGBoost | $0.91_{\pm0.01}$ | $0.95_{\pm0.00}$ | $0.93_{\pm0.04}$ | $0.33_{\pm0.02}$ | $0.81_{\pm0.01}$ | $0.11_{\pm0.00}$ | $0.93_{\pm0.04}$ | $0.40_{\pm0.01}$ |
| TabNAT | $0.66_{\pm0.00}$ | $0.72_{\pm0.00}$ | $0.44_{\pm0.02}$ | $0.06_{\pm0.01}$ | $0.82_{\pm0.01}$ | $0.44_{\pm0.01}$ | $0.68_{\pm0.02}$ | $0.28_{\pm0.01}$ |
| TabularARGN | $0.78_{\pm0.00}$ | $0.67_{\pm0.00}$ | $0.46_{\pm0.02}$ | $0.00_{\pm0.00}$ | $1.00_{\pm0.00}$ | $0.56_{\pm0.01}$ | $0.66_{\pm0.04}$ | $0.17_{\pm0.08}$ |
| **Foundation Model** | | | | | | | | |
| GReaT | $0.35_{\pm0.01}$ | $0.61_{\pm0.01}$ | $0.00_{\pm0.00}$ | $0.00_{\pm0.00}$ | $1.00_{\pm0.00}$ | $1.00_{\pm0.00}$ | $0.61_{\pm0.04}$ | $0.00_{\pm0.00}$ |
| TabPFN | $0.63_{\pm0.00}$ | $0.66_{\pm0.01}$ | $0.55_{\pm0.03}$ | $0.00_{\pm0.00}$ | $0.85_{\pm0.00}$ | $0.47_{\pm0.02}$ | $0.68_{\pm0.02}$ | $0.27_{\pm0.01}$ |
| TabDPT | $0.61_{\pm0.01}$ | $0.60_{\pm0.01}$ | $0.47_{\pm0.02}$ | $0.00_{\pm0.00}$ | $0.80_{\pm0.00}$ | $0.32_{\pm0.01}$ | $0.64_{\pm0.02}$ | $0.22_{\pm0.06}$ |
| Mitra | $0.63_{\pm0.01}$ | $0.59_{\pm0.01}$ | $0.32_{\pm0.05}$ | $0.01_{\pm0.00}$ | $0.76_{\pm0.00}$ | $0.31_{\pm0.02}$ | $0.50_{\pm0.02}$ | $0.21_{\pm0.06}$ |
| LimiX | $0.68_{\pm0.01}$ | $0.62_{\pm0.01}$ | $0.50_{\pm0.04}$ | $0.00_{\pm0.00}$ | $0.92_{\pm0.00}$ | $0.35_{\pm0.02}$ | $0.69_{\pm0.02}$ | $0.27_{\pm0.07}$ |
| TabICL | $0.78_{\pm0.00}$ | $0.75_{\pm0.01}$ | $0.65_{\pm0.03}$ | $0.08_{\pm0.02}$ | $0.82_{\pm0.01}$ | $0.27_{\pm0.02}$ | $0.70_{\pm0.01}$ | $0.49_{\pm0.05}$ |
| TabEBM | $0.92_{\pm0.01}$ | $0.93_{\pm0.01}$ | $0.69_{\pm0.02}$ | $0.02_{\pm0.00}$ | $0.97_{\pm0.01}$ | $0.25_{\pm0.00}$ | $0.92_{\pm0.01}$ | $0.57_{\pm0.02}$ |
| CTSyn | $0.49_{\pm0.00}$ | $0.67_{\pm0.00}$ | $0.39_{\pm0.02}$ | $0.09_{\pm0.02}$ | $0.79_{\pm0.02}$ | $0.45_{\pm0.01}$ | $0.67_{\pm0.02}$ | $0.26_{\pm0.01}$ |
| **TabFORGE (Ours)** | $0.95_{\pm0.00}$ | $0.97_{\pm0.00}$ | $0.96_{\pm0.01}$ | $0.40_{\pm0.03}$ | $0.72_{\pm0.02}$ | $0.09_{\pm0.01}$ | $0.97_{\pm0.01}$ | $0.75_{\pm0.02}$ |

*Table 53.* **Raw benchmark results of 23 tabular generators on the "Healthcare" dataset.** We report the mean $\pm$ std of each metric across 10 repeated data splits. For benchmark generators, "$-$" denotes failed convergence of a specific model or unexpected values in the synthetic data that caused the evaluation metric computation to crash. We highlight the **First**, **Second**, and **Third** best performances for each metric. TabFORGE generally achieves competitive performance against the benchmark generators while maintaining a reduced risk of overfitting.

| Generator | Density Estimation | | | | Privacy Preservation | | ML Efficacy | Structural Fidelity |
|---|---|---|---|---|---|---|---|---|
| | Shape ↑ | Trend ↑ | $\alpha$-precision ↑ | $\beta$-recall ↑ | Authenticity ↑ | DCR Score ↑ | Local utility ↑ | Global utility ↑ |
| **Real Data** | | | | | | | | |
| Real Data (Train) | $1.00_{\pm 0.00}$ | $1.00_{\pm 0.00}$ | $1.00_{\pm 0.00}$ | $1.00_{\pm 0.00}$ | $0.00_{\pm 0.00}$ | $0.00_{\pm 0.00}$ | $1.00_{\pm 0.00}$ | $0.87_{\pm 0.04}$ |
| Real Data (Holdout) | $0.96_{\pm 0.01}$ | $0.90_{\pm 0.03}$ | $0.96_{\pm 0.02}$ | $0.51_{\pm 0.02}$ | $0.64_{\pm 0.02}$ | $0.16_{\pm 0.01}$ | $-$ | $-$ |
| **Dataset-specific Model** | | | | | | | | |
| SMOTE | $0.90_{\pm 0.01}$ | $0.66_{\pm 0.02}$ | $0.83_{\pm 0.01}$ | $0.58_{\pm 0.02}$ | $0.51_{\pm 0.02}$ | $0.09_{\pm 0.01}$ | $0.92_{\pm 0.03}$ | $0.58_{\pm 0.03}$ |
| TabSDS | $0.63_{\pm 0.00}$ | $0.42_{\pm 0.05}$ | $0.71_{\pm 0.01}$ | $0.32_{\pm 0.01}$ | $0.58_{\pm 0.01}$ | $0.16_{\pm 0.01}$ | $0.67_{\pm 0.01}$ | $0.50_{\pm 0.03}$ |
| TVAE | $0.85_{\pm 0.01}$ | $0.69_{\pm 0.03}$ | $0.91_{\pm 0.03}$ | $0.33_{\pm 0.03}$ | $0.82_{\pm 0.02}$ | $0.30_{\pm 0.02}$ | $0.74_{\pm 0.04}$ | $0.61_{\pm 0.02}$ |
| GOGGLE | $0.63_{\pm 0.05}$ | $0.58_{\pm 0.16}$ | $0.60_{\pm 0.11}$ | $0.14_{\pm 0.03}$ | $0.94_{\pm 0.02}$ | $0.24_{\pm 0.04}$ | $0.39_{\pm 0.02}$ | $0.30_{\pm 0.04}$ |
| CTGAN | $0.74_{\pm 0.06}$ | $0.70_{\pm 0.08}$ | $0.85_{\pm 0.10}$ | $0.23_{\pm 0.05}$ | $0.89_{\pm 0.02}$ | $0.35_{\pm 0.07}$ | $0.66_{\pm 0.08}$ | $0.44_{\pm 0.05}$ |
| NFlow | $0.83_{\pm 0.02}$ | $0.79_{\pm 0.08}$ | $0.88_{\pm 0.06}$ | $0.29_{\pm 0.03}$ | $0.84_{\pm 0.02}$ | $0.37_{\pm 0.04}$ | $0.61_{\pm 0.04}$ | $0.48_{\pm 0.06}$ |
| ARF | $0.90_{\pm 0.00}$ | $0.35_{\pm 0.15}$ | $0.95_{\pm 0.02}$ | $0.39_{\pm 0.03}$ | $0.76_{\pm 0.02}$ | $0.29_{\pm 0.02}$ | $0.82_{\pm 0.03}$ | $0.46_{\pm 0.03}$ |
| TabDDPM | $0.77_{\pm 0.03}$ | $0.72_{\pm 0.03}$ | $0.79_{\pm 0.04}$ | $0.31_{\pm 0.03}$ | $0.83_{\pm 0.02}$ | $0.37_{\pm 0.04}$ | $0.77_{\pm 0.08}$ | $0.55_{\pm 0.02}$ |
| CDTD | $0.76_{\pm 0.01}$ | $0.56_{\pm 0.02}$ | $0.74_{\pm 0.02}$ | $0.28_{\pm 0.01}$ | $0.72_{\pm 0.01}$ | $0.32_{\pm 0.03}$ | $0.73_{\pm 0.01}$ | $0.50_{\pm 0.04}$ |
| TabSyn | $0.89_{\pm 0.01}$ | $0.86_{\pm 0.04}$ | $0.93_{\pm 0.04}$ | $0.47_{\pm 0.02}$ | $0.68_{\pm 0.02}$ | $0.20_{\pm 0.02}$ | $0.94_{\pm 0.03}$ | $0.68_{\pm 0.02}$ |
| TabDiff | $0.88_{\pm 0.01}$ | $0.67_{\pm 0.05}$ | $0.82_{\pm 0.05}$ | $0.24_{\pm 0.03}$ | $0.88_{\pm 0.02}$ | $0.56_{\pm 0.04}$ | $0.68_{\pm 0.02}$ | $0.62_{\pm 0.02}$ |
| NRGBoost | $0.86_{\pm 0.01}$ | $0.81_{\pm 0.06}$ | $0.87_{\pm 0.02}$ | $0.40_{\pm 0.03}$ | $0.65_{\pm 0.03}$ | $0.12_{\pm 0.03}$ | $0.90_{\pm 0.02}$ | $0.33_{\pm 0.03}$ |
| TabNAT | $0.74_{\pm 0.00}$ | $0.67_{\pm 0.03}$ | $0.69_{\pm 0.02}$ | $0.27_{\pm 0.02}$ | $0.67_{\pm 0.01}$ | $0.37_{\pm 0.03}$ | $0.60_{\pm 0.01}$ | $0.37_{\pm 0.04}$ |
| TabularARGN | $0.80_{\pm 0.00}$ | $0.67_{\pm 0.04}$ | $0.95_{\pm 0.02}$ | $0.25_{\pm 0.02}$ | $0.87_{\pm 0.01}$ | $0.43_{\pm 0.02}$ | $0.66_{\pm 0.02}$ | $0.39_{\pm 0.06}$ |
| **Foundation Model** | | | | | | | | |
| GReaT | $0.82_{\pm 0.01}$ | $0.70_{\pm 0.04}$ | $0.76_{\pm 0.05}$ | $0.17_{\pm 0.02}$ | $0.92_{\pm 0.02}$ | $0.67_{\pm 0.14}$ | $0.65_{\pm 0.02}$ | $0.26_{\pm 0.06}$ |
| TabPFN | $0.72_{\pm 0.01}$ | $0.57_{\pm 0.07}$ | $0.78_{\pm 0.03}$ | $0.24_{\pm 0.02}$ | $0.73_{\pm 0.01}$ | $0.35_{\pm 0.03}$ | $0.63_{\pm 0.02}$ | $0.41_{\pm 0.03}$ |
| TabDPT | $0.60_{\pm 0.03}$ | $0.57_{\pm 0.04}$ | $0.69_{\pm 0.06}$ | $0.17_{\pm 0.02}$ | $0.68_{\pm 0.01}$ | $0.26_{\pm 0.03}$ | $0.53_{\pm 0.04}$ | $0.38_{\pm 0.03}$ |
| Mitra | $0.58_{\pm 0.02}$ | $0.53_{\pm 0.08}$ | $0.62_{\pm 0.06}$ | $0.19_{\pm 0.02}$ | $0.75_{\pm 0.01}$ | $0.25_{\pm 0.02}$ | $0.57_{\pm 0.03}$ | $0.40_{\pm 0.04}$ |
| LimiX | $0.71_{\pm 0.01}$ | $0.56_{\pm 0.07}$ | $0.74_{\pm 0.04}$ | $0.24_{\pm 0.02}$ | $0.72_{\pm 0.01}$ | $0.28_{\pm 0.02}$ | $0.61_{\pm 0.03}$ | $0.40_{\pm 0.03}$ |
| TabICL | $0.70_{\pm 0.01}$ | $0.52_{\pm 0.06}$ | $0.80_{\pm 0.02}$ | $0.32_{\pm 0.02}$ | $0.63_{\pm 0.01}$ | $0.23_{\pm 0.02}$ | $0.74_{\pm 0.02}$ | $0.51_{\pm 0.02}$ |
| TabEBM | $0.91_{\pm 0.01}$ | $0.48_{\pm 0.05}$ | $0.92_{\pm 0.03}$ | $0.32_{\pm 0.03}$ | $0.80_{\pm 0.01}$ | $0.36_{\pm 0.01}$ | $0.86_{\pm 0.04}$ | $0.48_{\pm 0.05}$ |
| CTSyn | $0.70_{\pm 0.01}$ | $0.66_{\pm 0.02}$ | $0.72_{\pm 0.02}$ | $0.27_{\pm 0.01}$ | $0.76_{\pm 0.01}$ | $0.32_{\pm 0.04}$ | $0.65_{\pm 0.02}$ | $0.44_{\pm 0.04}$ |
| **TabFORGE (Ours)** | $0.86_{\pm 0.01}$ | $0.84_{\pm 0.02}$ | $0.94_{\pm 0.02}$ | $0.40_{\pm 0.04}$ | $0.77_{\pm 0.03}$ | $0.29_{\pm 0.02}$ | $0.84_{\pm 0.03}$ | $0.60_{\pm 0.05}$ |

*Table 54.* **Raw benchmark results of 23 tabular generators on the "House16H" dataset.** We report the mean ± std of each metric across 10 repeated data splits. For benchmark generators, "−" denotes failed convergence of a specific model or unexpected values in the synthetic data that caused the evaluation metric computation to crash. We highlight the **First**, **Second**, and **Third** best performances for each metric. TabFORGE generally achieves competitive performance against the benchmark generators while maintaining a reduced risk of overfitting.

| Generator | Density Estimation | | | | Privacy Preservation | | ML Efficacy | Structural Fidelity |
|---|---|---|---|---|---|---|---|---|
| | Shape ↑ | Trend ↑ | $\alpha$-precision ↑ | $\beta$-recall ↑ | Authenticity ↑ | DCR Score ↑ | Local utility ↑ | Global utility ↑ |
| **Real Data** | | | | | | | | |
| Real Data (Train) | $1.00_{\pm 0.00}$ | $1.00_{\pm 0.00}$ | $1.00_{\pm 0.00}$ | $1.00_{\pm 0.00}$ | $0.00_{\pm 0.00}$ | $0.00_{\pm 0.00}$ | $1.00_{\pm 0.00}$ | $0.99_{\pm 0.01}$ |
| Real Data (Holdout) | $0.99_{\pm 0.00}$ | $0.99_{\pm 0.00}$ | $0.99_{\pm 0.00}$ | $0.50_{\pm 0.01}$ | $0.75_{\pm 0.00}$ | $0.04_{\pm 0.00}$ | − | − |
| **Dataset-specific Model** | | | | | | | | |
| SMOTE | $\mathbf{0.87}_{\pm 0.01}$ | − | − | $\mathbf{0.69}_{\pm 0.01}$ | $0.44_{\pm 0.00}$ | $0.05_{\pm 0.00}$ | $\mathbf{0.93}_{\pm 0.01}$ | − |
| TabSDS | − | − | − | − | − | − | − | − |
| TVAE | $0.84_{\pm 0.01}$ | $0.91_{\pm 0.01}$ | $0.94_{\pm 0.02}$ | $0.23_{\pm 0.01}$ | $0.91_{\pm 0.00}$ | $0.07_{\pm 0.00}$ | $0.79_{\pm 0.02}$ | − |
| GOGGLE | $0.48_{\pm 0.03}$ | $0.64_{\pm 0.01}$ | $0.24_{\pm 0.12}$ | $0.01_{\pm 0.00}$ | $\mathbf{1.00}_{\pm 0.00}$ | $0.05_{\pm 0.00}$ | $0.39_{\pm 0.01}$ | $0.20_{\pm 0.02}$ |
| CTGAN | $0.81_{\pm 0.01}$ | $0.90_{\pm 0.01}$ | $\mathbf{0.96}_{\pm 0.01}$ | $0.16_{\pm 0.01}$ | $0.94_{\pm 0.01}$ | $0.08_{\pm 0.00}$ | $0.72_{\pm 0.02}$ | $\mathbf{0.48}_{\pm 0.03}$ |
| NFlow | $0.82_{\pm 0.02}$ | $0.80_{\pm 0.01}$ | $0.90_{\pm 0.04}$ | $0.08_{\pm 0.02}$ | $0.97_{\pm 0.01}$ | $\mathbf{0.10}_{\pm 0.01}$ | $0.54_{\pm 0.04}$ | $\mathbf{0.41}_{\pm 0.04}$ |
| ARF | $\mathbf{0.90}_{\pm 0.00}$ | $0.89_{\pm 0.01}$ | $0.94_{\pm 0.01}$ | $0.13_{\pm 0.01}$ | $0.96_{\pm 0.00}$ | $0.10_{\pm 0.00}$ | $0.82_{\pm 0.02}$ | − |
| TabDDPM | $\mathbf{0.87}_{\pm 0.00}$ | $\mathbf{0.98}_{\pm 0.01}$ | $\mathbf{0.97}_{\pm 0.00}$ | $\mathbf{0.37}_{\pm 0.01}$ | $0.83_{\pm 0.01}$ | $0.06_{\pm 0.00}$ | $\mathbf{0.90}_{\pm 0.01}$ | − |
| CDTD | − | − | − | − | − | − | − | − |
| TabSyn | − | − | − | − | − | − | − | − |
| TabDiff | − | − | − | − | − | − | − | − |
| NRGBoost | $0.86_{\pm 0.00}$ | $\mathbf{0.98}_{\pm 0.00}$ | $\mathbf{0.97}_{\pm 0.01}$ | $0.09_{\pm 0.00}$ | $0.97_{\pm 0.00}$ | $0.07_{\pm 0.00}$ | $0.69_{\pm 0.06}$ | − |
| TabNAT | − | − | − | − | − | − | − | − |
| TabularARGN | $0.85_{\pm 0.00}$ | $0.89_{\pm 0.01}$ | $0.94_{\pm 0.02}$ | $0.03_{\pm 0.00}$ | $\mathbf{0.99}_{\pm 0.00}$ | $0.08_{\pm 0.00}$ | $0.64_{\pm 0.03}$ | $0.39_{\pm 0.04}$ |
| **Foundation Model** | | | | | | | | |
| GReaT | $0.60_{\pm 0.01}$ | $0.65_{\pm 0.01}$ | $0.36_{\pm 0.02}$ | $0.01_{\pm 0.00}$ | $\mathbf{1.00}_{\pm 0.00}$ | $\mathbf{0.15}_{\pm 0.01}$ | $0.54_{\pm 0.03}$ | $0.04_{\pm 0.02}$ |
| TabPFN | $0.71_{\pm 0.01}$ | $0.68_{\pm 0.00}$ | $0.68_{\pm 0.02}$ | $0.07_{\pm 0.01}$ | $0.84_{\pm 0.00}$ | $0.10_{\pm 0.01}$ | $0.58_{\pm 0.03}$ | $0.35_{\pm 0.02}$ |
| TabDPT | $0.63_{\pm 0.01}$ | $0.71_{\pm 0.01}$ | $0.60_{\pm 0.03}$ | $0.07_{\pm 0.00}$ | $0.84_{\pm 0.00}$ | $0.07_{\pm 0.00}$ | $0.57_{\pm 0.01}$ | $0.39_{\pm 0.03}$ |
| Mitra | $0.49_{\pm 0.01}$ | $0.62_{\pm 0.00}$ | $0.48_{\pm 0.06}$ | $0.05_{\pm 0.01}$ | $0.83_{\pm 0.00}$ | $0.06_{\pm 0.00}$ | $0.58_{\pm 0.02}$ | $0.31_{\pm 0.03}$ |
| LimiX | $0.63_{\pm 0.01}$ | $0.66_{\pm 0.01}$ | $0.67_{\pm 0.03}$ | $0.07_{\pm 0.01}$ | $0.84_{\pm 0.00}$ | $0.08_{\pm 0.00}$ | $0.63_{\pm 0.02}$ | $0.41_{\pm 0.03}$ |
| TabICL | − | − | − | − | − | − | − | − |
| TabEBM | $0.85_{\pm 0.00}$ | $0.95_{\pm 0.01}$ | $0.92_{\pm 0.01}$ | $0.02_{\pm 0.00}$ | $0.99_{\pm 0.00}$ | $\mathbf{0.10}_{\pm 0.00}$ | $0.69_{\pm 0.02}$ | − |
| CTSyn | − | − | − | − | − | − | − | − |
| **TabFORGE (Ours)** | $0.83_{\pm 0.01}$ | $\mathbf{0.98}_{\pm 0.00}$ | $0.87_{\pm 0.01}$ | $\mathbf{0.34}_{\pm 0.02}$ | $0.77_{\pm 0.01}$ | $0.06_{\pm 0.00}$ | $\mathbf{0.86}_{\pm 0.02}$ | $\mathbf{0.72}_{\pm 0.02}$ |

*Table 55.* **Raw benchmark results of 23 tabular generators on the "Houses" dataset.** We report the mean $\pm$ std of each metric across 10 repeated data splits. For benchmark generators, "$-$" denotes failed convergence of a specific model or unexpected values in the synthetic data that caused the evaluation metric computation to crash. We highlight the **First**, **Second**, and **Third** best performances for each metric. TabFORGE generally achieves competitive performance against the benchmark generators while maintaining a reduced risk of overfitting.

| Generator | Density Estimation | | | | Privacy Preservation | | ML Efficacy | Structural Fidelity |
|---|---|---|---|---|---|---|---|---|
| | Shape ↑ | Trend ↑ | $\alpha$-precision ↑ | $\beta$-recall ↑ | Authenticity ↑ | DCR Score ↑ | Local utility ↑ | Global utility ↑ |
| **Real Data** | | | | | | | | |
| Real Data (Train) | $1.00_{\pm 0.00}$ | $1.00_{\pm 0.00}$ | $1.00_{\pm 0.00}$ | $1.00_{\pm 0.00}$ | $0.00_{\pm 0.00}$ | $0.00_{\pm 0.00}$ | $1.00_{\pm 0.00}$ | $1.00_{\pm 0.00}$ |
| Real Data (Holdout) | $0.99_{\pm 0.00}$ | $1.00_{\pm 0.00}$ | $0.99_{\pm 0.01}$ | $0.50_{\pm 0.01}$ | $0.68_{\pm 0.01}$ | $0.05_{\pm 0.00}$ | $-$ | $-$ |
| **Dataset-specific Model** | | | | | | | | |
| SMOTE | $0.97_{\pm 0.00}$ | $-$ | $-$ | $0.63_{\pm 0.01}$ | $0.48_{\pm 0.01}$ | $0.05_{\pm 0.00}$ | $0.90_{\pm 0.01}$ | $-$ |
| TabSDS | $-$ | $-$ | $-$ | $-$ | $-$ | $-$ | $-$ | $-$ |
| TVAE | $0.92_{\pm 0.01}$ | $0.83_{\pm 0.02}$ | $0.93_{\pm 0.01}$ | $0.19_{\pm 0.01}$ | $0.91_{\pm 0.00}$ | $0.11_{\pm 0.01}$ | $0.65_{\pm 0.02}$ | $-$ |
| GOGGLE | $0.46_{\pm 0.08}$ | $0.59_{\pm 0.03}$ | $0.56_{\pm 0.11}$ | $0.00_{\pm 0.00}$ | $1.00_{\pm 0.00}$ | $0.15_{\pm 0.06}$ | $0.23_{\pm 0.03}$ | $0.06_{\pm 0.03}$ |
| CTGAN | $0.87_{\pm 0.02}$ | $0.87_{\pm 0.01}$ | $0.92_{\pm 0.04}$ | $0.14_{\pm 0.02}$ | $0.93_{\pm 0.01}$ | $0.11_{\pm 0.01}$ | $0.63_{\pm 0.03}$ | $0.34_{\pm 0.03}$ |
| NFlow | $0.87_{\pm 0.02}$ | $0.74_{\pm 0.04}$ | $0.91_{\pm 0.02}$ | $0.11_{\pm 0.02}$ | $0.95_{\pm 0.01}$ | $0.15_{\pm 0.02}$ | $0.45_{\pm 0.03}$ | $0.28_{\pm 0.05}$ |
| ARF | $0.96_{\pm 0.00}$ | $0.95_{\pm 0.01}$ | $0.98_{\pm 0.00}$ | $0.21_{\pm 0.01}$ | $0.90_{\pm 0.00}$ | $0.11_{\pm 0.01}$ | $0.74_{\pm 0.01}$ | $-$ |
| TabDDPM | $0.98_{\pm 0.00}$ | $0.97_{\pm 0.02}$ | $0.98_{\pm 0.01}$ | $0.34_{\pm 0.01}$ | $0.80_{\pm 0.01}$ | $0.08_{\pm 0.01}$ | $0.88_{\pm 0.01}$ | $-$ |
| CDTD | $-$ | $-$ | $-$ | $-$ | $-$ | $-$ | $-$ | $-$ |
| TabSyn | $-$ | $-$ | $-$ | $-$ | $-$ | $-$ | $-$ | $-$ |
| TabDiff | $-$ | $-$ | $-$ | $-$ | $-$ | $-$ | $-$ | $-$ |
| NRGBoost | $0.97_{\pm 0.00}$ | $0.97_{\pm 0.01}$ | $0.98_{\pm 0.01}$ | $0.11_{\pm 0.00}$ | $0.94_{\pm 0.00}$ | $0.08_{\pm 0.01}$ | $0.71_{\pm 0.00}$ | $-$ |
| TabNAT | $-$ | $-$ | $-$ | $-$ | $-$ | $-$ | $-$ | $-$ |
| TabularARGN | $0.83_{\pm 0.00}$ | $0.95_{\pm 0.01}$ | $0.98_{\pm 0.00}$ | $0.07_{\pm 0.00}$ | $0.97_{\pm 0.00}$ | $0.10_{\pm 0.01}$ | $0.57_{\pm 0.02}$ | $0.37_{\pm 0.04}$ |
| **Foundation Model** | | | | | | | | |
| GReaT | $0.83_{\pm 0.00}$ | $0.59_{\pm 0.01}$ | $0.88_{\pm 0.01}$ | $0.03_{\pm 0.00}$ | $0.99_{\pm 0.00}$ | $0.19_{\pm 0.01}$ | $0.43_{\pm 0.01}$ | $0.03_{\pm 0.02}$ |
| TabPFN | $0.77_{\pm 0.01}$ | $0.68_{\pm 0.02}$ | $0.83_{\pm 0.01}$ | $0.11_{\pm 0.01}$ | $0.80_{\pm 0.01}$ | $0.12_{\pm 0.01}$ | $0.52_{\pm 0.02}$ | $0.26_{\pm 0.02}$ |
| TabDPT | $0.68_{\pm 0.02}$ | $0.69_{\pm 0.01}$ | $0.69_{\pm 0.03}$ | $0.08_{\pm 0.01}$ | $0.88_{\pm 0.01}$ | $0.10_{\pm 0.02}$ | $0.50_{\pm 0.02}$ | $0.21_{\pm 0.03}$ |
| Mitra | $0.51_{\pm 0.03}$ | $0.63_{\pm 0.02}$ | $0.66_{\pm 0.05}$ | $0.06_{\pm 0.01}$ | $0.81_{\pm 0.00}$ | $0.11_{\pm 0.03}$ | $0.40_{\pm 0.04}$ | $0.20_{\pm 0.04}$ |
| LimiX | $0.66_{\pm 0.01}$ | $0.61_{\pm 0.02}$ | $0.80_{\pm 0.02}$ | $0.11_{\pm 0.01}$ | $0.75_{\pm 0.01}$ | $0.12_{\pm 0.02}$ | $0.49_{\pm 0.02}$ | $0.28_{\pm 0.03}$ |
| TabICL | $-$ | $-$ | $-$ | $-$ | $-$ | $-$ | $-$ | $-$ |
| TabEBM | $0.93_{\pm 0.00}$ | $0.95_{\pm 0.01}$ | $0.89_{\pm 0.03}$ | $0.03_{\pm 0.00}$ | $0.99_{\pm 0.00}$ | $0.14_{\pm 0.01}$ | $0.59_{\pm 0.01}$ | $-$ |
| CTSyn | $-$ | $-$ | $-$ | $-$ | $-$ | $-$ | $-$ | $-$ |
| **TabFORGE (Ours)** | $0.97_{\pm 0.00}$ | $0.99_{\pm 0.01}$ | $0.97_{\pm 0.01}$ | $0.36_{\pm 0.01}$ | $0.74_{\pm 0.01}$ | $0.07_{\pm 0.00}$ | $0.84_{\pm 0.01}$ | $0.75_{\pm 0.03}$ |

*Table 56.* **Raw benchmark results of 23 tabular generators on the "JM1" dataset.** We report the mean $\pm$ std of each metric across 10 repeated data splits. For benchmark generators, "−" denotes failed convergence of a specific model or unexpected values in the synthetic data that caused the evaluation metric computation to crash. We highlight the **First**, **Second**, and **Third** best performances for each metric. TabFORGE generally achieves competitive performance against the benchmark generators while maintaining a reduced risk of overfitting.

| Generator | Density Estimation | | | | Privacy Preservation | | ML Efficacy | Structural Fidelity |
|---|---|---|---|---|---|---|---|---|
| | Shape ↑ | Trend ↑ | $\alpha$-precision ↑ | $\beta$-recall ↑ | Authenticity ↑ | DCR Score ↑ | Local utility ↑ | Global utility ↑ |
| **Real Data** | | | | | | | | |
| Real Data (Train) | $1.00_{\pm 0.00}$ | $1.00_{\pm 0.00}$ | $1.00_{\pm 0.00}$ | $1.00_{\pm 0.00}$ | $0.00_{\pm 0.00}$ | $0.00_{\pm 0.00}$ | $1.00_{\pm 0.00}$ | $0.88_{\pm 0.06}$ |
| Real Data (Holdout) | $0.99_{\pm 0.00}$ | $0.97_{\pm 0.01}$ | $0.99_{\pm 0.00}$ | $0.56_{\pm 0.01}$ | $0.60_{\pm 0.01}$ | $0.00_{\pm 0.00}$ | − | − |
| **Dataset-specific Model** | | | | | | | | |
| SMOTE | $0.80_{\pm 0.02}$ | $0.80_{\pm 0.01}$ | $0.91_{\pm 0.01}$ | $0.61_{\pm 0.02}$ | $0.51_{\pm 0.02}$ | $0.00_{\pm 0.00}$ | $0.99_{\pm 0.01}$ | $0.47_{\pm 0.04}$ |
| TabSDS | $0.60_{\pm 0.01}$ | $0.68_{\pm 0.02}$ | $0.76_{\pm 0.01}$ | $0.13_{\pm 0.01}$ | $0.70_{\pm 0.01}$ | $0.01_{\pm 0.00}$ | $0.80_{\pm 0.01}$ | $0.43_{\pm 0.03}$ |
| TVAE | $0.76_{\pm 0.02}$ | $0.73_{\pm 0.03}$ | $0.89_{\pm 0.02}$ | $0.03_{\pm 0.01}$ | $0.99_{\pm 0.00}$ | $0.01_{\pm 0.00}$ | $0.90_{\pm 0.03}$ | $0.45_{\pm 0.03}$ |
| GOGGLE | $0.38_{\pm 0.09}$ | $0.63_{\pm 0.01}$ | $0.55_{\pm 0.04}$ | $0.00_{\pm 0.00}$ | $1.00_{\pm 0.00}$ | $0.01_{\pm 0.00}$ | $1.00_{\pm 0.00}$ | $0.14_{\pm 0.04}$ |
| CTGAN | $0.76_{\pm 0.02}$ | $0.73_{\pm 0.04}$ | $0.92_{\pm 0.04}$ | $0.03_{\pm 0.01}$ | $0.99_{\pm 0.00}$ | $0.01_{\pm 0.00}$ | $0.91_{\pm 0.05}$ | $0.25_{\pm 0.04}$ |
| NFlow | $0.79_{\pm 0.01}$ | $0.67_{\pm 0.03}$ | $0.65_{\pm 0.06}$ | $0.00_{\pm 0.00}$ | $1.00_{\pm 0.00}$ | $0.05_{\pm 0.01}$ | $0.77_{\pm 0.03}$ | $0.17_{\pm 0.05}$ |
| ARF | $0.83_{\pm 0.01}$ | $0.91_{\pm 0.03}$ | $0.98_{\pm 0.01}$ | $0.14_{\pm 0.01}$ | $0.93_{\pm 0.01}$ | $0.01_{\pm 0.00}$ | $0.95_{\pm 0.02}$ | $0.42_{\pm 0.04}$ |
| TabDDPM | $0.77_{\pm 0.03}$ | $0.90_{\pm 0.01}$ | $0.91_{\pm 0.04}$ | $0.27_{\pm 0.01}$ | $0.87_{\pm 0.01}$ | $0.01_{\pm 0.00}$ | $0.93_{\pm 0.01}$ | $0.52_{\pm 0.05}$ |
| CDTD | $0.62_{\pm 0.01}$ | $0.72_{\pm 0.01}$ | $0.74_{\pm 0.02}$ | $0.11_{\pm 0.01}$ | $0.86_{\pm 0.00}$ | $0.02_{\pm 0.00}$ | $0.82_{\pm 0.01}$ | $0.33_{\pm 0.03}$ |
| TabSyn | $0.79_{\pm 0.02}$ | $0.94_{\pm 0.02}$ | $0.96_{\pm 0.03}$ | $0.31_{\pm 0.02}$ | $0.84_{\pm 0.01}$ | $0.00_{\pm 0.00}$ | $0.98_{\pm 0.01}$ | $0.49_{\pm 0.04}$ |
| TabDiff | $0.76_{\pm 0.01}$ | $0.68_{\pm 0.02}$ | $0.78_{\pm 0.04}$ | $0.00_{\pm 0.00}$ | $1.00_{\pm 0.00}$ | $0.03_{\pm 0.01}$ | $0.96_{\pm 0.05}$ | $0.52_{\pm 0.05}$ |
| NRGBoost | $0.76_{\pm 0.01}$ | $0.83_{\pm 0.04}$ | $0.94_{\pm 0.04}$ | $0.06_{\pm 0.01}$ | $0.98_{\pm 0.00}$ | $0.01_{\pm 0.00}$ | $0.93_{\pm 0.02}$ | $0.18_{\pm 0.01}$ |
| TabNAT | $0.61_{\pm 0.01}$ | $0.56_{\pm 0.01}$ | $0.68_{\pm 0.02}$ | $0.10_{\pm 0.01}$ | $0.84_{\pm 0.00}$ | $0.01_{\pm 0.00}$ | $0.87_{\pm 0.02}$ | $0.20_{\pm 0.01}$ |
| TabularARGN | $0.80_{\pm 0.01}$ | $0.81_{\pm 0.02}$ | $0.94_{\pm 0.02}$ | $0.03_{\pm 0.01}$ | $0.99_{\pm 0.00}$ | $0.01_{\pm 0.00}$ | $0.94_{\pm 0.04}$ | $0.33_{\pm 0.03}$ |
| **Foundation Model** | | | | | | | | |
| GReaT | $0.58_{\pm 0.01}$ | $0.66_{\pm 0.01}$ | $0.77_{\pm 0.01}$ | $0.01_{\pm 0.00}$ | $0.99_{\pm 0.00}$ | $0.01_{\pm 0.00}$ | $0.87_{\pm 0.03}$ | $0.16_{\pm 0.04}$ |
| TabPFN | $0.64_{\pm 0.01}$ | $0.67_{\pm 0.02}$ | $0.68_{\pm 0.03}$ | $0.03_{\pm 0.00}$ | $0.80_{\pm 0.00}$ | $0.03_{\pm 0.01}$ | $0.78_{\pm 0.02}$ | $0.21_{\pm 0.02}$ |
| TabDPT | $0.55_{\pm 0.03}$ | $0.61_{\pm 0.02}$ | $0.64_{\pm 0.02}$ | $0.02_{\pm 0.01}$ | $0.85_{\pm 0.01}$ | $0.01_{\pm 0.00}$ | $1.00_{\pm 0.00}$ | $0.23_{\pm 0.03}$ |
| Mitra | $0.46_{\pm 0.04}$ | $0.61_{\pm 0.01}$ | $0.57_{\pm 0.03}$ | $0.00_{\pm 0.00}$ | $0.81_{\pm 0.00}$ | $0.02_{\pm 0.01}$ | $1.00_{\pm 0.00}$ | $0.17_{\pm 0.03}$ |
| LimiX | $0.57_{\pm 0.02}$ | $0.56_{\pm 0.02}$ | $0.62_{\pm 0.03}$ | $0.04_{\pm 0.00}$ | $0.80_{\pm 0.00}$ | $0.02_{\pm 0.01}$ | $0.85_{\pm 0.02}$ | $0.28_{\pm 0.03}$ |
| TabICL | $0.64_{\pm 0.01}$ | $0.74_{\pm 0.02}$ | $0.80_{\pm 0.02}$ | $0.12_{\pm 0.01}$ | $0.78_{\pm 0.01}$ | $0.02_{\pm 0.00}$ | $0.82_{\pm 0.01}$ | $0.36_{\pm 0.02}$ |
| TabEBM | $0.72_{\pm 0.01}$ | $0.92_{\pm 0.02}$ | $0.82_{\pm 0.04}$ | $0.00_{\pm 0.00}$ | $0.99_{\pm 0.00}$ | $0.04_{\pm 0.01}$ | $0.94_{\pm 0.02}$ | $0.37_{\pm 0.04}$ |
| CTSyn | $0.54_{\pm 0.01}$ | $0.62_{\pm 0.01}$ | $0.75_{\pm 0.02}$ | $0.11_{\pm 0.01}$ | $0.71_{\pm 0.00}$ | $0.01_{\pm 0.00}$ | $0.81_{\pm 0.01}$ | $0.25_{\pm 0.03}$ |
| **TabFORGE (Ours)** | $0.79_{\pm 0.02}$ | $0.95_{\pm 0.03}$ | $0.98_{\pm 0.01}$ | $0.33_{\pm 0.05}$ | $0.79_{\pm 0.04}$ | $0.00_{\pm 0.00}$ | $0.98_{\pm 0.02}$ | $0.61_{\pm 0.10}$ |

*Table 57.* **Raw benchmark results of 23 tabular generators on the "Marketing" dataset.** We report the mean $\pm$ std of each metric across 10 repeated data splits. For benchmark generators, "−" denotes failed convergence of a specific model or unexpected values in the synthetic data that caused the evaluation metric computation to crash. We highlight the **First**, **Second**, and **Third** best performances for each metric. TabFORGE generally achieves competitive performance against the benchmark generators while maintaining a reduced risk of overfitting.

| Generator | Density Estimation | | | | Privacy Preservation | | ML Efficacy | Structural Fidelity |
|---|---|---|---|---|---|---|---|---|
| | Shape ↑ | Trend ↑ | $\alpha$-precision ↑ | $\beta$-recall ↑ | Authenticity ↑ | DCR Score ↑ | Local utility ↑ | Global utility ↑ |
| **Real Data** | | | | | | | | |
| Real Data (Train) | $1.00_{\pm0.00}$ | $1.00_{\pm0.00}$ | $1.00_{\pm0.00}$ | $1.00_{\pm0.00}$ | $0.00_{\pm0.00}$ | $0.00_{\pm0.00}$ | $1.00_{\pm0.00}$ | $0.70_{\pm0.03}$ |
| Real Data (Holdout) | $0.96_{\pm0.00}$ | $0.69_{\pm0.04}$ | $0.86_{\pm0.02}$ | $0.84_{\pm0.02}$ | $0.30_{\pm0.03}$ | $0.16_{\pm0.01}$ | − | − |
| **Dataset-specific Model** | | | | | | | | |
| SMOTE | $0.87_{\pm0.01}$ | − | − | $0.65_{\pm0.02}$ | $0.49_{\pm0.02}$ | $0.02_{\pm0.00}$ | $1.00_{\pm0.03}$ | − |
| TabSDS | − | − | − | − | − | − | − | − |
| TVAE | $0.82_{\pm0.00}$ | $0.59_{\pm0.04}$ | $0.83_{\pm0.05}$ | $0.28_{\pm0.02}$ | $0.85_{\pm0.01}$ | $0.28_{\pm0.02}$ | $0.89_{\pm0.05}$ | $0.42_{\pm0.02}$ |
| GOGGLE | $0.54_{\pm0.06}$ | $0.42_{\pm0.03}$ | $0.21_{\pm0.05}$ | $0.05_{\pm0.02}$ | $0.97_{\pm0.01}$ | $0.22_{\pm0.02}$ | $0.91_{\pm0.02}$ | $0.35_{\pm0.02}$ |
| CTGAN | $0.80_{\pm0.01}$ | $0.66_{\pm0.03}$ | $0.92_{\pm0.04}$ | $0.24_{\pm0.02}$ | $0.88_{\pm0.01}$ | $0.30_{\pm0.02}$ | $0.88_{\pm0.03}$ | $0.44_{\pm0.06}$ |
| NFlow | $0.83_{\pm0.01}$ | $0.61_{\pm0.02}$ | $0.83_{\pm0.09}$ | $0.15_{\pm0.03}$ | $0.92_{\pm0.01}$ | $0.35_{\pm0.03}$ | $0.81_{\pm0.04}$ | $0.36_{\pm0.04}$ |
| ARF | − | − | − | − | − | − | − | − |
| TabDDPM | $0.39_{\pm0.03}$ | $0.36_{\pm0.07}$ | $0.06_{\pm0.03}$ | $0.03_{\pm0.01}$ | $0.99_{\pm0.01}$ | $0.99_{\pm0.02}$ | $0.94_{\pm0.06}$ | $0.41_{\pm0.03}$ |
| CDTD | − | − | − | − | − | − | − | − |
| TabSyn | $0.85_{\pm0.01}$ | $0.69_{\pm0.03}$ | $0.92_{\pm0.04}$ | $0.37_{\pm0.05}$ | $0.79_{\pm0.04}$ | $0.27_{\pm0.02}$ | $0.96_{\pm0.04}$ | $0.50_{\pm0.04}$ |
| TabDiff | $0.81_{\pm0.02}$ | $0.59_{\pm0.03}$ | $0.32_{\pm0.09}$ | $0.05_{\pm0.02}$ | $0.97_{\pm0.01}$ | $0.51_{\pm0.03}$ | $0.88_{\pm0.02}$ | $0.46_{\pm0.03}$ |
| NRGBoost | $0.84_{\pm0.00}$ | $0.65_{\pm0.04}$ | $0.80_{\pm0.03}$ | $0.23_{\pm0.02}$ | $0.85_{\pm0.01}$ | $0.33_{\pm0.01}$ | $0.99_{\pm0.06}$ | − |
| TabNAT | $0.67_{\pm0.01}$ | $0.45_{\pm0.02}$ | $0.39_{\pm0.04}$ | $0.18_{\pm0.02}$ | $0.73_{\pm0.01}$ | $0.35_{\pm0.02}$ | $0.73_{\pm0.02}$ | $0.29_{\pm0.01}$ |
| TabularARGN | $0.76_{\pm0.00}$ | $0.57_{\pm0.03}$ | $0.90_{\pm0.04}$ | $0.76_{\pm0.03}$ | $0.45_{\pm0.03}$ | $0.31_{\pm0.02}$ | $0.90_{\pm0.03}$ | $0.47_{\pm0.03}$ |
| **Foundation Model** | | | | | | | | |
| GReaT | $0.61_{\pm0.01}$ | $0.42_{\pm0.03}$ | $0.33_{\pm0.06}$ | $0.43_{\pm0.04}$ | $0.73_{\pm0.03}$ | $0.27_{\pm0.01}$ | $0.88_{\pm0.04}$ | $0.12_{\pm0.03}$ |
| TabPFN | − | − | − | − | − | − | − | − |
| TabDPT | $0.71_{\pm0.01}$ | $0.51_{\pm0.02}$ | $0.54_{\pm0.02}$ | $0.23_{\pm0.01}$ | $0.75_{\pm0.01}$ | $0.22_{\pm0.01}$ | $0.92_{\pm0.02}$ | $0.38_{\pm0.03}$ |
| Mitra | $0.65_{\pm0.03}$ | $0.45_{\pm0.02}$ | $0.40_{\pm0.04}$ | $0.09_{\pm0.01}$ | $0.72_{\pm0.01}$ | $0.21_{\pm0.01}$ | $1.00_{\pm0.00}$ | $0.41_{\pm0.02}$ |
| LimiX | − | − | − | − | − | − | − | − |
| TabICL | − | − | − | − | − | − | − | − |
| TabEBM | − | − | − | − | − | − | − | − |
| CTSyn | $0.62_{\pm0.01}$ | $0.42_{\pm0.03}$ | $0.47_{\pm0.03}$ | $0.27_{\pm0.03}$ | $0.66_{\pm0.02}$ | $0.22_{\pm0.01}$ | $0.96_{\pm0.02}$ | $0.35_{\pm0.02}$ |
| **TabFORGE (Ours)** | $0.86_{\pm0.02}$ | $0.69_{\pm0.04}$ | $0.82_{\pm0.08}$ | $0.42_{\pm0.06}$ | $0.74_{\pm0.07}$ | $0.21_{\pm0.02}$ | $0.97_{\pm0.03}$ | $0.58_{\pm0.03}$ |

*Table 58.* **Raw benchmark results of 23 tabular generators on the "Maternal" dataset.** We report the mean $\pm$ std of each metric across 10 repeated data splits. For benchmark generators, "$-$" denotes failed convergence of a specific model or unexpected values in the synthetic data that caused the evaluation metric computation to crash. We highlight the **First**, **Second**, and **Third** best performances for each metric. TabFORGE generally achieves competitive performance against the benchmark generators while maintaining a reduced risk of overfitting.

| Generator | Density Estimation | | | | Privacy Preservation | | ML Efficacy | Structural Fidelity |
|---|---|---|---|---|---|---|---|---|
| | Shape ↑ | Trend ↑ | $\alpha$-precision ↑ | $\beta$-recall ↑ | Authenticity ↑ | DCR Score ↑ | Local utility ↑ | Global utility ↑ |
| **Real Data** | | | | | | | | |
| Real Data (Train) | $1.00_{\pm0.00}$ | $1.00_{\pm0.00}$ | $0.99_{\pm0.00}$ | $0.99_{\pm0.00}$ | $0.00_{\pm0.00}$ | $0.00_{\pm0.00}$ | $1.00_{\pm0.00}$ | $1.00_{\pm0.00}$ |
| Real Data (Holdout) | $0.96_{\pm0.01}$ | $0.90_{\pm0.02}$ | $0.96_{\pm0.01}$ | $0.67_{\pm0.04}$ | $0.40_{\pm0.04}$ | $0.00_{\pm0.00}$ | $-$ | $-$ |
| **Dataset-specific Model** | | | | | | | | |
| SMOTE | $0.93_{\pm0.01}$ | $0.75_{\pm0.02}$ | $0.85_{\pm0.02}$ | $0.50_{\pm0.05}$ | $0.65_{\pm0.04}$ | $0.02_{\pm0.01}$ | $0.91_{\pm0.02}$ | $0.60_{\pm0.03}$ |
| TabSDS | $0.70_{\pm0.01}$ | $0.55_{\pm0.02}$ | $0.67_{\pm0.01}$ | $0.13_{\pm0.01}$ | $0.69_{\pm0.01}$ | $0.14_{\pm0.02}$ | $0.72_{\pm0.02}$ | $0.52_{\pm0.03}$ |
| TVAE | $0.80_{\pm0.01}$ | $0.57_{\pm0.05}$ | $0.82_{\pm0.04}$ | $0.14_{\pm0.02}$ | $0.96_{\pm0.01}$ | $0.23_{\pm0.05}$ | $0.68_{\pm0.08}$ | $0.63_{\pm0.03}$ |
| GOGGLE | $0.63_{\pm0.04}$ | $0.34_{\pm0.06}$ | $0.70_{\pm0.11}$ | $0.07_{\pm0.02}$ | $0.98_{\pm0.01}$ | $0.23_{\pm0.03}$ | $0.31_{\pm0.02}$ | $0.18_{\pm0.03}$ |
| CTGAN | $0.62_{\pm0.08}$ | $0.51_{\pm0.08}$ | $0.70_{\pm0.07}$ | $0.06_{\pm0.02}$ | $0.98_{\pm0.01}$ | $0.30_{\pm0.09}$ | $0.58_{\pm0.07}$ | $0.36_{\pm0.05}$ |
| NFlow | $0.83_{\pm0.02}$ | $0.70_{\pm0.03}$ | $0.91_{\pm0.04}$ | $0.10_{\pm0.01}$ | $0.97_{\pm0.01}$ | $0.35_{\pm0.08}$ | $0.54_{\pm0.03}$ | $0.46_{\pm0.06}$ |
| ARF | $0.88_{\pm0.01}$ | $0.65_{\pm0.04}$ | $0.93_{\pm0.02}$ | $0.18_{\pm0.04}$ | $0.93_{\pm0.02}$ | $0.25_{\pm0.04}$ | $0.86_{\pm0.05}$ | $0.52_{\pm0.03}$ |
| TabDDPM | $0.73_{\pm0.04}$ | $0.67_{\pm0.03}$ | $0.66_{\pm0.06}$ | $0.09_{\pm0.02}$ | $0.97_{\pm0.01}$ | $0.39_{\pm0.08}$ | $0.78_{\pm0.03}$ | $0.57_{\pm0.03}$ |
| CDTD | $0.64_{\pm0.01}$ | $0.58_{\pm0.03}$ | $0.61_{\pm0.02}$ | $0.09_{\pm0.01}$ | $0.82_{\pm0.01}$ | $0.37_{\pm0.03}$ | $0.64_{\pm0.02}$ | $0.40_{\pm0.02}$ |
| TabSyn | $0.91_{\pm0.01}$ | $0.85_{\pm0.02}$ | $0.93_{\pm0.03}$ | $0.19_{\pm0.02}$ | $0.92_{\pm0.02}$ | $0.10_{\pm0.03}$ | $0.91_{\pm0.03}$ | $0.68_{\pm0.03}$ |
| TabDiff | $0.84_{\pm0.02}$ | $0.72_{\pm0.03}$ | $0.76_{\pm0.05}$ | $0.09_{\pm0.02}$ | $0.97_{\pm0.01}$ | $0.52_{\pm0.10}$ | $0.51_{\pm0.11}$ | $0.66_{\pm0.03}$ |
| NRGBoost | $0.69_{\pm0.01}$ | $0.80_{\pm0.02}$ | $0.88_{\pm0.03}$ | $0.17_{\pm0.02}$ | $0.90_{\pm0.02}$ | $0.12_{\pm0.03}$ | $0.88_{\pm0.04}$ | $0.31_{\pm0.02}$ |
| TabNAT | $0.78_{\pm0.01}$ | $0.63_{\pm0.03}$ | $0.58_{\pm0.03}$ | $0.09_{\pm0.01}$ | $0.85_{\pm0.01}$ | $0.47_{\pm0.05}$ | $0.48_{\pm0.04}$ | $0.30_{\pm0.02}$ |
| TabularARGN | $0.78_{\pm0.01}$ | $0.72_{\pm0.02}$ | $0.93_{\pm0.03}$ | $0.20_{\pm0.02}$ | $0.92_{\pm0.02}$ | $0.24_{\pm0.06}$ | $0.76_{\pm0.02}$ | $0.55_{\pm0.04}$ |
| **Foundation Model** | | | | | | | | |
| GReaT | $0.72_{\pm0.02}$ | $0.47_{\pm0.06}$ | $0.34_{\pm0.04}$ | $0.02_{\pm0.01}$ | $0.99_{\pm0.00}$ | $0.96_{\pm0.04}$ | $0.49_{\pm0.07}$ | $0.02_{\pm0.02}$ |
| TabPFN | $0.70_{\pm0.01}$ | $0.57_{\pm0.01}$ | $0.62_{\pm0.02}$ | $0.09_{\pm0.01}$ | $0.82_{\pm0.01}$ | $0.34_{\pm0.04}$ | $0.59_{\pm0.02}$ | $0.40_{\pm0.04}$ |
| TabDPT | $0.55_{\pm0.04}$ | $0.40_{\pm0.05}$ | $0.60_{\pm0.02}$ | $0.08_{\pm0.01}$ | $0.82_{\pm0.01}$ | $0.22_{\pm0.05}$ | $0.53_{\pm0.04}$ | $0.32_{\pm0.04}$ |
| Mitra | $0.64_{\pm0.02}$ | $0.47_{\pm0.04}$ | $0.70_{\pm0.06}$ | $0.08_{\pm0.01}$ | $0.78_{\pm0.00}$ | $0.24_{\pm0.04}$ | $0.49_{\pm0.02}$ | $0.34_{\pm0.04}$ |
| LimiX | $0.71_{\pm0.01}$ | $0.51_{\pm0.02}$ | $0.74_{\pm0.03}$ | $0.10_{\pm0.01}$ | $0.76_{\pm0.01}$ | $0.25_{\pm0.05}$ | $0.55_{\pm0.02}$ | $0.43_{\pm0.04}$ |
| TabICL | $0.73_{\pm0.01}$ | $0.61_{\pm0.01}$ | $0.73_{\pm0.01}$ | $0.12_{\pm0.01}$ | $0.70_{\pm0.01}$ | $0.21_{\pm0.04}$ | $0.64_{\pm0.02}$ | $0.55_{\pm0.04}$ |
| TabEBM | $0.81_{\pm0.01}$ | $0.70_{\pm0.04}$ | $0.92_{\pm0.03}$ | $0.09_{\pm0.02}$ | $0.96_{\pm0.01}$ | $0.34_{\pm0.04}$ | $0.85_{\pm0.05}$ | $0.55_{\pm0.02}$ |
| CTSyn | $0.63_{\pm0.01}$ | $0.51_{\pm0.03}$ | $0.49_{\pm0.03}$ | $0.09_{\pm0.01}$ | $0.85_{\pm0.01}$ | $0.44_{\pm0.02}$ | $0.50_{\pm0.02}$ | $0.31_{\pm0.02}$ |
| **TabFORGE (Ours)** | $0.93_{\pm0.01}$ | $0.87_{\pm0.04}$ | $0.96_{\pm0.01}$ | $0.22_{\pm0.01}$ | $0.87_{\pm0.01}$ | $0.03_{\pm0.01}$ | $0.92_{\pm0.04}$ | $0.78_{\pm0.05}$ |

*Table 59.* **Raw benchmark results of 23 tabular generators on the "Miami" dataset.** We report the mean ± std of each metric across 10 repeated data splits. For benchmark generators, "−" denotes failed convergence of a specific model or unexpected values in the synthetic data that caused the evaluation metric computation to crash. We highlight the **First**, **Second**, and **Third** best performances for each metric. TabFORGE generally achieves competitive performance against the benchmark generators while maintaining a reduced risk of overfitting.

| Generator | Density Estimation | | | | Privacy Preservation | | ML Efficacy | Structural Fidelity |
|---|---|---|---|---|---|---|---|---|
| | Shape ↑ | Trend ↑ | $\alpha$-precision ↑ | $\beta$-recall ↑ | Authenticity ↑ | DCR Score ↑ | Local utility ↑ | Global utility ↑ |
| **Real Data** | | | | | | | | |
| Real Data (Train) | $1.00_{\pm 0.00}$ | $1.00_{\pm 0.00}$ | $1.00_{\pm 0.00}$ | $1.00_{\pm 0.00}$ | $0.00_{\pm 0.00}$ | $0.00_{\pm 0.00}$ | $1.00_{\pm 0.00}$ | $1.00_{\pm 0.00}$ |
| Real Data (Holdout) | $0.99_{\pm 0.00}$ | $1.00_{\pm 0.00}$ | $0.99_{\pm 0.01}$ | $0.50_{\pm 0.01}$ | $0.68_{\pm 0.01}$ | $0.07_{\pm 0.00}$ | − | − |
| **Dataset-specific Model** | | | | | | | | |
| SMOTE | $\mathbf{0.96}_{\pm 0.00}$ | $0.80_{\pm 0.00}$ | $0.73_{\pm 0.01}$ | $\mathbf{0.67}_{\pm 0.01}$ | $0.44_{\pm 0.01}$ | $0.06_{\pm 0.00}$ | $\mathbf{0.89}_{\pm 0.01}$ | $\mathbf{0.60}_{\pm 0.01}$ |
| TabSDS | $0.64_{\pm 0.00}$ | $0.75_{\pm 0.01}$ | $0.66_{\pm 0.01}$ | $0.13_{\pm 0.01}$ | $0.71_{\pm 0.00}$ | $0.13_{\pm 0.00}$ | $0.61_{\pm 0.02}$ | $0.42_{\pm 0.01}$ |
| TVAE | $0.91_{\pm 0.00}$ | $0.85_{\pm 0.02}$ | $0.87_{\pm 0.03}$ | $0.04_{\pm 0.01}$ | $0.99_{\pm 0.00}$ | $0.28_{\pm 0.01}$ | $0.50_{\pm 0.03}$ | $0.53_{\pm 0.01}$ |
| GOGGLE | $0.56_{\pm 0.02}$ | $0.65_{\pm 0.02}$ | $0.34_{\pm 0.09}$ | $0.00_{\pm 0.00}$ | $\mathbf{1.00}_{\pm 0.00}$ | $0.27_{\pm 0.02}$ | $0.22_{\pm 0.01}$ | $0.07_{\pm 0.02}$ |
| CTGAN | $0.87_{\pm 0.02}$ | $0.93_{\pm 0.01}$ | $0.91_{\pm 0.04}$ | $0.03_{\pm 0.01}$ | $0.99_{\pm 0.00}$ | $0.26_{\pm 0.02}$ | $0.51_{\pm 0.03}$ | $0.27_{\pm 0.03}$ |
| NFlow | $0.88_{\pm 0.02}$ | $0.79_{\pm 0.03}$ | $0.87_{\pm 0.06}$ | $0.01_{\pm 0.00}$ | $1.00_{\pm 0.00}$ | $\mathbf{0.36}_{\pm 0.03}$ | $0.36_{\pm 0.04}$ | $0.19_{\pm 0.03}$ |
| ARF | $\mathbf{0.96}_{\pm 0.00}$ | $0.94_{\pm 0.02}$ | $\mathbf{0.98}_{\pm 0.01}$ | $0.12_{\pm 0.00}$ | $0.95_{\pm 0.00}$ | $0.21_{\pm 0.01}$ | $0.73_{\pm 0.03}$ | $0.46_{\pm 0.01}$ |
| TabDDPM | $\mathbf{0.96}_{\pm 0.01}$ | $0.97_{\pm 0.02}$ | $0.95_{\pm 0.01}$ | $0.19_{\pm 0.01}$ | $0.92_{\pm 0.00}$ | $0.16_{\pm 0.00}$ | $0.81_{\pm 0.02}$ | $0.58_{\pm 0.01}$ |
| CDTD | $0.69_{\pm 0.00}$ | $0.81_{\pm 0.01}$ | $0.51_{\pm 0.01}$ | $0.13_{\pm 0.01}$ | $0.84_{\pm 0.01}$ | $0.28_{\pm 0.00}$ | $0.53_{\pm 0.01}$ | $0.36_{\pm 0.01}$ |
| TabSyn | $0.95_{\pm 0.01}$ | $\mathbf{0.99}_{\pm 0.00}$ | $0.86_{\pm 0.03}$ | $\mathbf{0.34}_{\pm 0.02}$ | $0.82_{\pm 0.00}$ | $0.13_{\pm 0.00}$ | $\mathbf{0.84}_{\pm 0.02}$ | $\mathbf{0.68}_{\pm 0.02}$ |
| TabDiff | $0.92_{\pm 0.00}$ | $0.84_{\pm 0.02}$ | $0.69_{\pm 0.03}$ | $0.01_{\pm 0.00}$ | $\mathbf{1.00}_{\pm 0.00}$ | $\mathbf{0.40}_{\pm 0.01}$ | $0.47_{\pm 0.02}$ | $0.54_{\pm 0.01}$ |
| NRGBoost | $0.91_{\pm 0.00}$ | $\mathbf{0.98}_{\pm 0.00}$ | $\mathbf{0.96}_{\pm 0.03}$ | $0.09_{\pm 0.02}$ | $0.96_{\pm 0.01}$ | $0.16_{\pm 0.01}$ | $0.68_{\pm 0.03}$ | $0.28_{\pm 0.01}$ |
| TabNAT | $0.64_{\pm 0.00}$ | $0.64_{\pm 0.01}$ | $0.50_{\pm 0.02}$ | $0.09_{\pm 0.00}$ | $0.79_{\pm 0.00}$ | $0.29_{\pm 0.00}$ | $0.46_{\pm 0.01}$ | $0.24_{\pm 0.01}$ |
| TabularARGN | $0.80_{\pm 0.00}$ | $0.93_{\pm 0.02}$ | $\mathbf{0.96}_{\pm 0.01}$ | $0.01_{\pm 0.00}$ | $1.00_{\pm 0.00}$ | $0.26_{\pm 0.01}$ | $0.53_{\pm 0.02}$ | $0.22_{\pm 0.02}$ |
| **Foundation Model** | | | | | | | | |
| GReaT | $0.65_{\pm 0.01}$ | $0.64_{\pm 0.03}$ | $0.13_{\pm 0.01}$ | $0.00_{\pm 0.00}$ | $\mathbf{1.00}_{\pm 0.00}$ | $\mathbf{0.64}_{\pm 0.01}$ | $0.32_{\pm 0.03}$ | $0.03_{\pm 0.02}$ |
| TabPFN | $0.79_{\pm 0.01}$ | $0.64_{\pm 0.01}$ | $0.62_{\pm 0.02}$ | $0.03_{\pm 0.00}$ | $0.74_{\pm 0.00}$ | $0.29_{\pm 0.01}$ | $0.43_{\pm 0.02}$ | $0.23_{\pm 0.02}$ |
| TabDPT | $0.72_{\pm 0.01}$ | $0.70_{\pm 0.01}$ | $0.66_{\pm 0.02}$ | $0.02_{\pm 0.00}$ | $0.87_{\pm 0.01}$ | $0.23_{\pm 0.01}$ | $0.40_{\pm 0.01}$ | $0.19_{\pm 0.02}$ |
| Mitra | $0.55_{\pm 0.01}$ | $0.57_{\pm 0.01}$ | $0.58_{\pm 0.05}$ | $0.02_{\pm 0.00}$ | $0.89_{\pm 0.00}$ | $0.24_{\pm 0.01}$ | $0.37_{\pm 0.02}$ | $0.16_{\pm 0.02}$ |
| LimiX | $0.74_{\pm 0.01}$ | $0.67_{\pm 0.01}$ | $0.62_{\pm 0.03}$ | $0.04_{\pm 0.00}$ | $0.86_{\pm 0.00}$ | $0.26_{\pm 0.01}$ | $0.43_{\pm 0.02}$ | $0.24_{\pm 0.02}$ |
| TabICL | $0.77_{\pm 0.00}$ | $0.75_{\pm 0.01}$ | $0.86_{\pm 0.02}$ | $0.13_{\pm 0.00}$ | $0.80_{\pm 0.00}$ | $0.20_{\pm 0.01}$ | $0.57_{\pm 0.01}$ | $0.42_{\pm 0.01}$ |
| TabEBM | $0.90_{\pm 0.00}$ | $0.94_{\pm 0.01}$ | $0.75_{\pm 0.02}$ | $0.01_{\pm 0.00}$ | $0.99_{\pm 0.00}$ | $0.26_{\pm 0.00}$ | $0.64_{\pm 0.02}$ | $0.33_{\pm 0.02}$ |
| CTSyn | $0.65_{\pm 0.00}$ | $0.63_{\pm 0.01}$ | $0.41_{\pm 0.02}$ | $0.13_{\pm 0.01}$ | $0.82_{\pm 0.00}$ | $0.31_{\pm 0.01}$ | $0.45_{\pm 0.01}$ | $0.30_{\pm 0.01}$ |
| **TabFORGE (Ours)** | $0.95_{\pm 0.01}$ | $\mathbf{0.99}_{\pm 0.00}$ | $0.96_{\pm 0.02}$ | $\mathbf{0.39}_{\pm 0.03}$ | $0.69_{\pm 0.03}$ | $0.09_{\pm 0.01}$ | $\mathbf{0.82}_{\pm 0.03}$ | $\mathbf{0.60}_{\pm 0.03}$ |

*Table 60.* **Raw benchmark results of 23 tabular generators on the "NATICUSdroid" dataset.** We report the mean ± std of each metric across 10 repeated data splits. For benchmark generators, "−" denotes failed convergence of a specific model or unexpected values in the synthetic data that caused the evaluation metric computation to crash. We highlight the **First**, **Second**, and **Third** best performances for each metric. TabFORGE generally achieves competitive performance against the benchmark generators while maintaining a reduced risk of overfitting.

| Generator | Density Estimation | | | | Privacy Preservation | | ML Efficacy | Structural Fidelity |
|---|---|---|---|---|---|---|---|---|
| | Shape ↑ | Trend ↑ | α-precision ↑ | β-recall ↑ | Authenticity ↑ | DCR Score ↑ | Local utility ↑ | Global utility ↑ |
| **Real Data** | | | | | | | | |
| Real Data (Train) | $1.00_{\pm 0.00}$ | $1.00_{\pm 0.00}$ | $1.00_{\pm 0.00}$ | $1.00_{\pm 0.00}$ | $0.00_{\pm 0.00}$ | $0.00_{\pm 0.00}$ | $1.00_{\pm 0.00}$ | $0.71_{\pm 0.02}$ |
| Real Data (Holdout) | $0.99_{\pm 0.00}$ | $0.99_{\pm 0.00}$ | $0.99_{\pm 0.00}$ | $0.72_{\pm 0.01}$ | $0.55_{\pm 0.01}$ | $0.04_{\pm 0.00}$ | — | — |
| **Dataset-specific Model** | | | | | | | | |
| SMOTE | $0.99_{\pm 0.00}$ | — | — | $0.83_{\pm 0.01}$ | $0.39_{\pm 0.00}$ | $0.00_{\pm 0.00}$ | $0.99_{\pm 0.01}$ | — |
| TabSDS | — | — | — | — | — | — | — | — |
| TVAE | $0.97_{\pm 0.00}$ | $0.95_{\pm 0.01}$ | $0.67_{\pm 0.05}$ | $0.50_{\pm 0.01}$ | $0.77_{\pm 0.01}$ | $0.06_{\pm 0.00}$ | $0.91_{\pm 0.02}$ | — |
| GOGGLE | $0.91_{\pm 0.00}$ | $0.89_{\pm 0.01}$ | $0.05_{\pm 0.01}$ | $0.06_{\pm 0.01}$ | $0.97_{\pm 0.00}$ | $0.03_{\pm 0.00}$ | — | — |
| CTGAN | $0.94_{\pm 0.04}$ | $0.91_{\pm 0.08}$ | $0.73_{\pm 0.14}$ | $0.22_{\pm 0.12}$ | $0.90_{\pm 0.06}$ | $0.11_{\pm 0.05}$ | $0.85_{\pm 0.09}$ | $0.29_{\pm 0.03}$ |
| NFlow | $0.96_{\pm 0.01}$ | $0.93_{\pm 0.01}$ | $0.75_{\pm 0.08}$ | $0.18_{\pm 0.05}$ | $0.94_{\pm 0.02}$ | $0.14_{\pm 0.01}$ | $0.66_{\pm 0.09}$ | $0.26_{\pm 0.05}$ |
| ARF | $0.99_{\pm 0.00}$ | $0.98_{\pm 0.00}$ | $0.94_{\pm 0.01}$ | $0.55_{\pm 0.01}$ | $0.72_{\pm 0.01}$ | $0.09_{\pm 0.00}$ | $0.98_{\pm 0.01}$ | — |
| TabDDPM | $0.61_{\pm 0.00}$ | $0.40_{\pm 0.01}$ | $0.05_{\pm 0.01}$ | $0.00_{\pm 0.00}$ | $1.00_{\pm 0.00}$ | $0.94_{\pm 0.01}$ | $0.55_{\pm 0.06}$ | — |
| CDTD | — | — | — | — | — | — | — | — |
| TabSyn | — | — | — | — | — | — | — | — |
| TabDiff | — | — | — | — | — | — | — | — |
| NRGBoost | $0.97_{\pm 0.00}$ | $0.95_{\pm 0.00}$ | $0.82_{\pm 0.01}$ | $0.44_{\pm 0.01}$ | $0.78_{\pm 0.01}$ | $0.09_{\pm 0.00}$ | $0.99_{\pm 0.01}$ | — |
| TabNAT | — | — | — | — | — | — | — | — |
| TabularARGN | $0.86_{\pm 0.00}$ | $0.97_{\pm 0.01}$ | $0.94_{\pm 0.01}$ | $0.38_{\pm 0.02}$ | $0.83_{\pm 0.01}$ | $0.07_{\pm 0.02}$ | $0.97_{\pm 0.01}$ | $0.45_{\pm 0.02}$ |
| **Foundation Model** | | | | | | | | |
| GReaT | — | — | — | — | — | — | — | — |
| TabPFN | — | — | — | — | — | — | — | — |
| TabDPT | $0.81_{\pm 0.02}$ | $0.76_{\pm 0.04}$ | $0.47_{\pm 0.07}$ | $0.15_{\pm 0.05}$ | $0.82_{\pm 0.03}$ | $0.07_{\pm 0.02}$ | $0.69_{\pm 0.04}$ | $0.26_{\pm 0.01}$ |
| Mitra | $0.77_{\pm 0.00}$ | $0.78_{\pm 0.01}$ | $0.34_{\pm 0.03}$ | $0.12_{\pm 0.01}$ | $0.78_{\pm 0.00}$ | $0.07_{\pm 0.00}$ | $0.56_{\pm 0.04}$ | $0.27_{\pm 0.02}$ |
| LimiX | $0.77_{\pm 0.00}$ | $0.74_{\pm 0.01}$ | $0.54_{\pm 0.04}$ | $0.24_{\pm 0.02}$ | $0.70_{\pm 0.01}$ | $0.09_{\pm 0.01}$ | $0.63_{\pm 0.05}$ | $0.24_{\pm 0.03}$ |
| TabICL | — | — | — | — | — | — | — | — |
| TabEBM | $0.97_{\pm 0.00}$ | $0.93_{\pm 0.00}$ | $0.51_{\pm 0.01}$ | $0.11_{\pm 0.01}$ | $0.91_{\pm 0.01}$ | $0.09_{\pm 0.00}$ | $0.99_{\pm 0.01}$ | — |
| CTSyn | — | — | — | — | — | — | — | — |
| **TabFORGE (Ours)** | $0.96_{\pm 0.00}$ | $0.95_{\pm 0.00}$ | $0.58_{\pm 0.05}$ | $0.62_{\pm 0.02}$ | $0.67_{\pm 0.01}$ | $0.05_{\pm 0.01}$ | $0.96_{\pm 0.01}$ | $0.40_{\pm 0.02}$ |

*Table 61.* **Raw benchmark results of 23 tabular generators on the "Nomao" dataset.** We report the mean $\pm$ std of each metric across 10 repeated data splits. For benchmark generators, "$-$" denotes failed convergence of a specific model or unexpected values in the synthetic data that caused the evaluation metric computation to crash. We highlight the **First**, **Second**, and **Third** best performances for each metric. TabFORGE generally achieves competitive performance against the benchmark generators while maintaining a reduced risk of overfitting.

| Generator | Density Estimation | | | | Privacy Preservation | | ML Efficacy | Structural Fidelity |
|---|---|---|---|---|---|---|---|---|
| | Shape ↑ | Trend ↑ | $\alpha$-precision ↑ | $\beta$-recall ↑ | Authenticity ↑ | DCR Score ↑ | Local utility ↑ | Global utility ↑ |
| **Real Data** | | | | | | | | |
| Real Data (Train) | $1.00_{\pm0.00}$ | $1.00_{\pm0.00}$ | $1.00_{\pm0.00}$ | $1.00_{\pm0.00}$ | $0.01_{\pm0.00}$ | $0.00_{\pm0.00}$ | $1.00_{\pm0.00}$ | $0.99_{\pm0.01}$ |
| Real Data (Holdout) | $0.99_{\pm0.00}$ | $0.99_{\pm0.00}$ | $0.99_{\pm0.00}$ | $0.52_{\pm0.01}$ | $0.66_{\pm0.00}$ | $0.02_{\pm0.00}$ | $-$ | $-$ |
| **Dataset-specific Model** | | | | | | | | |
| SMOTE | $0.97_{\pm0.00}$ | $-$ | $-$ | $0.63_{\pm0.01}$ | $0.49_{\pm0.00}$ | $0.03_{\pm0.00}$ | $-$ | $-$ |
| TabSDS | $0.73_{\pm0.00}$ | $0.45_{\pm0.00}$ | $0.71_{\pm0.01}$ | $0.01_{\pm0.00}$ | $0.73_{\pm0.00}$ | $0.12_{\pm0.01}$ | $-$ | $0.31_{\pm0.01}$ |
| TVAE | $0.76_{\pm0.01}$ | $0.79_{\pm0.01}$ | $0.85_{\pm0.01}$ | $0.10_{\pm0.01}$ | $0.97_{\pm0.00}$ | $0.09_{\pm0.00}$ | $0.94_{\pm0.01}$ | $0.48_{\pm0.01}$ |
| GOGGLE | $-$ | $-$ | $-$ | $-$ | $-$ | $-$ | $-$ | $-$ |
| CTGAN | $0.70_{\pm0.02}$ | $0.77_{\pm0.01}$ | $0.94_{\pm0.03}$ | $0.02_{\pm0.00}$ | $1.00_{\pm0.00}$ | $0.13_{\pm0.01}$ | $0.88_{\pm0.05}$ | $0.27_{\pm0.02}$ |
| NFlow | $-$ | $-$ | $-$ | $-$ | $-$ | $-$ | $-$ | $-$ |
| ARF | $0.91_{\pm0.00}$ | $0.58_{\pm0.01}$ | $0.95_{\pm0.01}$ | $0.03_{\pm0.00}$ | $0.99_{\pm0.00}$ | $0.12_{\pm0.01}$ | $0.95_{\pm0.00}$ | $0.38_{\pm0.01}$ |
| TabDDPM | $0.54_{\pm0.00}$ | $0.39_{\pm0.00}$ | $0.06_{\pm0.00}$ | $0.00_{\pm0.00}$ | $1.00_{\pm0.00}$ | $1.00_{\pm0.00}$ | $0.54_{\pm0.06}$ | $0.51_{\pm0.01}$ |
| CDTD | $-$ | $-$ | $-$ | $-$ | $-$ | $-$ | $-$ | $-$ |
| TabSyn | $0.93_{\pm0.01}$ | $0.84_{\pm0.01}$ | $0.82_{\pm0.04}$ | $0.01_{\pm0.00}$ | $1.00_{\pm0.00}$ | $0.15_{\pm0.01}$ | $0.94_{\pm0.02}$ | $0.36_{\pm0.02}$ |
| TabDiff | $0.82_{\pm0.01}$ | $0.60_{\pm0.02}$ | $0.30_{\pm0.08}$ | $0.00_{\pm0.00}$ | $1.00_{\pm0.00}$ | $0.41_{\pm0.03}$ | $0.64_{\pm0.09}$ | $0.55_{\pm0.01}$ |
| NRGBoost | $0.66_{\pm0.01}$ | $0.69_{\pm0.01}$ | $0.71_{\pm0.01}$ | $0.00_{\pm0.00}$ | $1.00_{\pm0.00}$ | $0.26_{\pm0.02}$ | $-$ | $-$ |
| TabNAT | $-$ | $-$ | $-$ | $-$ | $-$ | $-$ | $-$ | $-$ |
| TabularARGN | $0.63_{\pm0.01}$ | $0.86_{\pm0.01}$ | $0.92_{\pm0.02}$ | $0.01_{\pm0.00}$ | $0.99_{\pm0.00}$ | $0.09_{\pm0.01}$ | $0.92_{\pm0.01}$ | $0.38_{\pm0.01}$ |
| **Foundation Model** | | | | | | | | |
| GReaT | $-$ | $-$ | $-$ | $-$ | $-$ | $-$ | $-$ | $-$ |
| TabPFN | $-$ | $-$ | $-$ | $-$ | $-$ | $-$ | $-$ | $-$ |
| TabDPT | $-$ | $-$ | $-$ | $-$ | $-$ | $-$ | $-$ | $-$ |
| Mitra | $-$ | $-$ | $-$ | $-$ | $-$ | $-$ | $-$ | $-$ |
| LimiX | $-$ | $-$ | $-$ | $-$ | $-$ | $-$ | $-$ | $-$ |
| TabICL | $-$ | $-$ | $-$ | $-$ | $-$ | $-$ | $-$ | $-$ |
| TabEBM | $-$ | $-$ | $-$ | $-$ | $-$ | $-$ | $-$ | $-$ |
| CTSyn | $-$ | $-$ | $-$ | $-$ | $-$ | $-$ | $-$ | $-$ |
| **TabFORGE (Ours)** | $0.96_{\pm0.00}$ | $0.96_{\pm0.00}$ | $0.99_{\pm0.01}$ | $0.26_{\pm0.02}$ | $0.83_{\pm0.02}$ | $0.04_{\pm0.00}$ | $0.98_{\pm0.01}$ | $0.57_{\pm0.01}$ |

*Table 62.* **Raw benchmark results of 23 tabular generators on the "Phoneme" dataset.** We report the mean $\pm$ std of each metric across 10 repeated data splits. For benchmark generators, "$-$" denotes failed convergence of a specific model or unexpected values in the synthetic data that caused the evaluation metric computation to crash. We highlight the **First**, **Second**, and **Third** best performances for each metric. TabFORGE generally achieves competitive performance against the benchmark generators while maintaining a reduced risk of overfitting.

| Generator | Density Estimation | | | | Privacy Preservation | | ML Efficacy | Structural Fidelity |
|---|---|---|---|---|---|---|---|---|
| | Shape $\uparrow$ | Trend $\uparrow$ | $\alpha$-precision $\uparrow$ | $\beta$-recall $\uparrow$ | Authenticity $\uparrow$ | DCR Score $\uparrow$ | Local utility $\uparrow$ | Global utility $\uparrow$ |
| **Real Data** | | | | | | | | |
| Real Data (Train) | $1.00_{\pm0.00}$ | $1.00_{\pm0.00}$ | $1.00_{\pm0.00}$ | $1.00_{\pm0.00}$ | $0.00_{\pm0.00}$ | $0.00_{\pm0.00}$ | $1.00_{\pm0.00}$ | $0.95_{\pm0.02}$ |
| Real Data (Holdout) | $0.98_{\pm0.00}$ | $0.84_{\pm0.06}$ | $0.98_{\pm0.01}$ | $0.50_{\pm0.01}$ | $0.65_{\pm0.01}$ | $0.06_{\pm0.00}$ | $-$ | $-$ |
| **Dataset-specific Model** | | | | | | | | |
| SMOTE | $\mathbf{0.97}_{\pm0.00}$ | $0.67_{\pm0.04}$ | $0.88_{\pm0.01}$ | $\mathbf{0.60}_{\pm0.01}$ | $0.50_{\pm0.01}$ | $0.04_{\pm0.00}$ | $\mathbf{0.97}_{\pm0.01}$ | $0.56_{\pm0.03}$ |
| TabSDS | $0.78_{\pm0.00}$ | $0.61_{\pm0.04}$ | $0.68_{\pm0.01}$ | $0.17_{\pm0.00}$ | $0.68_{\pm0.00}$ | $0.10_{\pm0.00}$ | $0.74_{\pm0.01}$ | $0.47_{\pm0.02}$ |
| TVAE | $0.88_{\pm0.01}$ | $0.59_{\pm0.06}$ | $0.88_{\pm0.04}$ | $0.12_{\pm0.01}$ | $0.95_{\pm0.00}$ | $0.24_{\pm0.01}$ | $0.74_{\pm0.02}$ | $0.58_{\pm0.03}$ |
| GOGGLE | $0.54_{\pm0.05}$ | $0.15_{\pm0.06}$ | $0.47_{\pm0.15}$ | $0.01_{\pm0.01}$ | $\mathbf{1.00}_{\pm0.00}$ | $0.32_{\pm0.16}$ | $0.65_{\pm0.11}$ | $0.17_{\pm0.05}$ |
| CTGAN | $0.72_{\pm0.07}$ | $0.50_{\pm0.05}$ | $0.83_{\pm0.08}$ | $0.06_{\pm0.03}$ | $\mathbf{0.98}_{\pm0.01}$ | $0.26_{\pm0.06}$ | $0.75_{\pm0.08}$ | $0.24_{\pm0.06}$ |
| NFlow | $0.88_{\pm0.02}$ | $0.77_{\pm0.05}$ | $0.91_{\pm0.09}$ | $0.09_{\pm0.01}$ | $0.96_{\pm0.01}$ | $0.30_{\pm0.02}$ | $0.67_{\pm0.05}$ | $0.28_{\pm0.09}$ |
| ARF | $0.95_{\pm0.00}$ | $0.76_{\pm0.10}$ | $\mathbf{0.98}_{\pm0.01}$ | $0.19_{\pm0.01}$ | $0.91_{\pm0.01}$ | $0.19_{\pm0.00}$ | $0.90_{\pm0.01}$ | $0.42_{\pm0.02}$ |
| TabDDPM | $\mathbf{0.96}_{\pm0.01}$ | $\mathbf{0.92}_{\pm0.02}$ | $\mathbf{0.97}_{\pm0.01}$ | $0.22_{\pm0.01}$ | $0.89_{\pm0.01}$ | $0.16_{\pm0.01}$ | $0.93_{\pm0.02}$ | $0.62_{\pm0.03}$ |
| CDTD | $0.77_{\pm0.00}$ | $0.67_{\pm0.05}$ | $0.72_{\pm0.01}$ | $0.13_{\pm0.01}$ | $0.88_{\pm0.01}$ | $0.20_{\pm0.01}$ | $0.77_{\pm0.01}$ | $0.35_{\pm0.03}$ |
| TabSyn | $0.96_{\pm0.00}$ | $0.84_{\pm0.06}$ | $0.92_{\pm0.03}$ | $0.30_{\pm0.01}$ | $0.83_{\pm0.01}$ | $0.13_{\pm0.01}$ | $0.93_{\pm0.01}$ | $\mathbf{0.69}_{\pm0.03}$ |
| TabDiff | $0.95_{\pm0.01}$ | $0.85_{\pm0.01}$ | $0.96_{\pm0.01}$ | $0.08_{\pm0.00}$ | $0.97_{\pm0.00}$ | $\mathbf{0.32}_{\pm0.02}$ | $0.70_{\pm0.04}$ | $\mathbf{0.65}_{\pm0.03}$ |
| NRGBoost | $0.95_{\pm0.01}$ | $\mathbf{0.90}_{\pm0.02}$ | $0.96_{\pm0.01}$ | $\mathbf{0.34}_{\pm0.01}$ | $0.77_{\pm0.01}$ | $0.09_{\pm0.00}$ | $\mathbf{0.95}_{\pm0.02}$ | $0.36_{\pm0.02}$ |
| TabNAT | $0.78_{\pm0.00}$ | $0.62_{\pm0.03}$ | $0.77_{\pm0.01}$ | $0.11_{\pm0.00}$ | $0.84_{\pm0.00}$ | $0.23_{\pm0.01}$ | $0.61_{\pm0.02}$ | $0.30_{\pm0.03}$ |
| TabularARGN | $0.82_{\pm0.00}$ | $0.76_{\pm0.05}$ | $0.96_{\pm0.01}$ | $0.07_{\pm0.01}$ | $0.97_{\pm0.00}$ | $\mathbf{0.33}_{\pm0.01}$ | $0.72_{\pm0.03}$ | $0.19_{\pm0.09}$ |
| **Foundation Model** | | | | | | | | |
| GReaT | $0.87_{\pm0.01}$ | $0.47_{\pm0.13}$ | $0.82_{\pm0.01}$ | $0.06_{\pm0.00}$ | $\mathbf{0.98}_{\pm0.00}$ | $\mathbf{0.36}_{\pm0.02}$ | $0.61_{\pm0.04}$ | $0.10_{\pm0.05}$ |
| TabPFN | $0.71_{\pm0.01}$ | $0.65_{\pm0.05}$ | $0.82_{\pm0.04}$ | $0.11_{\pm0.01}$ | $0.79_{\pm0.00}$ | $0.24_{\pm0.01}$ | $0.74_{\pm0.03}$ | $0.25_{\pm0.04}$ |
| TabDPT | $0.60_{\pm0.03}$ | $0.38_{\pm0.03}$ | $0.63_{\pm0.04}$ | $0.04_{\pm0.01}$ | $0.92_{\pm0.00}$ | $0.24_{\pm0.06}$ | $0.77_{\pm0.07}$ | $0.22_{\pm0.06}$ |
| Mitra | $0.55_{\pm0.03}$ | $0.41_{\pm0.04}$ | $0.54_{\pm0.08}$ | $0.07_{\pm0.00}$ | $0.78_{\pm0.00}$ | $0.27_{\pm0.08}$ | $0.92_{\pm0.09}$ | $0.28_{\pm0.07}$ |
| LimiX | $0.71_{\pm0.01}$ | $0.55_{\pm0.04}$ | $0.67_{\pm0.05}$ | $0.10_{\pm0.01}$ | $0.89_{\pm0.01}$ | $0.21_{\pm0.02}$ | $0.71_{\pm0.03}$ | $0.33_{\pm0.06}$ |
| TabICL | $0.86_{\pm0.01}$ | $0.63_{\pm0.04}$ | $0.75_{\pm0.02}$ | $0.17_{\pm0.01}$ | $0.77_{\pm0.01}$ | $0.19_{\pm0.01}$ | $0.75_{\pm0.01}$ | $0.45_{\pm0.04}$ |
| TabEBM | $0.92_{\pm0.00}$ | $0.82_{\pm0.03}$ | $0.91_{\pm0.02}$ | $0.07_{\pm0.00}$ | $0.97_{\pm0.00}$ | $0.29_{\pm0.01}$ | $0.87_{\pm0.02}$ | $0.36_{\pm0.03}$ |
| CTSyn | $0.73_{\pm0.01}$ | $0.48_{\pm0.06}$ | $0.69_{\pm0.02}$ | $0.13_{\pm0.01}$ | $0.77_{\pm0.00}$ | $0.23_{\pm0.02}$ | $0.73_{\pm0.02}$ | $0.33_{\pm0.02}$ |
| **TabFORGE (Ours)** | $\mathbf{0.97}_{\pm0.00}$ | $\mathbf{0.90}_{\pm0.03}$ | $\mathbf{0.98}_{\pm0.01}$ | $\mathbf{0.44}_{\pm0.02}$ | $0.66_{\pm0.02}$ | $0.05_{\pm0.01}$ | $\mathbf{0.95}_{\pm0.02}$ | $\mathbf{0.71}_{\pm0.04}$ |

*Table 63.* **Raw benchmark results of 23 tabular generators on the "Plants" dataset.** We report the mean $\pm$ std of each metric across 10 repeated data splits. For benchmark generators, "$-$" denotes failed convergence of a specific model or unexpected values in the synthetic data that caused the evaluation metric computation to crash. We highlight the **First**, **Second**, and **Third** best performances for each metric. TabFORGE generally achieves competitive performance against the benchmark generators while maintaining a reduced risk of overfitting.

| Generator | Density Estimation | | | | Privacy Preservation | | ML Efficacy | Structural Fidelity |
|---|---|---|---|---|---|---|---|---|
| | Shape $\uparrow$ | Trend $\uparrow$ | $\alpha$-precision $\uparrow$ | $\beta$-recall $\uparrow$ | Authenticity $\uparrow$ | DCR Score $\uparrow$ | Local utility $\uparrow$ | Global utility $\uparrow$ |
| Real Data | | | | | | | | |
| Real Data (Train) | $1.00_{\pm0.00}$ | $1.00_{\pm0.00}$ | $1.00_{\pm0.00}$ | $1.00_{\pm0.00}$ | $0.00_{\pm0.00}$ | $0.00_{\pm0.00}$ | $-$ | $0.98_{\pm0.01}$ |
| Real Data (Holdout) | $0.96_{\pm0.00}$ | $0.83_{\pm0.01}$ | $0.96_{\pm0.01}$ | $0.51_{\pm0.02}$ | $0.66_{\pm0.03}$ | $0.05_{\pm0.00}$ | $-$ | $-$ |
| Dataset-specific Model | | | | | | | | |
| SMOTE | $0.67_{\pm0.01}$ | $-$ | $-$ | $\mathbf{0.71}_{\pm0.02}$ | $0.38_{\pm0.02}$ | $0.03_{\pm0.00}$ | $-$ | $-$ |
| TabSDS | $-$ | $-$ | $-$ | $-$ | $-$ | $-$ | $-$ | $-$ |
| TVAE | $0.63_{\pm0.01}$ | $0.55_{\pm0.00}$ | $0.38_{\pm0.02}$ | $0.02_{\pm0.01}$ | $1.00_{\pm0.00}$ | $0.18_{\pm0.00}$ | $-$ | $-$ |
| GOGGLE | $0.41_{\pm0.01}$ | $0.36_{\pm0.01}$ | $0.08_{\pm0.00}$ | $0.00_{\pm0.00}$ | $\mathbf{1.00}_{\pm0.00}$ | $0.14_{\pm0.01}$ | $-$ | $-$ |
| CTGAN | $0.61_{\pm0.02}$ | $0.64_{\pm0.02}$ | $\mathbf{0.89}_{\pm0.04}$ | $0.01_{\pm0.00}$ | $1.00_{\pm0.00}$ | $0.20_{\pm0.01}$ | $-$ | $0.19_{\pm0.02}$ |
| NFlow | $0.69_{\pm0.01}$ | $0.60_{\pm0.01}$ | $0.83_{\pm0.07}$ | $0.01_{\pm0.01}$ | $1.00_{\pm0.00}$ | $0.22_{\pm0.01}$ | $-$ | $0.13_{\pm0.01}$ |
| ARF | $-$ | $-$ | $-$ | $-$ | $-$ | $-$ | $-$ | $-$ |
| TabDDPM | $0.40_{\pm0.02}$ | $0.48_{\pm0.01}$ | $0.00_{\pm0.00}$ | $0.00_{\pm0.00}$ | $\mathbf{1.00}_{\pm0.00}$ | $\mathbf{0.65}_{\pm0.05}$ | $-$ | $-$ |
| CDTD | $-$ | $-$ | $-$ | $-$ | $-$ | $-$ | $-$ | $-$ |
| TabSyn | $-$ | $-$ | $-$ | $-$ | $-$ | $-$ | $-$ | $-$ |
| TabDiff | $-$ | $-$ | $-$ | $-$ | $-$ | $-$ | $-$ | $-$ |
| NRGBoost | $0.63_{\pm0.00}$ | $\mathbf{0.69}_{\pm0.01}$ | $\mathbf{0.94}_{\pm0.02}$ | $\mathbf{0.10}_{\pm0.01}$ | $0.95_{\pm0.01}$ | $0.13_{\pm0.00}$ | $-$ | $-$ |
| TabNAT | $-$ | $-$ | $-$ | $-$ | $-$ | $-$ | $-$ | $-$ |
| TabularARGN | $\mathbf{0.76}_{\pm0.01}$ | $0.56_{\pm0.02}$ | $0.80_{\pm0.06}$ | $0.00_{\pm0.00}$ | $\mathbf{1.00}_{\pm0.00}$ | $\mathbf{0.22}_{\pm0.00}$ | $-$ | $\mathbf{0.18}_{\pm0.01}$ |
| Foundation Model | | | | | | | | |
| GReaT | $-$ | $-$ | $-$ | $-$ | $-$ | $-$ | $-$ | $-$ |
| TabPFN | $-$ | $-$ | $-$ | $-$ | $-$ | $-$ | $-$ | $-$ |
| TabDPT | $0.48_{\pm0.01}$ | $0.46_{\pm0.01}$ | $0.56_{\pm0.02}$ | $0.00_{\pm0.00}$ | $0.79_{\pm0.00}$ | $0.15_{\pm0.01}$ | $-$ | $0.17_{\pm0.01}$ |
| Mitra | $0.44_{\pm0.01}$ | $0.43_{\pm0.00}$ | $0.41_{\pm0.03}$ | $0.00_{\pm0.00}$ | $0.88_{\pm0.00}$ | $0.13_{\pm0.01}$ | $-$ | $0.14_{\pm0.01}$ |
| LimiX | $-$ | $-$ | $-$ | $-$ | $-$ | $-$ | $-$ | $-$ |
| TabICL | $-$ | $-$ | $-$ | $-$ | $-$ | $-$ | $-$ | $-$ |
| TabEBM | $\mathbf{0.72}_{\pm0.00}$ | $0.61_{\pm0.01}$ | $0.82_{\pm0.03}$ | $\mathbf{0.19}_{\pm0.01}$ | $0.82_{\pm0.01}$ | $0.12_{\pm0.00}$ | $-$ | $-$ |
| CTSyn | $-$ | $-$ | $-$ | $-$ | $-$ | $-$ | $-$ | $-$ |
| **TabFORGE (Ours)** | $0.52_{\pm0.02}$ | $\mathbf{0.72}_{\pm0.01}$ | $\mathbf{0.87}_{\pm0.02}$ | $0.02_{\pm0.01}$ | $1.00_{\pm0.00}$ | $0.16_{\pm0.00}$ | $-$ | $\mathbf{0.38}_{\pm0.02}$ |

*Table 64.* **Raw benchmark results of 23 tabular generators on the "SDSS17" dataset.** We report the mean $\pm$ std of each metric across 10 repeated data splits. For benchmark generators, "$-$" denotes failed convergence of a specific model or unexpected values in the synthetic data that caused the evaluation metric computation to crash. We highlight the **First**, **Second**, and **Third** best performances for each metric. TabFORGE generally achieves competitive performance against the benchmark generators while maintaining a reduced risk of overfitting.

| Generator | Density Estimation | | | | Privacy Preservation | | ML Efficacy | Structural Fidelity |
|---|---|---|---|---|---|---|---|---|
| | Shape ↑ | Trend ↑ | $\alpha$-precision ↑ | $\beta$-recall ↑ | Authenticity ↑ | DCR Score ↑ | Local utility ↑ | Global utility ↑ |
| **Real Data** | | | | | | | | |
| Real Data (Train) | $1.00_{\pm0.00}$ | $1.00_{\pm0.00}$ | $1.00_{\pm0.00}$ | $1.00_{\pm0.00}$ | $0.00_{\pm0.00}$ | $0.00_{\pm0.00}$ | $1.00_{\pm0.00}$ | $0.93_{\pm0.04}$ |
| Real Data (Holdout) | $0.97_{\pm0.00}$ | $0.58_{\pm0.07}$ | $0.98_{\pm0.00}$ | $0.72_{\pm0.01}$ | $0.49_{\pm0.01}$ | $0.26_{\pm0.06}$ | $-$ | $-$ |
| **Dataset-specific Model** | | | | | | | | |
| SMOTE | $0.95_{\pm0.00}$ | $-$ | $-$ | $0.61_{\pm0.01}$ | $0.45_{\pm0.00}$ | $0.43_{\pm0.10}$ | $0.99_{\pm0.00}$ | $-$ |
| TabSDS | $-$ | $-$ | $-$ | $-$ | $-$ | $-$ | $-$ | $-$ |
| TVAE | $0.88_{\pm0.01}$ | $0.49_{\pm0.09}$ | $0.99_{\pm0.01}$ | $0.11_{\pm0.00}$ | $0.95_{\pm0.00}$ | $0.47_{\pm0.11}$ | $0.92_{\pm0.01}$ | $0.62_{\pm0.03}$ |
| GOGGLE | $0.51_{\pm0.04}$ | $0.23_{\pm0.07}$ | $0.33_{\pm0.06}$ | $0.02_{\pm0.00}$ | $1.00_{\pm0.00}$ | $0.46_{\pm0.11}$ | $0.30_{\pm0.00}$ | $0.19_{\pm0.03}$ |
| CTGAN | $0.87_{\pm0.01}$ | $0.51_{\pm0.09}$ | $0.98_{\pm0.01}$ | $0.11_{\pm0.01}$ | $0.95_{\pm0.00}$ | $0.47_{\pm0.10}$ | $0.91_{\pm0.01}$ | $0.46_{\pm0.11}$ |
| NFlow | $0.80_{\pm0.07}$ | $0.47_{\pm0.08}$ | $0.84_{\pm0.16}$ | $0.09_{\pm0.02}$ | $0.97_{\pm0.01}$ | $0.50_{\pm0.10}$ | $0.78_{\pm0.04}$ | $0.36_{\pm0.12}$ |
| ARF | $-$ | $-$ | $-$ | $-$ | $-$ | $-$ | $-$ | $-$ |
| TabDDPM | $0.94_{\pm0.00}$ | $0.62_{\pm0.06}$ | $0.99_{\pm0.00}$ | $0.13_{\pm0.01}$ | $0.94_{\pm0.01}$ | $0.46_{\pm0.10}$ | $0.94_{\pm0.00}$ | $0.62_{\pm0.03}$ |
| CDTD | $-$ | $-$ | $-$ | $-$ | $-$ | $-$ | $-$ | $-$ |
| TabSyn | $0.92_{\pm0.00}$ | $0.52_{\pm0.11}$ | $0.96_{\pm0.02}$ | $0.13_{\pm0.01}$ | $0.94_{\pm0.00}$ | $0.46_{\pm0.10}$ | $0.93_{\pm0.01}$ | $0.59_{\pm0.04}$ |
| TabDiff | $0.91_{\pm0.00}$ | $0.52_{\pm0.07}$ | $0.90_{\pm0.03}$ | $0.10_{\pm0.01}$ | $0.96_{\pm0.00}$ | $0.51_{\pm0.11}$ | $0.54_{\pm0.05}$ | $0.58_{\pm0.03}$ |
| NRGBoost | $-$ | $-$ | $-$ | $-$ | $-$ | $-$ | $-$ | $-$ |
| TabNAT | $0.70_{\pm0.00}$ | $0.35_{\pm0.07}$ | $0.73_{\pm0.02}$ | $0.19_{\pm0.02}$ | $0.77_{\pm0.00}$ | $0.42_{\pm0.09}$ | $0.67_{\pm0.03}$ | $0.35_{\pm0.03}$ |
| TabularARGN | $0.70_{\pm0.00}$ | $0.42_{\pm0.11}$ | $0.81_{\pm0.02}$ | $0.14_{\pm0.01}$ | $0.90_{\pm0.01}$ | $0.46_{\pm0.10}$ | $0.92_{\pm0.01}$ | $0.50_{\pm0.06}$ |
| **Foundation Model** | | | | | | | | |
| GReaT | $0.83_{\pm0.00}$ | $0.27_{\pm0.09}$ | $0.67_{\pm0.02}$ | $0.42_{\pm0.08}$ | $0.60_{\pm0.01}$ | $0.52_{\pm0.11}$ | $0.83_{\pm0.02}$ | $0.31_{\pm0.04}$ |
| TabPFN | $-$ | $-$ | $-$ | $-$ | $-$ | $-$ | $-$ | $-$ |
| TabDPT | $0.70_{\pm0.01}$ | $0.35_{\pm0.07}$ | $0.63_{\pm0.02}$ | $0.08_{\pm0.00}$ | $0.84_{\pm0.00}$ | $0.35_{\pm0.08}$ | $0.63_{\pm0.01}$ | $0.34_{\pm0.06}$ |
| Mitra | $0.48_{\pm0.03}$ | $0.30_{\pm0.06}$ | $0.55_{\pm0.06}$ | $0.06_{\pm0.01}$ | $0.80_{\pm0.00}$ | $0.39_{\pm0.09}$ | $0.52_{\pm0.01}$ | $0.26_{\pm0.07}$ |
| LimiX | $-$ | $-$ | $-$ | $-$ | $-$ | $-$ | $-$ | $-$ |
| TabICL | $-$ | $-$ | $-$ | $-$ | $-$ | $-$ | $-$ | $-$ |
| TabEBM | $-$ | $-$ | $-$ | $-$ | $-$ | $-$ | $-$ | $-$ |
| CTSyn | $0.71_{\pm0.01}$ | $0.30_{\pm0.07}$ | $0.59_{\pm0.01}$ | $0.19_{\pm0.03}$ | $0.68_{\pm0.00}$ | $0.40_{\pm0.09}$ | $0.69_{\pm0.01}$ | $0.37_{\pm0.04}$ |
| **TabFORGE (Ours)** | $0.94_{\pm0.01}$ | $0.62_{\pm0.07}$ | $0.99_{\pm0.00}$ | $0.19_{\pm0.01}$ | $0.89_{\pm0.01}$ | $0.45_{\pm0.10}$ | $0.95_{\pm0.01}$ | $0.61_{\pm0.10}$ |

*Table 65.* **Raw benchmark results of 23 tabular generators on the "Satisfaction" dataset.** We report the mean $\pm$ std of each metric across 10 repeated data splits. For benchmark generators, "$-$" denotes failed convergence of a specific model or unexpected values in the synthetic data that caused the evaluation metric computation to crash. We highlight the **First**, **Second**, and **Third** best performances for each metric. TabFORGE generally achieves competitive performance against the benchmark generators while maintaining a reduced risk of overfitting.

| Generator | Density Estimation | | | | Privacy Preservation | | ML Efficacy | Structural Fidelity |
|---|---|---|---|---|---|---|---|---|
| | Shape $\uparrow$ | Trend $\uparrow$ | $\alpha$-precision $\uparrow$ | $\beta$-recall $\uparrow$ | Authenticity $\uparrow$ | DCR Score $\uparrow$ | Local utility $\uparrow$ | Global utility $\uparrow$ |
| Real Data | | | | | | | | |
| Real Data (Train) | $1.00_{\pm0.00}$ | $1.00_{\pm0.00}$ | $1.00_{\pm0.00}$ | $1.00_{\pm0.00}$ | $0.00_{\pm0.00}$ | $0.00_{\pm0.00}$ | $1.00_{\pm0.00}$ | $0.99_{\pm0.01}$ |
| Real Data (Holdout) | $1.00_{\pm0.00}$ | $0.99_{\pm0.00}$ | $1.00_{\pm0.00}$ | $0.50_{\pm0.00}$ | $0.73_{\pm0.00}$ | $0.21_{\pm0.00}$ | $-$ | $-$ |
| Dataset-specific Model | | | | | | | | |
| SMOTE | $0.94_{\pm0.00}$ | $-$ | $-$ | $0.64_{\pm0.00}$ | $0.47_{\pm0.00}$ | $0.29_{\pm0.01}$ | $0.99_{\pm0.00}$ | $-$ |
| TabSDS | $0.75_{\pm0.00}$ | $0.57_{\pm0.01}$ | $0.71_{\pm0.01}$ | $0.15_{\pm0.00}$ | $0.69_{\pm0.00}$ | $0.30_{\pm0.01}$ | $0.80_{\pm0.00}$ | $0.61_{\pm0.01}$ |
| TVAE | $0.92_{\pm0.01}$ | $0.85_{\pm0.03}$ | $0.95_{\pm0.00}$ | $0.15_{\pm0.01}$ | $0.94_{\pm0.00}$ | $0.46_{\pm0.01}$ | $0.94_{\pm0.00}$ | $0.72_{\pm0.01}$ |
| GOGGLE | $0.63_{\pm0.02}$ | $0.44_{\pm0.02}$ | $0.14_{\pm0.02}$ | $0.01_{\pm0.00}$ | $1.00_{\pm0.00}$ | $0.46_{\pm0.04}$ | $-$ | $-$ |
| CTGAN | $0.89_{\pm0.02}$ | $0.85_{\pm0.04}$ | $0.96_{\pm0.02}$ | $0.16_{\pm0.02}$ | $0.94_{\pm0.01}$ | $0.43_{\pm0.02}$ | $0.91_{\pm0.01}$ | $0.66_{\pm0.02}$ |
| NFlow | $-$ | $-$ | $-$ | $-$ | $-$ | $-$ | $-$ | $-$ |
| ARF | $0.96_{\pm0.00}$ | $0.86_{\pm0.03}$ | $0.99_{\pm0.00}$ | $0.20_{\pm0.01}$ | $0.91_{\pm0.00}$ | $0.45_{\pm0.01}$ | $0.95_{\pm0.00}$ | $0.63_{\pm0.01}$ |
| TabDDPM | $0.67_{\pm0.02}$ | $0.52_{\pm0.05}$ | $0.16_{\pm0.04}$ | $0.01_{\pm0.00}$ | $1.00_{\pm0.00}$ | $0.82_{\pm0.05}$ | $0.88_{\pm0.01}$ | $0.72_{\pm0.01}$ |
| CDTD | $-$ | $-$ | $-$ | $-$ | $-$ | $-$ | $-$ | $-$ |
| TabSyn | $0.91_{\pm0.01}$ | $0.86_{\pm0.01}$ | $0.94_{\pm0.03}$ | $0.26_{\pm0.01}$ | $0.89_{\pm0.00}$ | $0.38_{\pm0.01}$ | $0.96_{\pm0.00}$ | $0.78_{\pm0.02}$ |
| TabDiff | $0.79_{\pm0.01}$ | $0.58_{\pm0.02}$ | $0.54_{\pm0.03}$ | $0.00_{\pm0.00}$ | $1.00_{\pm0.00}$ | $0.78_{\pm0.02}$ | $0.63_{\pm0.04}$ | $0.79_{\pm0.01}$ |
| NRGBoost | $0.89_{\pm0.00}$ | $0.85_{\pm0.01}$ | $0.90_{\pm0.02}$ | $0.01_{\pm0.00}$ | $0.99_{\pm0.00}$ | $0.38_{\pm0.01}$ | $0.92_{\pm0.01}$ | $-$ |
| TabNAT | $-$ | $-$ | $-$ | $-$ | $-$ | $-$ | $-$ | $-$ |
| TabularARGN | $0.85_{\pm0.00}$ | $0.85_{\pm0.01}$ | $0.97_{\pm0.01}$ | $0.01_{\pm0.00}$ | $1.00_{\pm0.00}$ | $0.39_{\pm0.02}$ | $0.90_{\pm0.00}$ | $0.61_{\pm0.01}$ |
| Foundation Model | | | | | | | | |
| GReaT | $-$ | $-$ | $-$ | $-$ | $-$ | $-$ | $-$ | $-$ |
| TabPFN | $-$ | $-$ | $-$ | $-$ | $-$ | $-$ | $-$ | $-$ |
| TabDPT | $0.68_{\pm0.01}$ | $0.57_{\pm0.02}$ | $0.61_{\pm0.01}$ | $0.08_{\pm0.01}$ | $0.83_{\pm0.00}$ | $0.35_{\pm0.02}$ | $0.60_{\pm0.00}$ | $0.43_{\pm0.01}$ |
| Mitra | $-$ | $-$ | $-$ | $-$ | $-$ | $-$ | $-$ | $-$ |
| LimiX | $-$ | $-$ | $-$ | $-$ | $-$ | $-$ | $-$ | $-$ |
| TabICL | $-$ | $-$ | $-$ | $-$ | $-$ | $-$ | $-$ | $-$ |
| TabEBM | $-$ | $-$ | $-$ | $-$ | $-$ | $-$ | $-$ | $-$ |
| CTSyn | $-$ | $-$ | $-$ | $-$ | $-$ | $-$ | $-$ | $-$ |
| **TabFORGE (Ours)** | $0.94_{\pm0.00}$ | $0.93_{\pm0.03}$ | $1.00_{\pm0.00}$ | $0.50_{\pm0.01}$ | $0.57_{\pm0.00}$ | $0.33_{\pm0.01}$ | $0.98_{\pm0.00}$ | $0.89_{\pm0.01}$ |

*Table 66.* **Raw benchmark results of 23 tabular generators on the "Seismic" dataset.** We report the mean $\pm$ std of each metric across 10 repeated data splits. For benchmark generators, "$-$" denotes failed convergence of a specific model or unexpected values in the synthetic data that caused the evaluation metric computation to crash. We highlight the First, Second, and Third best performances for each metric. TabFORGE generally achieves competitive performance against the benchmark generators while maintaining a reduced risk of overfitting.

| Generator | Density Estimation | | | | Privacy Preservation | | ML Efficacy | Structural Fidelity |
|---|---|---|---|---|---|---|---|---|
| | Shape ↑ | Trend ↑ | $\alpha$-precision ↑ | $\beta$-recall ↑ | Authenticity ↑ | DCR Score ↑ | Local utility ↑ | Global utility ↑ |
| **Real Data** | | | | | | | | |
| Real Data (Train) | $1.00_{\pm 0.00}$ | $1.00_{\pm 0.00}$ | $1.00_{\pm 0.00}$ | $1.00_{\pm 0.00}$ | $0.00_{\pm 0.00}$ | $0.00_{\pm 0.00}$ | $1.00_{\pm 0.00}$ | $0.84_{\pm 0.03}$ |
| Real Data (Holdout) | $0.98_{\pm 0.00}$ | $0.93_{\pm 0.02}$ | $0.97_{\pm 0.01}$ | $0.50_{\pm 0.02}$ | $0.67_{\pm 0.02}$ | $0.01_{\pm 0.00}$ | $-$ | $-$ |
| **Dataset-specific Model** | | | | | | | | |
| SMOTE | $0.67_{\pm 0.01}$ | $0.78_{\pm 0.01}$ | $0.88_{\pm 0.01}$ | $\mathbf{0.63}_{\pm 0.01}$ | $0.47_{\pm 0.01}$ | $0.00_{\pm 0.00}$ | $1.00_{\pm 0.03}$ | $0.51_{\pm 0.03}$ |
| TabSDS | $0.50_{\pm 0.00}$ | $0.58_{\pm 0.02}$ | $0.73_{\pm 0.01}$ | $0.18_{\pm 0.01}$ | $0.61_{\pm 0.01}$ | $0.05_{\pm 0.00}$ | $0.73_{\pm 0.02}$ | $0.38_{\pm 0.03}$ |
| TVAE | $0.63_{\pm 0.02}$ | $0.75_{\pm 0.03}$ | $0.83_{\pm 0.04}$ | $0.20_{\pm 0.01}$ | $0.90_{\pm 0.01}$ | $0.03_{\pm 0.01}$ | $0.97_{\pm 0.03}$ | $\mathbf{0.52}_{\pm 0.04}$ |
| GOGGLE | $0.45_{\pm 0.02}$ | $0.47_{\pm 0.04}$ | $0.14_{\pm 0.02}$ | $0.02_{\pm 0.01}$ | $\mathbf{0.99}_{\pm 0.00}$ | $0.01_{\pm 0.00}$ | $1.00_{\pm 0.00}$ | $0.28_{\pm 0.02}$ |
| CTGAN | $0.59_{\pm 0.03}$ | $0.74_{\pm 0.05}$ | $0.88_{\pm 0.06}$ | $0.14_{\pm 0.04}$ | $0.93_{\pm 0.03}$ | $0.04_{\pm 0.01}$ | $0.97_{\pm 0.02}$ | $0.30_{\pm 0.06}$ |
| NFlow | $0.65_{\pm 0.02}$ | $0.73_{\pm 0.02}$ | $0.78_{\pm 0.06}$ | $0.11_{\pm 0.02}$ | $0.95_{\pm 0.01}$ | $0.08_{\pm 0.02}$ | $0.78_{\pm 0.02}$ | $0.28_{\pm 0.03}$ |
| ARF | $0.68_{\pm 0.01}$ | $0.78_{\pm 0.04}$ | $\mathbf{0.96}_{\pm 0.01}$ | $0.26_{\pm 0.02}$ | $0.86_{\pm 0.01}$ | $0.03_{\pm 0.01}$ | $0.99_{\pm 0.04}$ | $0.38_{\pm 0.02}$ |
| TabDDPM | $0.48_{\pm 0.03}$ | $0.66_{\pm 0.06}$ | $0.13_{\pm 0.12}$ | $0.00_{\pm 0.00}$ | $\mathbf{1.00}_{\pm 0.00}$ | $\mathbf{0.73}_{\pm 0.16}$ | $\mathbf{1.06}_{\pm 0.13}$ | $0.49_{\pm 0.03}$ |
| CDTD | $0.59_{\pm 0.00}$ | $0.64_{\pm 0.02}$ | $0.62_{\pm 0.01}$ | $0.17_{\pm 0.01}$ | $0.83_{\pm 0.01}$ | $0.03_{\pm 0.00}$ | $0.79_{\pm 0.02}$ | $0.35_{\pm 0.03}$ |
| TabSyn | $0.66_{\pm 0.01}$ | $\mathbf{0.91}_{\pm 0.03}$ | $\mathbf{0.95}_{\pm 0.02}$ | $\mathbf{0.43}_{\pm 0.02}$ | $0.72_{\pm 0.02}$ | $0.01_{\pm 0.00}$ | $1.00_{\pm 0.03}$ | $\mathbf{0.54}_{\pm 0.05}$ |
| TabDiff | $0.66_{\pm 0.01}$ | $0.64_{\pm 0.03}$ | $0.28_{\pm 0.03}$ | $0.02_{\pm 0.00}$ | $\mathbf{0.99}_{\pm 0.00}$ | $\mathbf{0.43}_{\pm 0.03}$ | $0.97_{\pm 0.03}$ | $\mathbf{0.54}_{\pm 0.04}$ |
| NRGBoost | $0.66_{\pm 0.00}$ | $\mathbf{0.86}_{\pm 0.02}$ | $0.63_{\pm 0.02}$ | $0.02_{\pm 0.01}$ | $0.98_{\pm 0.01}$ | $\mathbf{0.20}_{\pm 0.02}$ | $0.99_{\pm 0.04}$ | $0.24_{\pm 0.02}$ |
| TabNAT | $0.64_{\pm 0.01}$ | $0.57_{\pm 0.02}$ | $0.53_{\pm 0.02}$ | $0.14_{\pm 0.01}$ | $0.79_{\pm 0.01}$ | $0.15_{\pm 0.01}$ | $0.78_{\pm 0.02}$ | $0.23_{\pm 0.02}$ |
| TabularARGN | $\mathbf{0.76}_{\pm 0.01}$ | $0.72_{\pm 0.02}$ | $0.91_{\pm 0.02}$ | $0.14_{\pm 0.02}$ | $0.93_{\pm 0.02}$ | $0.07_{\pm 0.01}$ | $0.99_{\pm 0.02}$ | $0.35_{\pm 0.05}$ |
| **Foundation Model** | | | | | | | | |
| GReaT | $\mathbf{0.71}_{\pm 0.01}$ | $0.52_{\pm 0.05}$ | $0.70_{\pm 0.04}$ | $0.03_{\pm 0.01}$ | $0.98_{\pm 0.00}$ | $0.02_{\pm 0.00}$ | $0.96_{\pm 0.03}$ | $0.14_{\pm 0.03}$ |
| TabPFN | $0.61_{\pm 0.01}$ | $0.60_{\pm 0.02}$ | $0.79_{\pm 0.03}$ | $0.11_{\pm 0.01}$ | $0.73_{\pm 0.00}$ | $0.05_{\pm 0.01}$ | $0.80_{\pm 0.02}$ | $0.23_{\pm 0.02}$ |
| TabDPT | $0.57_{\pm 0.01}$ | $0.57_{\pm 0.02}$ | $0.64_{\pm 0.03}$ | $0.08_{\pm 0.02}$ | $0.80_{\pm 0.01}$ | $0.03_{\pm 0.01}$ | $1.00_{\pm 0.00}$ | $0.34_{\pm 0.04}$ |
| Mitra | $0.41_{\pm 0.01}$ | $0.47_{\pm 0.02}$ | $0.42_{\pm 0.02}$ | $0.05_{\pm 0.01}$ | $0.85_{\pm 0.00}$ | $0.03_{\pm 0.01}$ | $1.00_{\pm 0.00}$ | $0.33_{\pm 0.02}$ |
| LimiX | $0.56_{\pm 0.01}$ | $0.62_{\pm 0.02}$ | $0.60_{\pm 0.03}$ | $0.13_{\pm 0.01}$ | $0.78_{\pm 0.01}$ | $0.05_{\pm 0.01}$ | $0.91_{\pm 0.02}$ | $0.33_{\pm 0.02}$ |
| TabICL | $0.58_{\pm 0.01}$ | $0.71_{\pm 0.02}$ | $0.81_{\pm 0.02}$ | $0.24_{\pm 0.01}$ | $0.76_{\pm 0.01}$ | $0.03_{\pm 0.00}$ | $0.89_{\pm 0.03}$ | $0.37_{\pm 0.03}$ |
| TabEBM | $\mathbf{0.77}_{\pm 0.00}$ | $0.78_{\pm 0.04}$ | $0.82_{\pm 0.02}$ | $0.08_{\pm 0.01}$ | $0.92_{\pm 0.01}$ | $0.07_{\pm 0.00}$ | $\mathbf{1.01}_{\pm 0.03}$ | $0.47_{\pm 0.04}$ |
| CTSyn | $0.53_{\pm 0.01}$ | $0.57_{\pm 0.03}$ | $0.59_{\pm 0.02}$ | $0.16_{\pm 0.01}$ | $0.71_{\pm 0.01}$ | $0.01_{\pm 0.00}$ | $1.00_{\pm 0.00}$ | $0.31_{\pm 0.02}$ |
| **TabFORGE (Ours)** | $0.63_{\pm 0.01}$ | $\mathbf{0.84}_{\pm 0.03}$ | $\mathbf{0.94}_{\pm 0.03}$ | $\mathbf{0.29}_{\pm 0.03}$ | $0.83_{\pm 0.02}$ | $0.02_{\pm 0.00}$ | $0.99_{\pm 0.03}$ | $0.46_{\pm 0.04}$ |

*Table 67.* **Raw benchmark results of 23 tabular generators on the "Shipping" dataset.** We report the mean ± std of each metric across 10 repeated data splits. For benchmark generators, "−" denotes failed convergence of a specific model or unexpected values in the synthetic data that caused the evaluation metric computation to crash. We highlight the **First**, **Second**, and **Third** best performances for each metric. TabFORGE generally achieves competitive performance against the benchmark generators while maintaining a reduced risk of overfitting.

| Generator | Density Estimation | | | | Privacy Preservation | | ML Efficacy | Structural Fidelity |
|---|---|---|---|---|---|---|---|---|
| | Shape ↑ | Trend ↑ | $\alpha$-precision ↑ | $\beta$-recall ↑ | Authenticity ↑ | DCR Score ↑ | Local utility ↑ | Global utility ↑ |
| **Real Data** | | | | | | | | |
| Real Data (Train) | $1.00_{\pm0.00}$ | $1.00_{\pm0.00}$ | $1.00_{\pm0.00}$ | $1.00_{\pm0.00}$ | $0.00_{\pm0.00}$ | $0.00_{\pm0.00}$ | $1.00_{\pm0.00}$ | $0.68_{\pm0.05}$ |
| Real Data (Holdout) | $0.99_{\pm0.00}$ | $0.93_{\pm0.05}$ | $0.99_{\pm0.00}$ | $0.50_{\pm0.01}$ | $0.65_{\pm0.01}$ | $0.23_{\pm0.00}$ | − | − |
| **Dataset-specific Model** | | | | | | | | |
| SMOTE | $\mathbf{0.95}_{\pm0.00}$ | $0.74_{\pm0.04}$ | $0.77_{\pm0.01}$ | $\mathbf{0.66}_{\pm0.01}$ | $0.43_{\pm0.01}$ | $0.12_{\pm0.01}$ | $\mathbf{0.98}_{\pm0.01}$ | $0.47_{\pm0.03}$ |
| TabSDS | $0.68_{\pm0.01}$ | $0.51_{\pm0.02}$ | $0.71_{\pm0.01}$ | $0.30_{\pm0.01}$ | $0.56_{\pm0.00}$ | $0.28_{\pm0.00}$ | $0.72_{\pm0.01}$ | $0.39_{\pm0.03}$ |
| TVAE | $0.94_{\pm0.01}$ | $0.71_{\pm0.03}$ | $0.93_{\pm0.02}$ | $0.36_{\pm0.01}$ | $0.78_{\pm0.01}$ | $0.39_{\pm0.01}$ | $0.90_{\pm0.02}$ | $0.46_{\pm0.03}$ |
| GOGGLE | $0.64_{\pm0.03}$ | $0.24_{\pm0.05}$ | $0.15_{\pm0.03}$ | $0.11_{\pm0.02}$ | $\mathbf{0.95}_{\pm0.01}$ | $0.25_{\pm0.03}$ | $0.48_{\pm0.00}$ | $0.17_{\pm0.03}$ |
| CTGAN | $0.92_{\pm0.01}$ | $0.72_{\pm0.06}$ | $\mathbf{0.96}_{\pm0.01}$ | $0.34_{\pm0.02}$ | $0.78_{\pm0.01}$ | $0.40_{\pm0.01}$ | $0.96_{\pm0.03}$ | $\mathbf{0.47}_{\pm0.03}$ |
| NFlow | $0.90_{\pm0.02}$ | $\mathbf{0.85}_{\pm0.05}$ | $0.92_{\pm0.02}$ | $0.30_{\pm0.02}$ | $0.83_{\pm0.02}$ | $\mathbf{0.44}_{\pm0.02}$ | $0.81_{\pm0.03}$ | $0.38_{\pm0.03}$ |
| ARF | $\mathbf{0.97}_{\pm0.01}$ | $0.39_{\pm0.05}$ | $\mathbf{0.99}_{\pm0.00}$ | $\mathbf{0.40}_{\pm0.01}$ | $0.73_{\pm0.01}$ | $0.37_{\pm0.01}$ | $\mathbf{0.99}_{\pm0.02}$ | $0.45_{\pm0.02}$ |
| TabDDPM | $0.91_{\pm0.01}$ | $0.81_{\pm0.02}$ | $0.89_{\pm0.03}$ | $0.34_{\pm0.01}$ | $0.81_{\pm0.01}$ | $0.40_{\pm0.01}$ | $0.87_{\pm0.03}$ | $0.41_{\pm0.02}$ |
| CDTD | $0.80_{\pm0.01}$ | $0.55_{\pm0.02}$ | $0.77_{\pm0.01}$ | $0.24_{\pm0.00}$ | $0.72_{\pm0.00}$ | $0.32_{\pm0.00}$ | $0.74_{\pm0.02}$ | $0.42_{\pm0.03}$ |
| TabSyn | $\mathbf{0.96}_{\pm0.01}$ | $0.80_{\pm0.03}$ | $0.94_{\pm0.02}$ | $\mathbf{0.41}_{\pm0.01}$ | $0.73_{\pm0.01}$ | $0.35_{\pm0.01}$ | $\mathbf{0.98}_{\pm0.02}$ | $\mathbf{0.54}_{\pm0.04}$ |
| TabDiff | $0.91_{\pm0.01}$ | $\mathbf{0.83}_{\pm0.02}$ | $0.82_{\pm0.03}$ | $0.20_{\pm0.01}$ | $\mathbf{0.88}_{\pm0.00}$ | $\mathbf{0.57}_{\pm0.01}$ | $0.77_{\pm0.02}$ | $0.47_{\pm0.03}$ |
| NRGBoost | $0.89_{\pm0.00}$ | $\mathbf{0.93}_{\pm0.02}$ | $0.94_{\pm0.03}$ | $0.21_{\pm0.01}$ | $0.86_{\pm0.01}$ | $0.41_{\pm0.02}$ | $0.96_{\pm0.01}$ | $0.21_{\pm0.03}$ |
| TabNAT | $0.75_{\pm0.01}$ | $0.60_{\pm0.01}$ | $0.67_{\pm0.01}$ | $0.23_{\pm0.01}$ | $0.74_{\pm0.00}$ | $\mathbf{0.41}_{\pm0.01}$ | $0.68_{\pm0.02}$ | $0.30_{\pm0.02}$ |
| TabularARGN | $0.83_{\pm0.01}$ | $0.76_{\pm0.05}$ | $\mathbf{0.98}_{\pm0.00}$ | $0.16_{\pm0.01}$ | $\mathbf{0.90}_{\pm0.01}$ | $0.39_{\pm0.01}$ | $0.94_{\pm0.01}$ | $0.35_{\pm0.04}$ |
| **Foundation Model** | | | | | | | | |
| GReaT | $0.90_{\pm0.00}$ | $0.42_{\pm0.05}$ | $0.70_{\pm0.02}$ | $0.24_{\pm0.01}$ | $0.88_{\pm0.00}$ | $0.39_{\pm0.01}$ | $0.76_{\pm0.05}$ | $0.46_{\pm0.05}$ |
| TabPFN | $0.72_{\pm0.01}$ | $0.55_{\pm0.03}$ | $0.75_{\pm0.01}$ | $0.25_{\pm0.01}$ | $0.72_{\pm0.01}$ | $0.35_{\pm0.01}$ | $0.83_{\pm0.02}$ | $0.36_{\pm0.01}$ |
| TabDPT | $0.73_{\pm0.01}$ | $0.49_{\pm0.03}$ | $0.53_{\pm0.01}$ | $0.19_{\pm0.01}$ | $0.70_{\pm0.01}$ | $0.28_{\pm0.01}$ | $0.76_{\pm0.01}$ | $0.32_{\pm0.03}$ |
| Mitra | $0.62_{\pm0.02}$ | $0.51_{\pm0.04}$ | $0.48_{\pm0.01}$ | $0.20_{\pm0.02}$ | $0.79_{\pm0.01}$ | $0.30_{\pm0.01}$ | $0.70_{\pm0.01}$ | $0.28_{\pm0.03}$ |
| LimiX | $0.81_{\pm0.01}$ | $0.53_{\pm0.03}$ | $0.70_{\pm0.01}$ | $0.25_{\pm0.01}$ | $0.65_{\pm0.01}$ | $0.32_{\pm0.01}$ | $0.80_{\pm0.02}$ | $0.34_{\pm0.01}$ |
| TabICL | $0.85_{\pm0.01}$ | $0.53_{\pm0.02}$ | $0.85_{\pm0.01}$ | $0.35_{\pm0.01}$ | $0.74_{\pm0.01}$ | $0.35_{\pm0.01}$ | $0.81_{\pm0.01}$ | $0.42_{\pm0.02}$ |
| TabEBM | $0.91_{\pm0.00}$ | $0.55_{\pm0.04}$ | $0.89_{\pm0.02}$ | $0.20_{\pm0.00}$ | $0.85_{\pm0.00}$ | $0.39_{\pm0.00}$ | $0.96_{\pm0.02}$ | $0.33_{\pm0.03}$ |
| CTSyn | $0.71_{\pm0.01}$ | $0.47_{\pm0.02}$ | $0.61_{\pm0.01}$ | $0.26_{\pm0.01}$ | $0.76_{\pm0.00}$ | $0.30_{\pm0.00}$ | $0.68_{\pm0.02}$ | $0.42_{\pm0.02}$ |
| **TabFORGE (Ours)** | $0.92_{\pm0.01}$ | $0.78_{\pm0.05}$ | $0.95_{\pm0.02}$ | $0.36_{\pm0.01}$ | $0.75_{\pm0.01}$ | $0.36_{\pm0.01}$ | $0.96_{\pm0.01}$ | $\mathbf{0.51}_{\pm0.05}$ |

*Table 68.* **Raw benchmark results of 23 tabular generators on the "Space" dataset.** We report the mean $\pm$ std of each metric across 10 repeated data splits. For benchmark generators, "$-$" denotes failed convergence of a specific model or unexpected values in the synthetic data that caused the evaluation metric computation to crash. We highlight the **First**, **Second**, and **Third** best performances for each metric. TabFORGE generally achieves competitive performance against the benchmark generators while maintaining a reduced risk of overfitting.

| Generator | Density Estimation | | | | Privacy Preservation | | ML Efficacy | Structural Fidelity |
|---|---|---|---|---|---|---|---|---|
| | Shape ↑ | Trend ↑ | $\alpha$-precision ↑ | $\beta$-recall ↑ | Authenticity ↑ | DCR Score ↑ | Local utility ↑ | Global utility ↑ |
| **Real Data** | | | | | | | | |
| Real Data (Train) | $1.00_{\pm0.00}$ | $1.00_{\pm0.00}$ | $1.00_{\pm0.00}$ | $1.00_{\pm0.00}$ | $0.00_{\pm0.00}$ | $0.00_{\pm0.00}$ | $1.00_{\pm0.00}$ | $1.00_{\pm0.00}$ |
| Real Data (Holdout) | $0.96_{\pm0.01}$ | $1.00_{\pm0.00}$ | $0.97_{\pm0.01}$ | $0.50_{\pm0.02}$ | $0.64_{\pm0.02}$ | $0.06_{\pm0.01}$ | $-$ | $-$ |
| **Dataset-specific Model** | | | | | | | | |
| SMOTE | $0.96_{\pm0.01}$ | $-$ | $-$ | $0.64_{\pm0.02}$ | $0.47_{\pm0.02}$ | $0.03_{\pm0.00}$ | $0.93_{\pm0.02}$ | $-$ |
| TabSDS | $-$ | $-$ | $-$ | $-$ | $-$ | $-$ | $-$ | $-$ |
| TVAE | $0.84_{\pm0.01}$ | $0.83_{\pm0.02}$ | $0.80_{\pm0.02}$ | $0.11_{\pm0.02}$ | $0.94_{\pm0.01}$ | $0.15_{\pm0.02}$ | $0.73_{\pm0.03}$ | $-$ |
| GOGGLE | $0.51_{\pm0.07}$ | $0.56_{\pm0.04}$ | $0.64_{\pm0.14}$ | $0.01_{\pm0.00}$ | $1.00_{\pm0.00}$ | $0.17_{\pm0.07}$ | $0.35_{\pm0.04}$ | $0.14_{\pm0.01}$ |
| CTGAN | $0.76_{\pm0.04}$ | $0.92_{\pm0.04}$ | $0.79_{\pm0.10}$ | $0.04_{\pm0.01}$ | $0.98_{\pm0.01}$ | $0.19_{\pm0.05}$ | $0.67_{\pm0.08}$ | $0.40_{\pm0.06}$ |
| NFlow | $0.90_{\pm0.02}$ | $0.83_{\pm0.03}$ | $0.92_{\pm0.04}$ | $0.09_{\pm0.02}$ | $0.96_{\pm0.01}$ | $0.21_{\pm0.03}$ | $0.57_{\pm0.05}$ | $0.43_{\pm0.05}$ |
| ARF | $0.96_{\pm0.00}$ | $0.97_{\pm0.01}$ | $0.97_{\pm0.02}$ | $0.23_{\pm0.02}$ | $0.87_{\pm0.01}$ | $0.13_{\pm0.01}$ | $0.83_{\pm0.03}$ | $-$ |
| TabDDPM | $0.96_{\pm0.01}$ | $0.99_{\pm0.01}$ | $0.98_{\pm0.01}$ | $0.34_{\pm0.01}$ | $0.78_{\pm0.01}$ | $0.10_{\pm0.01}$ | $0.86_{\pm0.05}$ | $-$ |
| CDTD | $-$ | $-$ | $-$ | $-$ | $-$ | $-$ | $-$ | $-$ |
| TabSyn | $-$ | $-$ | $-$ | $-$ | $-$ | $-$ | $-$ | $-$ |
| TabDiff | $-$ | $-$ | $-$ | $-$ | $-$ | $-$ | $-$ | $-$ |
| NRGBoost | $0.90_{\pm0.02}$ | $0.99_{\pm0.01}$ | $0.93_{\pm0.03}$ | $0.53_{\pm0.01}$ | $0.61_{\pm0.01}$ | $0.06_{\pm0.01}$ | $0.86_{\pm0.08}$ | $-$ |
| TabNAT | $-$ | $-$ | $-$ | $-$ | $-$ | $-$ | $-$ | $-$ |
| TabularARGN | $0.79_{\pm0.00}$ | $0.62_{\pm0.02}$ | $0.86_{\pm0.02}$ | $0.02_{\pm0.01}$ | $0.99_{\pm0.00}$ | $0.33_{\pm0.04}$ | $0.59_{\pm0.03}$ | $0.30_{\pm0.03}$ |
| **Foundation Model** | | | | | | | | |
| GReaT | $0.63_{\pm0.01}$ | $0.54_{\pm0.01}$ | $0.01_{\pm0.01}$ | $0.00_{\pm0.00}$ | $1.00_{\pm0.00}$ | $1.00_{\pm0.00}$ | $0.59_{\pm0.04}$ | $0.02_{\pm0.02}$ |
| TabPFN | $0.79_{\pm0.01}$ | $0.66_{\pm0.01}$ | $0.65_{\pm0.02}$ | $0.09_{\pm0.01}$ | $0.74_{\pm0.01}$ | $0.29_{\pm0.02}$ | $0.54_{\pm0.03}$ | $0.39_{\pm0.03}$ |
| TabDPT | $0.64_{\pm0.03}$ | $0.60_{\pm0.02}$ | $0.63_{\pm0.05}$ | $0.02_{\pm0.01}$ | $0.82_{\pm0.00}$ | $0.20_{\pm0.04}$ | $0.57_{\pm0.04}$ | $0.27_{\pm0.03}$ |
| Mitra | $0.64_{\pm0.03}$ | $0.56_{\pm0.02}$ | $0.69_{\pm0.05}$ | $0.04_{\pm0.01}$ | $0.86_{\pm0.01}$ | $0.15_{\pm0.04}$ | $0.52_{\pm0.04}$ | $0.27_{\pm0.02}$ |
| LimiX | $0.75_{\pm0.01}$ | $0.74_{\pm0.01}$ | $0.79_{\pm0.02}$ | $0.10_{\pm0.01}$ | $0.80_{\pm0.01}$ | $0.15_{\pm0.03}$ | $0.63_{\pm0.03}$ | $0.38_{\pm0.02}$ |
| TabICL | $-$ | $-$ | $-$ | $-$ | $-$ | $-$ | $-$ | $-$ |
| TabEBM | $0.95_{\pm0.01}$ | $0.94_{\pm0.00}$ | $0.89_{\pm0.02}$ | $0.04_{\pm0.01}$ | $0.98_{\pm0.00}$ | $0.20_{\pm0.02}$ | $0.79_{\pm0.03}$ | $-$ |
| CTSyn | $-$ | $-$ | $-$ | $-$ | $-$ | $-$ | $-$ | $-$ |
| **TabFORGE (Ours)** | $0.96_{\pm0.00}$ | $1.00_{\pm0.00}$ | $0.97_{\pm0.01}$ | $0.51_{\pm0.01}$ | $0.60_{\pm0.01}$ | $0.04_{\pm0.00}$ | $0.89_{\pm0.04}$ | $0.83_{\pm0.02}$ |

*Table 69.* **Raw benchmark results of 23 tabular generators on the "Students" dataset.** We report the mean ± std of each metric across 10 repeated data splits. For benchmark generators, "−" denotes failed convergence of a specific model or unexpected values in the synthetic data that caused the evaluation metric computation to crash. We highlight the **First**, **Second**, and **Third** best performances for each metric. TabFORGE generally achieves competitive performance against the benchmark generators while maintaining a reduced risk of overfitting.

| Generator | Density Estimation | | | | Privacy Preservation | | ML Efficacy | Structural Fidelity |
|---|---|---|---|---|---|---|---|---|
| | Shape ↑ | Trend ↑ | $\alpha$-precision ↑ | $\beta$-recall ↑ | Authenticity ↑ | DCR Score ↑ | Local utility ↑ | Global utility ↑ |
| **Real Data** | | | | | | | | |
| Real Data (Train) | $1.00_{\pm0.00}$ | $1.00_{\pm0.00}$ | $1.00_{\pm0.00}$ | $1.00_{\pm0.00}$ | $0.00_{\pm0.00}$ | $0.00_{\pm0.00}$ | $1.00_{\pm0.00}$ | $0.77_{\pm0.05}$ |
| Real Data (Holdout) | $0.98_{\pm0.00}$ | $0.94_{\pm0.01}$ | $0.99_{\pm0.01}$ | $0.49_{\pm0.02}$ | $0.77_{\pm0.01}$ | $0.17_{\pm0.00}$ | − | − |
| **Dataset-specific Model** | | | | | | | | |
| SMOTE | $\mathbf{0.84}_{\pm0.00}$ | $0.66_{\pm0.01}$ | $0.72_{\pm0.01}$ | $\mathbf{0.64}_{\pm0.01}$ | $0.48_{\pm0.01}$ | $0.01_{\pm0.00}$ | $\mathbf{0.96}_{\pm0.01}$ | $\mathbf{0.53}_{\pm0.03}$ |
| TabSDS | − | − | − | − | − | − | − | − |
| TVAE | $0.73_{\pm0.02}$ | $0.66_{\pm0.02}$ | $0.54_{\pm0.09}$ | $0.04_{\pm0.02}$ | $0.99_{\pm0.01}$ | $0.41_{\pm0.02}$ | $0.85_{\pm0.02}$ | $0.51_{\pm0.03}$ |
| GOGGLE | $0.52_{\pm0.02}$ | $0.48_{\pm0.03}$ | $0.63_{\pm0.16}$ | $0.01_{\pm0.01}$ | $\mathbf{1.00}_{\pm0.00}$ | $0.33_{\pm0.05}$ | $0.35_{\pm0.02}$ | $0.24_{\pm0.02}$ |
| CTGAN | $0.71_{\pm0.03}$ | $0.74_{\pm0.02}$ | $0.43_{\pm0.15}$ | $0.01_{\pm0.01}$ | $1.00_{\pm0.00}$ | $0.43_{\pm0.03}$ | $0.86_{\pm0.02}$ | $0.40_{\pm0.03}$ |
| NFlow | $0.72_{\pm0.02}$ | $0.64_{\pm0.02}$ | $0.37_{\pm0.10}$ | $0.01_{\pm0.01}$ | $\mathbf{1.00}_{\pm0.00}$ | $\mathbf{0.49}_{\pm0.02}$ | $0.59_{\pm0.04}$ | $0.28_{\pm0.03}$ |
| ARF | − | − | − | − | − | − | − | − |
| TabDDPM | $0.42_{\pm0.02}$ | $0.55_{\pm0.02}$ | $0.00_{\pm0.00}$ | $0.00_{\pm0.00}$ | $\mathbf{1.00}_{\pm0.00}$ | $\mathbf{0.98}_{\pm0.02}$ | $0.72_{\pm0.05}$ | $0.46_{\pm0.02}$ |
| CDTD | $0.71_{\pm0.00}$ | $0.68_{\pm0.01}$ | $0.69_{\pm0.02}$ | $0.13_{\pm0.01}$ | $0.77_{\pm0.00}$ | $0.24_{\pm0.00}$ | $0.63_{\pm0.01}$ | $0.39_{\pm0.03}$ |
| TabSyn | $\mathbf{0.81}_{\pm0.01}$ | $\mathbf{0.87}_{\pm0.02}$ | $\mathbf{0.95}_{\pm0.02}$ | $0.22_{\pm0.03}$ | $0.92_{\pm0.01}$ | $0.29_{\pm0.01}$ | $\mathbf{0.93}_{\pm0.01}$ | $\mathbf{0.55}_{\pm0.05}$ |
| TabDiff | $0.79_{\pm0.01}$ | $0.73_{\pm0.02}$ | $0.40_{\pm0.07}$ | $0.03_{\pm0.01}$ | $0.99_{\pm0.00}$ | $\mathbf{0.50}_{\pm0.02}$ | $0.67_{\pm0.05}$ | $0.48_{\pm0.03}$ |
| NRGBoost | $0.77_{\pm0.01}$ | $\mathbf{0.87}_{\pm0.01}$ | $0.91_{\pm0.01}$ | $\mathbf{0.30}_{\pm0.01}$ | $0.87_{\pm0.01}$ | $0.35_{\pm0.01}$ | $\mathbf{0.94}_{\pm0.02}$ | $0.26_{\pm0.02}$ |
| TabNAT | $0.64_{\pm0.00}$ | $0.60_{\pm0.01}$ | $0.61_{\pm0.04}$ | $0.08_{\pm0.01}$ | $0.75_{\pm0.00}$ | $0.30_{\pm0.01}$ | $0.61_{\pm0.02}$ | $0.26_{\pm0.03}$ |
| TabularARGN | $0.79_{\pm0.00}$ | $\mathbf{0.81}_{\pm0.01}$ | $\mathbf{0.94}_{\pm0.01}$ | $\mathbf{0.23}_{\pm0.01}$ | $0.92_{\pm0.01}$ | $0.29_{\pm0.01}$ | $0.89_{\pm0.02}$ | $0.47_{\pm0.03}$ |
| **Foundation Model** | | | | | | | | |
| GReaT | $0.69_{\pm0.01}$ | $0.50_{\pm0.01}$ | $0.85_{\pm0.07}$ | $0.05_{\pm0.01}$ | $0.99_{\pm0.00}$ | $0.42_{\pm0.02}$ | $0.59_{\pm0.06}$ | $0.11_{\pm0.03}$ |
| TabPFN | − | − | − | − | − | − | − | − |
| TabDPT | $0.58_{\pm0.02}$ | $0.56_{\pm0.01}$ | $0.46_{\pm0.11}$ | $0.04_{\pm0.00}$ | $0.83_{\pm0.00}$ | $0.32_{\pm0.01}$ | $0.70_{\pm0.01}$ | $0.37_{\pm0.02}$ |
| Mitra | $0.54_{\pm0.01}$ | $0.52_{\pm0.02}$ | $0.44_{\pm0.08}$ | $0.02_{\pm0.00}$ | $0.82_{\pm0.00}$ | $0.33_{\pm0.02}$ | $0.59_{\pm0.03}$ | $0.31_{\pm0.02}$ |
| LimiX | − | − | − | − | − | − | − | − |
| TabICL | − | − | − | − | − | − | − | − |
| TabEBM | $\mathbf{0.83}_{\pm0.00}$ | $0.79_{\pm0.02}$ | $0.54_{\pm0.01}$ | $0.19_{\pm0.01}$ | $0.82_{\pm0.01}$ | $0.25_{\pm0.00}$ | $0.90_{\pm0.02}$ | $0.47_{\pm0.02}$ |
| CTSyn | $0.60_{\pm0.00}$ | $0.56_{\pm0.01}$ | $0.77_{\pm0.03}$ | $0.09_{\pm0.01}$ | $0.81_{\pm0.00}$ | $0.27_{\pm0.01}$ | $0.65_{\pm0.02}$ | $0.30_{\pm0.02}$ |
| **TabFORGE (Ours)** | $0.80_{\pm0.01}$ | $0.80_{\pm0.01}$ | $\mathbf{0.94}_{\pm0.03}$ | $0.20_{\pm0.02}$ | $0.94_{\pm0.01}$ | $0.31_{\pm0.01}$ | $0.91_{\pm0.02}$ | $\mathbf{0.54}_{\pm0.04}$ |

*Table 70.* **Raw benchmark results of 23 tabular generators on the "Supercond" dataset.** We report the mean ± std of each metric across 10 repeated data splits. For benchmark generators, "−" denotes failed convergence of a specific model or unexpected values in the synthetic data that caused the evaluation metric computation to crash. We highlight the **First**, **Second**, and **Third** best performances for each metric. TabFORGE generally achieves competitive performance against the benchmark generators while maintaining a reduced risk of overfitting.

| Generator | Density Estimation | | | | Privacy Preservation | | ML Efficacy | Structural Fidelity |
| --- | --- | --- | --- | --- | --- | --- | --- | --- |
| | Shape ↑ | Trend ↑ | $\alpha$-precision ↑ | $\beta$-recall ↑ | Authenticity ↑ | DCR Score ↑ | Local utility ↑ | Global utility ↑ |
| **Real Data** | | | | | | | | |
| Real Data (Train) | $1.00_{\pm0.00}$ | $1.00_{\pm0.00}$ | $1.00_{\pm0.00}$ | $1.00_{\pm0.00}$ | $0.02_{\pm0.00}$ | $0.00_{\pm0.00}$ | $1.00_{\pm0.00}$ | $1.00_{\pm0.00}$ |
| Real Data (Holdout) | $0.99_{\pm0.00}$ | $1.00_{\pm0.00}$ | $0.99_{\pm0.00}$ | $0.63_{\pm0.01}$ | $0.52_{\pm0.01}$ | $0.01_{\pm0.00}$ | − | − |
| **Dataset-specific Model** | | | | | | | | |
| SMOTE | $0.97_{\pm0.00}$ | $0.87_{\pm0.00}$ | $0.82_{\pm0.01}$ | $0.40_{\pm0.01}$ | $0.72_{\pm0.01}$ | $0.01_{\pm0.00}$ | $0.90_{\pm0.01}$ | $0.51_{\pm0.01}$ |
| TabSDS | $0.61_{\pm0.00}$ | $0.67_{\pm0.00}$ | $0.58_{\pm0.01}$ | $0.01_{\pm0.00}$ | $0.79_{\pm0.00}$ | $0.14_{\pm0.00}$ | $0.48_{\pm0.01}$ | $0.26_{\pm0.01}$ |
| TVAE | $0.88_{\pm0.01}$ | $0.88_{\pm0.01}$ | $0.90_{\pm0.01}$ | $0.00_{\pm0.00}$ | $1.00_{\pm0.00}$ | $0.24_{\pm0.01}$ | $0.58_{\pm0.03}$ | $0.43_{\pm0.01}$ |
| GOGGLE | − | − | − | − | − | − | − | − |
| CTGAN | $0.86_{\pm0.02}$ | $0.91_{\pm0.01}$ | $0.87_{\pm0.02}$ | $0.00_{\pm0.00}$ | $1.00_{\pm0.00}$ | $0.29_{\pm0.01}$ | $0.49_{\pm0.04}$ | $0.15_{\pm0.01}$ |
| NFlow | $0.86_{\pm0.01}$ | $0.72_{\pm0.02}$ | $0.61_{\pm0.02}$ | $0.00_{\pm0.00}$ | $1.00_{\pm0.00}$ | $0.46_{\pm0.02}$ | $0.33_{\pm0.02}$ | $0.09_{\pm0.02}$ |
| ARF | $0.96_{\pm0.00}$ | $0.96_{\pm0.00}$ | $0.96_{\pm0.01}$ | $0.02_{\pm0.01}$ | $1.00_{\pm0.00}$ | $0.15_{\pm0.01}$ | $0.74_{\pm0.01}$ | $0.34_{\pm0.01}$ |
| TabDDPM | $0.40_{\pm0.01}$ | $0.73_{\pm0.01}$ | $0.00_{\pm0.00}$ | $0.00_{\pm0.00}$ | $1.00_{\pm0.00}$ | $1.00_{\pm0.00}$ | $0.23_{\pm0.06}$ | $0.47_{\pm0.01}$ |
| CDTD | − | − | − | − | − | − | − | − |
| TabSyn | $0.92_{\pm0.01}$ | $0.98_{\pm0.01}$ | $0.90_{\pm0.04}$ | $0.01_{\pm0.01}$ | $1.00_{\pm0.00}$ | $0.12_{\pm0.02}$ | $0.70_{\pm0.03}$ | $0.40_{\pm0.03}$ |
| TabDiff | $0.94_{\pm0.01}$ | $0.72_{\pm0.01}$ | $0.60_{\pm0.02}$ | $0.00_{\pm0.00}$ | $1.00_{\pm0.00}$ | $0.45_{\pm0.01}$ | $0.39_{\pm0.01}$ | $0.50_{\pm0.01}$ |
| NRGBoost | $0.88_{\pm0.01}$ | $0.90_{\pm0.00}$ | $0.65_{\pm0.03}$ | $0.00_{\pm0.00}$ | $1.00_{\pm0.00}$ | $0.34_{\pm0.01}$ | $0.41_{\pm0.04}$ | $0.11_{\pm0.00}$ |
| TabNAT | − | − | − | − | − | − | − | − |
| TabularARGN | $0.82_{\pm0.00}$ | $0.92_{\pm0.00}$ | $0.89_{\pm0.01}$ | $0.00_{\pm0.00}$ | $1.00_{\pm0.00}$ | $0.26_{\pm0.01}$ | $0.53_{\pm0.02}$ | $0.15_{\pm0.01}$ |
| **Foundation Model** | | | | | | | | |
| GReaT | − | − | − | − | − | − | − | − |
| TabPFN | − | − | − | − | − | − | − | − |
| TabDPT | − | − | − | − | − | − | − | − |
| Mitra | − | − | − | − | − | − | − | − |
| LimiX | − | − | − | − | − | − | − | − |
| TabICL | $0.72_{\pm0.00}$ | $0.71_{\pm0.01}$ | $0.71_{\pm0.01}$ | $0.01_{\pm0.00}$ | $0.83_{\pm0.00}$ | $0.22_{\pm0.01}$ | $0.51_{\pm0.02}$ | $0.27_{\pm0.01}$ |
| TabEBM | $0.88_{\pm0.00}$ | $0.95_{\pm0.00}$ | $0.78_{\pm0.02}$ | $0.00_{\pm0.00}$ | $1.00_{\pm0.00}$ | $0.20_{\pm0.00}$ | $0.55_{\pm0.02}$ | $0.27_{\pm0.01}$ |
| CTSyn | − | − | − | − | − | − | − | − |
| **TabFORGE (Ours)** | $0.95_{\pm0.01}$ | $0.99_{\pm0.00}$ | $0.97_{\pm0.02}$ | $0.04_{\pm0.00}$ | $0.97_{\pm0.00}$ | $0.07_{\pm0.00}$ | $0.77_{\pm0.02}$ | $0.55_{\pm0.01}$ |

*Table 71.* **Raw benchmark results of 23 tabular generators on the "Transfusion" dataset.** We report the mean $\pm$ std of each metric across 10 repeated data splits. For benchmark generators, "$-$" denotes failed convergence of a specific model or unexpected values in the synthetic data that caused the evaluation metric computation to crash. We highlight the First, Second, and Third best performances for each metric. TabFORGE generally achieves competitive performance against the benchmark generators while maintaining a reduced risk of overfitting.

| Generator | Density Estimation | | | | Privacy Preservation | | ML Efficacy | Structural Fidelity |
| --- | --- | --- | --- | --- | --- | --- | --- | --- |
| | Shape $\uparrow$ | Trend $\uparrow$ | $\alpha$-precision $\uparrow$ | $\beta$-recall $\uparrow$ | Authenticity $\uparrow$ | DCR Score $\uparrow$ | Local utility $\uparrow$ | Global utility $\uparrow$ |
| **Real Data** | | | | | | | | |
| Real Data (Train) | $1.00_{\pm0.00}$ | $1.00_{\pm0.00}$ | $0.99_{\pm0.00}$ | $0.99_{\pm0.00}$ | $0.00_{\pm0.00}$ | $0.00_{\pm0.00}$ | $1.00_{\pm0.00}$ | $0.86_{\pm0.09}$ |
| Real Data (Holdout) | $0.95_{\pm0.01}$ | $0.94_{\pm0.04}$ | $0.95_{\pm0.02}$ | $0.62_{\pm0.06}$ | $0.48_{\pm0.05}$ | $0.04_{\pm0.01}$ | $-$ | $-$ |
| **Dataset-specific Model** | | | | | | | | |
| SMOTE | $0.87_{\pm0.02}$ | $0.79_{\pm0.03}$ | $0.76_{\pm0.01}$ | $0.47_{\pm0.05}$ | $0.63_{\pm0.04}$ | $0.02_{\pm0.00}$ | $0.99_{\pm0.06}$ | $0.57_{\pm0.06}$ |
| TabSDS | $0.56_{\pm0.01}$ | $0.65_{\pm0.03}$ | $0.61_{\pm0.01}$ | $0.22_{\pm0.02}$ | $0.59_{\pm0.01}$ | $0.06_{\pm0.02}$ | $0.63_{\pm0.03}$ | $0.48_{\pm0.04}$ |
| TVAE | $0.81_{\pm0.02}$ | $0.78_{\pm0.04}$ | $0.83_{\pm0.06}$ | $0.20_{\pm0.03}$ | $0.89_{\pm0.02}$ | $0.11_{\pm0.03}$ | $0.89_{\pm0.05}$ | $0.51_{\pm0.05}$ |
| GOGGLE | $0.65_{\pm0.02}$ | $0.60_{\pm0.03}$ | $0.80_{\pm0.11}$ | $0.11_{\pm0.03}$ | $0.95_{\pm0.02}$ | $0.20_{\pm0.07}$ | $0.44_{\pm0.03}$ | $0.13_{\pm0.03}$ |
| CTGAN | $0.65_{\pm0.05}$ | $0.73_{\pm0.09}$ | $0.75_{\pm0.14}$ | $0.11_{\pm0.04}$ | $0.95_{\pm0.02}$ | $0.14_{\pm0.04}$ | $0.88_{\pm0.09}$ | $0.33_{\pm0.10}$ |
| NFlow | $0.84_{\pm0.04}$ | $0.79_{\pm0.06}$ | $0.90_{\pm0.04}$ | $0.18_{\pm0.02}$ | $0.90_{\pm0.01}$ | $0.14_{\pm0.04}$ | $0.76_{\pm0.04}$ | $0.40_{\pm0.07}$ |
| ARF | $0.88_{\pm0.03}$ | $0.88_{\pm0.06}$ | $0.94_{\pm0.03}$ | $0.25_{\pm0.04}$ | $0.87_{\pm0.03}$ | $0.09_{\pm0.03}$ | $0.94_{\pm0.06}$ | $0.40_{\pm0.04}$ |
| TabDDPM | $0.84_{\pm0.02}$ | $0.94_{\pm0.03}$ | $0.81_{\pm0.03}$ | $0.26_{\pm0.03}$ | $0.83_{\pm0.02}$ | $0.09_{\pm0.03}$ | $0.97_{\pm0.05}$ | $0.59_{\pm0.06}$ |
| CDTD | $0.62_{\pm0.01}$ | $0.67_{\pm0.03}$ | $0.62_{\pm0.02}$ | $0.14_{\pm0.02}$ | $0.72_{\pm0.01}$ | $0.11_{\pm0.03}$ | $0.79_{\pm0.04}$ | $0.38_{\pm0.04}$ |
| TabSyn | $0.85_{\pm0.02}$ | $0.95_{\pm0.03}$ | $0.92_{\pm0.03}$ | $0.29_{\pm0.04}$ | $0.82_{\pm0.03}$ | $0.07_{\pm0.02}$ | $0.97_{\pm0.06}$ | $0.62_{\pm0.07}$ |
| TabDiff | $0.84_{\pm0.03}$ | $0.81_{\pm0.02}$ | $0.83_{\pm0.05}$ | $0.13_{\pm0.02}$ | $0.93_{\pm0.01}$ | $0.17_{\pm0.06}$ | $0.85_{\pm0.05}$ | $0.58_{\pm0.06}$ |
| NRGBoost | $0.83_{\pm0.01}$ | $0.94_{\pm0.03}$ | $0.90_{\pm0.02}$ | $0.38_{\pm0.02}$ | $0.68_{\pm0.02}$ | $0.03_{\pm0.01}$ | $0.94_{\pm0.05}$ | $0.32_{\pm0.03}$ |
| TabNAT | $0.62_{\pm0.01}$ | $0.65_{\pm0.02}$ | $0.64_{\pm0.03}$ | $0.12_{\pm0.01}$ | $0.70_{\pm0.01}$ | $0.12_{\pm0.04}$ | $0.72_{\pm0.04}$ | $0.28_{\pm0.04}$ |
| TabularARGN | $0.78_{\pm0.02}$ | $0.73_{\pm0.05}$ | $0.92_{\pm0.04}$ | $0.32_{\pm0.07}$ | $0.78_{\pm0.04}$ | $0.12_{\pm0.03}$ | $0.90_{\pm0.04}$ | $0.40_{\pm0.06}$ |
| **Foundation Model** | | | | | | | | |
| GReaT | $0.53_{\pm0.02}$ | $0.59_{\pm0.05}$ | $0.54_{\pm0.10}$ | $0.01_{\pm0.01}$ | $0.99_{\pm0.00}$ | $0.15_{\pm0.05}$ | $0.86_{\pm0.11}$ | $0.05_{\pm0.06}$ |
| TabPFN | $0.76_{\pm0.02}$ | $0.63_{\pm0.04}$ | $0.72_{\pm0.02}$ | $0.14_{\pm0.01}$ | $0.76_{\pm0.01}$ | $0.11_{\pm0.03}$ | $0.84_{\pm0.04}$ | $0.37_{\pm0.05}$ |
| TabDPT | $0.52_{\pm0.02}$ | $0.63_{\pm0.05}$ | $0.64_{\pm0.06}$ | $0.13_{\pm0.03}$ | $0.76_{\pm0.01}$ | $0.14_{\pm0.04}$ | $0.74_{\pm0.05}$ | $0.25_{\pm0.05}$ |
| Mitra | $0.69_{\pm0.02}$ | $0.54_{\pm0.03}$ | $0.74_{\pm0.04}$ | $0.14_{\pm0.02}$ | $0.79_{\pm0.01}$ | $0.14_{\pm0.05}$ | $0.67_{\pm0.03}$ | $0.25_{\pm0.04}$ |
| LimiX | $0.72_{\pm0.02}$ | $0.69_{\pm0.04}$ | $0.84_{\pm0.02}$ | $0.16_{\pm0.01}$ | $0.76_{\pm0.01}$ | $0.13_{\pm0.04}$ | $0.84_{\pm0.04}$ | $0.33_{\pm0.04}$ |
| TabICL | $0.71_{\pm0.01}$ | $0.75_{\pm0.03}$ | $0.81_{\pm0.02}$ | $0.23_{\pm0.03}$ | $0.78_{\pm0.02}$ | $0.09_{\pm0.03}$ | $0.79_{\pm0.03}$ | $0.45_{\pm0.04}$ |
| TabEBM | $0.85_{\pm0.02}$ | $0.90_{\pm0.06}$ | $0.89_{\pm0.04}$ | $0.12_{\pm0.02}$ | $0.93_{\pm0.02}$ | $0.18_{\pm0.05}$ | $0.99_{\pm0.07}$ | $0.49_{\pm0.07}$ |
| CTSyn | $0.55_{\pm0.01}$ | $0.56_{\pm0.02}$ | $0.58_{\pm0.04}$ | $0.13_{\pm0.02}$ | $0.74_{\pm0.01}$ | $0.11_{\pm0.03}$ | $0.85_{\pm0.06}$ | $0.26_{\pm0.04}$ |
| **TabFORGE (Ours)** | $0.87_{\pm0.02}$ | $0.97_{\pm0.02}$ | $0.95_{\pm0.02}$ | $0.37_{\pm0.03}$ | $0.72_{\pm0.03}$ | $0.03_{\pm0.01}$ | $0.96_{\pm0.06}$ | $0.71_{\pm0.09}$ |

*Table 72.* **Raw benchmark results of 23 tabular generators on the "Vehicle" dataset.** We report the mean $\pm$ std of each metric across 10 repeated data splits. For benchmark generators, "$-$" denotes failed convergence of a specific model or unexpected values in the synthetic data that caused the evaluation metric computation to crash. We highlight the **First**, **Second**, and **Third** best performances for each metric. TabFORGE generally achieves competitive performance against the benchmark generators while maintaining a reduced risk of overfitting.

| Generator | Density Estimation | | | | Privacy Preservation | | ML Efficacy | Structural Fidelity |
|---|---|---|---|---|---|---|---|---|
| | Shape ↑ | Trend ↑ | $\alpha$-precision ↑ | $\beta$-recall ↑ | Authenticity ↑ | DCR Score ↑ | Local utility ↑ | Global utility ↑ |
| **Real Data** | | | | | | | | |
| Real Data (Train) | $1.00_{\pm0.00}$ | $1.00_{\pm0.00}$ | $1.00_{\pm0.00}$ | $1.00_{\pm0.00}$ | $0.00_{\pm0.00}$ | $0.00_{\pm0.00}$ | $1.00_{\pm0.00}$ | $0.99_{\pm0.02}$ |
| Real Data (Holdout) | $0.95_{\pm0.01}$ | $0.95_{\pm0.01}$ | $0.94_{\pm0.03}$ | $0.50_{\pm0.04}$ | $0.66_{\pm0.02}$ | $0.13_{\pm0.01}$ | $-$ | $-$ |
| **Dataset-specific Model** | | | | | | | | |
| SMOTE | $0.92_{\pm0.01}$ | $0.78_{\pm0.00}$ | $0.74_{\pm0.02}$ | $0.70_{\pm0.03}$ | $0.38_{\pm0.02}$ | $0.06_{\pm0.00}$ | $0.94_{\pm0.03}$ | $0.55_{\pm0.01}$ |
| TabSDS | $0.68_{\pm0.00}$ | $0.71_{\pm0.01}$ | $0.67_{\pm0.02}$ | $0.22_{\pm0.02}$ | $0.61_{\pm0.02}$ | $0.16_{\pm0.01}$ | $0.59_{\pm0.03}$ | $0.40_{\pm0.02}$ |
| TVAE | $0.78_{\pm0.02}$ | $0.72_{\pm0.01}$ | $0.70_{\pm0.04}$ | $0.05_{\pm0.02}$ | $0.98_{\pm0.01}$ | $0.34_{\pm0.02}$ | $0.56_{\pm0.08}$ | $0.50_{\pm0.02}$ |
| GOGGLE | $0.60_{\pm0.02}$ | $0.54_{\pm0.01}$ | $0.79_{\pm0.05}$ | $0.00_{\pm0.00}$ | $1.00_{\pm0.00}$ | $0.50_{\pm0.02}$ | $0.21_{\pm0.02}$ | $0.04_{\pm0.01}$ |
| CTGAN | $0.72_{\pm0.04}$ | $0.85_{\pm0.03}$ | $0.83_{\pm0.09}$ | $0.01_{\pm0.01}$ | $1.00_{\pm0.00}$ | $0.42_{\pm0.03}$ | $0.48_{\pm0.08}$ | $0.18_{\pm0.03}$ |
| NFlow | $0.87_{\pm0.01}$ | $0.74_{\pm0.02}$ | $0.85_{\pm0.02}$ | $0.01_{\pm0.01}$ | $1.00_{\pm0.00}$ | $0.47_{\pm0.02}$ | $0.35_{\pm0.08}$ | $0.24_{\pm0.01}$ |
| ARF | $0.91_{\pm0.01}$ | $0.89_{\pm0.01}$ | $0.93_{\pm0.03}$ | $0.10_{\pm0.02}$ | $0.95_{\pm0.01}$ | $0.31_{\pm0.01}$ | $0.76_{\pm0.05}$ | $0.46_{\pm0.02}$ |
| TabDDPM | $0.62_{\pm0.04}$ | $0.80_{\pm0.03}$ | $0.37_{\pm0.05}$ | $0.11_{\pm0.02}$ | $0.94_{\pm0.01}$ | $0.57_{\pm0.09}$ | $0.73_{\pm0.07}$ | $0.57_{\pm0.01}$ |
| CDTD | $0.63_{\pm0.04}$ | $0.54_{\pm0.00}$ | $0.75_{\pm0.04}$ | $0.18_{\pm0.02}$ | $0.73_{\pm0.02}$ | $0.33_{\pm0.01}$ | $0.53_{\pm0.03}$ | $0.47_{\pm0.02}$ |
| TabSyn | $0.90_{\pm0.01}$ | $0.94_{\pm0.01}$ | $0.93_{\pm0.04}$ | $0.43_{\pm0.07}$ | $0.71_{\pm0.06}$ | $0.19_{\pm0.03}$ | $0.85_{\pm0.10}$ | $0.59_{\pm0.05}$ |
| TabDiff | $0.89_{\pm0.01}$ | $0.77_{\pm0.01}$ | $0.85_{\pm0.05}$ | $0.01_{\pm0.01}$ | $1.00_{\pm0.01}$ | $0.47_{\pm0.03}$ | $0.45_{\pm0.06}$ | $0.56_{\pm0.01}$ |
| NRGBoost | $0.82_{\pm0.01}$ | $0.87_{\pm0.02}$ | $0.67_{\pm0.05}$ | $0.48_{\pm0.03}$ | $0.63_{\pm0.02}$ | $0.04_{\pm0.00}$ | $0.89_{\pm0.05}$ | $0.25_{\pm0.02}$ |
| TabNAT | $0.66_{\pm0.03}$ | $0.45_{\pm0.01}$ | $0.61_{\pm0.05}$ | $0.12_{\pm0.01}$ | $0.83_{\pm0.02}$ | $0.45_{\pm0.00}$ | $0.44_{\pm0.04}$ | $0.34_{\pm0.02}$ |
| TabularARGN | $0.69_{\pm0.01}$ | $0.70_{\pm0.02}$ | $0.86_{\pm0.05}$ | $0.01_{\pm0.01}$ | $1.00_{\pm0.00}$ | $0.43_{\pm0.04}$ | $0.65_{\pm0.06}$ | $0.23_{\pm0.03}$ |
| **Foundation Model** | | | | | | | | |
| GReaT | $0.37_{\pm0.15}$ | $0.02_{\pm0.01}$ | $0.49_{\pm0.12}$ | $0.01_{\pm0.00}$ | $0.99_{\pm0.00}$ | $1.00_{\pm0.00}$ | $0.35_{\pm0.04}$ | $0.41_{\pm0.02}$ |
| TabPFN | $0.68_{\pm0.02}$ | $0.53_{\pm0.01}$ | $0.71_{\pm0.02}$ | $0.03_{\pm0.01}$ | $0.88_{\pm0.00}$ | $0.45_{\pm0.01}$ | $0.44_{\pm0.06}$ | $0.31_{\pm0.01}$ |
| TabDPT | $0.65_{\pm0.02}$ | $0.68_{\pm0.02}$ | $0.72_{\pm0.04}$ | $0.01_{\pm0.00}$ | $0.89_{\pm0.00}$ | $0.40_{\pm0.01}$ | $0.41_{\pm0.05}$ | $0.13_{\pm0.01}$ |
| Mitra | $0.62_{\pm0.01}$ | $0.53_{\pm0.01}$ | $0.69_{\pm0.03}$ | $0.04_{\pm0.00}$ | $0.73_{\pm0.00}$ | $0.39_{\pm0.01}$ | $0.38_{\pm0.04}$ | $0.11_{\pm0.01}$ |
| LimiX | $0.72_{\pm0.01}$ | $0.62_{\pm0.01}$ | $0.79_{\pm0.02}$ | $0.03_{\pm0.01}$ | $0.73_{\pm0.00}$ | $0.39_{\pm0.01}$ | $0.44_{\pm0.05}$ | $0.21_{\pm0.01}$ |
| TabICL | $0.74_{\pm0.01}$ | $0.74_{\pm0.01}$ | $0.71_{\pm0.01}$ | $0.14_{\pm0.02}$ | $0.77_{\pm0.01}$ | $0.30_{\pm0.01}$ | $0.65_{\pm0.05}$ | $0.38_{\pm0.01}$ |
| TabEBM | $0.90_{\pm0.01}$ | $0.89_{\pm0.01}$ | $0.88_{\pm0.04}$ | $0.15_{\pm0.02}$ | $0.89_{\pm0.02}$ | $0.25_{\pm0.01}$ | $0.85_{\pm0.06}$ | $0.41_{\pm0.02}$ |
| CTSyn | $0.50_{\pm0.06}$ | $0.37_{\pm0.00}$ | $0.62_{\pm0.05}$ | $0.14_{\pm0.02}$ | $0.72_{\pm0.02}$ | $0.49_{\pm0.01}$ | $0.44_{\pm0.03}$ | $0.37_{\pm0.02}$ |
| **TabFORGE (Ours)** | $0.91_{\pm0.01}$ | $0.94_{\pm0.00}$ | $0.91_{\pm0.05}$ | $0.44_{\pm0.02}$ | $0.60_{\pm0.02}$ | $0.11_{\pm0.01}$ | $0.93_{\pm0.04}$ | $0.63_{\pm0.04}$ |

*Table 73.* **Raw benchmark results of 23 tabular generators on the "Wine" dataset.** We report the mean $\pm$ std of each metric across 10 repeated data splits. For benchmark generators, "$-$" denotes failed convergence of a specific model or unexpected values in the synthetic data that caused the evaluation metric computation to crash. We highlight the **First**, **Second**, and **Third** best performances for each metric. TabFORGE generally achieves competitive performance against the benchmark generators while maintaining a reduced risk of overfitting.

| Generator | Density Estimation | | | | Privacy Preservation | | ML Efficacy | Structural Fidelity |
|---|---|---|---|---|---|---|---|---|
| | Shape ↑ | Trend ↑ | $\alpha$-precision ↑ | $\beta$-recall ↑ | Authenticity ↑ | DCR Score ↑ | Local utility ↑ | Global utility ↑ |
| **Real Data** | | | | | | | | |
| Real Data (Train) | $1.00_{\pm 0.00}$ | $1.00_{\pm 0.00}$ | $1.00_{\pm 0.00}$ | $1.00_{\pm 0.00}$ | $0.00_{\pm 0.00}$ | $0.00_{\pm 0.00}$ | $1.00_{\pm 0.00}$ | $1.00_{\pm 0.00}$ |
| Real Data (Holdout) | $0.98_{\pm 0.00}$ | $0.81_{\pm 0.04}$ | $0.98_{\pm 0.01}$ | $0.53_{\pm 0.01}$ | $0.64_{\pm 0.01}$ | $0.07_{\pm 0.01}$ | $-$ | $-$ |
| **Dataset-specific Model** | | | | | | | | |
| SMOTE | $0.96_{\pm 0.00}$ | $0.66_{\pm 0.02}$ | $0.74_{\pm 0.01}$ | $0.61_{\pm 0.01}$ | $0.55_{\pm 0.01}$ | $0.04_{\pm 0.00}$ | $0.94_{\pm 0.01}$ | $0.61_{\pm 0.01}$ |
| TabSDS | $0.69_{\pm 0.00}$ | $0.52_{\pm 0.02}$ | $0.60_{\pm 0.01}$ | $0.17_{\pm 0.01}$ | $0.63_{\pm 0.01}$ | $0.09_{\pm 0.01}$ | $0.61_{\pm 0.01}$ | $0.52_{\pm 0.01}$ |
| TVAE | $0.86_{\pm 0.00}$ | $0.65_{\pm 0.03}$ | $0.80_{\pm 0.04}$ | $0.16_{\pm 0.02}$ | $0.95_{\pm 0.01}$ | $0.17_{\pm 0.02}$ | $0.84_{\pm 0.01}$ | $0.65_{\pm 0.02}$ |
| GOGGLE | $0.50_{\pm 0.03}$ | $0.35_{\pm 0.04}$ | $0.33_{\pm 0.08}$ | $0.02_{\pm 0.01}$ | $1.00_{\pm 0.00}$ | $0.15_{\pm 0.02}$ | $0.48_{\pm 0.02}$ | $0.16_{\pm 0.01}$ |
| CTGAN | $0.84_{\pm 0.02}$ | $0.77_{\pm 0.01}$ | $0.91_{\pm 0.06}$ | $0.11_{\pm 0.03}$ | $0.97_{\pm 0.01}$ | $0.17_{\pm 0.02}$ | $0.80_{\pm 0.02}$ | $0.48_{\pm 0.02}$ |
| NFlow | $0.89_{\pm 0.01}$ | $0.78_{\pm 0.05}$ | $0.86_{\pm 0.10}$ | $0.08_{\pm 0.02}$ | $0.98_{\pm 0.01}$ | $0.22_{\pm 0.03}$ | $0.68_{\pm 0.03}$ | $0.39_{\pm 0.03}$ |
| ARF | $0.95_{\pm 0.00}$ | $0.60_{\pm 0.04}$ | $0.97_{\pm 0.01}$ | $0.18_{\pm 0.02}$ | $0.94_{\pm 0.01}$ | $0.16_{\pm 0.02}$ | $0.89_{\pm 0.02}$ | $0.49_{\pm 0.01}$ |
| TabDDPM | $0.97_{\pm 0.00}$ | $0.96_{\pm 0.01}$ | $0.98_{\pm 0.00}$ | $0.24_{\pm 0.01}$ | $0.91_{\pm 0.01}$ | $0.14_{\pm 0.02}$ | $0.89_{\pm 0.01}$ | $0.66_{\pm 0.01}$ |
| CDTD | $0.70_{\pm 0.00}$ | $0.61_{\pm 0.02}$ | $0.75_{\pm 0.01}$ | $0.16_{\pm 0.01}$ | $0.82_{\pm 0.01}$ | $0.15_{\pm 0.02}$ | $0.72_{\pm 0.01}$ | $0.48_{\pm 0.02}$ |
| TabSyn | $0.94_{\pm 0.01}$ | $0.84_{\pm 0.03}$ | $0.92_{\pm 0.03}$ | $0.34_{\pm 0.03}$ | $0.85_{\pm 0.02}$ | $0.12_{\pm 0.01}$ | $0.90_{\pm 0.01}$ | $0.73_{\pm 0.03}$ |
| TabDiff | $0.92_{\pm 0.01}$ | $0.82_{\pm 0.01}$ | $0.78_{\pm 0.02}$ | $0.06_{\pm 0.02}$ | $0.99_{\pm 0.01}$ | $0.26_{\pm 0.04}$ | $0.50_{\pm 0.01}$ | $0.69_{\pm 0.02}$ |
| NRGBoost | $0.91_{\pm 0.01}$ | $0.93_{\pm 0.01}$ | $0.92_{\pm 0.01}$ | $0.20_{\pm 0.01}$ | $0.92_{\pm 0.01}$ | $0.14_{\pm 0.01}$ | $0.74_{\pm 0.02}$ | $0.36_{\pm 0.01}$ |
| TabNAT | $0.67_{\pm 0.00}$ | $0.61_{\pm 0.01}$ | $0.77_{\pm 0.01}$ | $0.10_{\pm 0.01}$ | $0.82_{\pm 0.01}$ | $0.18_{\pm 0.02}$ | $0.61_{\pm 0.01}$ | $0.32_{\pm 0.02}$ |
| TabularARGN | $0.85_{\pm 0.00}$ | $0.70_{\pm 0.03}$ | $0.91_{\pm 0.01}$ | $0.05_{\pm 0.01}$ | $0.99_{\pm 0.00}$ | $0.22_{\pm 0.03}$ | $0.76_{\pm 0.02}$ | $0.32_{\pm 0.03}$ |
| **Foundation Model** | | | | | | | | |
| GReaT | $0.77_{\pm 0.01}$ | $0.48_{\pm 0.03}$ | $0.88_{\pm 0.02}$ | $0.04_{\pm 0.01}$ | $0.99_{\pm 0.00}$ | $0.24_{\pm 0.03}$ | $0.72_{\pm 0.03}$ | $0.04_{\pm 0.01}$ |
| TabPFN | $0.72_{\pm 0.01}$ | $0.51_{\pm 0.03}$ | $0.78_{\pm 0.04}$ | $0.08_{\pm 0.01}$ | $0.76_{\pm 0.00}$ | $0.17_{\pm 0.02}$ | $0.66_{\pm 0.02}$ | $0.38_{\pm 0.01}$ |
| TabDPT | $0.71_{\pm 0.01}$ | $0.52_{\pm 0.01}$ | $0.57_{\pm 0.03}$ | $0.07_{\pm 0.01}$ | $0.86_{\pm 0.01}$ | $0.14_{\pm 0.02}$ | $0.69_{\pm 0.01}$ | $0.36_{\pm 0.01}$ |
| Mitra | $0.50_{\pm 0.01}$ | $0.45_{\pm 0.03}$ | $0.45_{\pm 0.04}$ | $0.04_{\pm 0.01}$ | $0.79_{\pm 0.00}$ | $0.15_{\pm 0.02}$ | $0.66_{\pm 0.02}$ | $0.31_{\pm 0.01}$ |
| LimiX | $0.68_{\pm 0.01}$ | $0.56_{\pm 0.03}$ | $0.72_{\pm 0.04}$ | $0.09_{\pm 0.01}$ | $0.85_{\pm 0.00}$ | $0.17_{\pm 0.02}$ | $0.63_{\pm 0.02}$ | $0.39_{\pm 0.01}$ |
| TabICL | $0.78_{\pm 0.00}$ | $0.63_{\pm 0.03}$ | $0.80_{\pm 0.03}$ | $0.16_{\pm 0.01}$ | $0.79_{\pm 0.01}$ | $0.14_{\pm 0.02}$ | $0.76_{\pm 0.01}$ | $0.50_{\pm 0.02}$ |
| TabEBM | $0.92_{\pm 0.01}$ | $0.71_{\pm 0.04}$ | $0.89_{\pm 0.02}$ | $0.14_{\pm 0.01}$ | $0.94_{\pm 0.01}$ | $0.16_{\pm 0.02}$ | $0.87_{\pm 0.01}$ | $0.56_{\pm 0.01}$ |
| CTSyn | $0.65_{\pm 0.01}$ | $0.54_{\pm 0.02}$ | $0.68_{\pm 0.02}$ | $0.12_{\pm 0.01}$ | $0.72_{\pm 0.01}$ | $0.14_{\pm 0.02}$ | $0.68_{\pm 0.01}$ | $0.32_{\pm 0.01}$ |
| **TabFORGE (Ours)** | $0.90_{\pm 0.01}$ | $0.83_{\pm 0.03}$ | $0.87_{\pm 0.06}$ | $0.26_{\pm 0.02}$ | $0.90_{\pm 0.01}$ | $0.14_{\pm 0.02}$ | $0.86_{\pm 0.02}$ | $0.69_{\pm 0.02}$ |

*Table 74.* **Raw benchmark results of 23 tabular generators on the "Zernike" dataset.** We report the mean $\pm$ std of each metric across 10 repeated data splits. For benchmark generators, "$-$" denotes failed convergence of a specific model or unexpected values in the synthetic data that caused the evaluation metric computation to crash. We highlight the **First**, **Second**, and **Third** best performances for each metric. TabFORGE generally achieves competitive performance against the benchmark generators while maintaining a reduced risk of overfitting.

| Generator | Density Estimation | | | | Privacy Preservation | | ML Efficacy | Structural Fidelity |
|---|---|---|---|---|---|---|---|---|
| | Shape ↑ | Trend ↑ | $\alpha$-precision ↑ | $\beta$-recall ↑ | Authenticity ↑ | DCR Score ↑ | Local utility ↑ | Global utility ↑ |
| **Real Data** | | | | | | | | |
| Real Data (Train) | $1.00_{\pm0.00}$ | $1.00_{\pm0.00}$ | $1.00_{\pm0.00}$ | $1.00_{\pm0.00}$ | $0.00_{\pm0.00}$ | $0.00_{\pm0.00}$ | $1.00_{\pm0.00}$ | $1.00_{\pm0.00}$ |
| Real Data (Holdout) | $0.96_{\pm0.00}$ | $0.97_{\pm0.00}$ | $0.97_{\pm0.01}$ | $0.50_{\pm0.02}$ | $0.70_{\pm0.01}$ | $0.15_{\pm0.00}$ | $-$ | $-$ |
| **Dataset-specific Model** | | | | | | | | |
| SMOTE | $\mathbf{0.95}_{\pm0.01}$ | $0.74_{\pm0.00}$ | $0.76_{\pm0.02}$ | $\mathbf{0.73}_{\pm0.01}$ | $0.39_{\pm0.01}$ | $0.07_{\pm0.00}$ | $\mathbf{0.96}_{\pm0.01}$ | $\mathbf{0.69}_{\pm0.01}$ |
| TabSDS | $0.72_{\pm0.00}$ | $0.61_{\pm0.00}$ | $0.62_{\pm0.02}$ | $0.14_{\pm0.01}$ | $0.69_{\pm0.01}$ | $0.29_{\pm0.00}$ | $0.52_{\pm0.02}$ | $0.45_{\pm0.01}$ |
| TVAE | $0.83_{\pm0.01}$ | $0.73_{\pm0.01}$ | $0.53_{\pm0.03}$ | $0.02_{\pm0.01}$ | $1.00_{\pm0.00}$ | $0.43_{\pm0.01}$ | $0.38_{\pm0.06}$ | $0.59_{\pm0.01}$ |
| GOGGLE | $0.50_{\pm0.01}$ | $0.59_{\pm0.01}$ | $0.40_{\pm0.07}$ | $0.00_{\pm0.00}$ | $1.00_{\pm0.00}$ | $0.44_{\pm0.02}$ | $0.08_{\pm0.02}$ | $0.11_{\pm0.00}$ |
| CTGAN | $0.81_{\pm0.02}$ | $0.86_{\pm0.01}$ | $0.68_{\pm0.06}$ | $0.00_{\pm0.00}$ | $\mathbf{1.00}_{\pm0.00}$ | $0.51_{\pm0.02}$ | $0.44_{\pm0.05}$ | $0.32_{\pm0.01}$ |
| NFlow | $0.90_{\pm0.01}$ | $0.66_{\pm0.01}$ | $0.81_{\pm0.04}$ | $0.00_{\pm0.00}$ | $\mathbf{1.00}_{\pm0.00}$ | $0.56_{\pm0.01}$ | $0.11_{\pm0.02}$ | $0.23_{\pm0.01}$ |
| ARF | $\mathbf{0.94}_{\pm0.00}$ | $0.81_{\pm0.01}$ | $\mathbf{0.84}_{\pm0.02}$ | $0.01_{\pm0.00}$ | $1.00_{\pm0.00}$ | $0.47_{\pm0.01}$ | $0.69_{\pm0.05}$ | $0.37_{\pm0.01}$ |
| TabDDPM | $0.38_{\pm0.01}$ | $0.85_{\pm0.01}$ | $0.00_{\pm0.00}$ | $0.00_{\pm0.00}$ | $1.00_{\pm0.00}$ | $\mathbf{1.00}_{\pm0.00}$ | $0.42_{\pm0.04}$ | $0.60_{\pm0.01}$ |
| CDTD | $0.55_{\pm0.00}$ | $0.75_{\pm0.00}$ | $0.44_{\pm0.02}$ | $0.20_{\pm0.02}$ | $0.66_{\pm0.01}$ | $0.34_{\pm0.01}$ | $0.47_{\pm0.02}$ | $0.39_{\pm0.01}$ |
| TabSyn | $0.91_{\pm0.01}$ | $\mathbf{0.95}_{\pm0.01}$ | $0.77_{\pm0.07}$ | $\mathbf{0.30}_{\pm0.05}$ | $0.86_{\pm0.03}$ | $0.30_{\pm0.01}$ | $0.57_{\pm0.01}$ | $0.65_{\pm0.02}$ |
| TabDiff | $\mathbf{0.94}_{\pm0.00}$ | $0.74_{\pm0.01}$ | $0.79_{\pm0.02}$ | $0.00_{\pm0.00}$ | $1.00_{\pm0.00}$ | $0.54_{\pm0.01}$ | $0.17_{\pm0.03}$ | $\mathbf{0.65}_{\pm0.01}$ |
| NRGBoost | $0.92_{\pm0.01}$ | $0.91_{\pm0.01}$ | $\mathbf{0.93}_{\pm0.02}$ | $0.27_{\pm0.03}$ | $0.87_{\pm0.02}$ | $0.30_{\pm0.01}$ | $\mathbf{0.79}_{\pm0.04}$ | $0.37_{\pm0.00}$ |
| TabNAT | $0.56_{\pm0.00}$ | $0.67_{\pm0.00}$ | $0.48_{\pm0.02}$ | $0.08_{\pm0.01}$ | $0.78_{\pm0.01}$ | $0.47_{\pm0.01}$ | $0.22_{\pm0.02}$ | $0.27_{\pm0.01}$ |
| TabularARGN | $0.84_{\pm0.00}$ | $0.64_{\pm0.00}$ | $0.72_{\pm0.02}$ | $0.00_{\pm0.00}$ | $1.00_{\pm0.00}$ | $\mathbf{0.59}_{\pm0.01}$ | $0.15_{\pm0.02}$ | $0.21_{\pm0.00}$ |
| **Foundation Model** | | | | | | | | |
| GReaT | $0.11_{\pm0.00}$ | $0.59_{\pm0.01}$ | $0.00_{\pm0.00}$ | $0.00_{\pm0.00}$ | $\mathbf{1.00}_{\pm0.00}$ | $\mathbf{0.88}_{\pm0.02}$ | $0.12_{\pm0.03}$ | $0.01_{\pm0.00}$ |
| TabPFN | $0.67_{\pm0.00}$ | $0.55_{\pm0.01}$ | $0.56_{\pm0.02}$ | $0.00_{\pm0.00}$ | $0.85_{\pm0.00}$ | $0.43_{\pm0.01}$ | $0.25_{\pm0.02}$ | $0.20_{\pm0.00}$ |
| TabDPT | $0.62_{\pm0.01}$ | $0.61_{\pm0.00}$ | $0.50_{\pm0.04}$ | $0.00_{\pm0.00}$ | $0.87_{\pm0.00}$ | $0.41_{\pm0.01}$ | $0.25_{\pm0.02}$ | $0.24_{\pm0.01}$ |
| Mitra | $0.62_{\pm0.01}$ | $0.59_{\pm0.01}$ | $0.52_{\pm0.02}$ | $0.01_{\pm0.00}$ | $0.89_{\pm0.00}$ | $0.42_{\pm0.01}$ | $0.11_{\pm0.02}$ | $0.17_{\pm0.00}$ |
| LimiX | $0.77_{\pm0.00}$ | $0.61_{\pm0.01}$ | $0.66_{\pm0.01}$ | $0.00_{\pm0.00}$ | $0.85_{\pm0.00}$ | $0.40_{\pm0.01}$ | $0.25_{\pm0.02}$ | $0.23_{\pm0.00}$ |
| TabICL | $0.84_{\pm0.00}$ | $0.69_{\pm0.00}$ | $0.60_{\pm0.02}$ | $0.07_{\pm0.01}$ | $0.76_{\pm0.01}$ | $0.33_{\pm0.00}$ | $0.45_{\pm0.02}$ | $0.41_{\pm0.01}$ |
| TabEBM | $0.94_{\pm0.00}$ | $\mathbf{0.92}_{\pm0.01}$ | $0.81_{\pm0.03}$ | $\mathbf{0.30}_{\pm0.02}$ | $0.75_{\pm0.02}$ | $0.24_{\pm0.00}$ | $\mathbf{0.89}_{\pm0.02}$ | $0.50_{\pm0.01}$ |
| CTSyn | $0.40_{\pm0.00}$ | $0.61_{\pm0.00}$ | $0.34_{\pm0.03}$ | $0.12_{\pm0.02}$ | $0.89_{\pm0.01}$ | $0.46_{\pm0.01}$ | $0.28_{\pm0.02}$ | $0.30_{\pm0.01}$ |
| **TabFORGE (Ours)** | $0.93_{\pm0.00}$ | $\mathbf{0.94}_{\pm0.01}$ | $\mathbf{0.90}_{\pm0.04}$ | $0.13_{\pm0.02}$ | $0.95_{\pm0.01}$ | $0.35_{\pm0.01}$ | $0.77_{\pm0.04}$ | $\mathbf{0.66}_{\pm0.01}$ |

