# OpenReview forum: "A Generative Foundation Model for Heterogeneous Tabular Data"
_ICML.cc/2026/Workshop/FMSD — FMSD @ ICML 2026 Poster_

### Official Review · Reviewer_RAKu · 2026-05-15

**Rating:** 8
**Confidence:** 5

**Review:**

## Summary

This paper presents TabFORGE, a generative foundation model designed for tabular data. The model adapts an existing tabular foundation model (Real-TabPFN-2.5) to extract embeddings and then runs a diffusion model to generate new samples on top of these embeddings. The diffusion model and decoder are pre-trained as part of this work, whereas Real-TabPFN-2.5 is an existing pre-trained architecture. At generation time, the diffusion model and decoder are first further fine-tuned on the incoming dataset for a short time, and then new data can be generated. Results show strong performance on a newly-created benchmark across utility and distribution matching.

## Strengths

- Novel architecture
- Intuitive and clever idea to use existing TFM as an embedding model to capture generic properties to then build a generic generator on top
- Well-motivated: TFMs don't have well-established generative properties, unlike language or images
- Very strong results on fidelity and distribution matching
- Fast generation, I think

## Areas for Improvement

- Among baselines, TabFORGE is the most "memorized" of all. This may indicate it has the best fit -- and it isn't more "memorized" than the holdout set -- but it could also incur an unfavourable amount of similarity to the real data in some situations. I would recommend showing something like a Pareto front of, e.g., DCR vs. Utility, to demonstrate trade-offs between the level of memorization / copying and the quality of the resulting dataset produced. This would offer a more holistic view of the performance.
- Fine-tuning is hidden in the Appendix, which is somewhat important to bring up in my opinion. It slightly weakens the claim that this is not a dataset-specific generator (although it is still very light fine-tuning, the point still stands).
- Related to this, the paper is way too long for a workshop ... nearly 80 pages ...
- There is not much discussion of runtime. It is stated that TabFORGE is much faster than auto-regressive schemes, but this is not quantified anywhere that I can see. There is also the matter of pre-training time, which is worth talking about in the overall runtime, that is also somewhere in the very long appendix. Of course, pre-training time does get amortized across tasks, especially as the number of tasks increases, but some discussion is warranted.
- A new benchmark is produced by this work, but I don't see any detailed discussion on how these datasets were chosen, which should accompany the creation of a new benchmark. Compare against, e.g., the level of detail put into TabArena for selecting datasets (this level of detail is not required, especially for a workshop paper, but at least *some* discussion of how datasets were chosen besides "not leaking" would be helpful).

## Detailed Comments

- Feels like this is an unsubstantiated claim in the abstract: "A key reason is their misalignment with the distinctive causal structural prior of heterogeneous tabular data." This is not explicitly shown anywhere in the paper.
- Abstract is excessively long for a workshop paper
- In-line references not provided in chronological order
- "Mixed-type" not defined anywhere - what mixed types are we talking about? Presumably categorical and numerical, but what about text, time-related, etc.
- In the related work, it is worth mentioning that the training of a TFM on real data is related to existing work like TabDPT and Real-TabPFN-2.5 (I know it is mentioned that the corpus was taken from Real-TabPFN-2.5, but there remains an ongoing debate about training TFMs with synthetic or real data).


## Justification of Score

The paper is quite interesting overall and I would argue for its acceptance at the workshop. Despite my review being more verbose on the flaws, the strengths remain quite compelling: novelty, motivation, and good results are very much in its favour.

---

### Official Review · Reviewer_2osZ · 2026-05-19
**Novel Latent-Diffusion Approach, but Causal Framing and Implementation Details Need A Lot More Clarification**

**Rating:** 6
**Confidence:** 5

**Review:**

# Summary
This paper proposes TabFORGE, a generative foundation model for heterogeneous tabular data. It uses a frozen PFN encoder to map tables into a 3d latent embeddings, then it trains a score-based diffusion transformer in that latent space, and finally it uses a denoising-aligned decoder to map generated latents back to tabular values. By treating the target as an additional feature, the model claims to generate the full joint table distribution rather than only model a supervised prediction target.

# Strengths
The paper tackles an important problem: foundation models for tabular data generation. The proposed architecture is interesting, especially the combination of pre-trained tabular representations, latent diffusion, and denoising-aligned decoding. The evaluation covers 45 real-world datasets and 22 baselines, and the reported results are strong.

# Areas for Improvement
1. The main weakness is that the paper’s causal framing is too strong. The method does not explicitly learn causal graphs, interventions, or structural causal models. The reported global utility metric appears to measure cross-feature predictive/dependency preservation, not causal correctness.
2. The paper should also define global utility clearly in the main text, since it is central to Figure 2. Readers should not need to consult TabStruct to understand the headline metric.
3. Another important clarification is how the PFN/TabPFN-style encoder handles the target column. Standard TabPFN treats labels specially. The implementation details here are important.
4. The compute cost is substantial: the method requires expensive pre-training and still needs per-dataset fine-tuning, so the “foundation model” framing should be presented carefully.

# Detailed Comments
1. Please define global utility explicitly in the main paper, ideally with a formula and short explanation.
2. Please clarify what “causality-aware” means. The current evidence supports structure awareness more than causal understanding.
3. Please explain exactly how the target column is passed through the PFN encoder. Is it tokenized like an ordinary feature, or does it use a label-specific pathway?
4. Please clarify how TabPFN baselines are converted into generators, especially how the first feature is sampled in the autoregressive process (Does it start from Gaussian noise or something else?)
5. Please discuss compute more transparently, including pre-training cost and per-dataset fine-tuning cost.
Additional ablations with alternative or random encoders would help show whether the gains truly come from the claimed PFN representation.

# Justification of Score
I would give this paper a weak accept / borderline accept. The method is novel, relevant, and empirically strong, but the claims need clearer framing. In particular, the causal interpretation, global utility definition, PFN encoder interface, and compute tradeoffs should be clarified.

---

### Official Review · Reviewer_aUxn · 2026-05-20
**Instead of training a separate tabular generator from scratch for every dataset, the paper proposes using a frozen PFN encoder to map tabular data into a latent representation, applying a latent diffusion Transformer in that space, and then using a denoising-aligned decoder to generate synthetic tabular data. The overall framework is interesting, but I think the paper should more clearly justify the novelty and soundness of its main claims.**

**Rating:** 5
**Confidence:** 3

**Review:**

Latent-space diffusion models are becoming increasingly popular for time-series generation, forecasting, and imputation. While I was able to understand the general idea of the paper, I believe there are several ambiguities regarding the soundness and clarity of the proposed contributions.
1. The paper frequently uses terms such as “causal structure” and “causality-aware,” but it is not fully clear how causal relationships are explicitly identified or preserved. If the model mainly preserves statistical dependencies among features, the term “causal structure” may be stronger than what the method actually demonstrates.

2. The paper states that a pretrained PFN encoder maps heterogeneous tabular data into a causality-aware latent space(PFN should be expanded at its first appearance), More importantly, could the authors clarify how this encoder is pretrained, what causal assumptions are embedded in it, and why its early-to-middle-layer representations should be interpreted as causal rather than merely statistical?

3. The paper argues that autoregressive tabular models suffer from feature-order bias. However, Transformer self-attention can model interactions among all features. Could the authors clarify why attention alone is insufficient to mitigate this bias? Is the concern specifically due to positional encodings, autoregressive factorization, causal masking, or empirical sensitivity to column permutations?

4. The mapping from X∈R^(N×(D+1)) to H∈R^(N×(D+1) ×k) appears to expand each feature into a k-dimensional token embedding rather than compressing the data into a lower-dimensional latent space. Could the authors clarify whether this latent representation is intended as dimensionality reduction, feature tokenization, or representation enrichment? Since this mapping is central to the method, an ablation over the embedding dimension $k$ would strengthen the paper.

5. The paper cites several tabular diffusion models, including TabDDPM, TabDiff, continuous diffusion for mixed-type tabular data, and TabSyn, which is directly related to score-based diffusion in latent space. However, since TabFORGE relies on diffusion over learned latent feature embeddings, the related work could more explicitly discuss broader latent-diffusion literature. In particular, it would be useful to connect the method to studies explaining why diffusion can be more stable or effective in latent space than in observation space, and how latent dimensionality affects the tradeoff between representation capacity, denoising quality, and decoder reconstruction quality.

6. The training losses are not sufficiently clear. The paper should explicitly state the diffusion objective, the denoising target, and the decoder reconstruction loss for mixed-type features. Since the method has multiple stages, a complete training algorithm or pseudocode would significantly improve reproducibility.

7. The results are promising, especially for global utility and structural fidelity. However, the paper should clarify how much these metrics truly support the causal-structure claim, rather than only showing stronger statistical or distributional similarity between real and synthetic data.

Overall, my main concern is not the general direction of the work, which is interesting, but the soundness and clarity of the claimed contributions, particularly around causality, latent-space design, and the connection between the reported metrics and the paper’s main claims.